# Aspects of 4d supersymmetric dynamics and geometry

Shlomo S. Razamat[1], Evyatar Sabag[1,2], Orr Sela[1,3] and Gabi Zafrir[4,5,6]

**1** Department of Physics, Technion, Haifa, 32000, Israel
**2** Mathematical Institute, University of Oxford, Oxford, OX2 6GG, UK
**3** Mani L. Bhaumik ITP, Department of Physics and Astronomy, UCLA, CA 90095, USA
**4** Dipartimento di Fisica, Universit'a di Milano-Bicocca INFN, I-20126 Milano, Italy
**5** YITP, Stony Brook University, Stony Brook, NY 11794-3840, USA
**6** SCGP, Stony Brook University, Stony Brook, NY 11794-3840, USA

## Abstract

In this set of five Lectures we present a basic toolbox to discuss the dynamics of four dimensional supersymmetric quantum field theories. In particular we overview the program of geometrically engineering the four dimensional supersymmetric models as compactifications of six dimensional SCFTs. We discuss how strong coupling phenomena in four dimensions, such as duality and emergence of symmetry, can be naturally imbedded in the geometric constructions. The Lectures mostly review results which previously appeared in the literature but also contain some unpublished derivations.

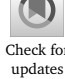

# 1 Introduction

Quantum field theory is a framework to study the dynamics of a wide variety of quantum systems. One of the interesting open problems in understanding the predictions of this framework is the question of strong coupling. Strong coupling problems have numerous avatars. For example, we can start from a free field theory and turn on relevant deformations flowing to strong coupling in the infra-red (IR). We then can ask questions about the deep IR physics which are hard to answer directly using the UV weakly coupled starting point.

Such issues become more tractable once we introduce enough supersymmetry. In 4d, which will be the focus of these Lectures, enough means four supercharges, *i.e.* minimal $\mathcal{N} = 1$ supersymmetry. With this amount of supersymmetry one can use a wide variety of techniques, ultimately related to holomorphy of various quantities [1], to deduce non pertur-

batively certain conclusions about strongly coupled phases of QFTs. Typically the power of holomorphy gives us a variety of quantities we can count: Most, if not all, of the exact results one can obtain about supersymmetric QFTs can be related to counting problems.[1]

### Dualities and emergence of symmetry

As an example of a strong coupling effect one can discuss is the notion of *duality*. A given CFT $A$ in the UV deformed by a relevant perturbation can flow in the IR to a theory (which can be free, gapped, or interacting CFT) which is equivalent to a theory obtained by staring with another CFT $B$ and some relevant deformation. When an RG flow is involved we will refer to such dualities as IR dualities. In case the deformations preserve conformality, that is they are exactly marginal, we will refer to these effects as conformal dualities. See Figure 1. In the last 25 years, following in particular [2], numerous examples of such dualities have been conjectured using a wide variety of exact supersymmetric techniques. The canonical example is the IR equivalence (in a certain range of parameters) of $SU(N)$ SQCD with $N_f$ flavors and $SU(N_f-N)$ SQCD with $N_f$ flavors, gauge singlet fields and a superpotential. The IR fixed point is fully determined (or labeled) by a choice of a pair, UV CFT and the relevant deformation. The statement of the duality is that this labeling is not unique and moreover might be not unique in an interesting way (*e.g.* the UV theories having different gauge groups). A canonical example of a conformal duality is the statement that $\mathcal{N} = 4$ SYM with gauge groups $G$ and $^L G$ (the Langlands dual of $G$) reside at two cusps of the same conformal manifold (see *e.g.* [3] for a detailed discussion).

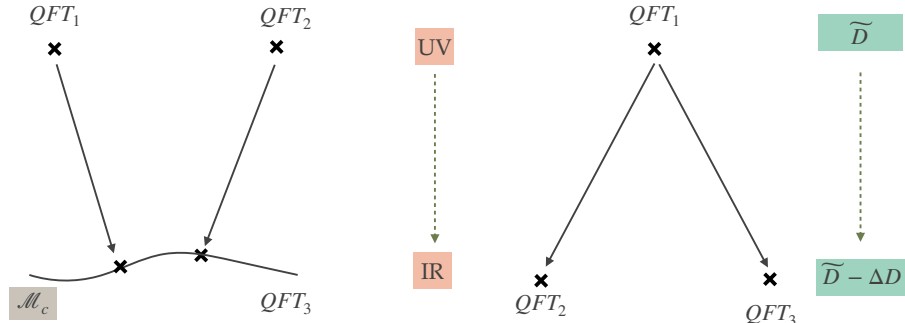

Figure 1: On the left we have a depiction of IR duality: Two different UV QFTs flowing to the same conformal manifold in the IR. In this paper we will refer to theories differing by exactly marginal deformations as equivalent. On the right we have a depiction of the notion of UV duality: Two different QFTs in the IR obtainable as two different deformations of a single CFT in the UV. This notions becomes very useful once one considers flows starting with a strongly coupled CFT leading to weakly coupled QFTs in the IR. Examples of these include compactifications on a circle of 6d SCFTs.

Another interesting strong coupling effect is the emergence of symmetry in the IR. The global symmetry of the IR fixed point can be larger than the one of the UV starting point and the interesting question is whether this can be understood directly in the UV. A simple example is $SU(2)$ with $N_f$ flavors and symmetry $SU(2N_f)$ describing in the IR the $SU(N_f - 2)$ SQCD with $N_f$ flavors with only the $SU(N_f)^2 \times U(1)$ symmetry visible in the UV of the latter, see *e.g.*

---

[1]The counting problem might be in the theory of our interest or maybe in a higher dimensional theory reducing to it. For example the $\mathbb{S}^d$ partition functions in $d$ dimensions can be related to various indices, $\mathbb{S}^d \times \mathbb{S}^1$ partition functions, in $d + 1$ dimensions.

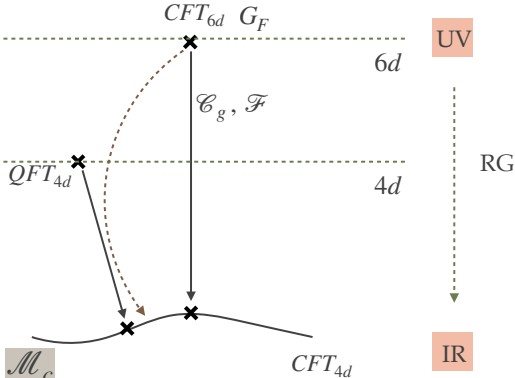

Figure 2: Depiction of a duality across dimensions. Here we give an example of a duality between 6d flow and a 4d flow. The 4d theory might be UV complete (asymptotically free) or UV completed by the 6d theory.

[4]. Other examples involve emergence of supersymmetry in the flow: *e.g.* using Intriligator-Seiberg duality [5][2] the $SU(2)$ $\mathcal{N} = 4$ SYM can be obtained by starting from $SU(2) \times SU(2)$ $\mathcal{N} = 1$ SQCD with the charged matter consisting of three bi-fundamentals.[3]

An interesting question is thus to bundle all of the scattered instances of interesting strong coupling dynamics into some uniform framework which will give some sort of an explanation or an understanding of when and how these phenomena occur. Yet another motivation for such a framework can be found by asking a sort of an inverse question: Given a strongly coupled SCFT with given properties to find a weakly couple UV theory which flows to it. The UV theory might exhibit less symmetry than the IR one, and in fact it often has to do so. For example, listing all the 4d $\mathcal{N} = 2$ Lagrangians leading to interacting SCFTs [9] one just does not find candidate descriptions of many of 4d $\mathcal{N} = 2$ SCFTs which can be obtained using a variety of more abstract techniques, *e.g.* coming from string theory.

## Dualities across dimensions

Typically one considers dualities between QFTs starting from two theories in the same number of dimensions and ending in the same number of dimensions. However, in recent years, following the seminal work of [8], it has been realized that much can be achieved, in particular answering the questions posed above, if one discusses flows across dimensions. Let us first then define the notion of an *IR duality across dimensions*. We can start from a higher dimensional CFT, in this paper we will start in $\widetilde{D} = 6$, and deform the theory by placing it on a compact geometry of dimension $D$, *e.g.* a Riemann surface with $D = 2$. Studying this setup at low energy, *i.e.* much lower than the scale set by the compact geometry, we will arrive at an effective $\widetilde{D} - D$ dimensional QFT. An interesting question is whether there exists a $\widetilde{D} - D$ dimensional *weakly coupled* QFT flowing to the same effective theory in the IR. If such a theory exists we will refer to the $\widetilde{D}$ deformed theory and the $\widetilde{D} - D$ one as being IR dual across dimensions. See Figure 2.

---

[2]See also [6] for recent discussion.

[3]Let us mention in passing here for the experts that this duality has a class $\mathcal{S}$ [7,8] interpretation. One can take two $T_N$ trinions and glue them together with $\mathcal{N} = 1$ vector multiplets; flip the two maximal punctures; close then one of the two punctures completely and the other one to a minimal puncture. This theory has the same geometric data as the one of genus one with a single minimal puncture and $\mathcal{N} = 2$ preserving flux, which is $\mathcal{N} = 4$ SYM with a decoupled hypermultiplet. The $N = 2$ case is then the Intriligator-Seiberg duality while for higher $N$ one has novel strongly coupled generalizations of this. *In our Lectures we will review the notions needed to understand the statements in this footnote.*

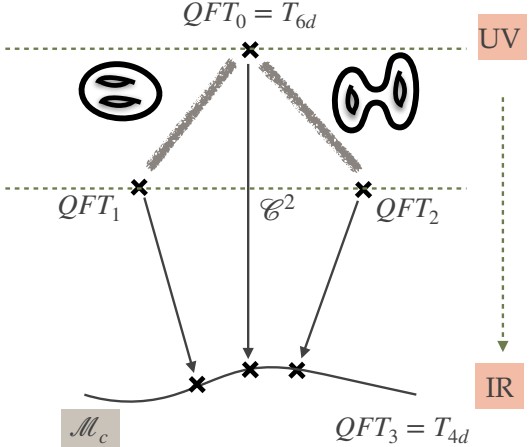

Figure 3: 4d dynamics from different decompositions of the compactification geometry.

Here we thus can also label the IR $\widetilde{D} - D$ dimensional QFT by a pair (CFT, deformation) but this might involve a higher dimensional CFT and a geometric deformation. As we will be mainly interested in 4d physics our starting points will be in 6d (and 5d) where interesting SCFTs are all strongly coupled and do not have a simple field theoretic definition, see *e.g.* [10]. As the starting point is strongly coupled, such geometric constructions often lead to 4d theories with properties which are hard or impossible to engineer directly in 4d *insisting on all the symmetries being manifest*. This in particular led to many such theories being referred to as *non-Lagrangian*. Across dimensional duals, if existing, thus would provide for a Lagrangian definition of such SCFTs. An example of such a duality is a geometric, class $\mathcal{S}$ [7,8], construction as compactification on punctured spheres of a $(2, 0)$ 6d SCFT of certain $\mathcal{N} = 2$ Argyres-Douglas SCFTs [11], and an alternative description of these SCFTs starting with a $\mathcal{N} = 1$ weakly coupled gauge theory in 4d [12].

As with in-dimension dualities some of the symmetries of the IR QFT might be explicit in the UV in one description but emergent in the other. Systematically understanding such across dimensional dualities will give us a handle over understanding of emergence of symmetry. In fact the appearance of geometry in the construction gives us a useful knob to start and build a systematic framework to understand emergence of symmetry and duality.

## 4d dynamics from across dimensional dualities

The main idea behind deriving 4d dualities and emergence of symmetry phenomena from 6d stems from the following factorization property of the constructions. One considers compactifying a given 6d SCFT, $T_{6d}$, on surface $\mathcal{C}$ such that the surface can be written as,

$$\mathcal{C} = \mathcal{C}_1 \oplus \mathcal{C}_2 \oplus \cdots \oplus \mathcal{C}_\ell \,. \tag{1}$$

Here $\mathcal{C}_i$ are punctured surfaces and $\oplus$ is the geometric operation of gluing two surfaces along a puncture. The compactification might be parametrized by additional geometric data, such as flux for a global symmetry supported on the surface. The operation $\oplus$ then associates to the combined surface a sum of these fluxes. We will call a set of surface $\mathcal{C}_i$ complete if any surface $\mathcal{C}$ can be constructed using these. We then first seek for across dimensional dualities associating a pair of 4d weakly coupled CFTs and deformations $(T_{\mathcal{C}_i}, \Delta_i)$ to $(T_{6d}, \mathcal{C}_i)$. Assuming that such a *dictionary* between a complete set of surfaces $\mathcal{C}_i$ and 4d theories could be found one can find

a theory dual across dimensions to $(T_{6d}, \mathcal{C})$,

$$(T_{\mathcal{C}}, \Delta) = (T_{\mathcal{C}_1}, \Delta_1) \otimes (T_{\mathcal{C}_2}, \Delta_2) \otimes \cdots \otimes (T_{\mathcal{C}_\ell}, \Delta_\ell). \tag{2}$$

Here $\otimes$ is a field theoretic operation in 4d. This can involve gauging symmetry associated with punctures (in a way we will discuss in detail in the paper), adding fields, and turning on various superpotentials couplings. The precise meaning of $\otimes$ will depend on the 6d SCFT and various other choices. For example, the complete set of $\mathcal{C}_i$ might not be unique; one can have different types of punctures (following from UV dualities exemplified in Figure 1), etc. In the statement (2) we have several RG flows, and thus hidden in it there is an assumption that these types of flows commute. Commutativity of flows is a non-trivial statement (see *e.g.* [13] and discussion below). However, assuming it and the statement (2) one can arrive at a large web of consistent results which supports the validity of the assumption.

The fact that such a construction exists is highly non-trivial: However as we will discuss in this review in various examples it can be worked out explicitly and by now somewhat systematically. An example of such a construction is the derivation of all compactifications of the $A_1$ $(2,0)$ 6d SCFT using tri-fundamentals of $SU(2)$ as across dimensional duals to compactifications on certain three-punctured spheres [8]. Given such a structure one can then generate various examples of dualities and instances of IR emergence of symmetry systematically. For example, if a given surface can be decomposed in more than one way,

$$\mathcal{C} = \mathcal{C}_1^{(1)} \oplus \mathcal{C}_2^{(1)} \oplus \cdots \oplus \mathcal{C}_\ell^{(1)} = \mathcal{C}_1^{(2)} \oplus \mathcal{C}_2^{(2)} \oplus \cdots \oplus \mathcal{C}_{\ell'}^{(2)}, \tag{3}$$

we will obtain different field theoretic descriptions of $(T_{\mathcal{C}}, \Delta)$ which should be IR equivalent by construction,

$$\begin{aligned}(T_{\mathcal{C}}, \Delta) &= (T_{\mathcal{C}_1^{(1)}}, \Delta_1^{(1)}) \otimes (T_{\mathcal{C}_2^{(1)}}, \Delta_2^{(1)}) \otimes \cdots \otimes (T_{\mathcal{C}_\ell^{(1)}}, \Delta_\ell^{(1)}), \\ &= (T_{\mathcal{C}_1^{(2)}}, \Delta_1^{(2)}) \otimes (T_{\mathcal{C}_2^{(2)}}, \Delta_2^{(2)}) \otimes \cdots \otimes (T_{\mathcal{C}_{\ell'}^{(2)}}, \Delta_{\ell'}^{(2)}).\end{aligned} \tag{4}$$

Moreover the geometry $\mathcal{C}$ might preserve more symmetry than some of the $\mathcal{C}_i$ building blocks. Then the corresponding theories $(T_{\mathcal{C}_i}, \Delta_i)$ would have less symmetry than $(T_{\mathcal{C}}, \Delta)$. However combining them together using (2) should give in the IR the expected symmetry, if the construction is correct. Building the necessary toolkit to discuss such correspondences and discussing the systematics of these constructions will be the main goal of this review.

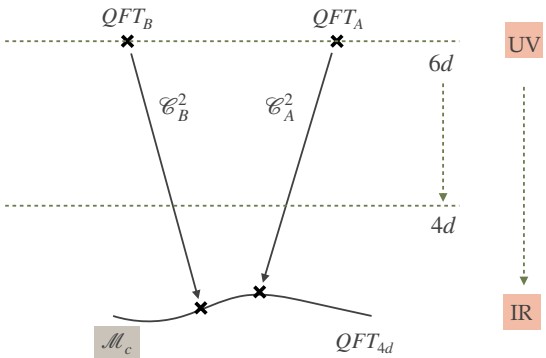

Figure 4: A depiction of a 6d duality. Starting from two different theories in 6d and deforming them by different geometries one can flow to equivalent theories in 4d.

We will discuss two explicit instances of collections of across dimension dualities, one starting with $T_{6d}$ which is described by pure $SU(3)$ SYM on its tensor branch and another with

$T_{6d}$ being the rank one E-string. These two cases turn out to be rather simple and amenable to a variety of ad hoc techniques, which we will discuss in detail. The systematic treatment of compactifications starting with a large class of more general $T_{6d}$ can be done understanding first compactifications on two punctured spheres. The problem of understanding reductions on such surfaces can be directly related to understanding duality domain walls in 5d [14]. We will discuss this procedure using again E-string as an explicit example. Understanding systematically compactifications on surfaces with more punctures can be done studying the interplay between 6d flows and across dimension flows, which we will soon describe.

## 6d dualities

Of course once we allow for across dimensional dualities we can consider starting and finishing in different combinations of dimensions. There has been a lot of work for example on compactifying 6d theories down to three dimensions and constructing 3d Lagrangians for these, see *e.g.* [15]. However, there is another interesting phenomenon that we want to mention here. One can start from two *different* 6d SCFTs, $T_{6d}^A$ and $T_{6d}^B$, deform by placing them on different geometries, $\mathcal{C}^A$ and $\mathcal{C}^B$, and consider the resulting four dimensional theories. In certain cases the 4d theories might be IR equivalent and we thus can refer to the pairs $(T_{6d}^A, \mathcal{C}^A)$ and $(T_{6d}^B, \mathcal{C}^B)$ as being 6d dual to each other. By now there are numerous examples of such dualities, though there is no systematic understanding of these. It is likely that to gain such an understanding one would need to exploit various string/M-theoretic constructions. For example, various compactifications of $(2,0)$ theories on spheres with punctures leading to $\mathcal{N} = 2$ theories in 4d turn out to be equivalent to compactifications on tori of different $(1,0)$ theories [16–18]; compactifications on certain punctured spheres of N5 branes probing $A$-type singularity is 6d dual to compactifications on tori with flux of M5 branes probing $D$-type singularity [19]. We will discuss yet another example of this phenomenon (Rank-one E-string on genus two surface without flux is the same as minimal $SU(3)$ 6d SCFT on a four punctured sphere).

## Interplay between flows in different dimensions

6d theories do not possess interesting supersymmetric relevant or marginal deformations [20]. However, given a 6d SCFT one can trigger an interesting flow by exploring the moduli space of vacua, *i.e.* turning on vacuum expectation values for some operators. One then can wonder how flows in six dimensions and across dimensions are interrelated. It turns out that the answer to this question is rather interesting and understanding it gives an explicit tool to derive compactifications on surfaces with more than two punctures. The basic idea is depicted in Figure 5. Let us consider two QFTs in 6d related by a flow triggered by a vev to some operator $\mathcal{O}_{6d}$, $QFT_1$ and $QFT_2$. Next we compactify $QFT_1$ on some surface $\mathcal{C}_A$ with some value of flux for the 6d global symmetry. Usually one can find the 4d analogue, $\mathcal{O}_{4d}$, of the operator $\mathcal{O}_{6d}$, the vev of which connects $QFT_1$ and $QFT_2$ in 6d. Turning on a vev to $\mathcal{O}_{4d}$ we flow then to a new QFT in 4d. A natural question is whether there is a surface $\mathcal{C}_B$ (and some value of flux) such that the same 4d QFT can be reached starting with $QFT_2$ in 6d. It turns out that the answer is positive but $\mathcal{C}_B$ differs from $\mathcal{C}_A$ by the number of punctures: The difference being determined by the value of flux for symmetries under which $\mathcal{O}_{6d}$ is charged. Thus understanding say compactifications on two punctured spheres for $QFT_1$ we can systematically derive compactifications on spheres with higher number of punctures for $QFT_2$. We will discuss in detail this procedure for a sequence of 6d SCFTs called $(D_{N+3}, D_{N+3})$ minimal conformal matter theories: $(1,0)$ SCFTs residing on a single M5 brane probing $D_{N+3}$ singularity in M-theory. Models with different values of $N$ are related by RG flows reducing the value of $N$. The simplest model with $N = 1$ turns out to be the rank one E-string. In particular this will give us yet another derivation of the three punctured spheres for the E-string theory.

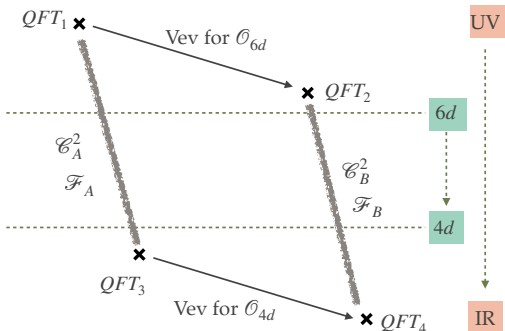

Figure 5: Commuting diagram of flows in 4d and in 6d triggered by vevs, and across dimensions triggered by geometry.

## Outline of the paper

The idea of these Lectures is to allow a reader familiar with the basics of supersymmetric dynamics in 4d (*e.g.* at the level of first chapters of [21]) to familiarize themselves in detail, and in a self contained way, with the techniques and results used to geometrically construct 4d dynamics. We tried to either explicitly review all the needed material, or to cite a paper discussing manifestly and in detail the needed points. Although string/M/F-theoretic considerations are very useful in understanding various aspects of our discussion, we restricted to purely field theoretic exposition for the sake of the Lectures being self contained. *These Lectures are not intended to be an exhaustive review of the subject but rather a pedagogical and self contained exposition of a particular slice of recent developments.*

The paper is structured as follows. In section 2 we overview the basic techniques to deal non-perturbatively with $\mathcal{N} = 1$ supersymmetric QFTs in 4d. This involves in particular the review of exact tools to extract information about the IR fixed point in 4d flows. In section 3 we discuss in detail examples of IR dualities, conformal dualities, and IR emergence of symmetry. We discuss several test cases to illustrate the applications of the techniques of section 2. Parts of Lecture II are mainly based on results reported in [22, 23]. Next in section 4 we analyze in full detail the compactification of 6d minimal $SU(3)$ SCFT to 4d. In particular we will discuss the 6d SCFT itself, its reduction to 5d, and the resulting 4d theories. This is a very tractable case (*e.g.* the analysis is simplified by the 6d theory not having continuous global (non-R) symmetry). We will illustrate various understandings following from compactifications using this example. This section is based on results of [24]. In section 5 we discuss in detail compactifications of the rank one E-string. Here we will use non generic but simple techniques to get to the result in a non-cumbersome manner. This section is mainly based on [25] as well as [22, 26]. In section 6 we discuss a systematic approach of studying 5d duality domain walls and commutative diagrams of RG flows. Here the discussion mainly follows the results derived in [27]. We use these techniques to arrive at the 4d models obtained in compactifications of the E-string theory. Finally in section 7 we discuss some generalities of geometrically engineering 4d SQFTs starting from 6d SCFTs, comment on subjects not covered in the Lectures, and make general remarks. Several appendices include pedagogically various topics in 4d/5d/6d supersymmetric physics as well as give more details of some of the computations in the bulk of the paper.

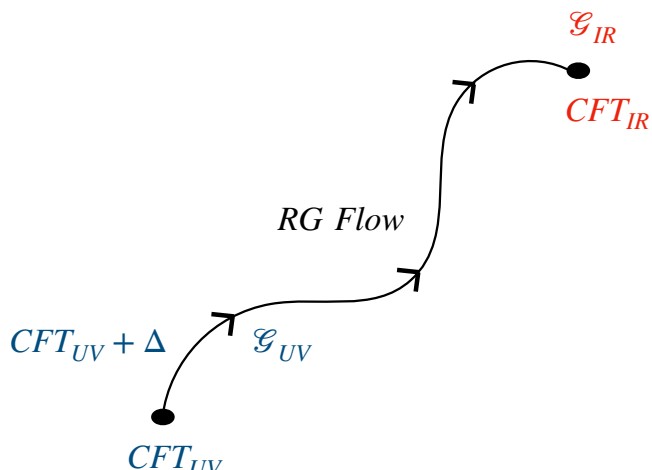

Figure 6: A depiction of an RG flow. We start with some CFT in the UV. This $CFT_{UV}$ can be a free theory or some non trivial strongly coupled theory. We deform it by a relevant deformation $\Delta$: It can be *e.g.* a relevant interaction, a vacuum expectation value, or turning on a non trivial background. We denote the symmetry of the theory after the deformation as $\mathscr{G}_{UV}$. In the IR the model flows to a new fixed point $CFT_{IR}$. The global symmetry of the fixed point $\mathscr{G}_{IR}$ might or might not be the same as $\mathscr{G}_{UV}$.

## 2 Lecture I: The basic toolbox

We wish to understand the following general set of question. Starting from a CFT in the UV ($CFT_{UV}$) and deforming it by introducing a scale, we trigger an RG flow. The properties of the deformed QFT depend on the energy scale at which we probe the physics. We are then interested to understand what happens when we go to extremely low energies, far below the energy scale set by the deformation, see Figure 6 for an illustration. This question is often hard to answer, since as we lower the energy scale at which we probe the physics the description in terms of the degrees of freedom of $CFT_{UV}$ can become strongly coupled. For example, as we will review soon, if we start from a free theory as $CFT_{UV}$ and turn on a relevant deformation, a variety of interesting behaviors can emerge in the IR: The theory can flow to an interacting strongly coupled conformal theory, it might flow to a weakly coupled gauge theory, and the flow can also be gapped with no propagating degrees of freedom remaining in the IR.

### 2.1 Symmetry, anomalies, and 't Hooft anomaly matching

Without simplifying assumptions, such as supersymmetry that we will soon introduce, answering the posed questions is rather hard. Nevertheless, we do have certain tools which we can use quite generally. Not surprisingly they have something to do with symmetry. Let us assume that after deforming $CFT_{UV}$ the global symmetry of the resulting QFT is $\mathscr{G}_{UV}$. Here we will focus on zero-form continuous symmetries, though one can consider generalizations of the discussion to higher form symmetries [28] and higher group structures [29]. Typically the deformation breaks explicitly some of the symmetry of $CFT_{UV}$ and some of the symmetry might also be spontaneously broken: Let us assume $\mathscr{G}_{UV}$ is the surviving fraction of the symmetry. This symmetry will be preserved during RG flow. Thus, we expect that the symmetry of the theory in the IR should be $\mathscr{G}_{UV}$. There are however two caveats. First, $\mathscr{G}_{UV}$ might not act faithfully in the IR. In particular in an extreme case it might not act at all, for example when the theory develops a gap in the IR. In this case, $\mathscr{G}_{UV}$ will not act manifestly in the IR.

Second, the symmetry in the IR can actually be bigger than $\mathcal{G}_{UV}$. Intuitively, some of the interactions/degrees of freedom which break a certain larger symmetry group, of which $\mathcal{G}_{UV}$ is a subgroup, might be washed away/gapped in the IR. In such situations we will say that there is an enhancement (or emergence) of the symmetry in the IR and the symmetry group $\mathcal{G}_{IR}$ might be bigger than $\mathcal{G}_{UV}$. We will denote by $\mathcal{G}_{IR}$ the symmetry group of the IR fixed point if the theory flows to an SCFT, or the symmetry of the weakly coupled gauge theory if it flows to such a theory.

Importantly, we can say something more about the symmetry. If a theory possesses a global symmetry with a corresponding conserved current[4] $J_\mu$, we can turn on background gauge fields $A_\mu$, valued in the Lie algebra of some sub-group of the symmetry group, coupled to this conserved current, and compute the effective action $\Gamma[A]$. As the current is conserved the gauge field comes with a gauge symmetry,

$$A \to A^g \equiv g \cdot A \cdot g^{-1} + (dg) \cdot g^{-1}, \tag{5}$$

where $g$ is an element of the symmetry (sub)group. Then we can try to promote $A_\mu$ to be dynamical fields. However, there might be an obstruction to doing so, which goes under the name of 't Hooft anomaly. The obstruction comes about as the effective action of the theory might or might not be invariant under (5),

$$\Gamma[A^g] \overset{?}{=} \Gamma[A]. \tag{6}$$

If the equality does not hold we say that the (sub)group of the symmetry has a 't Hooft anomaly. In particular this means that the symmetry cannot be gauged, *i.e.* the gauge fields $A_\mu$ cannot be promoted to dynamical fields. See [30] for a comprehensive introduction to the subject of anomalies and [31] for a more recent and general discussion. The important fact about 't Hooft anomalies is that they are quantifiable. There are various ways to understand this and let us here mention two of them. First, the anomaly of a continuous symmetry can be captured in $D = 2n$ dimensions by an $n+1$ point one loop amplitude involving the conserved currents. In particular, say in $D = 4$, this is proportional to,

$$a_{G_1,G_2,G_3} \equiv \text{Tr}_\mathfrak{R}\, G_1 G_2 G_3. \tag{7}$$

Here $\mathfrak{R}$ is the representation of the chiral fermions of the model (if we have a description of the theory in terms of a Lagrangian) under the product of the three groups $G_1 \times G_2 \times G_3$, which are the groups corresponding to the three currents. The trace is taken over this representation. For example, if $G_i$ all correspond to the same $U(1)$ symmetry then the 't Hooft anomaly is just given by $\sum_{l=1}^{N} q_l^3$, where $q_l$ are the charges of the $N$ Weyl fermions of the model.

Another way to think about the anomaly is as follows. The failure of the effective action to be invariant cannot take any form and is constrained by Wess-Zumino consistency conditions to be related to what is known as the anomaly polynomial $\mathcal{I}_{D+2}$ (where $D$ is the number of spacetime dimensions which is assumed to be even). $\mathcal{I}_{D+2}$ is a homogeneous polynomial of degree $D+2$ in characteristic classes built from gauge fields (including gravitational) corresponding to the symmetries of the system. The coefficients of the polynomial encode the 't Hooft anomalies of the theory. For example a term of the form,

$$n_{123} \frac{a_{G_1,G_2,G_3}}{6} \text{Tr}\, F_1 \wedge F_2 \wedge F_3 \subset \mathcal{I}_6, \tag{8}$$

captures the anomaly $a_{G_1,G_2,G_3}$ defined in (7). Here $n_{123}$ is a simple numerical factor depending on how many of the three $F_i$ correspond to different groups. If all are the same then $n_{123} = 1$,

---

[4]One can make this discussion more general and abstract by thinking about the conserved charge corresponding to a zero-form symmetry as certain co-dimension one topological operators in the QFT (and higher co-dimension when the symmetry is of a higher form) [28].

if two are the same $n_{123} = 3$ and if all are different it is equal to 6. See Appendix E for more details on the 6d anomaly polynomial.

't Hooft anomalies are useful for us since they don't change during the RG-flow and thus are the same in the UV and the IR: This fact goes under the name of 't Hooft anomaly matching condition. There are various ways to see this. First, we can add to the theory a number of Weyl fermions (called spectators), such that the combined anomaly of the theory of interest and the additional fermions is zero. Then we can weakly gauge the symmetry: That is turn on infinitesimally small gauge couplings. Assuming we can always stay in a regime where the gauge coupling is small, we can flow to the IR where we decouple the spectators. As there is no obstruction to the gauging in the IR, and we decouple the same fermions, the anomaly should not change in the process. Yet another argument for the non-renormalization is through what is called anomaly inflow. One way to render the symmetry non-anomalous is to realize the theory on the boundary of a $D + 1$ dimensional space with a background field $A$ living in $D + 1$ dimensions. Then, we can add a Chern-Simons term in the bulk that cancels the anomaly of the boundary theory. This will again remove the obstruction from gauging the symmetry which should hold in the UV as well as in the IR.

Thus, we learn that the anomaly polynomial $\mathcal{I}_6$ computed in the UV should be the same as the one computed in the IR, no matter what is the IR behavior of our deformation. This is true for the symmetries we can identify in the UV. At the IR fixed point new symmetries can emerge and as we cannot identify them in the UV their anomalies do not have to be zero. One way to understand this is that we can move out of the IR fixed point by irrelevant deformations which explicitly break the emergent symmetries and thus invalidate the 't Hooft anomaly matching argument. On the other hand, if a sub-group of the symmetry group does not act faithfully in the IR its anomaly has to be zero also in the UV due to the 't Hooft anomaly matching argument.[5]

## 2.2  $a$-maximization and superconformal R-symmetry

Until now our discussion did not involve supersymmetry at all. Indeed, without supersymmetry we do not have at the moment useful robust tools beyond matching symmetries and anomalies to understand the physics in the IR. However with supersymmetry there are more things that we can say. Let us start the discussion of supersymmetric theories in $D = 4$ with what the interplay between the supersymmetry and 't Hooft anomalies can give us.

We will assume throughout these Lectures that we are dealing with $\mathcal{N} = 1$ supersymmetric theories in 4d . Moreover we will assume that these theories possess an R-symmetry.[6] A $U(1)$ R-symmetry is a necessary ingredient of the $\mathcal{N} = 1$ superconformal group, as for example it appears on the right-hand side of anti-commutation relations between supercharges $Q_\alpha$ and their superconformal counterparts $S_\alpha$. See Appendix A for a summary of the $\mathcal{N} = 1$ superconformal group in 4d. The SCFT in the UV thus has an R-symmetry which is part of the superconformal group. We will only discuss deformations which preserve $\mathcal{N} = 1$ supersymmetry. We will also assume that some combination of this R-symmetry and an abelian subgroup of the global symmetry group of $CFT_{UV}$ is not broken by the deformation, though of course as we introduce a scale the conformal symmetry is broken. In the IR, if we arrive to a conformal fixed point, we again acquire the superconformal R-symmetry. However, the superconformal symmetry in the UV and in the IR might not be the same symmetry. Nevertheless, under certain assumptions that we will detail, the fact that the R-symmetry is intimately related to the superconformal group allows us to determine it in the IR.

---

[5]Note that this is not true for the case of spontaneous symmetry breaking. A symmetry with a non-trivial 't Hooft anomaly in the UV can be spontaneously broken in the IR.

[6]The R-symmetry can be broken by superpotentials or be anomalous in the UV. We thus will not deal with such situations.

Any conformal theory, supersymmetric or not, in four dimensions has two important numbers associated to it: These are referred to as the $a$ and the $c$ conformal anomalies. The conformal anomalies measure, among other things, the failure of the expectation value of the trace of the stress-energy tensor to vanish when the theory is placed on a curved background with metric $g_{\mu\nu}$,

$$\langle T^{\mu}{}_{\mu}\rangle = \frac{c}{16\pi^2}W^2 - \frac{a}{16\pi^2}E_4\,, \tag{9}$$

where $W$ is the Weyl tensor and $E_4$ is the Euler density, both built from certain combinations of the metric $g_{\mu\nu}$ and its derivatives [32].[7] In superconformal theories, as the stress energy tensor and the R-symmetry are part of the same symmetry algebra the various anomalies are interrelated. It is possible for example to utilize these relations to arrive at the following extremely useful statements [35],

$$a = \frac{9}{32}\operatorname{Tr}R^3 - \frac{3}{32}\operatorname{Tr}R\,, \qquad c = \frac{9}{32}\operatorname{Tr}R^3 - \frac{5}{32}\operatorname{Tr}R\,. \tag{10}$$

Here it is important that $R$ is the R-symmetry in the superconformal group. $\operatorname{Tr}R^3$ is the 't Hooft anomaly corresponding to $\operatorname{Tr}F^3$ term in the anomaly polynomial we have discussed above, with $F$ being the field strength for the R-symmetry background gauge field. Whereas $\operatorname{Tr}R$ is the mixed R-symmetry gravity anomaly, which in the anomaly polynomial appears as a coefficient of a term involving $\operatorname{Tr}F$ and a certain Pontryagin class computed using the background metric.

Exploiting the interplay between supersymmetry and the R-symmetry in a conformal theory one can also arrive at the conclusion that the mixed 't Hooft anomalies between any global $U(1)$ symmetry (which is not an R-symmetry) and the superconformal R-symmetry are also related to the mixed gravitational-$U(1)$ anomaly [36],

$$\operatorname{Tr}U(1) = 9\operatorname{Tr}U(1)R^2\,. \tag{11}$$

Moreover, the positivity of the two point function of the currents of the global $U(1)$ symmetry can be related to the negativity of yet another 't Hooft anomaly,

$$\operatorname{Tr}U(1)^2R < 0\,. \tag{12}$$

Combining all these observations Intriligator and Wecht arrived at a very simple procedure to determine the R-symmetry of the IR fixed point by knowing all the abelian symmetries and the R-symmetry preserved along the RG-flow [36]. One defines the following trial $a$ anomaly,

$$a(\{\alpha\}) = \frac{9}{32}\operatorname{Tr}(R + \alpha_i U(1)^{(i)})^3 - \frac{3}{32}\operatorname{Tr}(R + \alpha_i U(1)^{(i)})\,. \tag{13}$$

Here $\alpha_i$ are arbitrary real numbers associated to the $i$-th $U(1)$ symmetry: The summation over $i$ is implied in the equation. Now we notice that,

$$\frac{32}{3}\frac{\partial}{\partial\alpha_j}a(\{\alpha\}) = 9\operatorname{Tr}(R + \alpha_i U(1)^{(i)})^2 U(1)^{(j)} - \operatorname{Tr}U(1)^{(j)} = 0\,. \tag{14}$$

The last equality holds if $(R + \alpha_i U(1)^{(i)})$ is the superconformal R-symmetry following (11). On the other hand,

$$\frac{16}{27}\beta_j\frac{\partial}{\partial\alpha_j}\beta_k\frac{\partial}{\partial\alpha_k}a(\{\alpha\}) = \operatorname{Tr}(R + \alpha_i U(1)^{(i)})\beta_j U(1)^{(j)}\beta_k U(1)^{(k)} < 0\,, \tag{15}$$

---

[7]In any conformal theory the $a$ conformal anomaly is smaller at the IR fixed point relative to the UV fixed point [33, 34].

where $\beta_i$ are arbitrary real numbers and the last inequality follows from (12) assuming again $R + \alpha_i U(1)^{(i)}$) is the superconformal R-symmetry. This thus implies that:

*The superconformal R-symmetry maximizes the trial $a(\{\alpha\})$.*

The procedure of obtaining the superconformal R-symmetry discussed here is called *a*-maximization. This statement however has an important caveat. We assume that we have identified all the $U(1)$ symmetries that can possibly mix with the R-symmetry to produce the superconformal one. However, as we already discussed some of these symmetries can only emerge in the IR. This possibility should always be kept in mind, as it is usually hard to rule it out. In some cases, certain indications that this has to be the case can be derived. For example, using certain unitarity bounds, superconformal representation theory implies that the R-charge of any chiral operator in the theory has to be bigger or equal to $\frac{2}{3}$. The R-charge saturates this bound only if the operator is a free chiral field. If using $a$-maximization leads to certain chiral operators violating these bounds, then some of the assumptions going into the computation have to be wrong. A natural way for this to happen is if some of the abelian symmetries in the IR have been missed [37].

Using the relation (10) we can compute the conformal anomalies of free fields, which will be useful for us in what follows. A free chiral superfield $Q$ has an R-charge of $\frac{2}{3}$. This is the R-charge of the scalar component of the superfield. The R-charge of the Weyl fermion is shifted[8] by $-1$ and thus the conformal anomalies of the free chiral field are,

$$a = \frac{9}{32}\left(\frac{2}{3}-1\right)^3 - \frac{3}{32}\left(\frac{2}{3}-1\right) = \frac{1}{48}, \qquad c = \frac{9}{32}\left(\frac{2}{3}-1\right)^3 - \frac{5}{32}\left(\frac{2}{3}-1\right) = \frac{1}{24}. \quad (16)$$

For the free vector superfield the R-symmetry is zero, and thus the R-symmetry of the gaugino is $+1$. This implies the following anomalies,

$$a = \frac{9}{32} - \frac{3}{32} = \frac{3}{16} \equiv a_v, \qquad c = \frac{9}{32} - \frac{5}{32} = \frac{1}{8} \equiv c_v. \quad (17)$$

For future convenience we also define the contributions to the conformal anomalies of chiral fields of general R-charge $r$ to be,

$$a_R(r) = \frac{9}{32}(r-1)^3 - \frac{3}{32}(r-1), \qquad c_R(r) = \frac{9}{32}(r-1)^3 - \frac{5}{32}(r-1). \quad (18)$$

Exercise: Show that if the free R-charge assignment is consistent with all the superpotentials and is anomaly free in a general gauge theory, it solves the $a$-maximization problem.

We assume that a gauge group $\mathcal{G}$ is given, the number of chiral fields is $\dim \mathfrak{R}$, and that the assignment of R-charge $\frac{2}{3}$ to all chiral superfields is consistent with superpotentials and anomalies. We also assume that the theory has $L$ $U(1)_\ell$ symmetries under which the $i$-th chiral superfield has charges $Q_i^\ell$. Then we define a trial superconformal R-symmetry to be,

$$R_i^{tr}(\{\alpha\}) = \frac{2}{3} + Q_i^\ell \alpha_\ell, \quad (19)$$

---

[8]In superspace notations this is due to the fact that the superspace coordinates have a unit charge under the R-symmetry.

with $\alpha_\ell$ arbitrary real parameters. Computing the trial $a$-anomaly,

$$\widetilde{a}(\{\alpha\}) \equiv \frac{32}{3} a(\{\alpha\}) - a_v \dim \mathcal{G} = \sum_{i=1}^{\dim \mathfrak{R}} \left( 3(\frac{2}{3} + \alpha_\ell Q_i^\ell - 1)^3 - (\frac{2}{3} + \alpha_\ell Q_i^\ell - 1) \right) \tag{20}$$

$$= \frac{2}{9} \dim \mathfrak{R} + 3 \sum_{i=1}^{\dim \mathfrak{R}} \left( \alpha_\ell \, \alpha_{\ell'} \, \alpha_{\ell''} Q_i^\ell Q_i^{\ell'} Q_i^{\ell''} - \alpha_\ell \, \alpha_{\ell'} Q_i^\ell Q_i^{\ell'} \right),$$

we immediately see that the term linear in $\alpha_\ell$ cancels out. This implies that taking $\alpha_\ell = 0$ is a stationary point and it is then trivial to show that it is a local maximum. One way to see this is to take some arbitrary direction in $\{\alpha\}$ space parametrized as $\alpha_\ell = n_\ell t$ and see how $\widetilde{a}(\{\alpha\})$ changes with $t$,

$$\widetilde{a}(\{\alpha\}) = \frac{2}{9} \dim \mathfrak{R} + 3 \left( \sum_{i=1}^{\dim \mathfrak{R}} \Delta R_i^3 \right) t^3 - 3 \left( \sum_{i=1}^{\dim \mathfrak{R}} \Delta R_i^2 \right) t^2, \tag{21}$$

where we defined $\Delta R_i = Q_i^\ell n_\ell$. Obviously then $t = 0$ is a maximum for any choice of direction $n_\ell$ unless $\forall i \; \Delta R_i = 0$. However the latter case implies that $R_i^{tr}(\{n\,t\}) = \frac{2}{3}$, and thus the R-symmetry is still the free one.

$\square$

## 2.3 Beta functions, deformations, and conformal manifolds

An important question regarding CFTs is what is the collection of relevant and exactly marginal deformations of a model leading to an inequivalent fixed point. In particular for SCFTs we will be interested in such deformations which preserves $\mathcal{N} = 1$ supersymmetry.

One type of supersymmetric deformations one can add to an SCFT is the following superpotential term,

$$W = \lambda_\mathcal{O} \int d^4x d^2\theta \, \mathcal{O} + c.c., \tag{22}$$

where $\mathcal{O}$ is a chiral operator and $\lambda_\mathcal{O}$ is a complex number. Relevant supersymmetric deformations are given by chiral operators $\mathcal{O}$ with scaling dimension smaller than 3. For such deformations $\lambda_\mathcal{O}$ has a positive mass dimension and thus grows with the RG-flow to the IR. Remember that $d\theta$ has mass dimension $+\frac{1}{2}$. Note also, that for chiral operators the superconformal algebra relates the dimension to the superconformal R-charge, $\Delta = \frac{3}{2} R$; thus, the R-charge of relevant deformations $\mathcal{O}$ is smaller than 2.

The marginal superpotential deformations are given by chiral operators $\mathcal{O}$ of dimension exactly 3, or equivalently superconformal R-charge of exactly 2, and as we will soon discuss these can be either marginally irrelevant or exactly marginal.

Given an SCFT with a subgroup $G$ of the global symmetry group with vanishing 't Hooft anomaly we can couple it to dynamical gauge fields in the standard way. In particular we introduce the superpotential term,

$$W = \tau_G \int d^4x d^2\theta \, W_\alpha W^\alpha + c.c., \tag{23}$$

with $\tau$ being the complexified gauge coupling and $W_\alpha$ the chiral superfield with the field strength $F_{\mu\nu}$ as one of its components. As with other superpotential couplings $\tau$ can be either a relevant deformation, irrelevant, or marginal. To determine which of the three possibilities holds one needs to perform a one loop computation of the gauge beta-function: Classically

gauge-couplings are marginal in 4d. In supersymmetric theories the result of this computation can be expressed in an elegant form [38],[9]

$$\beta_G \propto -\operatorname{Tr} R_{\mathcal{P}} G^2 \,, \tag{24}$$

where $\mathcal{P}$ refers to the fixed point under consideration. That is the beta function is proportional to minus the 't Hooft anomaly of the superconformal R-symmetry, denoted here by $R_{\mathcal{P}}$, and two currents of the gauged symmetry. The proportionality coefficient is a positive number which will not play a role for us (and can be easily fixed by computing it for simple Lagrangians). The reason this expression can be derived is because the superconformal R-symmetry is in the same multiplet as the stress-energy tensor. In particular, if $\operatorname{Tr} R_{\mathcal{P}} G^2$ is positive the gauging is UV free, if $\operatorname{Tr} R_{\mathcal{P}} G^2$ is negative the gauging is IR free, and if $\operatorname{Tr} R_{\mathcal{P}} G^2 = 0$ the gauging is marginal. Note that this way of determining relevance of a gauging does not rely on a description of the theory in terms of weakly coupled fields and thus will turn out to be useful also for strongly coupled SCFTs. Note also that the marginality of gauging is determined by whether the superconformal R-symmetry at the fixed point is anomalous or not: That is whether it is consistent with the gauging. This is very analogous to marginality of superpotentials: Marginal superpotentials do not break R-symmetry.

Finally, we need to determine whether the marginal deformations we discussed are exactly marginal, marginally irrelevant, or marginally relevant. In fact for supersymmetric theories with supersymmetry preserving deformations the last option is not possible. The reason is as follows [40]. When deforming a theory by a supersymmetric marginal operator, in order for the deformation to cease being marginal its dimension needs to change after the RG-flow. However, such deformations are chiral and as such form shortened, protected, representations of the superconformal symmetry group. The only way for such a deformation to cease being marginal is for it to recombine during the flow with another shortened multiplet to form a long multiplet. Studying the representation theory of the superconformal group it is possible to show that the only multiplet with which the marginal deformations can recombine are conserved currents [41], see Appendix A. The primaries of the resulting long multiplet have dimensions which must be larger than three. As such, after such a recombination the dimension of the marginal deformation, which was 3 since it was the chiral primary of the short multiplet, must increase and thus it is marginally irrelevant. This implies in particular that there are no supersymmetric marginally relevant deformations. Another simple implication of this logic is that if a deformation does not break any symmetry it is exactly marginal. See Figure 7 for a depiction of exactly marginal, marginally irrelevant, and relevant deformations. Also, see Figure 8 for interesting cases of RG-flows.

A careful analysis of the above logic [40] (see also [42]) leads to the following conclusion. Given an SCFT $\mathcal{P}$ with a space of marginal operators $\mathcal{O}_{\mathcal{P}}^i$ and couplings $\lambda_{\mathcal{P}}^i$, the set of exactly marginal deformations is given by the following Kähler quotient,

$$\{\lambda_{\mathcal{P}}^i\}/(G_{\mathcal{P}})_{\mathbb{C}} \,, \tag{25}$$

where $G_{\mathcal{P}}$ is the global symmetry of the SCFT $\mathcal{P}$ and $(G_{\mathcal{P}})_{\mathbb{C}}$ is its complexification. When the marginal deformations involve also gauge couplings of some gauge group, we need to include in $G_{\mathcal{P}}$ also the symmetries which exist before the gauging but are anomalous once the gauging is introduced. The coupling which transforms well under the anomalous symmetry is $\exp(2\pi i \tau)$ with $\tau$ being the complexified gauge coupling. Note that in the exponentiation we took $\tau$ with the positive sign.[10] This procedure is very abstract and has again the advantage that it does not rely on having a description of a theory in terms of weakly coupled fields. One

---

[9]See also [39] for an important discussion.

[10]Usually in computations of Kähler quotients both signs are allowed. However, here taking the negative sign might lead to solutions with imaginary YM coupling. Formally in perturbation theory this would give a line of

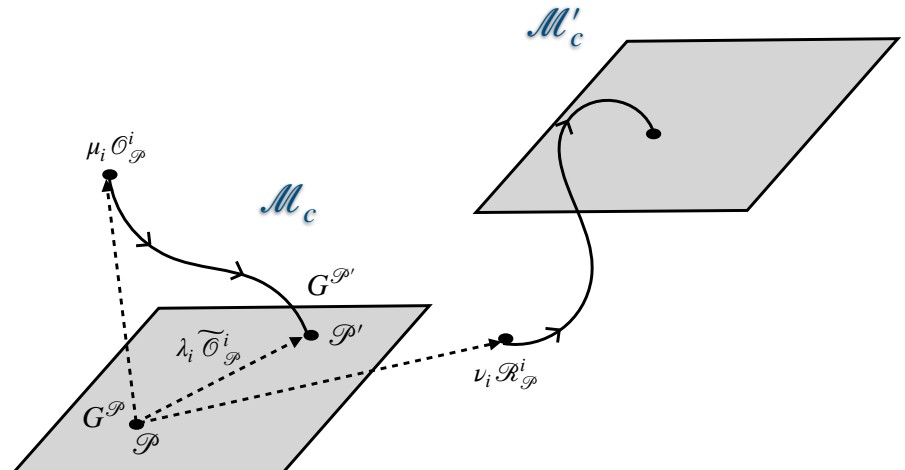

Figure 7: A depiction of the conformal manifold $\mathcal{M}_c$ and three deformations from the point $\mathscr{P}$. The exactly marginal deformation is related to $\lambda_i \widetilde{\mathcal{O}}_{\mathscr{P}}^i$, the marginally irrelevant deformation is related to $\mu_i \mathcal{O}_{\mathscr{P}}^i$, and the relevant deformation is related to $\nu_i \mathscr{R}_{\mathscr{P}}^i$.

way to compute this quotient is to list all the $(G_{\mathscr{P}})_{\mathbb{C}}$ invariant independent combinations of the couplings: Though often this is practically hard to do. We refer the reader to [48] for a detailed analysis of such quotients in many examples as well as a discussion of how to approach the problem in a general case.

> **Example 1: $SU(N)$ SQCD with $3N$ flavors**

First, a free R-symmetry assignment to all the matter fields is non anomalous,

$$\text{Tr } R_{\mathscr{P}} SU(N)^2 = N + \frac{1}{2}\left(\frac{2}{3} - 1\right) 3N + \frac{1}{2}\left(\frac{2}{3} - 1\right) 3N = 0. \tag{26}$$

Thus, according to the previous exercise it is also the superconformal one. Moreover, since this is also the superconformal R-symmetry of the free collection of chiral fields before gauging, the gauging is marginal. Since the superconformal R-charges at the free UV fixed point is $\frac{2}{3}$ we can turn on marginal superpotentials using chiral operators with R-charge 2 which are built from cubic combinations of the basic fields. However, for $N \neq 3$ no such operators exist. For $N = 3$ the operators $B = \epsilon_{ijk} Q_{(a}^i Q_b^j Q_{c)}^k$ and $\widetilde{B} = \epsilon^{ijk} \widetilde{Q}_i^{(a} \widetilde{Q}_j^b Q_k^{c)}$ are singlets under the gauged $SU(3)$ symmetry: These are the baryons and the anti-baryons. The analysis thus is different for $N \neq 3$ and $N = 3$. We start with the former.

The symmetry of the free fixed point is $G_{\mathscr{P}} = U(6N^2)$. We choose $G = SU(N)$ subgroup which we gauge. The commutant of $G$ in $G_{\mathscr{P}}$ is $G_{\mathscr{P}|\mathcal{G}} = U(3N) \times U(3N)$. The Kähler quotient we need to compute is thus,

$$\{\exp(2\pi i \tau_{SU(N)})\} / (U(3N) \times U(3N))_{\mathbb{C}}. \tag{27}$$

---

exactly marginal non-unitary SCFTs (See for similar effects *e.g.* [43]). However it is not clear whether this would make sense non perturbatively. In some cases, *e.g* AGT correspondence [44, 45] or 5d gauge theories [46, 47] (where gauge coupling is related to real mass for instanton symmetry), going to lower half plane for $\tau$ is allowed but it is interpreted as going to infinite coupling (and "beyond") where the gauge theory description breaks down. In our analysis we are dealing with weakly coupled gauge fields and thus will restrict to upper half-plane for $\tau$. We thank Z. Komargodski for discussions of this subtlety.

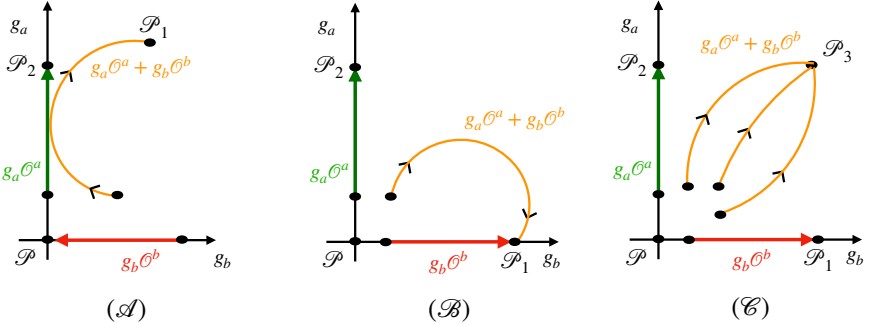

Figure 8: Several interesting RG flow patterns. ($\mathscr{A}$) Example of a dangerously irrelevant deformation. Turning on only $g_b$ we flow back to the original CFT. Turning on both $g_a$ and $g_b$, during the flow $g_a$ increases while $g_b$ decreases. However, once $g_a$ is large enough we cannot trust low orders in perturbation theory, and $g_b$ might start increasing again and we might flow to a new fixed point. We call $g_b$ in such a case a dangerously irrelevant deformation. ($\mathscr{B}$) An example of "dangerously relevant operator". Turning on either $g_a$ or $g_b$ we flow to a new CFT and thus these are relevant. However, if we first turn on $g_b$ and flow to the fixed point, $g_a$ then becomes an irrelevant deformation at the new fixed point (This is sometimes related to the so called chiral ring stability [49,50].) ($\mathscr{C}$) Turning on either $g_a$ or $g_b$ we flow to a new CFT and thus these are relevant. Also if we turn on the deformations sequentially they are still relevant at each step.

The gauge coupling is not charged under the $SU(3N) \times SU(3N) \times U(1)_B$ non-anomalous symmetry, but transforms under the $U(1)_A$ anomalous symmetry, with $\exp(2\pi i \tau_{SU(N)})$ having charge $3N$. Thus this quotient is empty and there are no exactly marginal deformations. The gauge coupling is marginally irrelevant and the theory is free.

Next we analyze the case of $N = 3$ where we have additional marginal operators $B$ and $\widetilde{B}$. These transform in $(\mathbf{84}, \mathbf{1})_{+3,+3}$ and $(\mathbf{1}, \mathbf{84})_{+3,-3}$ under $G_{\mathcal{P}|\mathcal{G}}$ which we parametrize as $(SU(9), SU(9))_{U(1)_A, U(1)_B}$, where $U(1)_A$ is the symmetry under which both quarks and antiquarks have charge $+1$ while $U(1)_B$ is the baryonic symmetry under which they are oppositely charged (that is with charges $\pm 1$). The representation $\mathbf{84}$ is the three index completely antisymmetric irrep of $SU(9)$. The gauge coupling transforms as $(\mathbf{1}, \mathbf{1})_{9,0}$. The charge of the gauge coupling $\tau$, or rather of $\exp(2\pi i \tau)$, is given by the 't Hooft anomaly Tr $U(1)_A G^2$. In this case it is $+1 \times \frac{1}{2} \times 18 = 9$. Note that the baryonic couplings have charge $-3$ as all the matter fields have charge $+1$. In particular this implies that we always can solve for the quotient with the anomalous $U(1)_A$. Thus the quotient we need to compute is,

$$\{(\mathbf{84}, \mathbf{1})_{+3}, (\mathbf{1}, \mathbf{84})_{-3}\}/(SU(9) \times SU(9) \times U(1)_B)_{\mathbb{C}}. \tag{28}$$

One way to construct the quotient is by finding all the independent monomial holomorphic combinations of the couplings invariant under the symmetries. In general this is a well defined group theoretic question which is nevertheless tricky to solve.

A way to proceed is to find a deformation which is exactly marginal. Understand what symmetry it breaks. Deform the theory by this deformation and repeat the process. In the given case a simple deformation which was found by Leigh and Strassler [51] (See Appendix B.) in the seminal paper on conformal manifolds is,

$$W = \lambda \left( Q_1 Q_2 Q_3 + Q_4 Q_5 Q_6 + Q_7 Q_8 Q_9 + \widetilde{Q}_1 \widetilde{Q}_2 \widetilde{Q}_3 + \widetilde{Q}_4 \widetilde{Q}_5 \widetilde{Q}_6 + \widetilde{Q}_7 \widetilde{Q}_8 \widetilde{Q}_9 \right). \tag{29}$$

This deformation breaks $U(1)_B$ and each $SU(9)$ is broken to $SU(3)^3$. Note that under this breaking,

$$\mathbf{84} \to \mathbf{1} + \mathbf{1} + \mathbf{1} + \mathbf{3}_1\mathbf{3}_2\mathbf{3}_3 + \sum_{i,j=1;\, i\neq j}^{3} \mathbf{3}_i\overline{\mathbf{3}}_j \,, \tag{30}$$

$$\mathbf{80} \to \mathbf{1} + \mathbf{1} + \mathbf{8}_1 + \mathbf{8}_2 + \mathbf{8}_3 + \sum_{i,j=1;\, i\neq j}^{3} \mathbf{3}_i\overline{\mathbf{3}}_j \,.$$

Note that the components of the currents (the $\mathbf{80}$) in representation $\sum_{i,j=1;\, i\neq j}^{3} \mathbf{3}_i\overline{\mathbf{3}}_j$ recombine with the same components of the marginal operators (the $\mathbf{84}$). Moreover the currents of the four $U(1)$s in the decomposition of $SU(9)^2$ to $SU(3)^6 \times U(1)^4$ as well as the current of $U(1)_B$ recombine with five out of the six singlets in the decomposition of the two $\mathbf{84}$s. We are thus left with marginal deformations in,

$$\mathbf{1} + \mathbf{3}_1\mathbf{3}_2\mathbf{3}_3 + \mathbf{3}_4\mathbf{3}_5\mathbf{3}_6 \,, \tag{31}$$

with the singlet being the exactly marginal deformation. We thus deduce that there is one direction on the conformal manifold on which the symmetry is $SU(3)^6$. Next we can turn on the marginal deformation $\mathbf{3}_1\mathbf{3}_2\mathbf{3}_3$. This will break $SU(3)^3$ down to diagonal $SU(3)$. Note that under this decomposition,

$$\mathbf{3}_1\mathbf{3}_2\mathbf{3}_3 \to \mathbf{1} + \mathbf{10} + \mathbf{8} + \mathbf{8} \,. \tag{32}$$

In particular the two $\mathbf{8}$ recombine with two of the three $SU(3)$ currents leaving only $SU(3)$ symmetry and marginal operators in,

$$\mathbf{1} + \mathbf{1} + \mathbf{10}_1 + \mathbf{3}_4\mathbf{3}_5\mathbf{3}_6 \,. \tag{33}$$

Thus we have a two dimensional conformal manifold on which the symmetry is $SU(3)^4$. Next we can break in the same manner using $\mathbf{3}_4\mathbf{3}_5\mathbf{3}_6$ another triplet of $SU(3)$ symmetries down to diagonal $SU(3)$. We will obtain a three dimensional conformal manifold with symmetry $SU(3)^2$ and marginal operators in,

$$\mathbf{1} + \mathbf{1} + \mathbf{1} + \mathbf{10}_1 + \mathbf{10}_2 \,. \tag{34}$$

*In Lecture III we will have a geometric interpretation of this conformal manifold as corresponding to complex structure moduli of a sphere with six marked points.* We can next continue exploring the conformal manifold by turning on operators in the $\mathbf{10}$. As from $\mathbf{10}$ one can build two independent invariant (the fourth and the sixth symmetric powers contain singlets), each gives two exactly marginal directions on which the relevant $SU(3)$ is completely broken. All in all we obtain a seven dimensional conformal manifold on which all the symmetry is broken.

Instead of starting with (29) we can start with the following,

$$W = \lambda \left( Q^1_{[i} Q^2_j Q^3_{k]} + \widetilde{Q}^1_{[i} \widetilde{Q}^2_j \widetilde{Q}^3_{k]} \right) . \tag{35}$$

Here we think of the nine quarks as forming $\mathbf{3} \times \mathbf{3}$ representation of the $SU(3) \times SU(3)$ subgroup of $SU(9)$. The lower flavor indices in (35) are antisymmetrized as well as the gauge indices are. This superpotential is exactly marginal. This can be understood either performing the LS analysis (all the fields have same anomalous dimensions but we have two couplings: Gauge coupling and $\lambda$), or performing the Kähler quotient. The two terms are charged oppositely under the $U(1)_B$ symmetry and we note that,

$$\mathbf{84} \to \mathbf{10}_1 + \mathbf{10}_2 + \mathbf{8}_1 \times \mathbf{8}_2 \,, \tag{36}$$

$$\mathbf{80} \to \mathbf{8}_1 + \mathbf{8}_2 + \mathbf{8}_1 \times \mathbf{8}_2 \,.$$

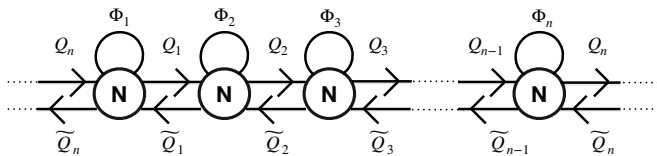

Figure 9: The necklace quiver of length $n$.

Moreover the characters of **10** and **8** are given by

$$\chi_{\mathbf{10}} = \chi_{\mathbf{8}} - 1 + z_1^3 + z_2^3 + \frac{1}{z_1^3 z_2^3}, \qquad \chi_{\mathbf{8}} = 1 + 1 + \sum_{i \neq j} z_i / z_j. \tag{37}$$

Using these observations we can conclude that (35) breaks the symmetry to $SU(3) \times SU(3) \times U(1)^4$ where the abelian symmetries are the Cartan generators of the $SU(3)$s broken by (35),[11]

$$1 + \mathbf{84} + \widetilde{\mathbf{84}} - 1 - 1 - \mathbf{80} - \widetilde{\mathbf{80}} \rightarrow 1 + \mathbf{10}_1 + \chi_{\mathbf{3}}((z_1^2)^3, (z_2^2)^3) + \widetilde{\mathbf{10}}_1 \tag{38}$$
$$+ \chi_{\mathbf{3}}((\widetilde{z}_1^2)^3, (\widetilde{z}_2^2)^3) - \mathbf{8}_1 - \widetilde{\mathbf{8}}_1 - 1 - 1 - 1 - 1.$$

Here $z_i^2$ are fugacities for the Cartan of $SU(3)_2$ and $\widetilde{z}_i^2$ are fugacities for the Cartan of $\widetilde{SU(3)}_2$. We can further break the rest of the symmetries. For example, the two $z_i^2$ can be broken together by turning on the three operators in $\chi_{\mathbf{3}}((z_1^2)^3, (z_2^2)^3)$ and the $SU(3)_1$ can be broken first to the Cartan and then completely by turning on operators in $\mathbf{10}_1$.  □

> Example 2: $\mathcal{N} = 2$ necklace quiver: Consider the quiver gauge theory of Figure 9. It is well known that this theory has $n$ exactly marginal deformations preserving $\mathcal{N} = 2$ supersymmetry (show it). Show that it also possesses an additional independent exactly marginal deformation preserving only $\mathcal{N} = 1$ supersymmetry.

The matter content is conformal, *i.e.* the one loop beta function vanishes, and thus all the fields have free R-charges. Let us start by discussing the symmetries of the theory. We have $n$ anomalous symmetries $U(1)_{A_i}$ under which the adjoint fields $\Phi_i$ have charge $+1$ and all the other fields are not charged. We have $n$ $U(1)_{\alpha_i}$ symmetries under which $Q_i$ has charge $+1$ and $\widetilde{Q}_i$ charge $-1$ with all the rest of the fields not charged. Finally we have $n$ $U(1)_{t_i}$ symmetries under which fields $Q_i$ and $\widetilde{Q}_i$ have charge $+1$, $\Phi_i$ and $\Phi_{i+1}$ charge $-1$ while the other fields are not charged. The $U(1)_\alpha$ and $U(1)_t$ symmetries are not anomalous. Let us consider the following superpotentials,

$$W = \sum_{i=1} \lambda_i \Phi_i Q_i \widetilde{Q}_i + \lambda_i' \Phi_i Q_{i-1} \widetilde{Q}_{i-1}. \tag{39}$$

Note that the couplings $\lambda_i$ have charges encoded in fugacities as $A_i^{-1} t_i^{-1} t_{i-1}$ while $\lambda_i'$ have charges $A_i^{-1} t_i t_{i-1}^{-1}$. Then $\lambda_i \lambda_i'$ dressed with appropriate power of $\exp(2\pi i \tau_i)$ is a singlet for each $i$. These thus give us $n$ independent exactly marginal parameters preserving in fact $\mathcal{N} = 2$ supersymmetry. Along these directions the $U(1)_{t_i}$ symmetries are broken to a diagonal combination. Note that as all of the couplings are charged under the anomalous symmetries with negative charge these can be taken care of by exponents of gauge couplings.

---

[11]This is very reminiscent of the $\mathcal{N} = 1$ $\beta$ deformation of $\mathcal{N} = 4$ SYM, see *e.g.* [52].

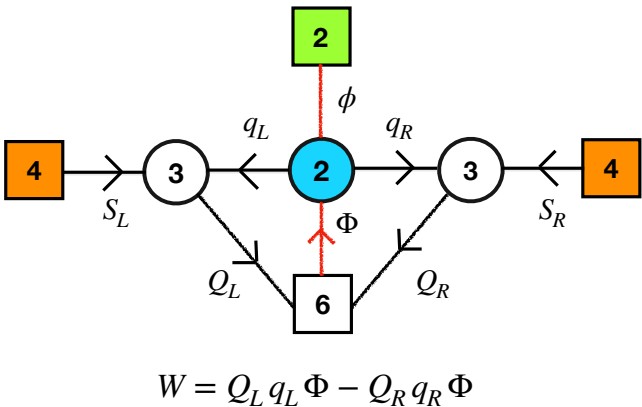

$$W = Q_L \, q_L \, \Phi - Q_R \, q_R \, \Phi$$

Figure 10: A quiver theory with the $SU(2)$ IR free gauge node in the UV.

Above we turned on $\lambda$ and $\lambda'$ couplings in pairs. Now consider turning on only $\lambda$ couplings. Then the combination $\prod_{i=1}^{n} \lambda_i$ (dressed by appropriate power of all $\exp(2\pi i \tau_i)$) is also a singlet of all the symmetries. and thus turning on only $\lambda_i$ gives us an exactly marginal deformation. Here also the $U(1)_{t_i}$ symmetries are broken to a diagonal combination. This deformation is only $\mathcal{N} = 1$ supersymmetric. Note that turning on only $\lambda'$ couplings also is exactly marginal, but it is a combination of the above and the $\mathcal{N} = 2$ exactly marginal couplings.

Note that (39) is not the most general cubic superpotential we can write. Specifically, for $N > 2$, we can also add the terms $\sum_{i=1} \hat{\lambda}_i \Phi_i^3$, where we use the fully symmetric cubic invariant polynomial of $SU(N)$ to contract the gauge indices and form a gauge invariant. The couplings $\hat{\lambda}_i$ then have the charges $A_i^{-3} t_i^3 t_{i-1}^3$. We can cancel the $U(1)_{A_i}$ charges by dressing with appropriate powers of the gauge couplings, but we cannot cancel all the $U(1)_{t_i}$ charges. As such, for generic $N$, these operators are marginally irrelevant. However, for $N = 3$, we further have the baryonic superpotential $\sum_{i=1} \lambda_i^B Q_i^3 + \lambda_i^{\tilde{B}} \widetilde{Q}_i^3$, where we use the epsilon tensor to contract the $SU(3)$ indices to a singlet. The $\lambda_i^B$ have the charges $t_i^{-3} \alpha_i^3$ while $\lambda_i^{\tilde{B}}$ have the charges $t_i^{-3} \alpha_i^{-3}$. We then see that the $\lambda_i^B \lambda_i^{\tilde{B}}$ combination of couplings is uncharged under $U(1)_{\alpha_i}$ and together with $\hat{\lambda}_i$ and the gauge couplings, can be used to form flavor symmetry singlets. Thus, we see that for $N = 3$ there are additional exactly marginal deformations that preserve only $\mathcal{N} = 1$ supersymmetry.

$\square$

> Exercise: Dangerously irrelevant deformations – Consider the quiver theory of Figure 10. The circles inscribing an integer $N$ denote $SU(N)$ gauge groups. The squares inscribing an integer $N$ denote $SU(N)$ flavor groups. The lines denote bifundamental chiral superfields: Fundamental representation of the group they point to and antifundamental of the group they emanate from. The theory has a superpotential denoted on the Figure. Note that the $SU(2)$ gauge group has 14 fundamental fields and thus naively is IR free. By sequentially analyzing the flows starting with the fixed point of the $SU(3)$ SQCD with six flavors and turning on first the superpotentials, show that the $SU(2)$ gauging becomes asymptotically free.

We start with the $SU(3)$ SQCD with six flavors. This theory is asymptotically free,

$$\text{Tr} R\, SU(3)^2 = 3 + \frac{1}{2}\left(\frac{2}{3} - 1\right) \times 2 \times 6 = 1 > 0 \,, \tag{40}$$

where we used the free assignment of R-charges. Assigning R-charge $\frac{1}{2}$ to all the chiral fields

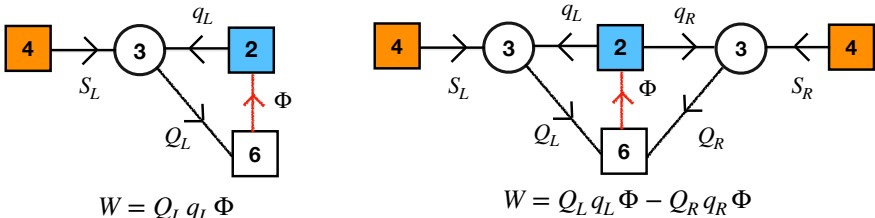

$$W = Q_L q_L \Phi \qquad\qquad W = Q_L q_L \Phi - Q_R q_R \Phi$$

Figure 11: We analyze the flow in three steps. First, we start with the IR fixed point of $SU(3)$ SQCD with six flavors, add 12 chiral fields $\Phi$ and turn on a superpotential (left figure). Then we couple a second copy of $SU(3)$ SQCD with six flavors to the theory through a superpotential (right figure). Finally, we add four more chiral fields $\phi$ and gauge the $SU(2)$ symmetry as in Figure 10.

is anomaly free,

$$\mathrm{Tr}\, R\, SU(3)^2 = 3 + \frac{1}{2}\left(\frac{1}{2} - 1\right) \times 2 \times 6 = 0\,, \tag{41}$$

and in fact this is the superconformal R-symmetry at the fixed point. We have one abelian $U(1)$ symmetry which is non-anomalous under which the fundamentals and the antifundamentals have opposite charges. We thus define a trial $a$-anomaly,

$$a(\alpha) = a_v \times 8 + a_R\left(\frac{1}{2} + \alpha\right) \times 18 + a_R\left(\frac{1}{2} - \alpha\right) \times 18 = \frac{3}{64}\left(41 - 324\alpha^2\right), \tag{42}$$

and the maximum is at $\alpha = 0$. Next we add the 12 fields $\Phi$ and couple them with superpotential as on the left of Figure 11. This interaction breaks some of the nonabelian symmetry. The R-charge of the superpotential at the UV fixed point is $\frac{1}{2} + \frac{1}{2} + \frac{2}{3} < 2$ and thus it is relevant and the theory flows in the IR to a fixed point where there is a non-anomalous R-symmetry under which the R-charges of $q_L$, $S_L$ and $Q_L$ are still $\frac{1}{2}$ but that of $\Phi$ is 1 (such that the R-charge of the superpotential is 2). We are still left with finding the superconformal R-symmetry at that IR fixed point. The theory has two non anomalous abelian symmetries $(U(1)_a^L, U(1)_b)$ so that the charges of the various fields are,

$$q_L : (1,2)\,, \quad S_L : (1,-1)\,, \quad Q_L : (-1,0)\,, \quad \Phi : (0,-2)\,. \tag{43}$$

We thus compute the trial anomaly depending on two parameters $\alpha_a$ and $\alpha_b$ corresponding to the two symmetries,

$$a(\alpha_a^L, \alpha_b) = a_v \times 8 + a_R(1 - 2\alpha_b) \times 12 + a_R\left(\frac{1}{2} + \alpha_a^L - \alpha_b\right) \times 12$$
$$+ a_R\left(\frac{1}{2} + \alpha_a^L + 2\alpha_b\right) \times 6 + a_R\left(\frac{1}{2} - \alpha_a^L\right) \times 18\,. \tag{44}$$

Here in the first line we have the contribution of the vectors and $\Phi$ and on the second line the charged matter fields. The charges of $\Phi$ are fixed by the superpotential as stated above. We compute the maximum of $a(\alpha_a^L, \alpha_b)$ to be at

$$(\alpha_a^L, \alpha_b) = (0.0045, 0.0672)\,,$$

approximately. Now we couple the second copy of the $SU(3)$ SQCD to the theory through a superpotential as on the right of Figure 11. The additional term in the superpotential at the new fixed point has R-charge

$$\frac{1}{2} + \frac{1}{2} + R_\Phi = 1 + (1 - 2 \times 0.0672) = 1.8657 < 2\,,$$

and thus it is relevant. The second copy of the SQCD adds another $U(1)$ symmetry and we compute the new trial $a$-anomaly to be,

$$a(\alpha_a^L, \alpha_b, \alpha_a^R) = a_v \times 8 \times 2 + a_R(1 - 2\alpha_b) \times 12 + a_R\left(\frac{1}{2} + \alpha_a^L - \alpha_b\right) \times 12 \tag{45}$$

$$+ a_R\left(\frac{1}{2} + \alpha_a^L + 2\alpha_b\right) \times 6 + a_R\left(\frac{1}{2} - \alpha_a^L\right) \times 18 + a_R\left(\frac{1}{2} + \alpha_a^R - \alpha_b\right) \times 12$$

$$+ a_R\left(\frac{1}{2} + \alpha_a^R + 2\alpha_b\right) \times 6 + a_R\left(\frac{1}{2} - \alpha_a^R\right) \times 18.$$

Here we used the fact that the superpotential identifies $U(1)_b^L$ with $U(1)_b^R$. We find the maximum of $a(\alpha_a^L, \alpha_b, \alpha_a^R)$ to be at,

$$\alpha_a^L = \alpha_a^R = 0.00135, \qquad \alpha_b = 0.03669.$$

Finally we compute the beta function of the $SU(2)$ gauging at the new fixed point,

$$\text{Tr } R SU(2)^2 = 2 + \frac{1}{2}(R_\phi - 1) \times 2 + \frac{1}{2}(R_\Phi - 1) \times 6 + \frac{1}{2}(R_{q_{L/R}} - 1) \times 6. \tag{46}$$

Now $R_\phi = \frac{2}{3}$ as these are free fields we add at this point and

$$R_{q_{L/R}} = \frac{1}{2} + \alpha_a^{L/R} + 2\alpha_b = 0.5747, \qquad R_\Phi = 1 - 2\alpha_b = 0.9266.$$

We thus deduce that the anomaly in (46) is

$$\text{Tr } R SU(2)^2 = 0.1707 > 0,$$

and thus at the new fixed point the $SU(2)$ gauging is asymptotically free although it has 14 fundamental fields.

This conclusion holds under the assumption that no emergent abelian symmetries appear in neither step of the flow invalidating the $a$-maximization argument. An indication that this assumption is wrong would be if some of the chiral operators violated the unitarity bounds at some of the steps. However, it is easy to verify that none do. For example the field $\Phi$ has R-charge $0.8657 > \frac{2}{3}$ after step one (I), and R-charge $0.9266 > \frac{2}{3}$ after step two (II). The mesons and baryons have the following charges,

$$\begin{array}{llllll}
\text{I:} & SQ = 0.9328, & qQ = 1.1343, & q^3 = 1.9165, & Q^3 = 1.48647, & S^3 = 1.31205, \\
\text{II:} & SQ = 0.9633, & qQ = 1.0734, & q^3 = 1.7242, & Q^3 = 1.4960, & S^3 = 1.3940,
\end{array} \tag{47}$$

all of which are above the unitarity bound.

A famous example of dangerously irrelevant deformations is the case of $SU(N)$ SQCD with $N_f$ fundamental flavors and a chiral field in adjoint representation $\Phi$ with superpotential $W = \text{Tr } \Phi^{n+1}$. For $n > 2$ the superpotential is irrelevant in the UV. However, first flowing with the gauge coupling and then turning on the superpotential it can be relevant with $n > 2$ for a range of choices of $N$ and $N_f$ [53, 54]. $\qquad\qquad\qquad\square$

## 2.4 Supersymmetric RG-flow invariants

We have discussed an invariant of RG flows which does not need supersymmetry, the anomaly polynomial. The supersymmetric flows have in fact many more RG-flow invariants. These invariants typically take a form of some version of the Witten index. Here we will discuss the simplest example of these, the supersymmetric index [55].

Let us briefly review the general construction of a Witten index. Consider the situation that we have two supercharges $Q$ and $Q^\dagger$ such that,

$$\{Q, Q^\dagger\} = \delta\,, \tag{48}$$

with $\delta$ being a combination of bosonic charges. Now consider $V_{\delta_0}$ to be the subspace of the linear space of states of the theory with $|\psi\rangle \in V_{\delta_0}$ satisfying $\delta \cdot |\psi\rangle = \delta_0 |\psi\rangle$ with $\delta_0 > 0$. Now from (48) it follows,

$$|\psi\rangle = \frac{1}{\delta_0}\{Q, Q^\dagger\}|\psi\rangle\,. \tag{49}$$

From here, since $Q$ and $Q^\dagger$ are nilpotent it follows that any state in $V_{\delta_0}$ is a linear combination of a state annihilated by $Q$ and a state annihilated by $Q^\dagger$, a fact we denote as $V_{\delta_0} = V_{\delta_0}^Q \oplus V_{\delta_0}^{Q^\dagger}$. The supercharges $Q$ and $Q^\dagger/\delta_0$ are a one to one map and its inverse from $V_{\delta_0}^{Q^\dagger}$ to $V_{\delta_0}^Q$. From here it follows that the index defined as the following trace over the space of states of the system $\mathcal{H}$,

$$\mathcal{I}(\{u\}) = \text{Tr}_{\mathcal{H}}(-1)^F e^{-\beta\,\delta} \prod_{i=1}^n u_i^{q_i}\,, \tag{50}$$

is independent of $\beta$ as only states in $\mathcal{H}$ with $\delta$ vanishing contribute to it. Here $q_i$ are the charges under the Cartan generators of the $n$ dimensional maximal bosonic subgroup of the symmetry group commuting with $Q$ and $Q^\dagger$. Moreover, this index is invariant under continuous deformations of the theory, and in particular is invariant under the RG flow as long as the symmetries used to define it are preserved along the flow. Let us now discuss a concrete example of such an index.

Consider an $\mathcal{N} = 1$ SCFT. It possesses a superconformal algebra and in particular one of the commutation relations defining the algebra reads (see Appendix A)

$$2\left\{\widetilde{Q}_{\dot\alpha}, \widetilde{S}^{\dot\beta}\right\} = \delta_{\dot\alpha}^{\dot\beta}\left(H - \frac{3}{2}R\right) + 2(J_2)_{\dot\alpha}^{\dot\beta}\,, \tag{51}$$

where $\dot\alpha, \dot\beta = \pm$ are spinorial indices. We think of the space as being $\mathbb{S}^3 \times \mathbb{R}$. $H$ is the operator whose eigenvalue is the scaling dimension $\Delta$, and which generates translations along $\mathbb{R}$. The operators $J_{i=1,2}$ are the generators of the $SU(2) \times SU(2)$ isometry of $\mathbb{S}^3$. Finally, the $\widetilde{Q}$'s are supercharges and the $\widetilde{S}$'s are their conformal counterparts, with $R$ being the superconformal R-symmetry. Now in radial quantization the hermitian conjugate of the $\widetilde{Q}$ supercharges are the $\widetilde{S}$ supercharges, $\widetilde{S}^{\dot\alpha} = \widetilde{Q}^{\dagger\dot\alpha}$. We take $Q = \widetilde{Q}_-$, such that under the Cartan generators $(j_i)$ of $J_i$ it has charges $(0, -\frac{1}{2})$ and has R-charge $+1$. The commutation relation above thus becomes,

$$2\{Q, Q^\dagger\} = H - 2j_2 - \frac{3}{2}R \equiv \delta\,. \tag{52}$$

The operators which commute with $Q$ and $Q^\dagger$ are in addition to $\delta$, $\pm j_1 + j_2 + \frac{1}{2}R$ and the Cartan generators of any global symmetry the theory might have (which we denote by $\mathfrak{Q}_i$). We then can define the superconformal index to be,

$$\mathcal{I}(q, p; \{u\}) = \text{Tr}_{\mathbb{S}^3}(-1)^F e^{-\beta\,\delta} q^{-j_1+j_2+\frac{1}{2}R} p^{j_1+j_2+\frac{1}{2}R} \prod_{i=1}^{\text{Rank}\,G_F} u_i^{\mathfrak{Q}_i}\,. \tag{53}$$

By the general logic of the Witten index described above this index is invariant under continuous deformations of the theory preserving the superconformal algebra, that is, it is invariant

on the conformal manifold $\mathcal{M}_c$ as long as we use only fugacities $u_i$ preserved by the exactly marginal deformations.

On the other hand we also want a quantity which is an invariant of RG-flows. The above definition of the index relies on the superconformal symmetry which is broken once the RG-flow is initiated and thus we cannot use it verbatim. However, there is a simple redefinition of the setup which allows us to define the same index without relying on superconformal symmetry [56] (See [21] Section 7.1 for a nice summary). Instead of thinking about the index as a counting problem we can compute it as a partition function on $\mathbb{S}^3 \times \mathbb{S}^1$ with supersymmetric boundary conditions along the $\mathbb{S}^1$. This is a curved background and analyzing carefully the supersymmetry algebra on it, one can find a supercharge $Q$ which satisfies the commutation relation (52). The charge $\Delta$ in this setup is a suitable combination of the R-symmetry and the translation along the $\mathbb{S}^1$. The construction needs to have an R-symmetry. The index defined in this way is sometimes called the supersymmetric index and it is invariant under the RG-flow as long as we use the R-symmetry which is preserved by the deformations used to start the flow. This R-symmetry does not have to be the one which becomes superconformal at the fixed point. Nevertheless, if we choose the R-symmetry which coincides with the superconformal symmetry in the IR, the supersymmetric index computed this way in the UV coincides with the superconformal index of the IR fixed point. The index as an invariant of RG-flows was first discussed in [57, 58].

The supersymmetric index captures a lot of very interesting RG-flow invariant information about the QFT. For example, let us understand in more detail what the index is counting. As the index is independent of $\beta$ the only states that actually contribute to it have $\delta = 0$. This implies that $\{Q, Q^\dagger\}$ annihilates these states and thus by unitarity both $Q$ and $Q^\dagger$ also annihilate it. States which are annihilated by some supercharges form short representations of the supersymmetry group. The number of short multiplets of a given type might change during the RG flow. However this only can happen if a collection of short multiplets forms a collection of long ones, or if a long multiplet decomposes into short ones. The index is an invariant under such recombinations [55]. We can then think of the index at the fixed point as a sum over the representations of the superconformal symmetry group $\mathfrak{R}_\ell$ and representations of the global symmetry group $G_F$ denoted by $\mathfrak{m}$,

$$\mathcal{I}(q, p; \{u\}) = \sum_{\ell, \mathfrak{m}} n_{\ell, \mathfrak{m}} \, \chi_{\mathfrak{R}_\ell}(q, p) \, \chi_{\mathfrak{m}}(\{u\}), \tag{54}$$

with the integers $n_{\ell, \mathfrak{m}}$ giving the multiplicities of these representations. These multiplicities can change with the flow or when one moves on the conformal manifold but the index is such that one can reorganize this sum in terms of equivalence classes of representations: Two representations are in the same equivalence class if they contribute to the index the same way, possibly up to a sign. This analysis was carefully performed in [41] Section 3. The conclusion is that in each equivalence class there is a finite number of representations and one can decompose the index in terms of net degeneracy numbers $\hat{n}_{\ell, \mathfrak{m}}$ where we now sum over the equivalence classes denoted by $[\mathfrak{R}]$,

$$\mathcal{I}(q, p; \{u\}) = \sum_{\ell, \mathfrak{m}} \hat{n}_{\ell, \mathfrak{m}} \, \chi_{[\mathfrak{R}]_\ell}(q, p) \, \chi_{\mathfrak{m}}(\{u\}). \tag{55}$$

Although in general we can extract information from the index only about these net degeneracies, it so happens that if one thinks of the index in terms of an expansion in the $q$ and $p$ fugacities, at low order in the expansion the multiplets which can contribute are very simple. As we are making use of the superconformal symmetry group the statements below are only true if one uses the superconformal R-symmetry to compute the index and we assume that the theory does not contain free fields.

- The only states that can contribute at order $(qp)^n$ with $n < 1$ are chiral operators of R-charge $R = 2n$. As $n < 1$ these are the relevant operators. Thus we can cleanly read off from the index the spectrum of relevant superpotential deformations.

- The only states which can contribute at order $qp$ are marginal operators, contributing with positive sign, and certain fermionic components from the conserved current multiplet, which contribute with a negative sign. Thus at this order we can extract the combination,

$$\chi_{Marginals}(\{u\}) - \chi_{adjoint\,G_F}(\{u\}). \tag{56}$$

In particular any negative contribution to the index at this order comes from conserved currents and has to be a part of the character of the adjoint representation of the global symmetry group. This is extremely useful to identify the symmetry of the IR fixed point. As we discussed this symmetry might be emergent and assuming we have identified the R-symmetry correctly the index computation can tell us reliably what that symmetry is.

We thus learn that the index provides us with a very non trivial probe about the deformations and the symmetry of the fixed point. As an aside comment, in fact the index also encodes some other non-trivial properties such as 't Hooft anomalies. To extract the anomalies one can send all of the fugacities, $\{q, p, u_i\}$, to 1 and study the leading divergent behavior of the index in this limit [13, 59–62] which turns out to neatly encode the 't Hooft anomalies of various symmetries. We know that the anomalies alone do not uniquely specify a CFT. The index is a rich observable however it also does not uniquely specify a theory, as for example different models on the same conformal manifold $\mathcal{M}_c$ will have the same index.[12]

The discussion till now was very general but the technology of computing the index is rather simple. We will not derive it here and only quote the results. The details can be found *e.g.* in [21, 65, 66]. The technology consists of two main ingredients.

- The index of a chiral field $Q$ with R-charge $R$ and representation $\mathfrak{R}$ of $G_F$ is given by,

$$\mathcal{I}_{R,\mathfrak{R}}(q, p; \{u\}) = \exp\left\{\sum_{m=1}^{\infty} \frac{1}{m} \frac{(qp)^{\frac{R}{2}m}\chi_{\mathfrak{R}}(u^m) - (qp)^{\frac{2-R}{2}m}\chi_{\overline{\mathfrak{R}}}(u^m)}{(1-q^m)(1-p^m)}\right\}. \tag{57}$$

Here $\chi_{\mathfrak{R}}$ is the character of the representation $\mathfrak{R}$. In particular this can be neatly written in terms of the so called elliptic Gamma functions [58],

$$\Gamma_e(z; q, p) = \exp\left\{\sum_{m=1}^{\infty} \frac{1}{m} \frac{z^m - (qp)^m z^{-m}}{(1-q^m)(1-p^m)}\right\} = \prod_{i,j=0}^{\infty} \frac{1 - q^{i+1}p^{j+1}z^{-1}}{1 - q^i p^j z}, \tag{58}$$

which can thus be interpreted as the index of a chiral field with R-charge zero and charge one under the $U(1)_z$ symmetry.

- Given an index of some SCFT $\mathcal{I}_1(q, p; \{u, v\})$ with global symmetry $G_u \times G_v$ we can compute the index of the theory with the $G_v$ symmetry gauged,

$$\mathcal{I}_2(q, p; \{u\}) = \frac{((q;q)(p;p))^{\text{Rank}\,G_v}}{|W_{G_v}|} \oint \prod_{i=1}^{\text{Rank}\,G_v} \frac{dv_i}{2\pi i v_i} \frac{\mathcal{I}_1(q, p; \{u, v\})}{\mathcal{I}_{0,(\text{adj.}\,G_v - \text{rank}\,G_v)}(q, p; \{v\})}. \tag{59}$$

---

[12]Theories differing solely by their higher form symmetries also might have the same index, say $SU(N)$ and $SU(N)/\mathbb{Z}_N$ $\mathcal{N} = 4$ SYM. However this difference can be captured by other types of partition functions, *e.g.* the Lens index [63, 64].

Here $W_{G_\nu}$ is the Weyl group of $G_\nu$. We also define $(z; y) = \prod_{i=0}^{\infty}(1 - zy^i)$. The contribution $1/\mathcal{I}_{0,(\text{adj.}\,G_\nu - \text{rank}\,G_\nu)}$ comes from the vector multiplets. It is equivalent to one over the contribution of chiral superfield with R-charge zero in adjoint representation without the contribution coming from the Cartan generator, $\chi_{\text{adj.}\,G_\nu - \text{rank}\,G_\nu} = \chi_{\text{adj.}\,G_\nu} - \text{rank}\,G_\nu$.

The superpotentials only effect the index through the restrictions they impose on flavor symmetry and R-symmetry. This index can be computed with any R-symmetry but it becomes the superconformal index of the fixed point only if the superconformal R-symmetry is used.

> Exercise: Consider the simplest QFT, two chiral superfields $Q$ and $\widetilde{Q}$ coupled with the superpotential $mQ\widetilde{Q}$. Compute the supersymmetric index and interpret it.

First we need to determine the symmetries and the charges of the model. At the free point the two chiral fields can be rotated by $U(2)$ but the superpotential breaks it to $U(1)$ under which the two fields have an opposite charge. We chose to normalize the charges to be $\pm 1$. The R-symmetry of the superpotential is 2 and thus the sum of the R-charges of the two fields is $R_Q + R_{\widetilde{Q}} = 2$. We then use the first entry in the technology of computing the index above to write the index of the system to be,

$$\mathcal{I}_{R_Q,+1}(q,p;u)\,\mathcal{I}_{2-R_Q,-1}(q,p;u) = \Gamma_e\left((qp)^{\frac{R_Q}{2}}u;q,p\right)\Gamma_e\left((qp)^{\frac{2-R_Q}{2}}u^{-1};q,p\right) = 1\,. \tag{60}$$

The last equality is derived by direct evaluation using the definition of the elliptic Gamma function (58). Let us try to interpret the result. The fact that the index is 1 means that only the identity operator corresponding to the vacuum contributes to it and there are no other protected operators (or to be more precise the spectrum of operators is such that all protected ones can recombine into long ones). The theory is massive and thus it is gapped and in the IR we will not have any propagating degrees of freedom. The theory has a single state, the vacuum, and thus the index is consistent with this. Note that this is a trivial example where the UV $U(1)_u$ symmetry does not act faithfully in the IR. For this reason it is also not important what the superconformal R-symmetry is. In particular the trial a-anomaly is identically zero,

$$a(\alpha) = a_R(R_Q + \alpha) + a_R(2 - R_Q - \alpha) = 0\,. \tag{61}$$

This is a trivial example, however in the more interesting cases the computations are not much more complicated and the physics can be extracted in a similar manner.

$\square$

We are now ready to study interesting physical systems using the simple toolkit of non-perturbative techniques we have reviewed.

# 3 Lecture II: Examples of strong coupling dynamics

Let us start our discussion of the dynamics of supersymmetric field theories with several examples of interesting strongly coupled behavior. We will review the phenomenon of IR duality, discuss the interplay between duality and emergence of symmetry in the IR, and discuss a simple algorithm to look for different weakly coupled theories residing conjecturally on the same conformal manifold. The purpose of this Lecture is to introduce various possible physical scenarios and effects. In later Lectures we will discuss a way to more systematically discuss such effects using geometric constructions.

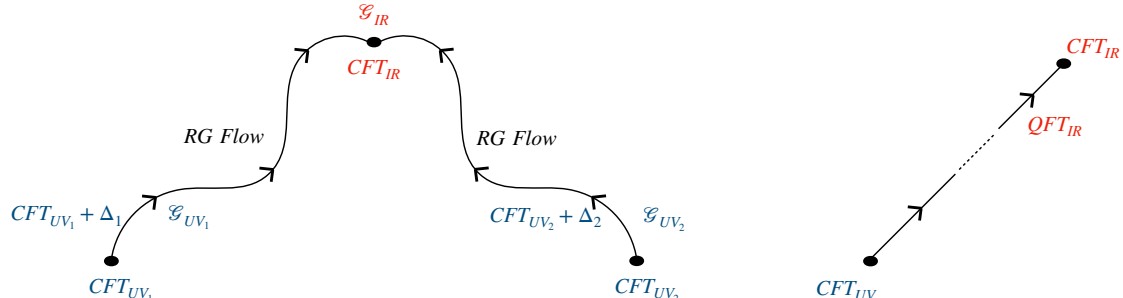

Figure 12: A depiction of IR dualities. On the left we have two different UV CFTs deformed to flow to the same IR fixed point. On the right we have a UV CFT deformed (*e.g.* a weakly coupled asymptotically free gauge theory) flowing in the IR to a QFT which close to the IR fixed point can be described by yet another weakly coupled theory (*e.g.* an IR free gauge theory).

## 3.1 IR dualities

Let us consider some of the basic examples of IR dualities discovered by Seiberg [2]. First, let us consider $\mathcal{N} = 1$ $SU(N)$ gauge theories (SQCD) with $N_f$ fundamental chiral fields and $N_f$ antifundamental ones: This is referred to as the theory having $N_f$ flavors. The matter content is non-anomalous for any $N$. For $N > 2$ we should worry about cubic anomalies $\mathrm{Tr} \, SU(N)^3$, which vanishes because the matter representation is real. For $N = 2$ there is no difference between fundamentals and antifundamentals but we have to have an even number of these so that the theory will be free of a Witten anomaly [67]. For $N_f \geq 3N$ ($N_f > 9$ for $N = 3$) the theory is IR free and thus we need to worry about UV completing the theory, as the couplings grow when we flow back to the UV. For $N_f < 3N$ ($N_f \leq 9$ for $N = 3$) these models are asymptotically free and thus can be thought of as deformations of Gaussian fixed points in the UV. The beta function is such that when flowing to the IR the gauge coupling grows and we are interested in understanding what is the effective description in the IR. We will be in particular interested in the case of $N_f > N$ as here the dynamics turn out to be rather interesting.[13] Here is the basic statement.

- For $\frac{3}{2}N < N_f < 3N$ $SU(N)$ SQCD with $N_f$ flavors, $Q_i$ and $\widetilde{Q}_i$, and no superpotential flows to an interacting SCFT in the IR. Moreover, the $SU(N_f - N)$ SQCD with $N_f$ flavors, $q_i$ and $\widetilde{q}_i$, and $N_f^2$ gauge singlet chiral fields, $M_{ij}$, with a superpotential $W = M \cdot q \cdot \widetilde{q}$, flows to exactly the same fixed point. The phenomenon of two different UV theories flowing to the same IR fixed point is called IR duality. See Figure 12 on the left for an illustration. This range of parameters is called *the conformal window*.

- For $N + 1 \leq N_f \leq \frac{3}{2}N$ $SU(N)$ SQCD with $N_f$ flavors, $Q_i$ and $\widetilde{Q}_i$, and no superpotential flows to an IR free theory. The effective description in the IR is that of $SU(N_f - N)$ SQCD with $N_f$ flavors, $q_i$ and $\widetilde{q}_i$, and $N_f^2$ gauge singlet chiral fields, $M_{ij}$, with a superpotential $W = M \cdot q \cdot \widetilde{q}$. Note that the latter theory satisfies in the given range of parameters $N_f > 3(N_f - N)$ and thus the theory is IR free. In other words the $SU(N_f - N)$ SQCD is UV completed here by the $SU(N)$ SQCD. See Figure 12 on the right for an illustration. In addition, see Figure 13 for a quiver description of Seiberg duality.

---

[13]For $N_f \leq N$ the dynamics becomes trickier, *e.g.* it involves effects such as runaway vacua and quantum deformed moduli spaces, that we will not discuss here [2]. See [21] for an excellent review.

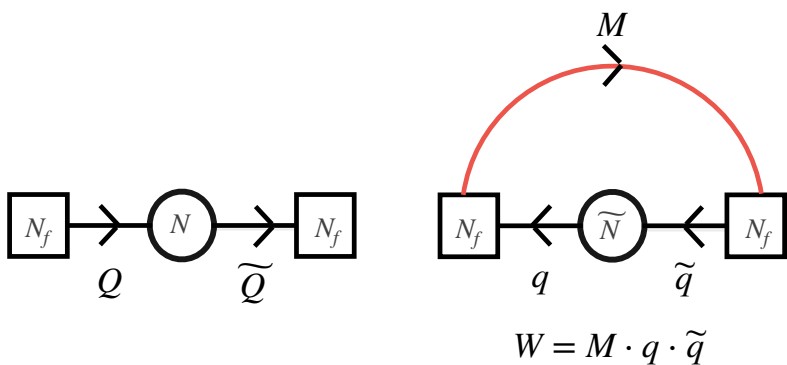

$$W = M \cdot q \cdot \widetilde{q}$$

Figure 13: A depiction of Seiberg duality with $SU(N)$ gauge groups.

Let us discuss some evidence for these statements. First, one can check the 't Hooft anomalies of the various symmetries: In the first case of the two different UV theories and in the second case of the weakly coupled theories in the UV and in the IR. We leave this as a simple exercise. Second, one can check that the supersymmetric indices of the relevant theories agree. In fact, there is a mathematical proof due to Rains that they do [68]. The proof is rather non trivial so let us here quote a simple computation for the simplest duality, $SU(2)$ SQCD with $N_f = 3$ dual to a WZ model (See Figure 14). Following the general rules we have outlined the duality implies the following identity of indices,

$$(q;q)(p;p) \oint \frac{du}{4\pi i u} \frac{\prod_{i=1}^{6} \Gamma_e((qp)^{\frac{1}{6}} u^{\pm 1} a_i; q, p)}{\Gamma_e(u^{\pm 2}; q, p)} = \prod_{i<j} \Gamma_e((qp)^{\frac{1}{3}} a_i a_j; q, p). \tag{62}$$

On the left we have the index of $SU(2)$ SQCD. The numerator comes from the six fundamental fields which have anomaly free R-charge of $1/3$. The symmetry is $SU(6)$ and $a_i$ parametrize the Cartan of this symmetry, $\prod_{i=1}^{6} a_i = 1$. The denominator comes from the vector superfield. The $\pm$ in the integral is a shorthand notation for the following, $f(x^{\pm 1}) \triangleq f(x) \cdot f(x^{-1})$. On the right hand side we have a WZ model with $\frac{6 \times 5}{2} = 15$ chiral fields with R-charge $2/3$ which are coupled with a cubic superpotential. The $SU(2)$ SQCD with $N_f = 3$ flows in the IR to a WZ model of 15 chiral fields with cubic superpotential, which flows to a free theory. The identity above was originally obtained by S. Spiridonov [69] and dubbed elliptic Beta function as in certain degeneration limits of parameters it becomes the well known Beta function integral identity.

Next consider a duality in the conformal window, $SU(2)$ SQCD with $N_f = 4$ dual to $SU(2)$ SQCD with $N_f = 4$ and a collection of 16 chiral fields. The index of the two dual theories is,

$$(q;q)(p;p) \oint \frac{du}{4\pi i u} \frac{\prod_{i=1}^{8} \Gamma_e((qp)^{\frac{1}{4}} u^{\pm 1} a_i; q, p)}{\Gamma_e(u^{\pm 2}; q, p)} = \prod_{i,j=1}^{4} \Gamma_e((qp)^{\frac{1}{2}} a_i a_{4+j}; q, p) \tag{63}$$

$$\times (q;q)(p;p) \oint \frac{du}{4\pi i u} \frac{\prod_{i=1}^{4} \Gamma_e((qp)^{\frac{1}{4}} u^{\pm 1} a a_i^{-1}; q, p) \prod_{i=5}^{8} \Gamma_e((qp)^{\frac{1}{4}} u^{\pm 1} a^{-1} a_i^{-1}; q, p)}{\Gamma_e(u^{\pm 2}; q, p)}.$$

Here the superconformal R-charge on both sides is $\frac{1}{2}$ for all the fundamental chiral superfields. The symmetry on the left is $SU(8)$ parametrized by $a_i$, $\prod_{i=1}^{8} a_i = 1$. On the right the symmetry in the UV is $SU(4) \times SU(4) \times U(1)$. The $U(1)$ is parametrized by $a^{\frac{1}{2}} \equiv \left(\prod_{i=1}^{4} a_i\right)^{\frac{1}{4}}$. The two $SU(4)$'s are parametrized by $\{a^{\frac{1}{2}} a_i^{-1}\}_{i=1}^{4}$ and $\{a^{-\frac{1}{2}} a_i^{-1}\}_{i=5}^{8}$. Note that $SU(4) \times SU(4) \times U(1)$

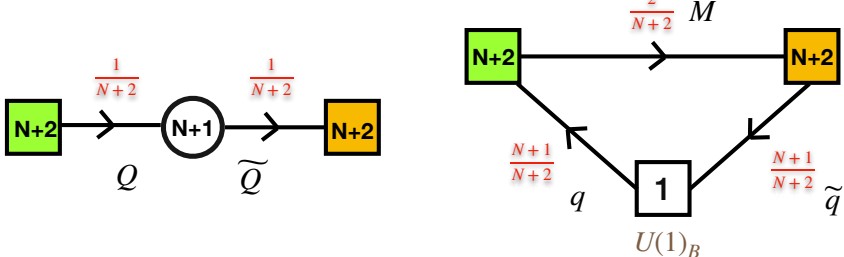

Figure 14: When the number of flavors is one more than the number of colors the theory in the IR is a WZ model with a superpotential $q\widetilde{q}M + M^{N+2}$ term. This phenomenon is called S-confinement. In the IR the WZ models flows to a collection of free chiral fields.

is a subgroup of $SU(8)$ and thus if this duality is correct the symmetry has to enhance to $SU(8)$ in the IR. This is a simple example of emergence of symmetry. As we have discussed the supersymmetric relevant operators are invariant under RG flows and thus have to match across the duality. Indeed the operatots $Q_i\widetilde{Q}_j$ match with $M_{ij}$, $Q_iQ_j$ match with $q_iq_j$, and $\widetilde{Q}_i\widetilde{Q}_j$ match with $\widetilde{q}_i\widetilde{q}_j$.

> Exercise: Compute the index of the $N_f = 4$ $SU(2)$ SQCD up to order $qp$ using the superconformal R-symmetry. What is the representation of the marginal operators under the $SU(8)$ global symmetry? This theory is extremely interesting. In fact it has (at least) 72 different dual descriptions [70]: (The number 72 is interesting: It is the ratio of the dimension of the Weyl group of $E_7$ and $SU(8)$. There is an $E_7$ lurking behind this theory. To see it one needs to do some work [71] (See also [72]).) The two Seiberg dual theories here are just 2 out of the 72 different duality frames. Can you find another 34?

We can use the integral expressions for the index above to compute the contribution at order $qp$,

$$\chi_{\mathbf{336}}(\{a\}) - \chi_{\mathbf{63}}(\{a\}). \tag{64}$$

Here $\mathbf{336}$ irrep of $SU(8)$ appears in decomposition of $\mathrm{Sym}^2\mathbf{28} = \mathbf{336} + \mathbf{70}$. The $\mathbf{28}$ is the representation of $Q \cdot Q$ and the marginal operators come from $(Q \cdot Q)^2$. We note in passing that $SU(8)$ is a maximal subgroup of $E_7$ with $\mathbf{133}_{E_7} = \mathbf{70} + \mathbf{63}$. The above can be written as,

$$\chi_{\mathrm{Sym}^2\mathbf{28}}(\{a\}) - \chi_{\mathbf{70}}(\{a\}) - \chi_{\mathbf{63}}(\{a\}) = \chi_{\mathbf{336}}(\{a\}) - \chi_{\mathbf{63}}(\{a\}). \tag{65}$$

The $E_7$ is not however a symmetry of the theory in the IR, the $-\chi_{\mathbf{70}}$ comes from trace relations and not from a conserved current. We see that $E_7$ is lurking under the surface, and in the next Lecture we will see some geometric importance of this.[14] The positive contributions are the marginal operators and the negative are the conserved currents. Note that we can here identify the positive and the negative contributions as these have to be characters of representations of $SU(8)$. Note also that there is no direction on the conformal manifold preserving the full $SU(8)$ symmetry as the marginal operators lack a singlet of this group.

Regarding the second part of the question: Note that to construct the dual theory we need to split the eight fundamentals into fundamentals and anti-fundamentals, which is an arbitrary procedure for $SU(2)$. We thus have $\frac{1}{2} \times (8!/4!^2) = 35$ different ways to do so. This

---

[14]There is an interesting interplay between kinematic constraints, more generally chiral ring relations, and enhanced symmetries which we will not review here [73].

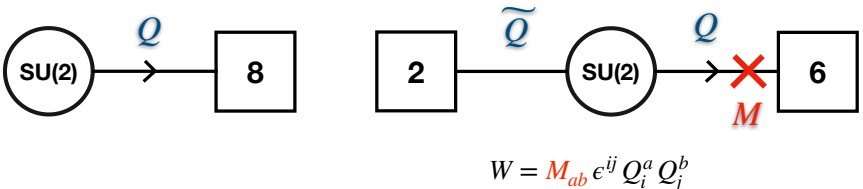

$$W = M_{ab}\, \epsilon^{ij}\, Q_i^a\, Q_j^b$$

Figure 15: On the left $SU(2)$ SQCD with eight fundamental fields. On the right a relevant deformation thereof.

immediately gives us 35 inequivalent possibilities of Seiberg duality. There are in fact another $35 + 1$ dualities which were discussed in [74, 75]. □

The $SU(2)$ SQCD with various amounts of flavors are probably the simplest supersymmetric gauge theories and already these exhibit extremely rich dynamics. We will now analyze yet another interesting strong coupling effect that can be derived starting from $SU(2)$ SQCD with $N_f = 4$ [23].

## 3.2 Emergence of symmetry in the IR

Let us consider $SU(2)$ SQCD with $N_f = 4$. We split the eight chiral fields into six ($Q$) and two ($\widetilde{Q}$). We also introduce gauge invariant operators $M$ coupling through a superpotential as,

$$W = M_{ab}\epsilon^{ij}Q_i^a Q_j^b\,. \tag{66}$$

Note that this superpotential is relevant as at the SQCD fixed point the R-charge of the quarks is $\frac{1}{2}$ and the R-charge of the gauge singlets, which are free fields at the fixed point, is $\frac{2}{3}$. The quiver theory is depicted in Figure 15 and charges of the various fields under the symmetries of the model are detailed in the table below. The gauge singlet fields $M$ and the superpotential break the symmetry of the model from $SU(8)$ down to $SU(6) \times SU(2) \times U(1)_h$. We will show eventually that this symmetry enhances to $E_6 \times U(1)_h$. We also note that $SU(8)$ is not a subgroup of $E_6$.

| Field | $SU(2)_G$ | $SU(2)$ | $SU(6)$ | $U(1)_h$ | $U(1)_{\hat{r}}$ |
|---|---|---|---|---|---|
| $Q$ | **2** | **1** | **6** | $\frac{1}{2}$ | $\frac{5}{9}$ |
| $\widetilde{Q}$ | **2** | **2** | **1** | $-\frac{3}{2}$ | $\frac{1}{3}$ |
| $M$ | **1** | **1** | $\overline{\mathbf{15}}$ | $-1$ | $\frac{8}{9}$ |
| $\widetilde{M}$ | **1** | **1** | **1** | $3$ | $\frac{4}{3}$ |

In the table $U(1)_{\hat{r}}$ is the superconformal R-symmetry obtained by $a$-maximization [76] and the conformal anomalies are $c = \frac{29}{24}, a = \frac{13}{16}$. Note that the operator $\widetilde{Q}\widetilde{Q}$ has superconformal R-symmetry $\frac{2}{3}$ and thus is a free field in the IR. This means that in particular we have an emergent $U(1)$ symmetry under which this field, and only it, is charged. Emergence of abelian symmetries might invalidate the $a$-maximization, however this is not the case here. The operator does not violate the bound but rather saturates it and thus following a version of one of the previous exercises taking into account the emergent symmetry will not violate the conclusion. We are interested however only in the interacting part of the IR SCFT and thus we can remove the free field simply by "flipping" it [77] (See also [50].). The operation of flipping [15] an operator $\mathcal{O}$ is preformed by introducing a chiral field $\Phi_\mathcal{O}$ and turning on a superpotential

$$W = \Phi_\mathcal{O}\,\mathcal{O}\,.$$

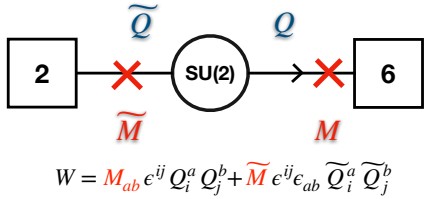

$$W = M_{ab}\,\epsilon^{ij}\,Q_i^a\,Q_j^b + \widetilde{M}\,\epsilon^{ij}\epsilon_{ab}\,\widetilde{Q}_i^a\,\widetilde{Q}_j^b$$

Figure 16: The model with $E_6 \times U(1)$ global symmetry.

In our case since $\mathcal{O} = \widetilde{Q}\widetilde{Q}$ is a free field the flipping is just a mass term for $\mathcal{O}$, both $\Phi_{\mathcal{O}} = \widetilde{M}$ and $\mathcal{O}$ become massive and decouple in the IR. Using this superconformal R-symmetry we compute the index after introducing $\widetilde{M}$ (see Figure 16),

$$1 + \overline{\mathbf{27}}h^{-1}(qp)^{\frac{4}{9}} + h^3(qp)^{\frac{2}{3}} + \cdots + (-\mathbf{78} - 1)qp + \dots \tag{67}$$

The bold-face numbers are representations of $E_6$ as we will elaborate momentarily, and $h$ is the fugacity for $U(1)_h$. We remind the reader that the power of $qp$ is half the R-charge for scalar operators and we observe that all the operators are above the unitarity bound. Let us count some of the operators contributing to the index. The relevant operators of the model are $Q\widetilde{Q}$ and $M$ which comprise the $(\mathbf{2}, \mathbf{6})$ and $(\mathbf{1}, \overline{\mathbf{15}})$ of $(SU(2), SU(6))$, which gives $\overline{\mathbf{27}}$ of $E_6$. We also have $\widetilde{M}$, a singlet of non abelian symmetries. At order $qp$, as we have discussed, assuming the theory flows to an interacting conformal fixed point, the index gets contributions only from marginal operators minus conserved currents for global symmetries. The operators contributing at order $qp$ are gaugino bilinear $\lambda\lambda$ $((\mathbf{1},\mathbf{1}))$, $Q\overline{\psi}_Q$ $((\mathbf{1},\mathbf{35}+\mathbf{1}))$, $\widetilde{Q}\overline{\psi}_{\widetilde{Q}}$ $((\mathbf{3}+\mathbf{1},\mathbf{1}))$: These operators give the contribution,

$$1 - (\mathbf{1},\mathbf{35}) - 1 - (\mathbf{3},\mathbf{1}) - 1\,,$$

which gives the conserved currents for the symmetry we see in the Lagrangian. Here $\overline{\psi}_F$ is the complex conjugate Weyl fermion in the chiral multiplet of the scalar $F$. We also have operators $\overline{\psi}_M M$, $\overline{\psi}_{\widetilde{M}}\widetilde{M}$, $\widetilde{M}\widetilde{Q}\widetilde{Q}$, and $QQM$, which cancel out in the computation since the first two are fermionic while the second two are bosonic, but are both in the same representation of the flavor symmetry pairwise. Finally we have $Q^3\widetilde{Q}$ $((\mathbf{2},\mathbf{70}))$ and $Q\overline{\psi}_M\widetilde{Q}$ $((\mathbf{2},\mathbf{20}+\mathbf{70}))$. These two contribute

$$-(\mathbf{2},\mathbf{20})$$

to the index, which, combined with the above, forms the character of the adjoint representation of the $E_6 \times U(1)_h$ symmetry. We emphasize that the fact we see $-\mathbf{78}$ at order $qp$ of the index is a proof following from representation theory of the superconformal algebra that the symmetry of the theory enhances to $E_6$ (See the discussion around (56).), where the only assumption is that the theory flows to an interacting fixed point. We also observe that the conformal manifold here is a point as we do not have any positive contributions to the index. As the index at order $qp$ has positive contributions from marginal operators and negative from conserved currents, cancellations can occur. However, this would imply that the symmetry of the IR fixed point is even bigger: Adding marginal operators we have to add also currents. We cannot rule out this possibility but we have no evidence for its existence. So, under the assumption that we have identified the IR symmetry correctly the conformal manifold is a point.

The enhancement of symmetry to $E_6$ follows from Seiberg duality of $SU(2)$ SQCD with $N_f = 4$. Note that we can reorganize the gauge charged matter into two groups of four chiral fields. We take four out of the six $Q$s and call them fundamentals and combine the other two with $\widetilde{Q}$ and call those anti-fundamentals. This also decomposes the symmetry $SU(6)$ to

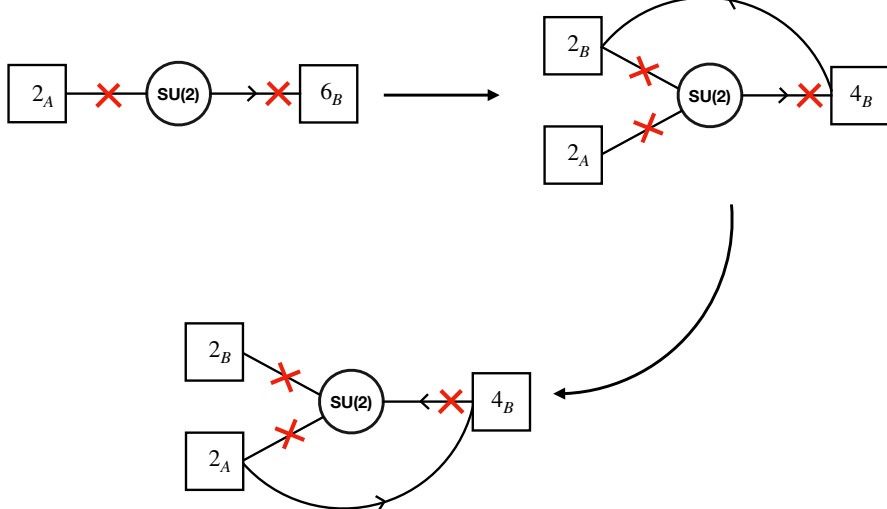

Figure 17: Explanation of $SU(6) \times SU(2) \rightarrow E_6$ enhancement from Seiberg duality.

$SU(4) \times SU(2) \times U(1)_{h'}$ with a combination of $U(1)_{h'}$ and $U(1)_h$ being the baryonic symmetry, see Figure 17. IR duality [2] without the gauge singlet fields will then map the baryonic symmetry to itself while conjugating the two $SU(4)$ symmetries and adding sixteen gauge singlet mesonic operators. With our choice of gauge singlet fields, the flipper fields of 17 are flipping eight of the baryons and the bifundamental gauge invariant operators form half of the mesons. Thus, the duality removes half of the mesons which connect $SU(2)$ with $SU(4)$ and attaches the other half between the $SU(4)$ and the other $SU(2)$. This transformation acts only on the symmetry leaving the quiver structurally unchanged. The action on the symmetry is as the Weyl transformation which enhances $SU(6) \times SU(2)$ symmetry to $E_6$. Note that in general dualities take two different UV theories to the same conformal manifold in the IR, but here as the conformal manifold is a point they actually are part of the symmetry group of the IR theory.

This is an example of a generic phenomenon of the interplay between symmetry and duality which we will encounter in several guises in what follows. In the last Lecture we will have a geometric explanation of the enhancement of the symmetry to $E_6$ in this example.

## 3.3 Conformal dualities

Next, we consider yet another interesting strong coupling phenomenon, which however does not involve an RG-flow. We want to consider the case when a given SCFT $T_1$ resides on a non trivial conformal manifold. We assume that there is an interesting/useful definition of this particular point of the conformal manifold. The conformal manifold is then spanned by exactly marginal deformations at $T_1$. One thing that can happen, and often does happen, is that there is another locus of the conformal manifold, $T_2$, where we have a completely different description of the theory. The theory $T_1$ then can be thought of as an exactly marginal deformation of $T_2$, and vice versa. This multitude of descriptions is called a conformal duality. A prototypical example is $\mathcal{N} = 4$ SYM with gauge group $G$ which has $\mathcal{N} = 4$ preserving one dimensional conformal manifold parameterized by complexified gauge coupling $\tau$. At strong coupling a dual weakly coupled description emerges in terms of $\mathcal{N} = 4$ SYM, but now with a Langlands dual gauge group $^L G$ [3] (for $SU(N)$ the dual is $SU(N)/\mathbb{Z}_N$). In general, such a duality might relate two weakly coupled gauge theories as in the case of $\mathcal{N} = 4$ SYM, a weakly coupled gauge theory and a strongly coupled one defined in a certain way (say geometrically), or two strongly coupled theories which have certain independent definition (see Figure 18).

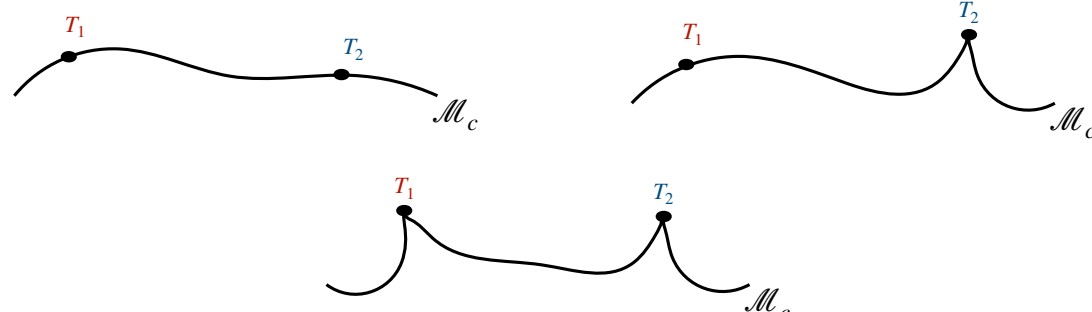

Figure 18: Conformal dualities. Theories residing on the same conformal manifold are usually called conformal dual to each other. A theory $T_1$ can have its own description or can be thought of as an exactly marginal deformation of $T_2$, and vice versa. This tautological definition of duality becomes interesting when indeed the two theories $T_i$ have a rather different independent definition. For example, one, or both, are given by a weakly coupled gauge theory. We denote here weakly coupled points on the conformal manifolds as cusps as these are singular loci on the conformal manifold.

We will be interested in addressing the following question [22]. Given a conformal theory $T_1$ which has a conformal manifold $\mathcal{M}_c$ we can define certain observables of $T_1$ which are invariant on $\mathcal{M}_c$. For example, we have already discussed that the 't Hooft anomalies of symmetries preserved on the conformal manifold are such an invariant, and for supersymmetric theories also the $a$ and $c$ anomalies, and various indices are invariant.[15] In particular the dimension of the conformal manifold $\dim \mathcal{M}_c$ and the symmetry preserved on a generic locus of the conformal manifold $G_F$, are such invariants.[16] Given $T_1$ and the set of its $\mathcal{M}_c$ invariants we will systematically seek for a dual *conformal gauge theory* $T_2$. Following our discussion we will be either able to prove that such a dual does not exist, or find a *conjecture* for such a dual. The conjecture will be supported by comparing robust supersymmetric properties and various anomalies. Even given that these agree the conjecture might in principle still be wrong.

The algorithm for seeking for a dual proceeds as follows. We first write down the $a$ and the $c$ anomalies of the theory $T_1$. These are quantities which for supersymmetric theories do not change on the conformal manifold. We parametrize the result as,

$$a = a_v \dim \mathcal{G} + a_\chi \dim \mathcal{R}, \quad c = c_v \dim \mathcal{G} + c_\chi \dim \mathcal{R}. \tag{68}$$

On the right hand side of each expression we have denoted by $(a_\chi, c_\chi) = (\frac{1}{48}, \frac{1}{24})$ and $(a_v, c_v) = (\frac{3}{16}, \frac{1}{8})$ the contributions to the $a$ and $c$ anomalies of free chiral and vector fields respectively. The anomalies of the free chiral field are computed assigning to it R-charge $\frac{2}{3}$. Next, the variables $\dim \mathcal{G}$ and $\dim \mathcal{R}$ can be thought of as the "effective number" of vector multiplets and chiral multiplets of our theory. The $a$ and $c$ anomalies are two independent numbers which we can parametrize uniquely by the two independent numbers $\dim \mathcal{G}$ and $\dim \mathcal{R}$.

The putative dual conformal gauge theory $T_2$ should have the same number of vector and chiral fields as we assume it is a weakly coupled conformal deformation of a free theory. *If a and c imply that these numbers are fractional, a conformal dual gauge theory does not exist.*

---

[15] For non supersymmetric theories as there is no relation between 't Hooft and conformal anomalies and the $c$ anomaly in principle can change on the conformal manifold [78] (see also [79]).

[16] As we have discussed the exactly marginal operators correspond to certain short multiplets, number of which cannot change as we vary continuous parameters.

Moreover, the number of vector fields $\dim \mathcal{G}$ must be a sum of dimensions of non-abelian gauge groups. Given the solution to the above problem exists, we then write down all the conformal gauge theories with the given number of free vectors and free chiral fields with an additional demand that the one loop beta function of all gauge couplings is zero. Importantly, the number of theories in this step is finite. If we find theories which have (one-loop) marginal gauge couplings we need to understand whether these theories are free or possess a non trivial conformal manifold. This conformal manifold also has to have the same dimension, $\dim \mathcal{M}_c$, and same symmetry on a generic locus, $G_F$, as $T_1$. Here the computation is performed by computing the Kähler quotient [40] as we have discussed in the previous Lecture: The number and properties of holomorphic combinations of couplings which are singlets under the full global symmetry group. The final step of the algorithm is to take all the theories with matching conformal manifolds and symmetries on the general locus and do finer checks: These can include, but are not limited to, various supersymmetric partition functions and 't Hooft anomalies. Any theory passing the last step qualifies as a *conjecturally conformal dual model to* $T_1$. Importantly if a conformal dual to $T_1$ exists we are guaranteed to find it using this procedure.

We will now discuss in detail a simple application of this algorithm to produce a duality between two weakly coupled conformal gauge theories [22]. As theory $T_1$ we will take $\mathcal{N} = 1$ SQCD with gauge group $G_2$, three fundamental fields (**7**) $Q_i$, and one chiral field in the **27**, $\widetilde{Q}$ (see Figure 19). The irrep **27** of $G_2$ appears in $\mathrm{Sym}^2 \mathbf{7} = \mathbf{27} + \mathbf{1}$. The one loop gauge beta function of this model vanishes,

$$\mathrm{Tr}\, R\,(G_2)^2 = 4 + 1 \times \left(\frac{2}{3} - 1\right) \times 3 + 9 \times \left(\frac{2}{3} - 1\right) = 0\,, \tag{69}$$

which determines then the superconformal R-charges of all the chiral fields to be the free ones. In the above computation 4 is the Dynkin index of the adjoint representation coming from the gauginos, 1 is the Dynkin index of the fundamental representation coming from the $Q_i$'s, and 9 is the Dynkin index of **27** and coming from $\widetilde{Q}$.

The non-anomalous global symmetry of the free locus of this SQCD is $U(1) \times SU(3)$. The fundamental fields $Q_i$ are in a three dimensional representation of $SU(3)$ (with no loss of generality we can take it to be the fundamental) and we will assign these $U(1)$ charge $-3$. The field in **27** is the assigned charge $+1$, which renders the $U(1)$ symmetry non anomalous,

$$\mathrm{Tr}\, U(1)\,(G_2)^2 = 1 \times (-3) \times 3 + 9 \times (+1) = 0\,. \tag{70}$$

Next, we analyze the marginal operators. The fundamental irrep of $G_2$ contains a singlet in its cubic antisymmetric power. Thus, operator $\gamma\, \epsilon^{ijk} Q_i Q_j Q_k$ is gauge invariant and is a marginal one. Moreover, this operator is a singlet of the $SU(3)$ global symmetry and has $U(1)$ charge $-9$. We can also build marginal operators from $\widetilde{Q}$: An interesting group theory fact is that $\mathrm{Sym}^3 \mathbf{27}$ has two independent invariants. These two then provide two additional marginal operators which we will denote as $\gamma_1 (\widetilde{Q})_1^3$ and $\gamma_2 (\widetilde{Q})_2^3$. Both of these operators are of course singlets of $SU(3)$ and have $U(1)$ charge $+3$. We have one additional marginal operator which contains both types of fields, $\gamma^{ij} \widetilde{Q} Q_{[i} Q_{j]}$. Unlike the other operators this transforms under $SU(3)$ in the symmetric six dimensional representation. The $U(1)$ charge of this operator is $-5$. This theory has one gauge coupling and an anomalous symmetry under which all the fields can be chosen to have the same positive charge. Thus all the marginal operators will have a positive charge, implying that the associated coupling all have negative charge. However, the gauge coupling carry positive charge. Thus, in computations of the conformal manifold in this case we can always account for the anomalous symmetry by an appropriate factor involving the gauge coupling and we will not discuss this further.

Next we analyze the dimension and properties of the conformal manifold of this SQCD. To do so we need to compute the Kähler quotient [40],

$$\{\gamma, \gamma_{1,2}, \gamma^{ij}\}/(SU(3) \times U(1))_{\mathbb{C}}\,.$$

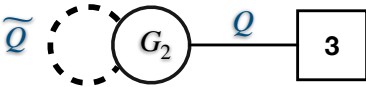

Figure 19: The $G_2$ conformal SQCD.

First let us take the couplings which preserve the $SU(3)$, $\{\gamma, \gamma_1, \gamma_2\}$. From these three we can construct *two* independent singlets ($\gamma_i^3 \gamma$) giving rise to two exactly marginal directions. The conformal manifold is thus not empty and on this two dimensional locus we break the $U(1)$ symmetry while preserving the $SU(3)$ one.

We have however an additional exactly marginal direction which involves also the operator charged under the $SU(3)$ symmetry. For example let us consider exploring the two dimensional manifold above. Now as only the $SU(3)$ symmetry is preserved we need to find whether we can build a singlet from powers of $\gamma^{ij}$. A natural singlet is a determinant of this symmetric matrix,

$$\epsilon_{i_1 i_2 i_3} \epsilon_{j_1 j_2 j_3} \gamma^{i_1 j_1} \gamma^{i_2 j_2} \gamma^{i_3 j_3} .$$

Since the $\gamma^{ij}$ transforms under the $SU(3)$ symmetry it is broken along this direction. However, it is not completely broken: There is an $SO(3)$ symmetry preserved. We can deduce this for example by noting that if we imbed $SO(3)$ in $SU(3)$ such that the adjoint of the former is the fundamental of the latter the six dimensional irrep of $SU(3)$ decomposes into a singlet and a five dimensional irrep of $SO(3)$. The adjoint of $SU(3)$ decomposes into a five dimensional irrep and the adjoint of $SO(3)$. Thus, the five dimensional fraction of the conserved current can combine with the five dimensional part of the marginal operator to leave behind the adjoint of $SO(3)$ (the conserved current) and the singlet *exactly marginal* deformation. The only 't Hooft anomaly involving the $SO(3)$ symmetry we need to compute is,

$$\text{Tr}\, R\, SO(3)^2 = \left(\frac{2}{3} - 1\right) \times 7 \times 2 = -\frac{14}{3} . \tag{71}$$

Here 2 is the Dynkin index of the adjoint of $SO(3)$ of which we have 7 coming from the fundamentals of $G_2$. All in all we thus have a three dimensional conformal manifold on general locus of which the $G_F = SO(3)$ symmetry is preserved.

We next ask whether we can find a conformal gauge theory $T_2$ with exactly same properties as above. Since $T_1$ itself is conformal gauge theory we know that any such conformal dual has to have

$$\dim \mathcal{G} = \dim G_2 = 14 , \qquad \dim \mathcal{R} = 27 + 3 \times 7 = 48 .$$

The only combinations of dimensions of simple compact Lie groups which give us 14 are the $G_2$ and $SU(3) \times SU(2) \times SU(2)$. We will seek a dual with the latter choice. We need to build a theory with $SU(3) \times SU(2) \times SU(2)$ gauge group and total of 48 chiral fields forming some representation of the gauge group. The matter content should be such that the one loop beta function for the gauge couplings vanishes. One of the finite number of choices accomplishing this is depicted in Figure 20. The fields $X$ and $\widetilde{X}$ are in the $\mathbf{6}$ and $\overline{\mathbf{6}}$ representations of $SU(3)$. We will next argue that it is plausible that the quiver theory of Figure 20 is in fact a conformal dual to the $G_2$ SQCD.

To support our claim we need first to compute the dimension of the conformal manifold and the symmetry on the generic locus of this manifold. At the free locus the quiver theory has $SU(3) \times SU(2)^2 \times U(1)^2$ as its non-anomalous global symmetry group. The non-anomalous $U(1)^2$ can be chosen as follows: We assign charges $(+1, 0)$ to the bifundamentals (the $q$ fields) between the two $SU(2)$'s; the fields $\phi$ and $\widetilde{\phi}$ are assigned charges $(-1, 0)$; and finally the

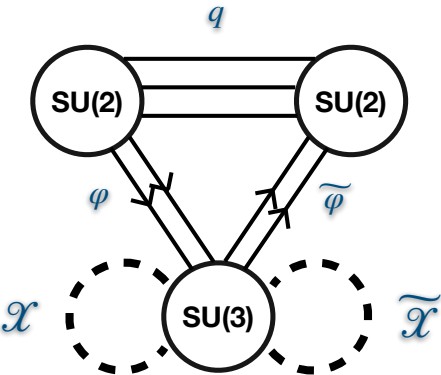

Figure 20: A quiver theory which is conjectured to be conformally dual to the $G_2$ SQCD. The superpotential of this model includes all gauge invariant marginal terms one can construct from the fields.

$X$ and $\widetilde{X}$ fields have charges $(\frac{4}{5}, \pm 1)$ respectively. One can verify that this charge assignment is not anomalous with respect to all three simple gauge group factors. Let us also comment in detail on the anomalous symmetries. We have three such symmetries which we denote by $U(1)_{A_i}$. Under $U(1)_{A_1}$ and $U(1)_{A_2}$ the fields $q$ have charge $+1$ while $(\varphi, \widetilde{\varphi})$ has charge $(+1/-1, -1/+1)$ for the former/latter. Uner $U(1)_{A_3}$ the fields $q$ have charge $-1$ while $\varphi$ and $\widetilde{\varphi}$ have charge $+1$. The fields in the symmetric representation are not charged under these symmetries. Note then that the gauge coupling of the $SU(2)$ on the left of Figure 20 only transforms under $U(1)_{A_1}$. Similarly the gauge coupling of the $SU(2)$ on the right transforms only under $U(1)_{A_2}$, and the $SU(3)$ gauge coupling transforms only under $U(1)_{A_3}$. The gauge couplings are charged positively under the anomalous symmetries.

Having analyzed the symmetries we discuss next the marginal operators. First one, which we will denote by $\lambda$, is built from triangles in the quiver. These marginal operators are in $(\mathbf{3}, \mathbf{2}, \mathbf{2})_{(-1,0)}$ of $(SU(3), SU(2), SU(2))_{U(1)^2}$. The second type of marginal operators, which we will denote by $\lambda_{\pm}$, corresponds to $X^3$ and $\widetilde{X}^3$, which have charges $(1, 1, 1)_{(\frac{12}{5}, \pm 3)}$ respectively. The third and last type of operators, which we will denote by $\lambda'_{\pm}$, are $X\phi^2$ and $\widetilde{X}\widetilde{\phi}^2$ which have charges $(1, 1, 1)_{(-\frac{6}{5}, \pm 1)}$. Under all three anomalous symmetries the $\lambda$ couplings have charge $-1$ and the $\lambda_{\pm}$ couplings are not charged. The $\lambda'_{\pm}$ couplings are charge $-2$ under $U(1)_{A_3}$, $\pm 2$ under $U(1)_{A_1}$ and $\mp 2$ under $U(1)_{A_2}$.

Let us first consider turning on only marginal deformations $\lambda_{\pm}$ and $\lambda'_{\pm}$. These couplings are singlets under the non-abelian symmetries. Under the non-anomalous symmetries the following two independent combinations of the couplings are not charged, $x_+ = (\lambda'_+)^{12}(\lambda_+)(\lambda_-)^5$ and $x_- = (\lambda'_-)^{12}(\lambda_-)(\lambda_+)^5$. Considering the anomalous symmetries the combination $x_+ x_-$ is charged only under $U(1)_{A_3}$ negatively and thus this can be offset by an appropriate power of the exponent of the gauge coupling of the $SU(3)$ gauge group. We thus have a one dimensional conformal manifold on which only $SU(3)$ gauge coupling is non vanishing and $\lambda_{\pm}$ and $\lambda'_{\pm}$ are turned on. These deformations break all the non-anomalous abelian symmetries as well as $U(1)_{A_3}$ and an off-diagonal combinations of $U(1)_{A_2}$ and $U(1)_{A_1}$. Along this direction the non-abelian $SU(3) \times SU(2)^2$ symmetry is preserved.

We can build two additional exactly marginal operator using the deformation $\lambda$. This will break all the symmetry but the diagonal combination of the two $SU(2)$s and $SO(3)$ in $SU(3)$. This comes about again as the decomposition of the representation of the marginal operator is $(\mathbf{3}, \mathbf{2}, \mathbf{2}) \rightarrow 2 \times \mathbf{3} + \mathbf{5} + \mathbf{1}$ and the decomposition of the conserved currents is $(\mathbf{8}, \mathbf{1}, \mathbf{1}) + (\mathbf{1}, \mathbf{3}, \mathbf{1}) + (\mathbf{1}, \mathbf{1}, \mathbf{3}) \rightarrow 3 \times \mathbf{3} + \mathbf{5}$. Two of the three $\mathbf{3}$ and the $\mathbf{5}$ recombine with

the relevant components of the marginal operators leaving behind a single singlet of $SU(2)$. We have broken all the abelian symmetries but the diagonal combination of the anomalous symmetries $U(1)_{A_2}$ and $U(1)_{A_1}$ by turning on $\lambda_\pm$ and $\lambda'_\pm$ and the $SU(3)$ gauge coupling. Under this diagonal combination $\lambda$ has a negative charge which can be offset by appropriate powers of the exponents of one of the two $SU(2)$ gauge couplings giving two additional exactly marginal deformation.

Thus, as we obtained for $T_1$, also $T_2$ has a three dimensional conformal manifold with $G_F = SO(3)$. Note that both duality frames have loci with enhanced symmetry which are however different. This is not a contradiction of the duality as the two do not have to intersect.

Let us next do finer checks of the duality. First we can compute the 't Hooft anomaly involving the $SO(3)$ symmetry on the quiver side,

$$\operatorname{Tr} R\, SO(3)^2 = \left(\frac{2}{3} - 1\right)\left(6 \times 2 \times \frac{1}{2} + 4 \times 2\right) = -\frac{14}{3}. \tag{72}$$

Here $\frac{1}{2}$ is the Dynkin index of the fundamentals and 2 is the index of the adjoint. The adjoint comes from the triplet of $SU(2)$ bi-fundamentals (there are thus four adjoints), while the fundamentals come from the bi-fundamentals of $SU(2)$ and $SU(3)$ (and thus there are $6 \times 2$ of those). Note that this precisely matches (71): The matter is very different but anomaly is exactly the same providing a rather non-trivial check of the proposed duality.

Finally, we can try to match the supersymmetric indices on the two sides of the putative duality. Following the general technology of computing the index detailed in the previous Lecture we have for the $G_2$ SQCD,

$$I_{G_2} = 1 + (h^2 + \frac{1}{h^6}\mathbf{6}_{SU(3)})(pq)^{\frac{2}{3}} + \left(2h^3 + \frac{1}{h^5}\mathbf{6}_{SU(3)} + \frac{1}{h^9} - \mathbf{8}_{SU(3)} - 1\right)pq + \cdots,$$

where we have refined the index with the full non-anomalous symmetry of the free theory. The fugaicty $h$ is the fugacity for the single $U(1)$ symmetry while the bold face numbers encode irreps of $SU(3)$. To compute the index we need the characters of various representations of $G_2$ and for completeness we give them here,

$$\chi_7(\{z\}) = z_1 + z_2 + \frac{1}{z_1 z_2} + \frac{1}{z_1} + \frac{1}{z_2} + z_1 z_2 + 1, \tag{73}$$

$$\chi_{adj.=\mathbf{14}}(\{z\}) = \frac{1}{2}(\chi_7(\{z\})^2 - \chi_7(\{z^2\})) - \chi_7(\{z\}),$$

$$\chi_{\mathbf{27}}(\{z\}) = \frac{1}{2}(\chi_7(\{z\})^2 + \chi_7(\{z^2\})) - 1.$$

On the other hand the index of the putative quiver dual is,

$$I_{quiver} = 1 + (a^{\frac{8}{5}} + a^2 \mathbf{6}_{SU(3)})(pq)^{\frac{2}{3}} \tag{74}$$

$$+ \left(a^{\frac{12}{5}}(b^3 + \frac{1}{b^3}) + \frac{1}{a}\mathbf{2}_{SU(2)_1}\mathbf{2}_{SU(2)_2}\mathbf{3}_{SU(3)} + \frac{1}{a^{\frac{6}{5}}}(b + \frac{1}{b}) - \mathbf{3}_{SU(2)_1} - \mathbf{3}_{SU(2)_2} - \mathbf{8}_{SU(3)} - 2\right)pq + \cdots.$$

We again computed the index at the free locus of the quiver refining it with the full non-anomalous symmetry group. The $a$ and $b$ fugacities parametrize the two non-anomalous $U(1)$s while the bold face numbers denote irreps of the corresponding non-abelian groups. The indices of the two theories at the free locus, (73) and (74), look rather different: This is to be expected as at these points new non-generic symmetries emerge. We should make a comparison only refining the index with symmetries appearing on a generic locus of the conformal manifold. In our case this is the $SO(3)$ symmetry. This implies that we need to set to 1 the

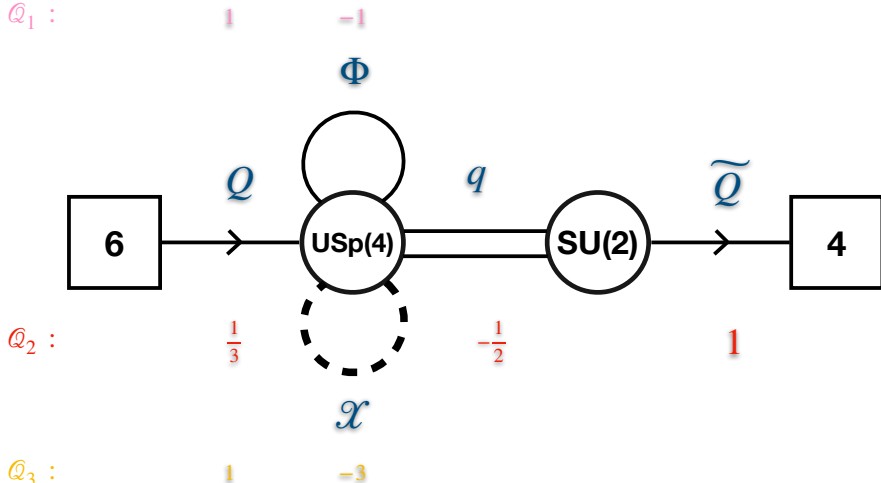

Figure 21: The conformal dual theory, where $X$ is in the traceless antisymmetric representation ($\mathbf{5}$) and $\Phi$ is in the symmetric representation ($\mathbf{10}$) of $USp(4)$.

fugacities for all the abelian symmetries and furthermore take,

$$\mathbf{3}_{SU(3)} \to y^2 + y^{-2} + 1 = \mathbf{3}_{SO(3)}\,, \tag{75}$$

$$\mathbf{2}_{SU(2)_1} = \mathbf{2}_{SU(2)_2} = y + y^{-1} = \mathbf{2}_{SO(3)}\,. \tag{76}$$

With this the two indices match on the nose,

$$I_{G_2} = I_{quiver} = 1 + (2 + \mathbf{5}_{SO(3)})(pq)^{\frac{2}{3}} + (3 - \mathbf{3}_{SO(3)})\,p\,q + \cdots. \tag{77}$$

This equality had to hold if the duality claim is correct.[17] We emphasize that this is not a proof of a duality. One can perform more checks (one of which is detailed in [22]). The two different gauge theories are expected to describe two different weakly coupled cusps of the same conformal manifold.

In the following Lectures we will see additional examples of such conformal dualities. Those examples will have a simple geometrical explanation, though for the duality presented here no such interpretation exists at the moment.

## 3.4 Exercise: Lagrangian dual of a class $\mathcal{S}$ theory

Exercise: Given an SCFT in 4d with $a = \frac{15}{4}$ and $c = \frac{17}{4}$, 26 supersymmetric relevant deformations, and 33 dimensional conformal manifold on a generic locus of which there are no global (non-$R$) symmetries, find a candidate conformal Lagrangian description. (In $A_2$ class $\mathcal{S}$ [7, 8] these are the properties of an $\mathcal{N} = 2$ theory corresponding to a sphere with three maximal and one minimal punctures. See Appendix D.)

The conformal anomalies are such that $\dim \mathcal{R} = 63$ and $\dim \mathcal{G} = 13$. The only choice of gauge group which gives $\dim \mathcal{G} = 13$ is $su(2) \times usp(4)$. Then going over possible matter content such that $\dim \mathcal{R} = 63$ and the one loop gauge beta functions vanish a candidate quiver

---

[17]We do not have a proof of this equality to all order in expansion in parameters but rather this has been checked to some high but finite order. We need to stress here that matching the indices to any finite order in expansion in fugacities does not constitute a proof of matching the indices precisely. There is a plethora of examples, see *e.g.* [80, 81], where the indices of two theories match to arbitrary high order but the theories are different.

theory is depicted in Figure 21. Let us count the supersymmetric relevant deformations. As the theory is free the relevant deformations are the ones given by quadratic gauge singlets,

$$Q^2 : (\mathbf{15}, \mathbf{1}, \mathbf{1})_{2, \frac{2}{3}, 2}, \quad \widetilde{Q}^2 : (\mathbf{1}, \mathbf{1}, \mathbf{6})_{0, 2, 0}, \quad q^2 : (\mathbf{1}, \mathbf{3}, \mathbf{1})_{0, -1, 0},$$
$$\Phi^2 : (\mathbf{1}, \mathbf{1}, \mathbf{1})_{-2, 0, 0}, \quad X^2 : (\mathbf{1}, \mathbf{1}, \mathbf{1})_{0, 0, -6}. \tag{78}$$

Here $(\mathcal{R}_1, \mathcal{R}_2, \mathcal{R}_3)_{q_1, q_2, q_3}$ are representations/charges under the global symmetry of the free theory, $(SU(6), SU(2), SU(4))_{U(1)_1, U(1)_2, U(1)_3}$. Note that the total number of relevant operators is 26 as needed.

Next, we write the most general marginal superpotential,

$$W = \lambda_1 \Phi^2 X + \lambda_2 Q^2 X + \lambda_3 Q^2 \Phi + \lambda_4 q^2 \Phi + \lambda_5 q^2 X + \lambda_6 Q q \widetilde{Q}. \tag{79}$$

We then list the representations under the global symmetry of the operators the couplings $\lambda_i$ couple to (the couplings are in the conjugate representations),

$$\lambda_1 : (\mathbf{1}, \mathbf{1}, \mathbf{1})_{-2, 0, -3}, \quad \lambda_2 : (\mathbf{15}, \mathbf{1}, \mathbf{1})_{2, \frac{2}{3}, -1}, \quad \lambda_3 : (\mathbf{21}, \mathbf{1}, \mathbf{1})_{1, \frac{2}{3}, 2},$$
$$\lambda_4 : (\mathbf{1}, \mathbf{1}, \mathbf{1})_{-1, -1, 0}, \quad \lambda_5 : (\mathbf{1}, \mathbf{3}, \mathbf{1})_{0, -1, -3}, \quad \lambda_6 : (\mathbf{6}, \mathbf{2}, \mathbf{4})_{1, \frac{5}{6}, 1}. \tag{80}$$

We also have two symmetries $U(1)_{SU(2)}$ and $U(1)_{USp(4)}$, which are anomalous under the corresponding gauge symmetries. We define the charges under these as,

$$U(1)_{SU(2)} : \quad [Q] = -\frac{1}{3}, \quad [q] = \frac{1}{2}, \quad [\widetilde{Q}] = 1,$$
$$U(1)_{USp(4)} : \quad [Q] = \frac{1}{3}, \quad [q] = \frac{1}{2}, \quad [\widetilde{Q}] = -1. \tag{81}$$

First, assuming the symmetry is completely broken on the conformal manifold, the dimension is given by the number of marginal operators minus the number of currents,

$$1 + 15 + 21 + 1 + 3 + 6 \times 2 \times 4 - 35 - 3 - 15 - 1 - 1 - 1 = 33, \tag{82}$$

which is the needed dimension of the conformal manifold. Let us now compute the Kähler quotient

$$\{\lambda_i, e^{2\pi i \tau_{SU(2)}}, e^{2\pi i \tau_{USp(4)}}\} / (SU(6) \times SU(2) \times SU(4) \times U(1)^3 \times U(1)_{SU(2)} \times U(1)_{USp(4)})_{\mathbb{C}}. \tag{83}$$

First,

$$\Lambda_0 \equiv \lambda_1 \lambda_4 \lambda_5 (\lambda_3)^3, \tag{84}$$

has charge zero under all the non-anomalous abelian symmetries and under $U(1)_{SU(2)}$. Under $U(1)_{USp(4)}$ it has a negative charge and thus dressing this with appropriate power of $e^{2\pi i \tau_{USp(4)}}$ this charge can be offset to be zero. Moreover $\Lambda_0^2$ is a singlet of all the non-abelian symmetries also. This follows from the fact that $\Lambda_0$ involves only couplings which are singlets of $SU(4)$; the symmetric square of adjoint of $SU(2)$ contains a singlet; the sixth symmetric power of $\mathbf{21}$ of $SU(6)$ contains a singlet. Thus we establish that the Kähler quotient is not empty and the theory is a non trivial SCFT. Along this direction the $SU(2)$ gauge coupling is zero. What is the symmetry preserved by the $USp(4)$ gauge couplings and $\lambda_{1,3,4,5}$? All the non-anomalous abelian symmetries and the $U(1)_{USp(4)}$ are broken as the couplings are charged under all of them. The $SU(4)$ symmetry is preserved. The Cartan of the $SU(2)$ is preserved as the only deformation charged under the $SU(2)$ is a single adjoint. Finally, the $SU(6)$ symmetry is broken

to $SO(6)$ such that $\mathbf{6} \to \mathbf{6}$. In particular the $\mathbf{21}$ decompose to $\mathbf{20'} + \mathbf{1}$ and adjoint to $\mathbf{15} + \mathbf{20'}$ and thus $\mathbf{20'}$ of the marginal operators recombines with the same representation of the conserved currents and we are left with a singlet of $SO(6)$. All in all we have one dimensional conformal manifold the marginal operators are in representations,

$$(\mathbf{1}, \mathbf{1})_0 + (\mathbf{6}, \mathbf{4})_{+1} + (\mathbf{6}, \mathbf{4})_{-1} + (\mathbf{15}, \mathbf{1})_0 \,, \tag{85}$$

and in addition we have $SU(2)$ gauge coupling. Here we write representations/charges in terms of the preserved $(SO(6), SU(4))_{U(1)}$ subgroup of the global symmetry, with the $U(1)$ being the Cartan of the $SU(2)$ at the origin of the conformal manifold.

We have gauge coupling for the $SU(2)$ left as well as the deformations $\lambda_2$ and $\lambda_6$. We can build an invariant under $SU(4)$ taking the baryonic combinations $(\lambda_6^{\pm})^4$ which are in the adjoint of the $SO(6)$. As $\lambda_6$ has opposite charge under anomalous symmetry $U(1)_{SU(2)}$ to the $SU(2)$ gauge coupling we then can build an invariant $(\lambda_6^+)^4 (\lambda_6^-)^4$ (dressed with a proper factor of gauge coupling) under all symmetries. This deformation will break the anomalous symmetry $U(1)_{SU(2)}$, the remaining Cartan of the $SU(2)$ symmetry, as well as the two non abelian symmetries $SU(4)$ and $SO(6)$. The couplings $\lambda_2$ will be exactly marginal. We thus are left with conformal manifold of dimension 33 with all the symmetry broken on a generic locus. This completes the solution of the exercise.

$\square$

The algorithm detailed here for the search of conformal dualities can be generalized to search for non-conformal dualities involving flows, as long as we can assume that the spectrum of R-charges is constraints. For examples of such generalizations see [82, 83] where a variant of the algorithm was applied to find IR duals of an $E_6$ Minahan-Nemeschansky model [84] and of some $\mathcal{N} = 3$ theories.

# 4 Lecture III: Across dimensional dualities, an example

In the previous Lecture we have discussed several scattered observations about IR physics of simple gauge theories. The question we want to ask now is whether this scattered plethora of facts has any interesting organizing principle. We will show that such an underpinning can be found by thinking about 4d QFTs geometrically. The logic is as follows.

One can consider discussing higher dimensional SCFTs. Following the notorious theorem of Nahm [85] (see also [86]) the maximal number of dimensions in which an interacting superconformal theory can reside is six dimensions. In higher dimensions a superconformal algebra (with no higher spin currents) just does not mathematically exist. There are two types of superconformal algebras in 6d, $(1,0)$ and $(2,0)$, differing by the amount of supersymmetry: The former one having 8 real supercharges and the latter 16 real supercharges. The $(2,0)$ SCFTs are conjectured to be classified by an ADE algebra. $A$ type $(2,0)$ theories can be engineered for example as the low energy effective theory residing on M5 branes in M-theory constructions. On the other hand there is a quite huge plethora of $(1,0)$ SCFTs. Here also there are various classification approaches [87, 88] (See [10] for a recent review.). One important fact about the 6d SCFTs is that all of these are strongly coupled. In particular, due to dimensional analysis the gauge couplings and all superpotential interactions in 6d are irrelevant; thus, gauge theories flow to free theories in 6d. Phrasing this more abstractly, 6d SCFTs do not possess any Lorentz preserving supersymmetric relevant or exactly marginal deformations [20].

As we are interested here in SCFTs in 4d we can then start from some given 6d $(1,0)$ theory (with the $(2,0)$ being just a more supersymmetric special case) and place it on a compact two dimensional space. Such compact spaces are called Riemann surfaces and are classified

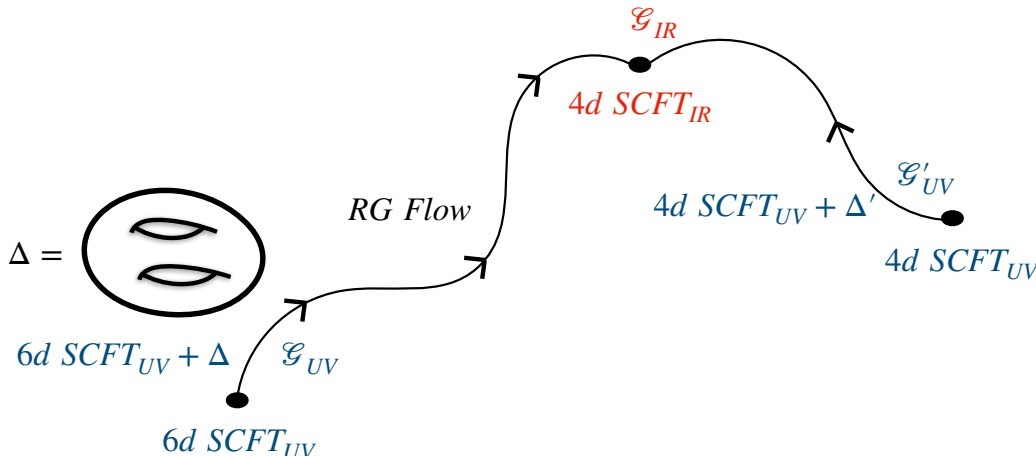

Figure 22: A 6d CFT can be placed on a compact geometry. At energy scales far below the scale set by the geometry the theory should be effectively described by a four dimensional model, which would then flow to some fixed point in the IR. The geometry can be thought of as a deformation $\Delta$ of the 6d fixed point and triggers an RG flow across dimensions. The resulting IR SCFT in 4d might or might not be a fixed point of an RG flow starting from a Gaussian fixed point in 4d deformed by a relevant deformation. If such a theory exists we have an IR duality between deformations of a 6d theory and a 4d one.

by their genus $g$. We will also allow to decorate the surfaces with marked points (which will have certain physical meaning in terms of either defects or boundary conditions). Although such a space is curved for $g \neq 1$ and thus breaks supersymmetry, it is possible to perform the compactification with a certain twist, which we will soon review, such that half of the supersymmetry is preserved. Half of eight supercharges of $(1,0)$ will lead to four supercharges, see Appendix F. One way to think about the geometry is as a certain deformation of the 6d SCFT. In the IR, far below the energy scale set by the geometry, the theory becomes effectively four dimensional. Preserving four supercharges we will obtain an $\mathcal{N} = 1$ theory in four dimensions. This effective theory might be a strongly coupled SCFT, a theory of free chiral fields, a weakly coupled gauge theory, or a combination of these options.

An interesting question then is whether we can find a description of this effective theory directly in four dimensions. That is whether there is a four dimensional Lagrangian which either flows to this theory or directly describes the conformal fixed point. Such a geometric engineering of the four dimensional models produces a huge plethora of theories. Some of these theories have properties such that no four dimensional model is known to produce directly, and thus sometimes these are called non-Lagrangian. *If a Lagrangian description is found one can think of the setup then as a novel type of an IR duality between a 6d theory deformed by geometry and a 4d Lagrangian theory*, see Figure 22. One way to think about such a set of dualities is as map, a dictionary, between the set of 6d SCFTs and two dimensional geometries into the set of four dimensional supersymmetric quantum field theories,

$$T^{4d}\left[T^{6d}, \mathcal{C}\right], \tag{86}$$

where $\mathcal{C}$ collectively labels the geometry and $T^{6d}$ the choice of the 6d starting point. The whole program can be of course imbedded in the larger structure of M-theory and there is a lot of benefit to be extracted from this, but we will concentrate on working within the paradigm of local quantum field theory. Our goal in the following two Lectures will be to derive some

entries in this dictionary and exemplify the utility of the procedure for understanding non trivial physics in four dimensions.

The fact that there is a geometry behind the construction leads to numerous very powerful understandings about the four dimensional physics: *e.g,* many of the dualities and emergence of symmetry phenomena of the type we have discussed can be explained, and in fact predicted, from such constructions. The prototypical example of the geometric constructions is compactifications of $(2,0)$ theories leading to $\mathcal{N} = 2$ theories in 4d. A lot has been understood about such models following the seminal work of Gaiotto [8]. Here, however we will concentrate on some simple $(1,0)$ examples. Let us first consider the compactification of one of the simplest interacting 6d SCFTs, the so called $SU(3)$ minimal SCFT [89, 90] (or $SU(3)$ non-higgsable cluster [91]). We will mainly follow the discussion in [24].

## 4.1 Six dimensions

One way to construct an SCFT in six dimensions is as follows. We consider a gauge theory, which as we mentioned is necessarily IR free. Without going into many details (see Appendix E and the review [10] for more details), a generic such theory can be built from three types of $(1,0)$ multiplets: A vector multiplet, a tensor multiplet, and a hyper-multiplet. We will not need the hypermultiplets for our discussion but let us mention that the vector multiplet contains of course a gauge field and a fermion: In terms of representations of the little group $SU(2) \times SU(2)$ these massless states are in $(\frac{1}{2}, \frac{1}{2})$ and $(0, \frac{1}{2})$, while the tensor multiplet contains a tensor field (in irrep $(1,0)$ which is a two-form with self-dual field strength), a fermion (in irrep $(\frac{1}{2}, 0)$), and importantly a scalar (in irrep $(0,0)$ of course). The scalar in the tensor multiplet (See Appendix E for discussion of the multiplets.) can be naturally coupled to the field strength (schematically),

$$\int d^6x \, \phi \, F \wedge \star F \,, \tag{87}$$

and thus the gauge coupling in six dimensions can be thought of as an expectation value of the scalar field residing in the tensor multiplet, $\langle \phi \rangle \sim 1/g_{YM}^2$. In particular one can ask what happens if this vev is set to zero. In certain cases it is believed that the theory one obtains is a strongly coupled CFT. Conversely, a strongly coupled SCFT might contain a moduli space of vacua, called the tensor branch, on which certain scalar operators acquire a vev and the effective description in the IR is in terms of an IR free gauge theory consisting of vector multiplets, tensors, and maybe hypermultiplets. This is very reminiscent of the $\mathcal{N} = 2$ dynamics in four dimensions. There an $\mathcal{N} = 2$ vector multiplet contains a scalar (which in terms of $\mathcal{N} = 1$ language corresponds to an adjoint scalar in an $\mathcal{N} = 1$ chiral superfield in the $\mathcal{N} = 2$ vector superfield), and giving a vev to this scalar we move on the Coulomb branch of the theory on which typically the description is in terms of abelian, IR free, gauge theories. Switching off the vev would take us back to the SCFT point.

The effective gauge theory description on the tensor branch is constrained to be non-anomalous. As the vector multiplet contains fermions, for example, just taking $(1,0)$ vectors usually leads an anomalous theory. However, the theory with a vector and a tensor in some cases can be made to be non anomalous. This comes about as naturally due to supersymmetry the term of the form (87) is accompanied by,

$$\alpha \int d^6x B \wedge \operatorname{Tr} F \wedge F \,, \tag{88}$$

with $B$ being the tensor (see Appendix E) and $\alpha$ being some real constant. Such a term actually contributes to the eight-form anomaly polynomial of the 6d theory due to the Green-Schwarz

mechanism [92, 93] a term of the form (see Appendix E for more details)

$$\alpha^2 \left(\operatorname{Tr} F \wedge F\right)^2 , \tag{89}$$

and this can be used to cancel the term of a similar structure coming from the vector multiplet.

Now let us then consider a gauge theory in 6d with a simple gauge group $G$ and a tensor multiplet. The contribution to the eight form anomaly polynomial with four field strengths can come in two different forms,

$$\operatorname{Tr} F \wedge F \wedge F \wedge F \qquad \text{and} \qquad \left(\operatorname{Tr} F \wedge F\right)^2 , \tag{90}$$

with the difference being different contractions of the indices. Specifically, the second term contracts the indices using the Cartan form squared, while the first terms uses the quartic Casimir, which is a completely symmetric invariant polynomial of order four. For most groups, the two terms are independent, but there are some groups that do not possess an independent quartic Casimir, in which case the two terms become equivalent and there is only one type of gauge anomalies. The groups for which this happens are: $SU(2)$, $SU(3)$, $G_2$, $F_4$, $E_6$, $E_7$ and $E_8$.

For a theory to be non-anomalous both types of anomalies must vanish. The second type of structure can be canceled by the introduction of a Green-Schwarz term of the form (89), provided the contribution of the vector multiplet to the anomaly polynomial comes with a negative coefficient, as in fact it does. However, the first type of structure can only be canceled by matter contributions, specifically, by introducing charged hypermultiplets. As such, if we insist on a theory containing only tensor and vector multiplets, we must limit ourselves to gauge groups that either: A) don't have an independent quartic Casimir or b) the contribution of an adjoint fermion to this anomaly is zero. We have already listed the options realizing a) and it turns out that there is a single option realizing b), $SO(8)$.[18] Additionally we also have to worry about the existence of a Witten anomaly [67] related to $\pi_6(G)$ being non-trivial, which is the case for $SU(2)$, $SU(3)$ and $G_2$. This anomaly leads to the pure $SU(2)$ and $G_2$ theories being inconsistent, though miraculously, an adjoint fermion of $SU(3)$ turns out to contribute trivially to the anomaly so a pure $SU(3)$ gauge theory is consistent [90]. Overall, the pure gauge theories with a single tensor that do not suffer from anomalies and thus can be consistent tensor branch descriptions of some SCFT, are: $SU(3)$, $SO(8)$, $F_4$, $E_6$, $E_7$ and $E_8$ [89, 90]. These theories are sometimes called minimal SCFTs in 6d.

We expect that the resulting SCFTs do not have continuous flavor symmetries as on the tensor branch we do not see any. We can evaluate the 't Hooft anomalies of these SCFTs, using the gauge theory description, and collect the results in an anomaly polynomial 8-form.

The 't Hooft anomalies receive contributions from three sources: The chiral fermion in the vector multiplet, the self-dual tensor and chiral fermion in the tensor multiplet, and the Green-Schwarz term required to cancel the $\operatorname{Tr}(F \wedge F)^2$ terms (including the mixed gauge-gravity anomalies). Let us quote the contributions of the vector multiplet for gauge group $G$ [94],

$$-\frac{Tr(F^4)_{Adj}}{24} - \frac{C_2(R)Tr(F^2)_{Adj}}{4} - \frac{d_G C_2^2(R)}{24} - \frac{d_G C_2(R)p_1(T)}{48}$$
$$-\frac{Tr(F^2)_{Adj}p_1(T)}{48} - d_G \frac{(7p_1^2(T) - 4p_2(T))}{5760} . \tag{91}$$

Here we use $C_2(R)$ for the second Chern class of the $SU(2)$ R-symmetry bundle in the doublet representation, and $p_1(T), p_2(T)$ for the first and second Pontryagin classes of the tangent

---

[18]$SO(8)$ has the unique property of having two different independent quartic Casimirs. As such in this case there are actually three different anomaly structures. The existence of the two structures and the reason why the contribution to both for an adjoint fermion vanishes is related to the special triality automorphism of $SO(8)$, see the discussion in [24].

bundle respectively. The constant $d_G$ stands for the dimension of the group $G$. Since we do not have a quartic Casimir the $Tr(F^4)_{Adj}$ can be expressed as,

$$Tr(F^4)_{Adj} = \lambda_G \, Tr(F^2)_{Adj}^2 \,. \tag{92}$$

We will be interested here only in the group $G = SU(3)$ for which $d_G = 8$ and $\lambda_G = \frac{1}{4}$. The tensor multiplet contributes,

$$\frac{C_2^2(R)}{24} + \frac{C_2(R)p_1(T)}{48} + \frac{(23p_1^2(T) - 116p_2(T))}{5760} \,, \tag{93}$$

and to cancel all gauge anomalies we need to introduce the Green-Schwarz term which contributes,

$$\frac{\lambda_G}{24} \left( Tr(F^2)_{Adj} + \frac{3}{\lambda_G} C_2(R) + \frac{1}{4\lambda_G} p_1(T) \right)^2 \,. \tag{94}$$

Summing up all the terms, we find that the eight-form anomaly polynomial is,

$$\begin{aligned} I^{6d} = & \frac{1}{24}(\frac{9}{\lambda_G} - d_G + 1)C_2^2(R) + \frac{1}{48}(\frac{3}{\lambda_G} - d_G + 1)C_2(R)p_1(T) \\ & + \frac{(\frac{15}{\lambda_G} - 7d_G + 23)p_1^2(T) + (4d_G - 116)p_2(T)}{5760} \,. \end{aligned} \tag{95}$$

## 4.2 Compactification to 4d

Given the 6d SCFT which on the tensor branch is described by the $SU(3)$ gauge theory, we want to understand what happens when we compactify it on a closed Riemann surface. The compactifications on lower genus surfaces, $g = 0$ and $g = 1$, do not follow the general pattern we will want to discuss and thus we will refrain from discussing these here. Compactifying on $g > 1$ surfaces we will be able to derive some very general statements.

As the Riemann surface with $g > 1$ is curved, supersymmetry is naively broken. To avoid this, we twist the $SU(2)$ R-symmetry bundle so as to cancel the curvature of the Riemann surface for some of the supercharges which are charged under it. The supercharges transform under the Lorentz group and under the R-symmetry so we need to turn on a certain R-symmetry bundle so that at least some supercharges will not feel the curved background and thus will remain invariant. We can do so preserving at most 4 supercharges, corresponding to $\mathcal{N} = 1$ in 4d. The twist, the non-trivial bundle for the R-symmetry, also breaks the $SU(2)$ R-symmetry to its $U(1)$ Cartan sub-group, which becomes an R-symmetry in 4d. See Appendix F for more details on the twisting procedure.

Next we will want to deduce the various 't Hooft anomalies of the 4d theories. We have an RG flow across dimensions and thus we need to deduce the six-form anomaly polynomial of the 4d theory from the eight-form anomaly polynomial of the 6d theory. The way to do so is to integrate the anomaly polynomial of the six dimensional SCFT on the compact Riemann surface in the presence of all the non-trivial bundles we have turned on [38] (see also Subsection 3.1 of [95]),

$$\mathcal{I}^{4d} = \int_{\mathcal{C}} \mathcal{I}^{6d}[\mathcal{F}] \,, \tag{96}$$

where by $\mathcal{F}$ we collectively denote the values of the background fields. In the current case the only such fields are due to the non-trivial bundle for the R-symmetry due to twisting, but in more general cases one also might have non trivial bundles for the global symmetries. This

should lead to the anomaly polynomial of the 4d theory, which contains the 't Hooft anomalies of the 4d theory, at least those for symmetries descending from 6d.

In our case, we need to integrate (95) on the Riemann surface, but first we need to take the twist into account. This is done by setting (see Appendix F) $C_2(R) = -C_1(R)^2 + 2(1-g)\hat{t} C_1(R) + \ldots$, where $C_1(R)$ is the first Chern class of the $U(1)$ Cartan of the $SU(2)$ R-symmetry and $\hat{t}$ is a unit-flux 2-form on the Riemann surface. Inserting this into (95) and integrating we find,

$$I^{4d} = \frac{1}{6}\left(\frac{9}{\lambda_G} - d_G + 1\right)(g-1)C_1^3(R) - \frac{1}{24}\left(\frac{3}{\lambda_G} - d_G + 1\right)(g-1)C_1(R)p_1(T). \quad (97)$$

From the coefficient of $C_1^3(R)$ we deduce the $\operatorname{Tr} R^3$ anomaly and from the $\frac{1}{24}C_1(R)p_1(T)$ term we deduce the $\operatorname{Tr} R$ anomaly,

$$\operatorname{Tr} R^3 = \left(\frac{9}{\lambda_G} - d_G + 1\right)(g-1), \qquad \operatorname{Tr} R = \left(\frac{3}{\lambda_G} - d_G + 1\right)(g-1). \quad (98)$$

Finally using the relation between 't Hooft anomalies of the R-symmetry and the conformal anomalies (10) we deduce that these are,

$$a = \frac{3}{16}\left(\frac{12}{\lambda_G} - d_G + 1\right)(g-1), \quad c = \frac{1}{8}\left(\frac{33}{2\lambda_G} - d_G + 1\right)(g-1). \quad (99)$$

In particular for the $SU(3)$ case at hand we get,

$$a = \frac{123}{16}(g-1), \qquad c = \frac{59}{8}(g-1). \quad (100)$$

We thus have a prediction for the existence of 4d theories labeled by $g$ with the above conformal anomalies. These models are also expected not to have any global symmetries. Of course global symmetries might emerge in principle in the IR, invalidating both of these statements. We will assume that this does not happen and will seek the corresponding four dimensional theories.

Let us here perform a quick computation: We assume as in the previous Lecture that these 4d theories have a conformal gauge theory description. This might or might not be true, and we will eventually argue that it is true. With this assumption we now deduce what would be the dimension of the gauge group in 4d and the dimension of the representation of the matter fields. We need to solve,

$$a_v \dim \mathfrak{G} + a_\chi \dim \mathfrak{R} = \frac{123}{16}(g-1), \qquad c_v \dim \mathfrak{G} + c_\chi \dim \mathfrak{R} = \frac{59}{8}(g-1), \quad (101)$$

to get $\dim \mathfrak{G} = 32(g-1)$ and $\dim \mathfrak{R} = 81(g-1)$. We note that these numbers are integer and thus the conjecture might be correct.[19] Moreover, a natural interpretation of 32 is as four factors of $SU(3)$ and of 81 as three tri-fundamentals of $SU(3)$. We will soon see how to construct such a gauge theory. However, first it will be worthwhile to understand what one would expect from compactifications on punctured Riemann surfaces, and that entails going through an intermediate step between 6d and 4d, a reduction to 5d.

---

[19]If one repeats the same exercise for other groups for some the dimensions turn out to be integer while for others they do not. For example for $F_4$ $(\lambda_G, d_G) = (\frac{5}{108}, 52)$ and $(\dim \mathfrak{G}, \dim \mathfrak{R}) = (\frac{798}{5}, \frac{2187}{5})(g-1)$ but for $E_6$ $(\lambda_G, d_G) = (\frac{1}{32}, 78)$ and $(\dim \mathfrak{G}, \dim \mathfrak{R}) = (235, 648)(g-1)$.

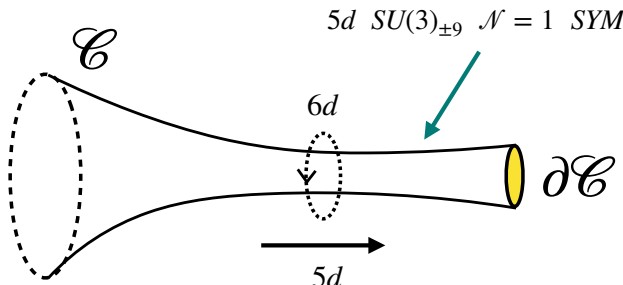

Figure 23: Compactifying the 6d $SU(3)$ minimal SCFT on a circle with the complex conjugation twist is conjectured to be described in 5d by an effective theory which is $SU(3)$ SYM with level ±9 CS term. The UV completion of the 5d theory is the 6d SCFT. Here $\mathcal{C}$ is the Riemann surface and $\partial\mathcal{C}$ is non empty, that is we have a puncture. Each puncture is associated with a global symmetry corresponding to the 5d gauge symmetry, which is $SU(3)$ in this case.

## 4.3 Reduction to 5d and punctures

Let us discuss how one can think about punctured Riemann surfaces. As the 6d theory at the CFT point, which we compactify, is strongly coupled it is hard to analyze the punctures directly in 6d. However, one way to think about the punctures is to elongate the region near a puncture to a long thin cylinder and first analyze this region. This amounts to compactifying first on an infinite cylinder of small radius and obtaining an effective 5d theory, and then cutting the cylinder with a specification of a boundary conditions in 5d. If the 5d theory is still strongly coupled we did not achieve much. However, in certain cases, starting with certain 6d SCFTs and compactifying on a circle to 5d with certain holonomies/twists, it is conjectured that one obtains an effective theory in 5d which is a gauge theory.[20] As in 6d these gauge theories are IR free and there are two possibilities for their UV completion: First, it is possible that they are relevant deformations of non trivial CFTs in 5d, and second, the UV completion might be given by a 6d SCFT. A canonical example is the ADE $(2, 0)$ theory which is conjectured to give the $\mathcal{N} = 2$ ADE SYM when compactified to 5d with no twists [108, 109].

If one indeed obtains a gauge theory in 5d with gauge group $G^{5d}$, one next can study supersymmetry preserving boundary conditions at the four dimensional boundary of the geometry. The details of the choices of the boundary conditions depend on the 5d theory, however there is a canonical set of choices, which is usually called *maximal*, giving a Dirichlet boundary condition to all the vector fields of the 5d theory and then whatever supersymmetry demands for the other fields. See Appendix G for more details, as well as [26, 110] (and recent similar discussion in 3d in [111]). Importantly, since the Dirichlet boundary condition breaks the gauge symmetry to the one which is constant along the boundary, we acquire a global symmetry which is given by the gauge group $G^{5d}$ associated with each boundary component, that is with each puncture.

Let us specialize now this general discussion to the case at hand. It is conjectured [112] that a pure $\mathcal{N} = 1$ $SU(3)$ gauge theory with CS term at level ±9 is obtained by compactifying the minimal $SU(3)$ SCFT in 6d on a circle with a twist by the complex conjugation symmetry in 6d, see Figure 23.[21] Twisting here means that upon compactification of the circle (defined

---

[20]There is an ongoing vigorous research into classifying such effects, see *e.g.* [16, 17, 96–107] for a partial list of references.

[21]The fact that the theories in 6d and in 5d are associated to the same group $SU(3)$ is not a generic feature. For example minimal $SO(8)$ SCFT upon compactification with certain twist reduces to an $SU(4)_{\pm 8}$ gauge theory in 5d [24].

by angle $\theta \in [0, 2\pi]$) we identify the configurations at $\theta = 0$ and $\theta = 2\pi$ with an action of the discrete symmetry. Note that the 6d theory does not have any continuous global symmetries and thus barring accidental appearance of symmetry in 5d, upon compactification we do not expect to obtain any symmetry beyond the one associated to the KK symmetry of the circle: The latter symmetry is identified with the topological (instantonic) symmetry of the 5d gauge theory. Thus we expect a pure gauge theory in 5d. Matching the moduli spaces of the 6d and 5d theories we obtain that if there is a gauge theory in 5d it should be $SU(3)$: In 6d the moduli space is three dimensional but the twist by complex conjugation reduces it to two (projecting out the dimension three Coulomb branch operator of the $SU(3)$), while in 5d $SU(3)$ has a two dimensional Coulomb branch. Finally, we need to fix the level of the $SU(3)$ gauge group; following various string theoretic arguments [112], for levels smaller than $\pm 9$ it is believed that the theory is a deformation of a 5d SCFT, for level $\pm 9$ it is UV completed by the 6d SCFT, and for higher level the theory might not have a UV completion.[22]

Finally, we need to specify the maximal boundary conditions for the theory at hand. As we are preserving $\mathcal{N} = 1$ 4d supersymmetry on the boundary, we decompose the 5d fields on the boundary in terms of the 4d multiplets: Which are $\mathcal{N} = 2$ 4d vector fields as the number of supersymmetries in 5d is eight. We choose Dirichlet for the vector fields and Neumann for the adjoint chiral (see Appendix G). This will imply that we have Dirichlet boundary conditions for chiral fermions and Neumann for anti-chiral. From this we can compute the anomaly inflow contribution of the puncture to four dimensions. This comes from several sources: The anti-chiral fermions with Neumann boundary conditions and the CS term. The former are in the adjoint of the 5d gauge group $SU(3)$, have R-charge $-1$, and contribute half of the contribution of four dimensional fermions.[23] This gives the anomaly contributions

$$\operatorname{Tr} R = -1 \times \frac{1}{2} \times \dim SU(3) = -4, \qquad \operatorname{Tr} R^3 = (-1)^3 \times \frac{1}{2} \times \dim SU(3) = -4,$$

$$\text{and} \qquad \operatorname{Tr} R \, SU(3)^2 = -1 \times 3 \times \frac{1}{2} = -\frac{3}{2},$$

coming from every puncture. To compute the full anomaly polynomial we also need to take into account the computation of the integral of the 6d anomaly polynomial on the Riemann surface which is the same as above but with $g - 1 \to g - 1 + s/2$ where $s$ is the number of punctures, and we specialize to the case of the 6d $SU(3)$ SCFT. These give for surfaces with punctures the anomalies,

$$\begin{aligned} a &= \frac{123}{16}\left(g - 1 + \frac{s}{2}\right) + \left(\frac{9}{32}(-4) - \frac{3}{32}(-4)\right)s = \frac{3}{32}(82(g-1) + 33s), \\ c &= \frac{59}{8}\left(g - 1 + \frac{s}{2}\right) + \left(\frac{9}{32}(-4) - \frac{5}{32}(-4)\right)s = \frac{1}{16}(118(g-1) + 51s). \end{aligned} \tag{102}$$

In addition the CS term contributes $\operatorname{Tr} SU(3)^3 = \pm 9$, where the sign is determined by the sign of the CS term.

Next we consider how we should glue two punctures in 4d. Gluing punctures geometrically corresponds to bringing two cylinders with boundaries together and un-doing the boundary conditions, identifying in some way the fields in the two cylinders. Field theoretically thus we need to gauge in 4d the symmetry associated to the punctures, $SU(3)$ in our case. Note that the R-symmetry is non anomalous as far as this gauging is concerned as

$$\operatorname{Tr} R \, SU(3)^2 = 3 + \left(-\frac{3}{2}\right) + \left(-\frac{3}{2}\right) = 0, \tag{103}$$

---

[22]The consistency of our analysis below can be viewed as additional evidence in favor of the conjecture that level nine theory is UV completed in 6d.

[23]A way to see that the half is needed is that if we compactify a 5d fermion on a finite interval with Neumann boundary condition on both ends, we will get a fermion in 4d. Thus, each boundary will contribute half of the anomaly.

where the first term comes from the vector fields and the second and third come from the two punctures. Moreover, as the gauging should be non-anomalous, $\mathrm{Tr}\, SU(3)^3 = 0$, we should glue a puncture with $+9$ CS term to the one with $-9$ CS term. Note that the anomalies (102) are of course self-consistent under this gluing: Adding anomalies with $(g_1, s_1)$ and $(g_2, s_2)$, and a vector multiplet anomalies gives anomalies with $(g_1 + g_2, s_1 + s_2 - 2)$. Conversely, we could have derived the anomalies of the surface with a pair of punctures and genus $g$ starting with surface without them and genus $g + 1$ and subtracting the contribution of the vector multiplet of $SU(3)$.

Note that we have already discussed that the anomalies on a closed surface might be consistent with free fields, and in particular $SU(3)$ gauge groups and tri-fundamentals. The anomalies of punctures support this observation. The $\mathrm{Tr}\, SU(3)^3 = \pm 9$ anomaly is consistent with having 9 (anti)fundamentals of $SU(3)$, which is the number of free flavors needed for a conformal gauging of $SU(3)$. Also $\mathrm{Tr}\, R\, SU(3)^2 = -\frac{3}{2} = \frac{1}{2} \times (\frac{2}{3} - 1) \times 9$, and thus interpretable as an anomaly of 9 (anti)fundamental free fields. We will momentarily turn to constructing in detail the 4d theories, but let us comment here on the implications of the twist we have performed upon compactification to 5d.

The twists in general can be turned on along any cycle on the surfaces. These particularly include those surrounding the punctures. For the special case of a sphere with $s$ punctures, these are the only cycles. Furthermore, the twists around different punctures and cycles are not independent: On a sphere the holonomy around $s - 1$ punctures should be the inverse of that around the remaining puncture. In our case, since the punctures we consider must incorporate a twist, this leads to constraints on the possible theories. As the punctures have a $\mathbb{Z}_2$ (complex conjugation) twist, we cannot have a sphere with odd number of such punctures. In principle there might be punctures without a twist associated to them, however as we defined punctures above using a 5d gauge theory description the untwisted punctures have to be of a different nature. We will say more about this when discussing the 4d theories. On Riemann surfaces without punctures, the twists can still be incorporated on the cycles of the Riemann surface. When the latter is built from punctured spheres then whenever we glue two twisted punctures we get the associated twist along the cycle spanned by the punctures.

## 4.4 Four dimensions

We want to find four dimensional models which will match the predictions coming from six dimensions we discussed above. In particular, these models should have factors of $SU(3)$ global symmetry, which we will associate to punctures, and the anomalies of these symmetries are $\mathrm{Tr}\, SU(3)^3 = \pm 9$, $\mathrm{Tr}\, R\, SU(3)^2 = -\frac{3}{2}$. As we already mentioned we will conjecture that we can build such theories from free fields and conformal gaugings. We have discovered that constructing field theories such that they have nine free chiral fields in the fundamental of $SU(3)$ produces the right anomalies. The question is then how we build models from these collections of fields. We will next discuss some conjectures which reproduce the expectations we have derived from six dimensions.

The main conjecture is depicted in Figure 24: We associate to the four punctured sphere a quiver theory with two $SU(3)$ gauge groups, three tri-fundamental fields, and a baryonic superpotential. We have a single superpotential term which is charged under the baryonic symmetry: Thus following [40] (which we reviewed in previous sections) this interaction is marginally irrelevant if we perturb the free theory with it. We can turn on additional baryonic superpertentials rendering the theory conformal, but these will break some of the $SU(3)$ symmetries associated to the punctures. As we have discussed in Example 1 (discussion around (35)), $SU(3)$ SQCD with nine flavors has a locus on its conformal manifold preserving $(SU(3)_1 \times (U(1)^2)_2) \times SU(3)_3$ symmetry where the $(U(1)^2)_2$ is the Cartan of an $SU(3)_2$

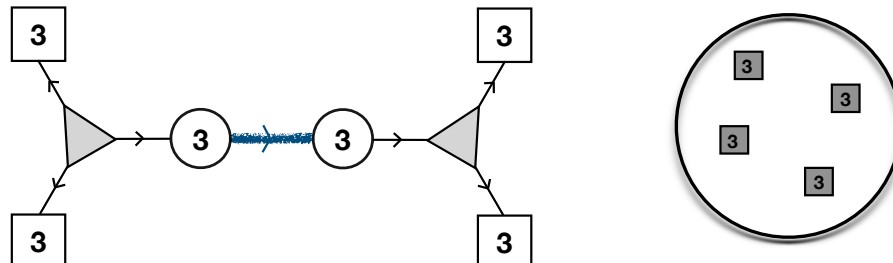

Figure 24: The four punctured sphere and its field theoretic description. The three bifundamental fields in the middle link of the quiver have a baryonic superpotential (which is encoded in the quiver by the fact the corresponding line is wavy). The various quiver notations are further explained in Figure 25. We have four factors of $SU(3)$ global symmetry we associate to the four punctures.

such that the **9** of $SU(9)$ flavor symmetry is $(\mathbf{3}, \mathbf{3})$ under the two $SU(3)_1 \times SU(3)_2$ symmetries. Our four punctured sphere thus has directions on the conformal manifold along which we can preserve two $SU(3)$ symmetries explicitly and the two other ones are broken to the Cartan. The baryonic symmetry is also broken. We then conjecture that somewhere on the conformal manifold the Cartan symmetry enhances again to $SU(3)$s but the baryonic symmetry does not appear. We will soon give evidence for this conjecture.

Once we defined the four-punctured sphere we can construct theories corresponding to arbitrary surfaces by gauging puncture symmetries, as we have discussed. For example in Figure 26 we depict theories corresponding to genus two surface with no punctures. As we have an explicit field theoretic description of the models one can verify explicitly that the conformal anomalies derived from six dimensions (102) match the anomalies of the four dimensional models. Note that as we conjectured above, the four punctured sphere has all the $SU(3)$ puncture symmetries only at some strongly coupled locus of the conformal manifold, to construct general theories we thus need to gauge *emergent symmetries*. We also stress that as we constructed a four punctured sphere but not a three punctured one, we only can construct surfaces with even number of punctures.[24]

As a further check of the conjecture we can compute the supersymmetric index and determine from it the supersymmetric relevant and marginal operators. For generic surfaces the index takes the following general form,[25]

$$\mathcal{I}_{g,s}(\{\mathbf{a}_i\}) = 1 + \left( (3g - 3 + s) + \sum_{j=1}^{s} (-\chi_{\mathbf{8}_j}(a_j) + \chi_{\mathbf{10}_j}(a_j)) \right) q\,p + \cdots . \tag{104}$$

Here $\mathbf{a}_i$ are fugacities for the $SU(3)$ symmetries associated to the punctures. There are no powers of $q\,p$ smaller than one and thus there are no supersymmetric relevant operators. At order $q\,p$ we have the marginal deformations minus the conserved currents. We have $3g - 3 + s$ terms with positive sign which are singlets of the flavor group associated to the punctures and thus these correspond to exactly marginal deformations: We expect such deformations to correspond to the complex structure moduli of the compactification surface. One way to think about these operators is to consider the reduction on a surface of the 6d stress-energy tensor. A careful application of the Riemann-Roch theorem then determines that the stress-energy tensor will lead to $3g - 3$ exactly marginal deformations on a closed Riemann surface [113], and the above is a natural generalization including the punctures. We have $-\chi_{\mathbf{8}_j}(a_j)$ factors

---

[24]See [24] for further discussion of this issue and its relation to the fact that in order to define punctures we have compactified the 6d SCFT to 5d with twist lines.

[25]For low numbers of punctures and low genus there might be deviations from this structure.

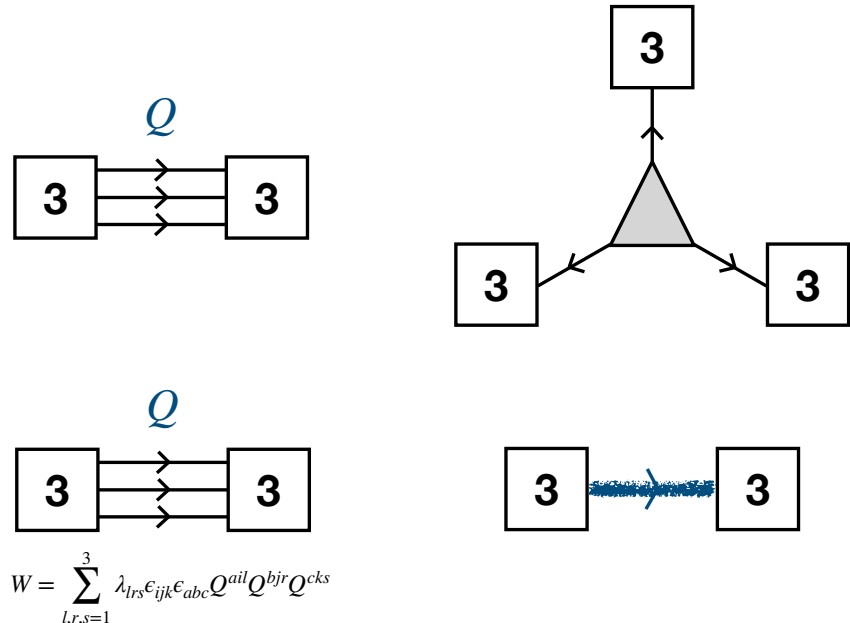

$$W = \sum_{l,r,s=1}^{3} \lambda_{lrs} \epsilon_{ijk} \epsilon_{abc} Q^{ail} Q^{bjr} Q^{cks}$$

Figure 25: Two types of tri-fundamental chiral fields we use in our constructions. First we have the tri-fundamental with no superpotential, and then trifundamental with one of the three $SU(3)$ symmetries broken by a baryonic superpotential.

corresponding to the conserved currents of each puncture. Finally, for each puncture we have marginal operators in the ten dimensional (symmetric three index) representation of $SU(3)$ puncture symmetry. Turning these on we can break completely the puncture symmetry and acquire two additional exactly marginal directions. The spectrum of protected states that our four dimensional theories possess is thus also consistent with the expectations from 6d.

We have discussed what is the theory corresponding to a four punctured sphere and how to glue two theories corresponding to two surfaces together along a puncture. The resulting theories are labeled by the combined surface. However, one can combine the same surface by performing gluings in different ways. For example, there are two different ways to glue the punctures of a four-punctured sphere together to obtain a genus two surface, see Figure 26.

*The fact that we conjecture that all the different ways to glue surfaces so that the topology will be the same correspond to compactifications on the same surface implies that the different ways to glue should be equivalent. More precisely these should reside on the same conformal manifold. In other words, the different quantum field theories corresponding to the different ways to construct the same surface are expected to be conformally dual to each other.*

Usually this fact is phrased as equivalence of different pairs-of-pants decomposition of a surface, even though here we decompose into four punctured spheres. The geometric construction thus systematically produces a collection of theories which are conformally dual to each other. This does not explain the duality directly from the 4d point of view, but rather by imbedding the theories in the higher dimensional setup it renders the dualities a trivial consequence of the geometrical construction, if this is correct.

The anomalies of the different decompositions are the same by construction. However, the indices of the different decompositions a priori might be different as these might be very different looking theories. For the conjecture to be correct combining the four puncture spheres to form surfaces of the same topology in different ways should give equivalent theories up to the action of dualities. In particular, the protected spectrum of the theory has to be invariant

under the exchange of the different factors of $SU(3)$ symmetries associated to the punctures. This is a highly non obvious fact. In particular if this holds for the four puncture sphere it will hold for any surface. One can check this by explicit evaluation of the index in a series expansion. In fact we find that a stronger statement from that which we need appears to hold true. Namely, gluing two tri-fundamentals and ignoring the baryonic symmetry, which is broken for general surfaces on the conformal manifold, the index is invariant under exchanging the four $SU(3)$ symmetries. The index is given by,

$$(q;q)^2 (p;p)^2 \frac{1}{6} \oint \prod_{\ell=1}^{2} \frac{dz_\ell}{2\pi i z_\ell} \frac{\prod_{i,j,\ell=1}^{3} \Gamma_e((qp)^{\frac{1}{3}} a_i^{\mathbf{1}} a_j^{\mathbf{2}} z_\ell^{-1}) \Gamma_e((qp)^{\frac{1}{3}} a_i^{\mathbf{3}} a_j^{\mathbf{4}} z_\ell)}{\prod_{\ell_1 \neq \ell_2} \Gamma_e(z_{\ell_1}/z_{\ell_2})} . \tag{105}$$

Here $\prod_{\ell=1}^{3} z_\ell = 1$ with $z_\ell$ being the parameters of the gauged $SU(3)$. Also $\prod_{j=1}^{3} a_j^{\mathbf{I}} = 1$ with $a^{\mathbf{I}}$ parameterizing the four $SU(3)$ flavors associated to the punctures. Although we do not have a mathematical proof, the above can be found to be invariant under permutations of the four $a^{\mathbf{I}}$ in perturbative expansion in the fugacities $q$ and $p$.

Let us next make several comments on the theory corresponding to the four punctured sphere. We can count the dimension of the conformal manifold of the theory near weak coupling. This is a special theory: The full symmetry of the theory is broken on the conformal manifold and the dimension is *252*. The large dimension of the conformal manifold is due to the fact that in addition to the baryons, which are marginal, we have marginal operators winding from one end to the other on the quiver.[26] We have already mentioned that the superpotential we turn on preserving the puncture symmetries is built from fields having some baryonic charge and it is marginally irrelevant at weak coupling and thus there is no conformal manifold passing through zero coupling and having the symmetries we are interested to have, four $SU(3)$s and no baryonic symmetry present. However, we have conjectured that somewhere on the conformal manifold the required symmetries do appear. We can utilize the supersymmetric index to check this conjecture. In particular, as we can explore the conformal manifold preserving the Cartan generators of all the puncture symmetries we do not loose any parameters we would want to have at the locus where all the $SU(3)$ symmetries appear. The index assuming the symmetry we are interested in takes the following form,

$$1 + q\, p \left( 3\,(\mathbf{3}_1, \mathbf{3}_2, \mathbf{3}_3, \mathbf{3}_4) + \mathbf{1} + \sum_{j=1}^{4} (-\mathbf{8}_j + \mathbf{10}_j) \right) + \cdots . \tag{106}$$

Let us count the exactly marginal deformations then assuming we have the $SU(3)$ symmetries: we can break completely the symmetry on the conformal manifold as $\mathbf{10}$ of $SU(3)$ has non trivial invariants. This means that we obtain *252* exactly marginal deformations. This is consistent with our conjectures.

We expect to have a duality acting on the line of the conformal manifold we find. This duality should permute different symmetry factors. How this duality acts on the couplings is a very interesting question to try and answer.

> *Exercise:* Consider the two different ways to construct genus two surface starting with a four punctured sphere. See Figure 26. The two models should be dual to each other. Analyze the superpotentials in the two frames, show that the theories are conformal, that all the symmetry is broken on the conformal manifold. Compute the dimension of the conformal manifold and check that the superconformal index in the two duality frames agrees in expansion in fugacities.

---

[26]In the next Lecture we will recover the same model in a completely different way and there this large conformal manifold will have a different meaning giving us an example of a 6d duality.

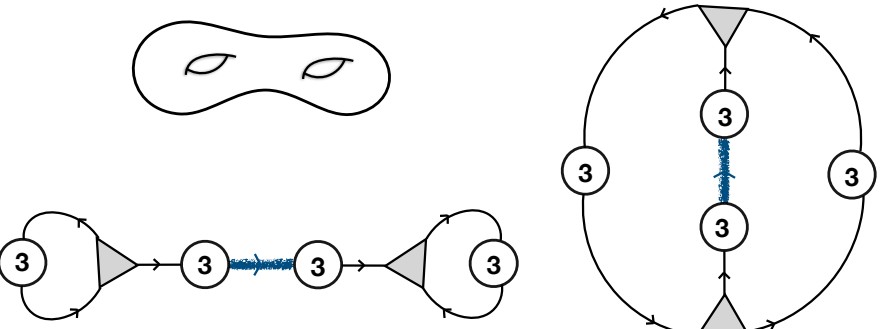

Figure 26: The two duality frames corresponding to genus two surface without punctures. If the conjecture is correct the two models have to be dual to each other on their conformal manifold. Note that the theory on the left has an interesting structure. The two gauge groups on the edges have matter content which is comprised of three adjoint fields, $\mathcal{N} = 4$ SYM. Then, one gauges an $SU(3)$ subgroup of the $\mathcal{N} = 4$ $SU(4)$ R-symmetry with the addition of extra fields. This theory has non generic features, in particular the conformal manifold has a non-generic structure.

This is a special case so let us analyze it in detail. We start with the "Θ" shaped quiver on the right of Figure 26. We have three tri-fundamentals of $SU(3)$, the top $Q_t$, the bottom $Q_b$, and the middle $Q_m$. We also have three $U(1)$ symmetries. We define $U(1)_t$ such that $Q_t$ is charged $-1$ and $Q_{m,b}$ $+1$. Similarly we define two additional symmetries $U(1)_b$ and $U(1)_m$. Note that $U(1)_m$ is anomalous under the two $SU(3)$ gauge symmetries on the sides of "Θ". The $U(1)_t$ is anomalous under the bottom $SU(3)$ and $U(1)_b$ is anomalous under top $SU(3)$. The fields $Q_m$ form a triplet under flavor $SU(3)$. The marginal operators are the baryon $\lambda Q_m^3$ (which form a **10** of flavor $SU(3)$) and the product of all three fields, $\tilde{\lambda} Q_t Q_m Q_b$ (which forms a **3** of flavor $SU(3)$). The latter has charge $+1$ under the three anomalous symmetries. We need to build a singlet out of the couplings under all symmetries. However, note that there are five independent singlets we can build, $\{\lambda^4, \lambda^6, \lambda^3 \tilde{\lambda}^3, \tilde{\lambda}^3 \lambda^5, \tilde{\lambda}^6 \lambda^4\}$. All of these have positive charge under $U(1)_m$ and thus we cannot form a singlet under it as the exponent of the gauge couplings is also charged positively. Thus we deduce that this theory has no conformal manifold in the vicinity of the weak coupling. However, we have conjectured that the four punctured sphere has a locus with four $SU(3)$ symmetries and *no* baryon symmetry. We can thus gauge these symmetries in pairs to form the "Θ" shape. As we do not have any other flavor symmetries except for the puncture ones we obtain an SCFT with a conformal manifold dimension of which is simply given by counting marginal operators minus the currents at zero coupling: 10 $\lambda$ plus 3 $\tilde{\lambda}$ plus 4 gauge couplings minus the currents of $SU(3) \times U(1)^3$. This gives us a six dimensional conformal manifold with no symmetries preserved. Note that for genus 2 we would generally expect $3g - 3 = 3$ deformation but here we find twice that amount.

Let us now analyze the dumbbell quiver on the right of Figure 26. The analysis is rather different than the above. Note that we can think of this model as two copies of $SU(3)$ $\mathcal{N} = 4$ SYM glued to a tri-fundamental with additional two fundamental chiral fields, where we gauge $SU(3)$ subgroup of the $SU(4)$ R-symmetry. Here the two copies of $\mathcal{N} = 4$ SYM give an exactly marginal deformation each preserving the $SU(3)$ flavor (the $\mathcal{N} = 4$ coupling). We gauge two additional $SU(3)$ symmetries under which the tri-fundamental in the middle and the two chiral fields are charged. We have three $U(1)$ symmetries: $U(1)_{left}$ under which one fundamental is charged $+1$, the trifundamental is charged $+1/9$, and the other fundamental charged $-1$; $U(1)_{right}$ reversing the roles of the two fundamentals; and finally $U(1)_M$ under which the two fundamentals have charge $+1$ and the trifundamental has charge $-1/9$. The

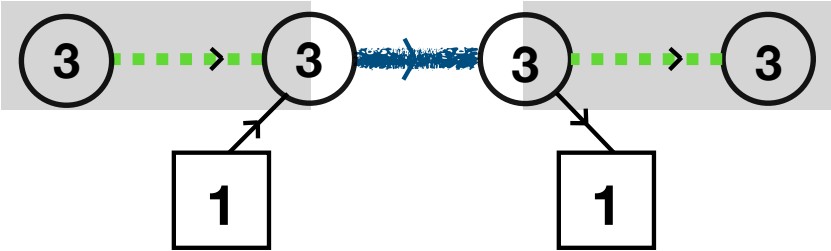

Figure 27: An equivalent way to depict the dumbbell quiver. The two shaded sub-quivers are $\mathcal{N}=4$ SYM. As $\mathbf{3}\times\bar{\mathbf{3}}=\mathbf{8}\oplus\mathbf{1}$ two trifundamental fields split to $(\mathbf{8},\mathbf{3})$ and $(\mathbf{1},\mathbf{3})$ of $SU(3)\times SU(3)$. The dotted lines represent a triplet of adjoints of $SU(3)$.

$U(1)_{left/right}$ symmetries are anomalous under the left/right $SU(3)$ gauge symmetries respectively, while $U(1)_M$ is non anomalous. The marginal deformations are the baryon built from the tri-fundamental and the operator built from the product of the two fundamentals and the tri-fundamental. Under the two anomalous symmetries both operators are charged positively and under the non-anomalous symmetry they have charges of opposite signs, $-1/3$ and $+17/9$. Moreover some of the five singlets of $SU(3)$ detailed above have positive and some negative charge under the non anomalous symmetry. We thus can form a singlet under all symmetries obtaining a conformal manifold under which all the symmetries are broken: We have ten couplings from the baryon, three from the other deformation and two gauge coupling with eleven currents (coming from $SU(3)$ of the tri-fundamental and the three $U(1)$ symmetries). This gives us $10+3+2-8-3=4$ which together with the two deformations coming from the $\mathcal{N}=4$ pieces gives us a six dimensional conformal manifold as expected.

The two models then behave in a strikingly different manner, where while for the dumbbell quiver the conformal manifold passes through weak coupling, it does not do so for the "$\Theta$" shaped quiver. We can understand the difference in this behavior as follows. To get the quiver associated with the genus two surface from the four punctured sphere we need to gauge two $SU(3)$ gauge groups that are embedded as diagonal symmetries in the $SU(3)^4 \subset SU(9)^2$ subgroup of the flavor symmetry of the quiver associated with the four punctured sphere. The two distinct quivers correspond to two different ways to perform this gauging. In one we embed the two $SU(3)$ groups as $\mathbf{9}_{SU(9)_1} \to (\mathbf{3}_{SU(3)_1},\mathbf{3}_{SU(3)_2})$, $\bar{\mathbf{9}}_{SU(9)_2} \to (\bar{\mathbf{3}}_{SU(3)_1},\bar{\mathbf{3}}_{SU(3)_2})$. Gauging $SU(3)_1$ and $SU(3)_2$ now leads to the "$\Theta$" shaped quiver. Alternatively, we can embed the two $SU(3)$ groups as $\mathbf{9}_{SU(9)_1} \to \mathbf{3}_{SU(3)_1} \times \bar{\mathbf{3}}_{SU(3)_1} \to 1+\mathbf{8}_{SU(3)_1}$ and similarly for $SU(9)_2$ and $SU(3)_2$ groups.[27] Gauging $SU(3)_1$ and $SU(3)_2$ now leads to the dumbbell quiver. It is possible to show that there is a subspace of the conformal manifold passing through weak coupling, that preserves the symmetries gauged in the second embedding. This suggests that the dumbbell quiver indeed sits on the conformal manifold of the genus two compactifcation. However, that is not the case for the first embedding, as shown by the analysis done here. In that case the Lagrangian "$\Theta$" shaped quiver is an IR free theory and is not really a compactification of the 6d SCFT on a genus two surface. To preform the gluing in this case, we need to go to the locus on the conformal manifold where only the four $SU(3)$ groups associated with the punctures are preserved. This occurs at strong coupling and requires breaking some of the flavor symmetry that is to be gauged to get the "$\Theta$" shaped quiver, and hence the lack of a 6d interpretation in this case.

□

---

[27]There is also a $U(1)$ that commutes with the $SU(3)$ group for each $SU(9)$ group, but that would not play a role here.

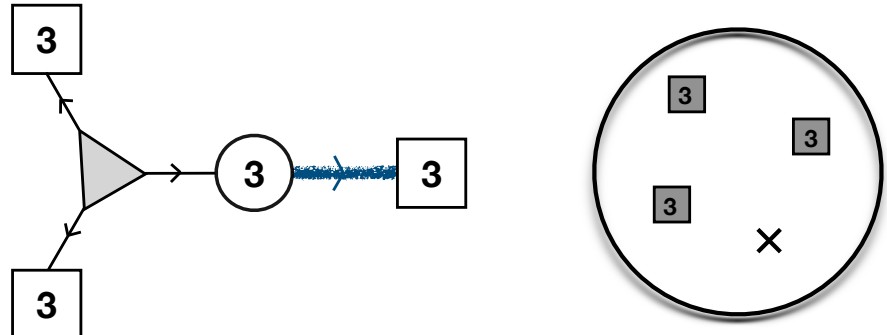

Figure 28: Closing a puncture of the four punctured sphere. The squares are $SU(3)$ punctures and crosses are empty ones. On the conformal manifold at most two $SU(3)$ symmetries are preserved. On a general locus all symmetry is broken and the model can be interpreted as a model with ten empty punctures.

### Closing punctures

We have understood what is the geometric interpretation of gauging $SU(3)$ factors of the global symmetry. We can also discuss other operations, such as turning on relevant deformations (which are absent here), or exactly marginal ones (which we discussed), and giving vacuum expectation values for various operators. Let us turn our attention now to the latter operation.

We will discuss turning on vacuum expectation values (vevs) for operators in a manner that preserves the R-symmetry: We only consider R-symmetry preserving geometric constructions since we wish to find a geometric interpretation of the vev. This in particular implies that the operator we give a vev to should be charged under some global symmetry. The only such symmetry is the puncture symmetry. We have thus a natural candidate for such an operator: These are the marginal operators in the **10** of the puncture symmetries. Such flows will break the symmetry of the puncture and will leave us with punctures of different type. Since the construction of the punctures we have discussed involves twists, it is not possible to close them completely: The twist has to be supported on some cycle.

As we have concrete Lagrangian QFTs corresponding to the geometric construction we can analyze such vevs in complete detail. Parametrizing the character of the fundamental of $SU(3)$ as $\mathbf{3} = b_1 + b_2 + b_3$, with $b_1 b_2 b_3 = 1$, we have (see Appendix C)

$$\mathbf{10} = \sum_{i \neq j} b_i / b_j + \sum_{i=1}^{3} b_i^3 + 1 \,. \tag{107}$$

In index notations we give expectation value to operator contributing with weight $q\, p\, b_1^3$ setting this combination of fugacities to one, that is we give a vacuum expectation value to a single bi-fundamental field. The effect of the flow is simple, the trifundamental associated to our puncture is removed from the theory with the symmetry, to which we gave a vacuum expectation value, and the two additional $SU(3)$ groups are identified: The vev is to a baryonic operator which Higgses one of the gauge $SU(3)$ symmetries. We interpret this procedure as closing a puncture to a different puncture which has no flavor symmetry and we will refer to that as an empty puncture.

Let us add $s'$ empty punctures by starting from a theory with $s' + s$ punctures and closing $s'$ of those. The anomaly is easy to compute and we obtain that empty punctures behave as

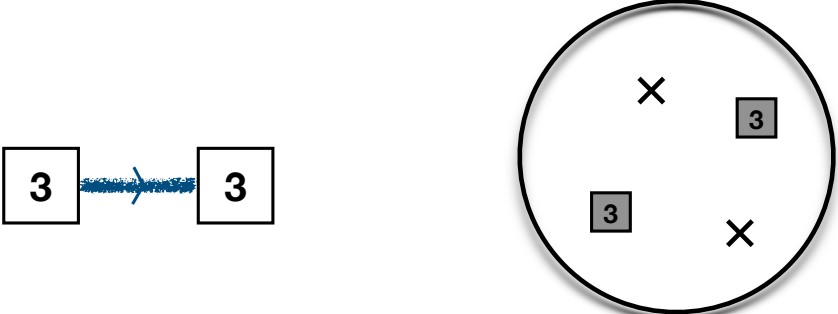

Figure 29: Closing $SU(3)$ puncture of Fig. 5 we obtain a Wess-Zumino theory with two $SU(3)$ and two empty punctures.

one third of the $SU(3)$ puncture. We have,

$$a = \frac{3}{32}(82(g-1) + 33s + 11s'),$$ (108)

$$c = \frac{1}{16}(118(g-1) + 51s + 17s').$$

We can also compute the index and find that it is,

$$1 + \left( 3g - 3 + s + s' + \sum_{j=1}^{s} (-\mathbf{8}_j + \mathbf{10}_j) \right) q\, p + \cdots.$$ (109)

The dimension of the conformal manifold preserving puncture symmetries is $3g - 3 + s + s'$ which is the number of complex structure moduli, as expected. Note that the above results also imply that triplets of empty punctures are equivalent on the conformal manifold to a full puncture. To break maximal punctures into empty ones we turn on the most general baryonic superpotentials. For example, the theory of Figure 28, the $SU(3)$ SQCD with nine flavors, then can be re-interpreted as a compactification on a sphere with ten empty punctures. Thus it is expected to have $3g - 3 + s = 3 \times 0 - 3 + 10 = 7$ dimensional conformal manifold, as indeed it does [40, 51].

The final picture is thus as follows. Turning on the most general exactly marginal superpotentials in the theories we have constructed corresponds to surfaces of some genus and some number of empty punctures. On special loci of the conformal manifold the empty punctures can collide in groups of three and form a maximal puncture with an $SU(3)$ symmetry associated to it.[28] Let us comment in passing that this picture can be further checked by giving vacuum expectation values to derivatives of the operators in $\mathbf{10}$ which will introduce surface defects into the theory [119]. Such constructions are related intimately to integrable models, and the corresponding indices should satisfy interesting sets of properties [120] which can be mathematically proven [121] providing more evidence for the conjectures.

In this Lecture we have discussed a derivation of the dictionary between a theory in 6d with no continuous global symmetry and 4d Lagrangians. Next we will discuss such a dictionary once the 6d theory does possess continuous global symmetry, which introduces more knobs and handles to produce interesting interplays between geometric constructions, emergence of symmetry and duality.

---

[28]The fact that collisions of punctures of one type can form punctures of different type has appeared in various contexts [25, 114–118]. For example in [116] such effects were dubbed a-typical degenerations.

# 5 Lecture IV: Compactifications of the E-string

We will now repeat the analysis of the previous Lecture but in a richer setup of compactifications of the rank one E-string theory.

## 5.1 Six and five dimensions

Let us consider one of the simplest and most studied 6d SCFTs, the rank one E-string theory. This model can be engineered in string theory in various ways. One of them is by taking a single M5 brane to probe the end of the world M9 brane. Another way is by taking a single M5 brane to probe a $\mathbb{C}^2/D_4$ singularity. Instead of taking a single brane, taking $N$ branes, in the former description one obtains the rank $N$ E-string. In the latter description taking one M5 brane to probe $\mathbb{C}^2/D_{N+3}$ singularity one obtain what is called minimal $(D_{N+3}, D_{N+3})$ conformal matter. Thus the $N = 1$ case is a starting point of several sequences of theories and in fact is one of the basic building blocks of constructing most general 6d SCFTs [87]. The rank one theory has a rank one tensor branch on which we have a single tensor multiplet: So in this respect this model is even simpler than the one we discussed in the previous Lecture.

There is only a limited amount of information we will need about this model. One piece of information is that the theory has an $E_8$ global symmetry. Another is the anomaly polynomial, which was computed by now using a variety of techniques [80,94,122]. Lastly, we will need the conjecture that upon compactification to 5d, with a certain holonomy breaking the $E_8$ symmetry, one obtains an effective 5d description as an $\mathcal{N} = 1$ $SU(2)$ SQCD with 8 fundamental hypermultiplets [123]. This effective theory is UV completed by the 6d SCFT. We remind the reader that an holonomy corresponds to a non-trivial background, here for a connection to a global symmetry, such that $F$ is zero but $A$ has a non-trivial integral over a closed cycle, here the compactification circle. The value of the integral of $A$ gives some element of the group, and its presence leads to the breaking of the group to the subgroup commuting with the given element. For instance, the holonomy $diag(e^{i\theta}, e^{-i\theta}, 1, 1, \ldots, 1)$ in $SU(N)$ generically breaks the group to $U(1)^2 \times SU(N-2)$.

We will use this 6d information, as in the previous Lecture, to come up with a prediction for the existence of a class of 4d theories: In particular we will predict their anomalies and symmetries. The anomaly polynomial of the 6d theory is [94,122][29] (we use notations of [26]),

$$I^{6d} = \frac{13}{24} C_2^2(R) - \frac{11}{48} C_2(R)p_1(T) - \frac{1}{60} C_2(R)C_2(E_8)_{\mathbf{248}} + \frac{1}{7200} C_2^2(E_8)_{\mathbf{248}}$$
$$+ \frac{1}{240} p_1(T)C_2(E_8)_{\mathbf{248}} + 29 \frac{7p_1(T)^2 - 4p_2(T)}{5760}. \tag{110}$$

We use the notation $C_2(R)$ for the second Chern class in the fundamental representation of the $SU(2)_R$ 6d R-symmetry. We also employ the notation $C_2(G)_{\mathbf{R}}$ for the second Chern class of the global symmetry $G$, evaluated in the representation $\mathbf{R}$, and $p_1(T), p_2(T)$ for the first and second Pontryagin classes respectively.

The anomaly polynomial of the 4d theory compactified on a closed Riemann surface of genus $g > 1$ with no flux for subgroups of $E_8$ is obtained by twisting the theory, defining $C_2(R) = -C_1(R)^2 + 2(1-g) t C_1(R) + \ldots$ as before, and integrating the 6d anomaly polynomial on the surface. The result is [26],

$$a = \frac{75}{16}(g-1), \qquad c = \frac{43}{8}(g-1), \qquad \text{Tr } R(E_8)^2 = -(g-1). \tag{111}$$

All the rest of the anomalies vanish.

---

[29]See Appendix E.3.

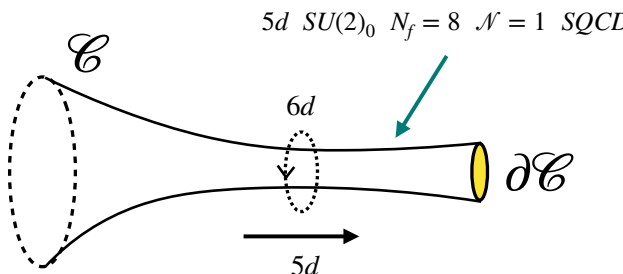

Figure 30: Compactifying the 6d rank one E-string on a circle with a certain holonomy breaking the $E_8$ symmetry to $SO(16)$ is conjectured to be described in 5d by an effective theory which is $SU(2)$ SQCD with eight flavors. The UV completion of the 5d theory is the 6d SCFT. Here $\mathcal{C}$ is the Riemann surface and $\partial\mathcal{C}$ is non empty, we have a puncture.

Now, we can compactify while also turning on fluxes for various abelian subgroups of $E_8$. For a flux in a single $U(1) \subset E_8$ this entails substituting $C_1(U(1)) = -z\,t + C_1(U(1)_F)$ which naturally generalizes to many $U(1)$s. Here $z$ is the value of the flux, $\int_{\mathcal{C}} C_1(U(1)) = -z$, and by $U(1)_F$ we denote the first Chern class of the corresponding symmetry in 4d . This leads to a variety of theories in 4d labeled by fluxes to the various abelian symmetries. We will not quote the most general results here. Let us just mention that the most general compactification will be labeled by the flux for the Cartan generators of the $E_8$ symmetry. To specify this flux we should choose some basis of $U(1)$ symmetries. It is convenient to do so by considering the $SO(16)$ maximal subgroup of $E_8$, $\mathbf{248} \rightarrow \mathbf{120} + \mathbf{128}$. The choice of such an $SO(16)$ subgroup will be later discussed in more details. Then we parametrize the flux by stating its values for the natural $SO(2)$ subgroups of $SO(16)$ and encode it in a vector, an octet, of fluxes we denote by $\mathcal{F}$.

Turning on fluxes will break the $E_8$ symmetry to the subgroup commuting with the choice of flux. In particular we will have abelian symmetries in 4d. As usual in such cases the superconformal symmetry of the 4d fixed point will be some mixture of the Cartan of the 6d R-symmetry, preserved by twisting, and all the abelian symmetries in 4d.

Finally let us discuss the punctures. We specify the maximal boundary conditions giving the 5d $SU(2)$ gauge fields Dirichlet boundary conditions. Here, unlike in the pure $SU(3)$ case of the previous section, we also have matter fields. In terms of $\mathcal{N} = 1$ 4d supersymmetry these are an octet of hypermultiplets, that is 16 chiral fields. We give eight of the chiral fermions in this multiplet Neumann boundary conditions and the remaining ones get Dirichlet boundary condition. These fermions are not transforming under the $SU(2)_R$ R-symmetry coming from six dimensions: This becomes the $SU(2)_R$ R-symmetry of 4d $\mathcal{N} = 2$ under which the fermions in the hypermultiplet do not transform. Thus the inflow contribution to the anomalies involving the R-symmetry only comes from the vector multiplet and gives,

$$\text{Tr}\,R = -1 \times \frac{1}{2} \times \dim SU(2) = -\frac{3}{2}, \qquad \text{Tr}\,R^3 = (-1)^3 \times \frac{1}{2} \times \dim SU(2) = -\frac{3}{2},$$

$$\text{and} \qquad \text{Tr}\,R\,SU(2)^2 = -1 \times 2 \times \frac{1}{2} = -1\,. \tag{112}$$

The matter fields do contribute to the anomalies of the subgroups of $E_8$ global symmetry. The octet of hypermultiplets transforms as a fundamental of the $U(8)$ subgroup of $SO(16) \subset E_8$. Thus the anomalies of a given $U(1)$ will be determined by the inflow of each one of the fermions which obtained Neumann boundary condition. Say we choose a $U(1) \subset E_8$ such that the eight

fermions with Neumann boundary condition have charges $q_{a=1\cdots 8}$, Then, the contribution to the anomaly of this $U(1)$ from the inflow is,

$$\text{Tr } U(1) = \frac{1}{2} \times 2 \times \sum_{a=1}^{8} q_a \,, \quad \text{Tr } U(1)^3 = \frac{1}{2} \times 2 \times \sum_{a=1}^{8} (q_a)^3 \,, \quad \text{Tr } U(1) SU(2)^2 = \frac{1}{2} \times \frac{1}{2} \times \sum_{a=1}^{8} q_a \,.$$

Here the source of $\frac{1}{2}$ is explained in footnote 23, the 2 comes from the fact that the fermions are doublets of the puncture $SU(2)$, and the additional factor of $\frac{1}{2}$ in the last anomaly comes from $\text{Tr } SU(2)^2$ in the fundamental representation. Let us mention here that we have a choice of which eight out of the sixteen chirals gets a Neumann boundary condition. All these choices are in principle equivalent up to the action of the Weyl group of $SO(16)$. However if we have different choices for different punctures on the same surface, the relative difference becomes physically significant. These different choices when defining the types of punctures will be referred to as different *colors* of punctures.

The octet of the fundamental chiral fields with Neumann boundary conditions is expected to give rise to an octet of chiral operators in the fundamental representation of the $SU(2)$ puncture symmetry in 4d. These operators will have R-charge +1 under the R-symmetry inherited from 6d. We will refer to such operators as "moment maps". The name is motivated by analogy with compactifications of the $(2,0)$ theory. If one repeats this procedure in the case of compactifications of the $(2,0)$ theory, say $A_{N-1}$ type, the model in 5d is $SU(N) \, \mathcal{N} = 2$ SYM, and in 5d $\mathcal{N} = 1$ language the matter content has an adjoint hypermultiplet. We again give Neumann boundary conditions to a chiral half and Dirichlet to the other one. In 4d the symmetry associated to the puncture is $SU(N)$ (the gauge symmetry of the 5d theory) and the chiral in the adjoint representation with the Neumann boundary condition becomes the moment map operator in the $\mathcal{N} = 2$ flavor current multiplet in 4d. In the case at hand there is only $\mathcal{N} = 1$ supersymmetry in 4d, but the appearance of the "moment map" operators has the same origin, and hence the name.

Let us use the above to construct the anomalies with punctures,

$$\begin{aligned}
a &= \frac{75}{16} \left( g - 1 + \frac{s}{2} \right) + \left( \frac{9}{32} \left( -\frac{3}{2} \right) - \frac{3}{32} \left( -\frac{3}{2} \right) \right) s = \frac{3}{16} (25(g-1) + 11s) \,, \\
c &= \frac{43}{8} \left( g - 1 + \frac{s}{2} \right) + \left( \frac{9}{32} \left( -\frac{3}{2} \right) - \frac{5}{32} \left( -\frac{3}{2} \right) \right) s = \frac{1}{8} (43(g-1) + 20s) \,.
\end{aligned} \tag{113}$$

Note that in general since the punctures break the symmetry and introduce a $U(1) \subset U(8)$, and fluxes also introduce $U(1)$s, the above are not the superconformal anomalies but rather the ones computed with the 6d R-charge, and these can be viewed just as encoding the 't Hooft anomalies of the 6d R-symmetry using (10),

$$\text{Tr } R^3 = -13(1-g) + 5s \,, \qquad \text{Tr } R = 11(1-g) - 7s \,. \tag{114}$$

With this concrete information about the 6d and the 5d theories we are ready to search for the 4d theories corresponding to the compactifications.

## 5.2   4d theories from genus $g$ Riemann surface with zero flux

Let us now conjecture that the 4d theories obtained by compactifying the rank one E-string on a closed genus $g > 1$ surface with no flux can be described by a conformal Lagrangian. From the anomalies (111) we obtain that,

$$a = \frac{75}{16}(g-1) = \left( 16 a_v + 81 a_\chi \right)(g-1) \,, \qquad c = \frac{43}{8}(g-1) = \left( 16 c_v + 81 c_\chi \right)(g-1) \,. \tag{115}$$

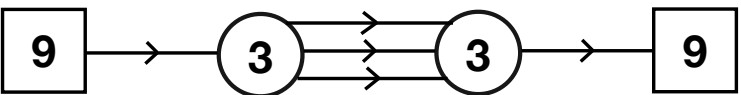

Figure 31: The conformal dual of rank one E-string compactified on genus two surface with no flux. We can turn on any exactly marginal superpotential.

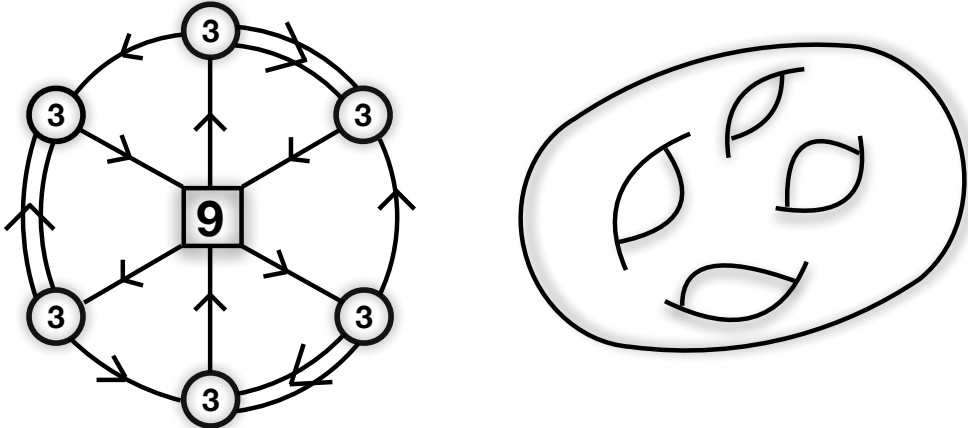

Figure 32: The conformal dual of rank one E-string compactified on genus $g$ surface with no flux. The number of gauge nodes is $2g - 2$. Here we have the example of $g = 4$. The Figure is taken from [25].

The number of vectors is thus $\dim \mathfrak{G} = 16(g-1)$ and the dimension of the representation is $\dim \mathfrak{R} = 81(g-1)$. These numbers are integers and thus there is a chance that the conjecture is correct.

The above value for $\dim \mathfrak{G}$ for genus two surface can be obtained by taking either two $SU(3)$ groups or an $USp(4)$ and two $SU(2)$ groups: We will find a candidate dual with the former option. See Figure 31. Note that this is the same model we obtained in Section 4.4 to correspond to a four punctured sphere compactification of minimal $SU(3)$ SCFT, and here we give it a different interpretation as genus $g = 2$ compactification of rank one E-string with no flux.[30]

We already have discussed properties of this model in previous sections, so let us here just remind that this theory has a 252 dimensional conformal manifold: A fact that we will soon relate to various geometric objects, such as complex structure moduli of the corresponding Riemann surface and the dimension of the space of flat connections on that surface.

For genus $g > 2$ we will seek a simple generalization of the above, and the simplest guess is to have $2g - 2$ $SU(3)$ gauge groups which will amount to $\dim \mathfrak{G} = 16(g-1)$. The quiver theory is depicted in Figure 32. The number of fields is as expected, $\dim \mathfrak{R} = 81(g-1)$. The marginal operators can be built from baryons and gauge invariant triangles on the quiver. The Kähler quotient computing the dimension of the conformal manifold is not empty: On a generic locus of the conformal manifold all the non-R global symmetry is broken and the dimension of the

---

[30]Such equivalences of compactifications are common but deep understanding of them is lacking at the moment. For example, the rank $N$ E-string on a torus with no flux is the $E_8$ Minahan-Nemeschansky theory [124] which can be interpreted as an $A_{6N-1}$ $(2,0)$ theory compactified on a sphere with three punctures of certain types [17]. Spheres with punctures of 2 M5 branes probing a $\mathbb{Z}_k$ singularity [125] can be mapped to tori compactifications with flux of minimal $(D_{N+3}, D_{N+3})$ conformal matter with the number of punctures and $k$ on one side maps to certain combinations of flux and $N$ on the other side [19, 126].

conformal manifold is

$$\dim \mathcal{M}_c = 251(g-1) = 3g - 3 + 248(g-1).$$

This general counting fails for the case of the genus two quiver and has to be done more carefully: We obtain an additional *accidental* marginal deformation as above, see [22].

The above expression for the dimension of the conformal manifold is what one would expect from 6d. First, we have the $3g-3$ deformations we would associate to complex structure moduli. Second, we have $248(g-1)$ additional deformations which have a natural meaning. Note that 248 is the dimension of $E_8$. These deformations then can be associated with flat connections for the $E_8$ global symmetries one can turn on upon compactification on the surface. These flat connections are parameters of the compactification, which in this case turn out to be exactly marginal. A different way to phrase this is that we can consider the proper "KK" reduction of the conserved current of the $E_8$ symmetry in six dimensions, and performing the Riemann-Roch analysis one gets the $248(g-1)$ exactly marginal deformations [113].[31]

*Thus we conjecture that the quiver of Figure 32 describes the compactification of rank one E-string on genus g surface with zero flux. In particular we expect the symmetry to enhance to $E_8$ somewhere on the conformal manifold.*[32]

Although on generic locus of the conformal manifold all the symmetry is broken, turning on non-generic superpotentials we can preserve some symmetry on sub-manifolds of the conformal manifold. For example, turning on cubic superpotentials corresponding to various triangles in the quiver identifies the various $SU(9)$ symmetries. To have a non vanishing Kähler quotient we also turn on baryonic superpotentials for the bifundamental fields. This breaks all the symmetries but a diagonal combination of all the $SU(9)$s. The Kähler quotient is not empty as the baryons and the triangle superpotentials have opposite charges under the baryonic $U(1)$. The baryons charged under the $SU(2)$s of the free point are in **4** and thus have a singlet in the fourth symmetric power. Finally, the triangle superpotentials are in a bifundamental of two $SU(9)$ groups and thus a di-baryon of these is an invariant. We note that under these deformations only, the marginal operators minus the conserved currents take the form,

$$3g + 3 + \left(\mathbf{84} + \overline{\mathbf{84}} + \mathbf{80}\right)(g-1), \tag{116}$$

where $g-1$ **84** and $\overline{\mathbf{84}}$ come from baryons and $g-1$ **80** coming from operators winding between $SU(9)$ groups. We note that $SU(9)$ is a maximal subgroup of $E_8$ such that

$$\mathbf{248}_{E_8} \to \mathbf{84}_{SU(9)} + \overline{\mathbf{84}}_{SU(9)} + \mathbf{80}_{SU(9)}.$$

All the representations of $SU(9)$ appearing in the index are expected to combine into $E_8$ representations under this branching rule. This can be verified at least in expansion of the index in the fugacities. We thus obtain as expected $(g-1)\mathbf{248}_{E_8}$ exactly marginal operators. Furthermore,

$$\mathrm{Tr}\, R\, SU(9)^2 = \frac{1}{2} \times (\frac{2}{3} - 1) \times 3 \times (2g-2) = -(g-1),$$

---

[31]In the $g=2$ case we have one additional deformation which is not explained by the general geometric logic: We do not know what the origin of it might be, but one possible explanation is that the 6d SCFT has certain surface operators which we can wrap on the Riemann surface and obtain local operators in 4d. For low genus these operators might have low charges and for example be exactly marginal. It will be very interesting to understand these issues in detail.

[32]There are other constructions of these models in 4d [26,118]. The other constructions are not completely Lagrangian as they rely on gauging certain emergent symmetries in the IR. Nevertheless they can be used to compute protected quantities such as indices, and one can check that the indices in all the descriptions agree. This serves as a consistency check of the various conjectures.

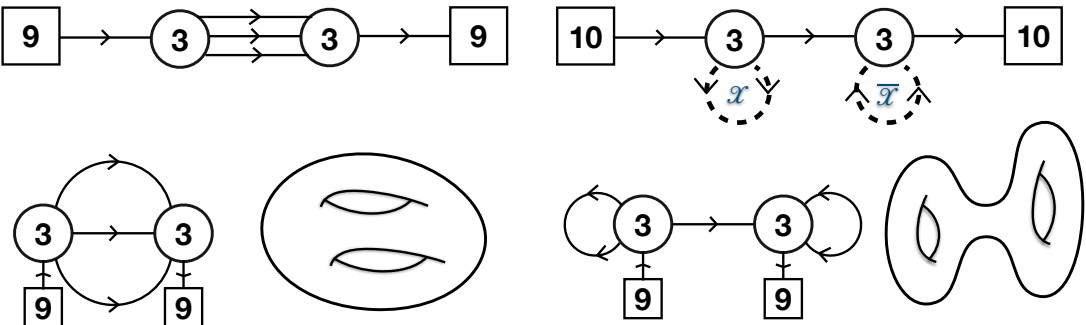

Figure 33: A conformal duality which we want to interpret as two different pair of pants decompositions of a genus two surface. The fields denoted by $X$ and $\overline{X}$ are in two index symmetric representation of $SU(3)$.

which matches the result expected from the strongly coupled theory where the $SU(9)$ is embedded inside $E_8$ (111) (Note that the imbedding index of $SU(9)$ in $E_8$ is 1). Note that actually if the theory has an $E_8$ point on the conformal manifold, or rather a $3g-3$ dimensional locus of such, with $(g-1)\mathbf{248}_{E_8}$ marginal operators, we cannot go out of it preserving $SU(9)$, but rather only the Cartan subgroup: This is what the Kähler quotient tells us, we can construct invariants out of powers of the adjoint which preserve the Cartan but break the non-abelian structure. Thus, all we can say is that the $SU(9)$ preserving sub-manifold of the conformal manifold passing through the weak coupling point might intersect the Cartan preserving sub-manifold passing through the $E_8$ point.

Finally, one can wonder whether the quiver theory we have found is the only conformal gauge theory corresponding to the given compactification. The answer is actually that there are more, and all of these are conformal dual to each other. Let us give here the example of genus two in some detail. We can match the anomalies with the two quivers in Figure 33: The one on the left we have already discussed, but also the one on the right fits the bill. The conformal anomalies work simply because it is again a conformal theory with two $SU(3)$ gauge groups and the number of fields is $2 \times 3 \times 10 + 2 \times 6 + 3 \times 3 = 81$. The gauge nodes are indeed conformal as the beta functions vanish,

$$\mathrm{Tr}\, SU(3)^3 = (-1) \times 10 + (+1) \times 3 + (+7) = 0\,,$$
$$\mathrm{Tr}\, R\, SU(3)^2 = 3 + \frac{1}{2} \times \left(\frac{2}{3} - 1\right)(10 + 3) + \frac{5}{2} \times \left(\frac{2}{3} - 1\right) = 0\,. \tag{117}$$

We used the fact that the cubic Casimir of the two index symmetric representation of $SU(N)$ is $4 + N$ and the Dynkin index is $\frac{N+2}{2}$. We can turn on baryonic superpotentials, the cubic symmetric power of the fields $X$ and $\overline{X}$ and the deformation running across the quiver. We leave the analysis of the Kähler quotient as an exercise. We just mention that the symmetry can be broken completely on the conformal manifold. The number of marginal operators at the free point is $120 \times 2 + 1 + 2 + 55 \times 2 + 10 \times 10 = 453$. The non-anomalous symmetry at the free point is $SU(10) \times SU(10) \times U(1)^3$, dimension of which is $99 + 99 + 3 = 201$. All in all the dimension of the conformal manifold is then $452 - 201 = 252$ as expected. We thus conjecture that the two quivers are dual to each other.[33] Note that as in the $G_2$ SQCD dual to a quiver theory we discussed in Lecture II, the two weakly coupled cusps are very different: They have different symmetry and different number of marginal operators. However going on the conformal manifold of both there is a chance that we can get from one to the other, and that is what the conjecture of the duality is about.

---

[33]This duality can be related to a sequence of Seiberg dualities [127].

In fact there is a natural interpretaion of the two theories being two different pair of pants decompositions of the genus two surface, as depicted in Figure 33. The point is that we can think of a line ending on the same gauge node with same orientation as a two index symmetric plus an antifundamental, $\mathbf{3} \times \mathbf{3} \rightarrow \mathbf{6} \oplus \overline{\mathbf{3}}$ (this is in contrast to a bi-fundamental ending on the same node which is an adjoint and a singlet). This duality thus suggests that we should be able to "cut" the bifundamental lines in the quiver and obtain theories corresponding to pairs-of-pants: Three puncture spheres. This is what we will do next, and by doing so we will discover how to compactify the rank one E-string on an arbitrary surface with an arbitrary amount of flux.

## 5.3 Decomposing the surface into pairs of pants

Let us start with the quiver in Figure 31 and try to decompose it into a gluing of two three punctured spheres. We have discussed that a puncture of rank one E-string compactifications should correspond to an $SU(2)$ symmetry. Thus we should rewrite the quiver as an $SU(2)^3$ gluing of two SCFTs. It is straightforward to do so using the S-confining Seiberg duality of Figure 14. We detail this rewriting in Figure 34. The quiver contains triangular blocks with superpotential terms of the form, $h \cdot \widetilde{h} \cdot Q_i + Q_i^3$. Such triangular blocks can be exchanged using the S-confining duality by an $SU(2)$ SQCD with $N_f = 3$. We have three such blocks corresponding to the three bi-fundamental fields $Q_i$, and thus can perform this procedure for each one of them. In the end we obtain the last quiver in the chain of dualities of Figure 34 having a superpotential which is a combination of quartic and sextic terms.

Next we conjecture that each one of the two $SU(3)$ $N_f = 6$ SQCD blocks, which are glued together by gauging the $SU(2)^3$ subgroup of their global symmetry and turning on the superpotential, corresponds to a three punctured sphere compactification with a certain amount of flux to be determined soon. Let us then discuss these SQCD models in detail.

**The three punctured spheres**

We analyze next the $\mathcal{N} = 1$ $SU(3)$ $N_f = 6$ SQCD interpreted as a compactification of the rank one E-string on three punctured sphere. The model is depicted in Figure 35. The global symmetry of the model is $SU(6) \times SU(6) \times U(1)$. However, following our disucssion in the previous section we should think about this symmetry in terms of specific sub-groups. In particular, taking one of the $SU(6)$ factors (say the one rotating the fundamentals) we decompose it into $SU(2)^3 \times U(1)^2$ maximal subgroup. We then associate the three $SU(2)$ factors with the three maximal punctures. The three remaining $U(1)$s (two coming from the decomposition of $SU(6)$ and the baryonic one) together with the remaining $SU(6)$ will form a maximal subgroup of the six dimensional $E_8$ symmetry. This statement is motivated by the discussion in the previous section and we will test it more soon, but here let us mention that the embedding of this subgroup in $E_8$ is as follows,

$$E_8 \rightarrow E_7 \times U(1) \rightarrow (SU(6) \times SU(3)) \times U(1) \rightarrow (SU(6) \times U(1)^2) \times U(1). \tag{118}$$

The superconformal R-symmetry of all the chiral fields is $\frac{1}{2}$. However this is not the R-symmetry which we obtained decomposing the genus two surface into trinions. The R-symmetry we will use is obtained by mixing the baryonic $U(1)$ symmetry with the superconformal one. In particular, we assign an R-charge 1/3 to the fundamentals and 2/3 to the anti-fundamental chiral fields. As this is the R-symmetry we obtained decomposing higher genus surfaces with zero flux it coincides with the Cartan generator of the $su(2)$ R-symmetry of the six dimensional E-string theory.

If we are to interpret the $SU(2)$ symmetries identified above with puncture symmetries we also need to find the associated octet of moment map operators. As we have discussed in



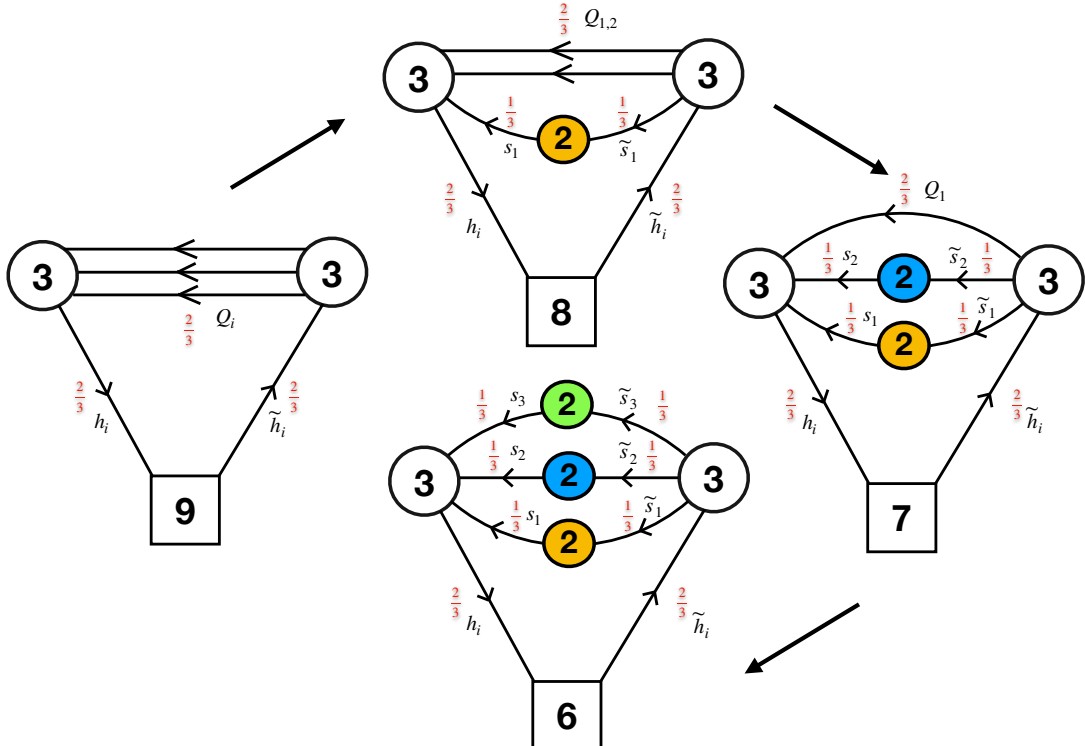

Figure 34: Cutting the edges of the quiver step by step employing the S-confining Seiberg duality of Figure 14. In the first step, going from the first to the second quiver, the field $h_9$ maps to $s_1^2$, $\widetilde{h}_9$ to $\widetilde{s}_1^2$, and $Q_1$ to $s_1\widetilde{s}_1$. The superpotential of the second quiver involves terms of the form $(s_1\widetilde{s}_1)^2 Q_i + (s_1\widetilde{s}_1)Q_i^2 + s_1^2\widetilde{s}_1^2(Q_1+Q_2) + s_1\widetilde{s}_1 h_i h^i$. In the second step we exchange yet another triangle in the quiver with an $SU(2)$ $N_f = 3$ sector, and similarly in the last step. The final superpotential has quartic terms of the form $(s_i h_j)(\widetilde{s}^i h^j)$ and sextic terms of the form $(s_i s_j^2)(\widetilde{s}^i(\widetilde{s}^j)^2)$. The numbers in red denote a choice of R-symmetry. As this is the superconformal R-symmetry for the first quiver and it follows from duality for the rest, it is the superconformal choice here for all the quivers.

Subsection 5.1, we expect the moment map operators to be in the fundamental representation of only one $SU(2)$ puncture symmetry and have R-charge $+1$. Indeed we have a variety of operators with these properties: For each puncture $SU(2)$ we have six mesons and two baryons with these properties. In more detail, the charges of the moment map operators (encoded in terms of their fugacities) are given by,

$$
\begin{aligned}
M_x &= \mathbf{2}_x \ \otimes \ \left( \mathbf{6}_{u^4/v^2w^2} \oplus \mathbf{1}_{u^6v^{12}} \oplus \mathbf{1}_{u^6w^{12}} \right), \\
M_y &= \mathbf{2}_y \ \otimes \ \left( \mathbf{6}_{v^4/u^2w^2} \oplus \mathbf{1}_{v^6u^{12}} \oplus \mathbf{1}_{v^6w^{12}} \right), \\
M_z &= \mathbf{2}_z \ \otimes \ \left( \mathbf{6}_{w^4/u^2v^2} \oplus \mathbf{1}_{w^6u^{12}} \oplus \mathbf{1}_{w^6v^{12}} \right).
\end{aligned}
\tag{119}
$$

We denoted the three octets of moment maps by $M_{x,y,z}$ with $x$, $y$, and $z$ denoting the Cartan generators of the corresponding $SU(2)$ puncture symmetries. Further, $\mathbf{6}$ denotes the character of the fundamental representation of the $SU(6)$ symmetry. The sextets of the $SU(6)$ are built from the mesons and the singlets from baryons. The baryon which has charges $\mathbf{1}_{u^6v^{12}}$ is built from two quarks charged under $SU(2)_y$ and one quark charged under $SU(2)_x$. Importantly, as can be seen from (119) the three punctures of the trinion have moment map operators charged differently under the various symmetries. These punctures thus are of different types: Such

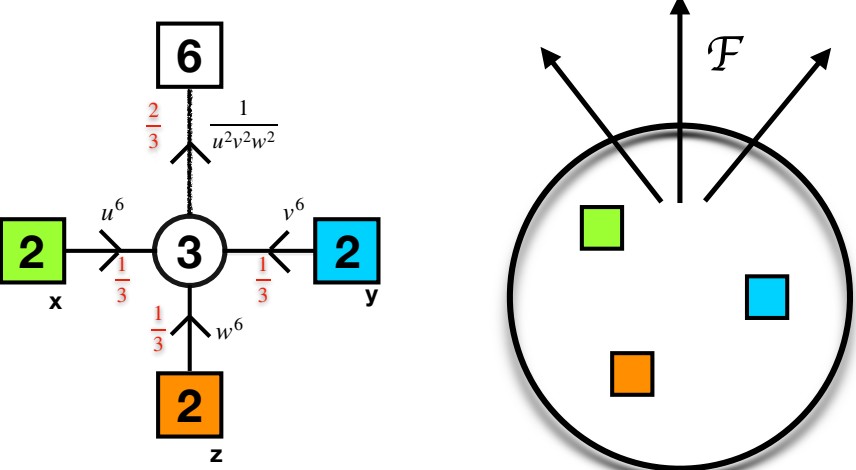

Figure 35: The $SU(3)$ SQCD in the middle of the conformal window reinterpreted as a 4d theory dual across dimensions to the rank one E-string placed on a three punctured sphere with a particular value of flux. The assignment of R-charges (denoted in red) follows from the decomposition of higher genus surfaces into trinions. This also can be identified with the Cartan of the six dimensional R-symmetry. The Figure is taken from [25].

types of punctures are usually called *colors* and, as we have discussed in Subsection 5.1, can be attributed to different choices in defining the boundary conditions in 5d.

Finally, let us compute the various 't Hooft anomalies of the model involving the R-symmetry,

$$\text{Tr } R = 8 + \left(\frac{2}{3} - 1\right) \times 18 + \left(\frac{1}{3} - 1\right) \times 18 = -10\,,$$

$$\text{Tr } R^3 = 8 + \left(\frac{2}{3} - 1\right)^3 \times 18 + \left(\frac{1}{3} - 1\right)^3 \times 18 = 2\,, \tag{120}$$

$$\text{Tr } R\,SU(2)^2_{x,y,z} = \frac{1}{2} \times \left(\frac{1}{3} - 1\right) \times 3 = -1\,.$$

These anomalies match the predictions for a three punctured sphere in (112) and (114). As the anomalies above are insensitive to the choice of flux we can match them without specifying the flux associated to the three punctured sphere we discuss.

## Gluing back to closed surfaces: S-gluing

We have obtained a 4d theory dual across dimensions to the E-string compactified on a three punctured sphere with some value of flux, value of which we need still to determine. Let us now discuss how to combine these trinions to obtain theories corresponding to more general surfaces, and in particular closed surfaces. We know what needs to be done as we obtained three punctured spheres by decomposing a closed surface, but it is worth to understand the gluing more systematically. The gluing we discuss first is called an S-gluing following the nomenclature of [26].[34]

---

[34]S-gluing can be defined for compactifications of any 6d SCFTs, as long as we have a notion of the maximal puncture: see *e.g.* [117, 125, 128, 129].

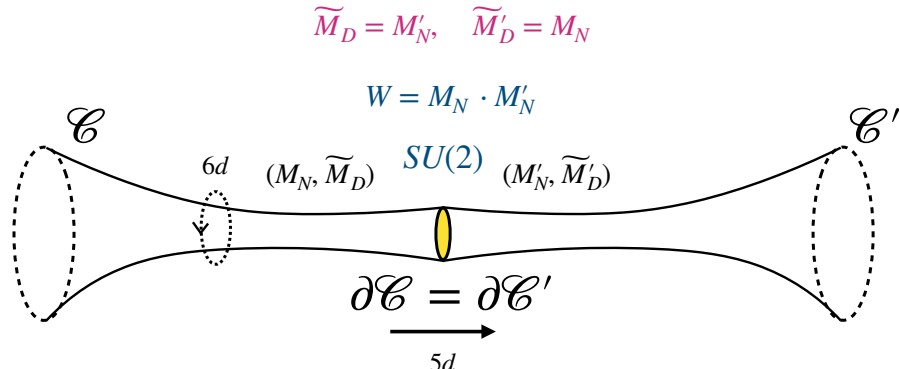

Figure 36: Explanation of S-gluing in 5d. We have the 16 fields on the left $(M_N, \widetilde{M}_D)$ and on the right, $(M'_N, \widetilde{M}'_D)$. We gave eight of these Dirichlet (label $D$) and eight Neumann (label $N$) boundary conditions. Now we are undoing the boundary conditions by gauging the $SU(2)$ symmetry and identifying $M'_D$ with $M_N$ and $M_D$ with $M'_N$, and thus having the full set of dynamical fields at the four dimensional boundary given by $M_N$ and $M'_N$.

The basic idea of gluing is to undo the 5d boundary conditions. This amounts to turning the background vector fields coupled to the flavor symmetry at the boundary to be dynamical. This corresponds to the gauging of the puncture $SU(2)$ symmetry here. However we also have the matter fields of the 5d theory, some of which we lost due to Dirichlet boundary conditions and thus need to reintroduce. In S-gluing we identify the operators of one puncture as the ones corresponding to the fields obtaining the Dirichlet boundary condition in the other puncture, and vice versa. The fields obtaining Dirichlet and Neumann boundary conditions have opposite charges and that is exactly what the superpotential above does for us. See Figure 36 for detailed illustration.

The S-gluing procedure for the E-string is thus performed as follows. Given two maximal punctures one gauges the diagonal combination of the two puncture $SU(2)$ symmetries and couples the moment maps of the two punctures (the octets of operators $M$ and $M'$ ) with a superpotential,

$$W = M \cdot M' \equiv \sum_{i=1}^{8} M_i M'_i. \tag{121}$$

Our field theoretic construction some of the moment maps are mesonic (quadratic in fields) and some are baryonic (cubic in fields). Thus the superpotential above has quartic and sextic interactions in fields. As the moment maps have R-charge one the superpotential is consistent with our choice of R-symmetry. Moreover, it is easy to verify that this R-symmetry is also not anomalous.

If we take two identical theories and S-glue them to each other because of the superpotential the charges of the moment maps of the two theories are identified with conjugation. In particular that means that if we associate some flux to one of the theories we need top associate a negated flux to the other one. The flux of the associated to the surface after S-gluing is zero. Here there is an important assumption that the flux associated to the combined surface is the sum of fluxes associated to the ingredients: This is an assumption motivated by consistency of the arguments following from it.

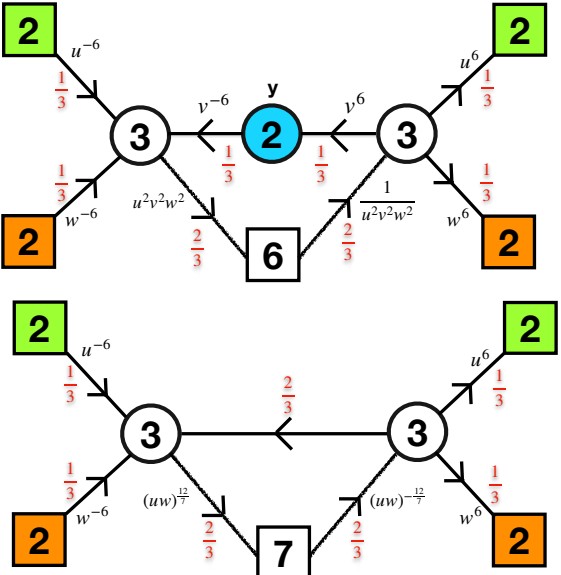

Figure 37: On the left we have two trinions S-glued to each other. There are various superpotential terms consistent with the charges denoted on the Figure. On the right we utilize the S-confining Seiberg duality of $SU(2)$ SQCD with three flavors to simplify the quiver theory. The Figure is taken from [25].

Exercise: Analyze the dynamics of gluing the two three punctured spheres (trinions) into a genus two surface.

As the gluing procedure contains quartic and sextic superpotential terms one shoul wonder whether these are relevant interactions or not. One can study this problem step by step, as we have done in Lecture II is by turning on only a single interaction at each step so that this interactions is relevant. We will not do this in detail here and leave the analysis as an exercise. The solution can be found in [25]. Let us just mention that the sequence of interactions is: Gauging the puncture symmetry first, quartic superpoential second, and finally the sextic superpotential. See Figure 37 for the illustration of S-gluing. We then can continue gluing the rest of punctures in pairs to obtain a genus two surface. One needs to be careful to glue together punctures so that no global stmmetry is broken by the interactions: This implies S-gluing together punctures so that the corresponding moment maps have opposite charges under global symmetries. Gluing two different surfaces together this can be always done as the procedure simply amounts to identifying the symmetries of the two surfaces in certain way. However when we glue punctures on the same surface we have no room anymore for arbitrary identifications: We need to glue punctures of the same "color" together. Field theoretically one can flue different punctures together, however in this case some of the symmetries will be broken. The geometric reason for this can be traced to issues with *discrete fluxes and twists* [26, 130]: We will not discuss such issues here. The procedure can be generalized to constructing general genus $g$ surfaces as depicted in Figure 38.

$\square$

## Gluing back to closed surfaces: Φ-gluing

The gluing we have discussed identifies the charges of the moment map operators of the two glued punctures with conjugation. Let us now discuss a different gluing, the Φ-gluing in the nomenclature of [26], which in particular identifies the symmetries of the moment maps of

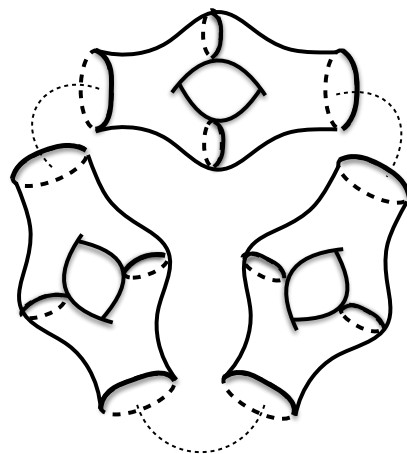

Figure 38: S-gluing trinions into a genus $g$ surface with no flux ($g = 4$ in this example). The Figure is taken from [25].

the two punctures without conjugation. In particular when $\Phi$-gluing two theories of the same kind together the corresponding fluxes are summed and not subtracted.

The $\Phi$-gluing includes gauging the puncture $SU(2)$ symmetry and identifying the moment map operators of the two punctures using the superpotential,

$$W = \sum_{i=1}^{8} \left( M_i - M_i' \right) \Phi_i \, , \tag{122}$$

where $\Phi_i$ is an octet of fields in the fundamental representation of the gauged symmetry which are added to the model. The 5d explanation of this procedure is as follows. In this gluing the missing fields, the ones which obtained Dirichlet boundary conditions, are the octet of fundamental chiral fields $\Phi$ while the fields obtaining the Neumann boundary condition are the moment maps of the two punctures which are identified, see Figure 39. Note that $\Phi$ and the moment maps have opposite charges as should be the case for chiral halves of the hypermultiplet in 5d. The $\Phi$-gluing is analogous to the $\mathcal{N} = 2$ gluing in the compactifications of $(2, 0)$ theory [8]. This field there can be viewed as the $\mathcal{N} = 1$ chiral adjoint superfield in the $\mathcal{N} = 2$ vector multiplet.

Let us make here two comments. First, from the discussion above it is clear that there is a simple and useful way to relate the two types of gluings, and in fact generalize them. First, let us consider taking a theory with a puncture and corresponding moment maps $M$ and deform the theory with the superpotential,

$$W = \Phi_M \cdot M \, , \tag{123}$$

where $\Phi_M$ is an octet of chiral fields in the fundamental representation of the puncture $SU(2)$ symmetry. This superpotential removes $M$ from the chiral ring of the theory and the procedure is usually called "flipping M" (with the field $\Phi_M$ being the flip field) [15]. Now notice that the field $\Phi_M$ has opposite charges to $M$ and becomes the moment map of the new theory.[35] In particular the procedure of say S-gluing two punctures can be thought of as first flipping one of the punctures and then $\Phi$-gluing them,

$$W = \Phi_M \cdot M + \Phi \cdot \left( M' - \Phi_M \right) \quad \rightarrow \quad W = M \cdot M' \, , \tag{124}$$

---

[35]Such flips of punctures, or "signs" of punctures, are very common in various geometric constructions. See for example [12, 131].

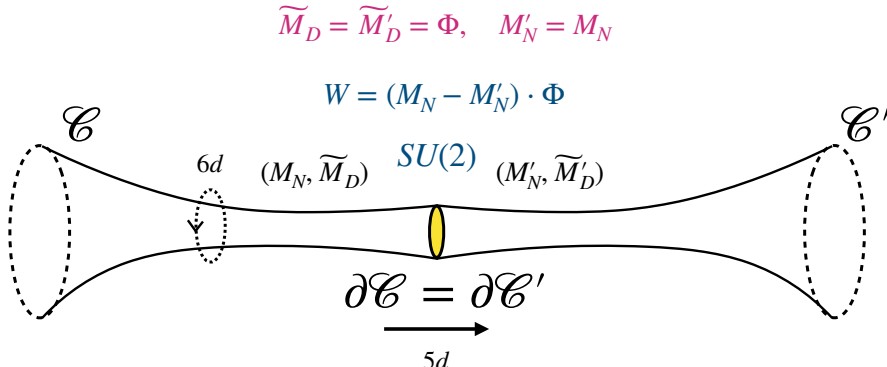

Figure 39: Explanation of $\Phi$-gluing in 5d. We have the 16 fields on the left $(M_N, \widetilde{M}_D)$ and on the right, $(M'_N, \widetilde{M}'_D)$. We gave eight of these Dirichlet (label $D$) and eight Neumann (label $N$) boundary conditions. Now we are undoing the boundary conditions by gauging the $SU(2)$ symmetry and identifying $M'_D$ with $M_D$ and $\Phi$ and $M_N$ with $M'_N$, and thus having the full set of dynamical fields at the four dimensional boundary given by $\Phi$ and $M_N = M'_N$.

where in the last step we have integrated out the massive fields $\Phi$ and $\Phi_M$. In a similar way the $\Phi$-gluing can be thought of as first flipping one of the punctures and then S-gluing.

Finally, the procedure can be generalized when we $\Phi$-glue some of the components of the moment maps and S-glue the remaining ones. That is we glue by gauging the puncture $SU(2)$ symmetry and turn on the superpotential,

$$W = \sum_{i=1}^{k} M_i M'_{\sigma(i)} + \sum_{i=k+1}^{8} \Phi_i \cdot \left( M_i - M'_{\sigma(i)} \right), \tag{125}$$

where $k$ is some number between zero and eight and $\sigma \in S_8$ is a permutation of the eight moment maps. This procedure is subject to some global obstructions having to do with consistency of various fluxes of the glued theories. In field theory these obstructions will appear concretely as the demand that gaugings should be free of Witten or chiral anomalies. We will not discuss these subtleties here.

Let us note that if the vector of fluxes of the two glued theories is $\mathcal{F}$ and $\mathcal{F}'$, the components of the vector of fluxes of the glued theories are $\mathcal{F}_i + (-1)^{ind(i)} \mathcal{F}'_{\sigma(i)}$ where $ind(i)$ is zero if the component $i$ is $\Phi$-glued and 1 if it is S-glued.

> Exercise: Analyze the dynamics of the $\Phi$-gluing of two three punctured spheres into a four punctured sphere.

The dynamics of the $\Phi$-gluing is more intricate than that of the S-gluing. For example gauging the puncture $SU(2)$ symmetry we have fourteen fundamental chiral fields which naively renders it IR free. However, one again can seek for a particular sequence in which to turn on all the interactions so that at each step the interactions will be relevant. In fact we have already analyzed most of this dynamics in an exercise in Lecture I (see Figures 10 and 11), and we will leave the rest as an exercise. See [25] for more details. Let us stress that in principle it might happen that the interactions turn out to be irrelevant: This by itself does not invalidate the identification of the across dimension duals. What that would imply is that the field theoretic description in 4d is an effective one and we should think of the 6d construction as of its UV completion. □

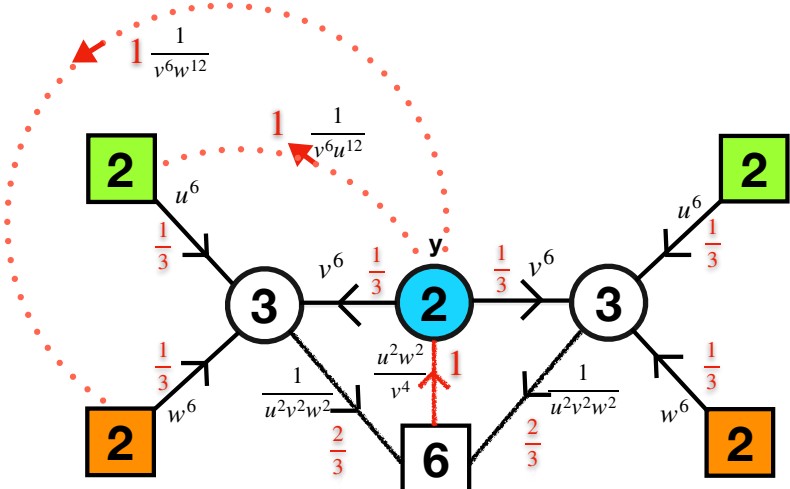

Figure 40: The field theoretic description of two trinions $\Phi$-glued to each other. The fields $\Phi_i$ flipping the moment maps are denoted by red lines: The dotted lines flip the baryonic operators with charges $v^6u^{12}$ and $v^6w^{12}$. The notation is that the dotted fields are in the singlet representation of the symmetry they point to and in the fundamental representation of the symmetry they emanate from. There are two sets of flipped baryon operators coming from each one of the two glued trinions with the superpotential given in (125). Note the way the quiver is drawn it might appear that the dotted fields couple asymmetrically to the two trinions, however because of (125) the coupling is symmetric. The Figure is taken from [25].

Using the $\Phi$-gluing we can identify what is the flux we should associate to the theory we conjectured is obtained by a compactification on the three punctured sphere. We can $\Phi$-glue the punctures of two three punctured spheres in pairs and obtain a genus two surface. Doing so we obtain that the superconformal R-symmetry of the theory is,

$$R_{sc} = R + \frac{1}{54}(\sqrt{5}-1)(\mathfrak{q}_u + \mathfrak{q}_v + \mathfrak{q}_w), \tag{126}$$

where $\mathfrak{q}_{u,v,w}$ denote charges under $U(1)_{u,v,w}$. This is done by performing a-maximization and assuming as usual that there are no accidental emergent $U(1)$ symmetries in the IR. Here the a-maximization gives a non-trivial result, as opposed to S-gluing, since the charges of various abelian symmetries in the field theory do not appear symmetrically to conjugation. Using this R-symmetry we can compute the conformal anomalies,

$$a = \frac{5\sqrt{5}}{4} + \frac{47}{16}, \qquad c = \frac{3\sqrt{5}}{2} + \frac{27}{8}. \tag{127}$$

One can wonder whether there is a compactification of the E-string theory on a genus two surface with *some value of flux* giving these conformal anomalies after a-maximization. In fact there is: We can take the anomaly polynomial of the E-string theory and integrate it over genus two surface with one unit of flux in a $U(1)$ which breaks the $E_8$ global symmetry of the 6d SCFT to $E_7 \times U(1)$. We will not perform the detailed analysis here as it requires a significant build up, but it can be read off easily from the 6d results in [26] (See Appendix F.3.). It is then natural to associate half a unit of this flux to each three punctured sphere we have obtained.

Finally we should detail what is the combination of the abelian symmetry explicitly visible in the Lagrangian of our SQCD model which corresponds to the $U(1)$ with the flux. A natural

such combination is a diagonal $U(1)$ of $U(1)_u$, $U(1)_v$, and $U(1)_w$. This is also supported by the structure of mixing of the three symmetries with the R-symmetry in (126). Then two other combinations of the abelian symmetries together with the $SU(6)$ should build the $E_7$. In fact $E_7$ has an $SU(3) \times SU(6)$ maximal sub-group and the precise imbedding of various symmetries is,

$$E_8 \to E_7 \times SU(2)_{u^6 v^6 w^6} \to SU(6) \times SU(3)_{\frac{u^8}{w^4 v^4}, \frac{v^8}{w^4 u^4}} \times SU(2)_{u^6 v^6 w^6}, \tag{128}$$

with the $U(1)$s in (118) being the Cartan generators of $SU(3)$ and $SU(2)$. The flux we have derived preserves the $E_7 \times U(1)$ subgroup of $E_8$, however the three punctured sphere only manifests a subgroup of this. The reason is that we have punctures and these come also with a choice of a subgroup of $E_8$: Choice of $U(1) \times SU(8)$ under which the moment maps are charged. This choices comes about, as we discussed, since we need to specify boundary conditions of the puncture in 5d. The choice of the boundary conditions breaks further the symmetry of the surface. However, if we consider surfaces without punctures, the symmetry of the theory should be the one consistent with the choice of flux: In our genus two case we thus would expect an $E_7 \times U(1)$. This symmetry is not manifest in the Lagrangian but we expect the conformal manifold of the IR theory to have a locus with this symmetry. One way to test this is to compute the supersymmetric index from the explicit field theoretic description. Computing the index of a genus two surface using $\Phi$ gluing for all punctures we obtain,

$$\mathcal{I}_{g=2, \mathcal{F}_{U(1) \times E_7} = 1} = 1 + \left( 3 + \{ 1 + \mathbf{133}_{E_7} \} + 3 \frac{1}{u^{12} v^{12} w^{12}} + 2 \frac{1}{u^6 v^6 w^6} \mathbf{56}_{E_7} \right) q p + \cdots . \tag{129}$$

This expression is written using the 6d R-symmetry. Because the superconformal R-symmetry is different, see (126), the operators with negative $uvw$ charge are actually relevant and contributions with zero charge are marginal. We observe that all the contributions combine into $E_7$ representations in the index. This is a rather non-trivial further evidence that our identification of the three punctured sphere and the flux is correct. Let us note in passing that in addition to having representations of $E_7$ the multiplicities of the various operators (*e.g.* $\frac{1}{u^{12} v^{12} w^{12}}$ appearing thrice above) can be also predicted from six dimensions [113] and for general genus the field theoretic computations agree with such predictions.

Finally, a theory corresponding to a four punctured sphere is expected to have a duality group acting on its conformal manifold exchanging punctures of the same type. These are the different pairs of pants decompositions we already discussed in the previous Lecture. One way to check that the duality holds is to consider the index of the four punctures sphere and verify that it is invariant under exchanging fugacities parametrizing the symmetries of punctures of the same type. This exercise can be performed for both S- and $\Phi$−gluings, and indeed at least in expansion of fugacities (as was also done in the previous Lecture) the result is consistent with such dualitis. We leave this as an exercise here and the reader can also consult [25] for more details.

## 5.4 Closing punctures and comments

As we have discussed in Lecture III there is yet another field theoretic operation which has a natural geometric meaning: Giving vevs to operators charged under puncture symmetries and closing the punctures. Before addressing this issue here let us expand a bit on the definition of the flux. To specify the flux of the system in a more general way than we have done so far we need to specify the flux of all the Cartan generators of $E_8$. There are many choices of this octet of $U(1)$ symmetries. Given a puncture a natural choice is in terms of the eight $U(1)$s rotating each one of the components of the moment map. Thus the flux is specified by a vector $\mathcal{F}$ which has eight components. A natural choice of these eight $U(1)$ symmetries is to take the Cartans

of $U(8) \subset SO(16) \subset E_8$. The choices of these $U(1)$s depend on the choices of the punctures. Different choices are related by an appropriate linear transformations of the $U(1)$s.

Given the octet of fluxes we can understand what is the sub-group of $E_8$ that it preserves. The preserved symmetry is generated by the roots of $E_8$ which are orthogonal to the vector of flux. See *e.g.* Appendix A of [132] for a detailed discussion. In our parametrization we write the flux in terms of the $SO(16)$ maximal subgroup of $E_8$ with the roots of the $SO(16)$ subgroup being $\frac{1}{2}(\pm 1, \pm 1, 0, 0, 0, 0, 0, 0)$ and permutations thereof. To find the symmetry we also need to complete the $SO(16)$ to $E_8$ by adding the spinor weights, which are $\frac{1}{4}(\pm 1, \pm 1, \pm 1, \pm 1, \pm 1, \pm 1, \pm 1, \pm 1)$: Here we need to make a choice and either allow only even number of minus signs (corresponding to a spinor) or only odd number (corresponding to a co-spinor). For the constructions we discuss here all odd turns out to be the relevant choice. We will soon discuss this more and here proceed with making this choice. We then check which of the roots of $SO(16)$ and which spinorial weights are orthogonal to the given vector of fluxes: These roots and weights will build the root system of the preserved symmetry.

Let us give a relevant example. Consider the vector of fluxes,

$$\mathcal{F} = (-1, -1, 0, 0, 0, 0, 0, 0). \tag{130}$$

The roots of $SO(16)$ $\pm \frac{1}{2}(1, -1, 0, 0, 0, 0, 0, 0)$ and $(0, 0, \cdots)$ (with ellipses standing for any possible choice) are orthogonal to $\mathcal{F}$. These roots span an $SU(2) \times SO(12)$ subgroup of $SO(16)$. The spinorial weights which are orthogonal to $\mathcal{F}$ are of the form $\pm \frac{1}{4}(1, -1, \cdots)$ with odd number of minuses. These form the representation $(\mathbf{2}, \mathbf{32})$ of $(SU(2), SO(12))$. Such spinorial weights, along with the Cartan generators, build the root system of $E_7 \times U(1)$. Thus the flux vector (130) preserves $E_7 \times U(1)$ subgroup of the $E_8$ symmetry of the 6d theory.

> Exercise: Show that the flux $\mathcal{F} = (-1, -1, 2, 0, 0, 0, 0, 0)$ preserves $E_6 \times SU(2) \times U(1)$ and $\mathcal{F} = (\frac{1}{2}, \frac{1}{2}, \frac{1}{2}, \frac{1}{2}, \frac{1}{2}, \frac{1}{2}, \frac{1}{2}, -\frac{1}{2})$ preserves $E_7 \times U(1)$. The latter vector of fluxes is equivalent by Weyl transformation to (130).

The roots of $SO(16)$ orthogonal to $\mathcal{F} = (-1, -1, 2, 0, 0, 0, 0, 0)$ are $\frac{1}{2}(0, 0, 0, \pm 1, 0, \cdots, \pm 1, \cdots)$ which span $SO(10)$, as well $\pm \frac{1}{2}(1, -1, 0, \cdots)$ which span $SU(2)$. The co-spinor weights orthogonal to $\mathcal{F}$ are $\pm \frac{1}{4}(1, 1, -1, \pm 1, \cdots)$ where in the brackets we have an even number of negative choices of $\pm$ signs. These weights form a $\mathbf{16}$ and a $\overline{\mathbf{16}}$ of $SO(10)$ charged oppositely under a $U(1)_a$ symmetry. The adjoint of the preserved symmetry thus is in the following representation of $(SU(2), SO(10))_{U(1)_a, U(1)_b}$,

$$(\mathbf{1}, \mathbf{1})_{0,0} + (\mathbf{1}, \mathbf{1})_{0,0} + (\mathbf{3}, \mathbf{1})_{0,0} + (\mathbf{1}, \mathbf{45})_{0,0} + (\mathbf{1}, \mathbf{16})_{+1,0} + (\mathbf{1}, \overline{\mathbf{16}})_{-1,0}, \tag{131}$$

where we added all the Cartan generator. The $(\mathbf{1}, \mathbf{1})_{0,0} + (\mathbf{1}, \mathbf{45})_{0,0} + (\mathbf{1}, \mathbf{16})_{+1,0} + (\mathbf{1}, \overline{\mathbf{16}})_{-1,0}$ builds the $\mathbf{78}$ representation of $E_6$ and thus the symmetry is $E_6 \times SU(2) \times U(1)$.

Let us look at $\mathcal{F} = (\frac{1}{2}, \frac{1}{2}, \frac{1}{2}, \frac{1}{2}, \frac{1}{2}, \frac{1}{2}, \frac{1}{2}, -\frac{1}{2})$. The roots orthogonal to it are $\pm \frac{1}{2}(1, -1, 0 \cdots, 0)$ and all permutations excluding the last element as well as $\pm \frac{1}{2}(1, 0 \cdots, 1)$ and all permutations excluding the last element. This spans the roots of $SU(8)$. The spinorial weights orthogonal to the flux are $\pm \frac{1}{4}(\pm 1, \pm 1, \cdots, 1)$ with three negative choices of $\pm$ signs. This builds the $2 \times 7!/(4!3!) = 70$ dimensional representation of $SU(8)$. The representations $\mathbf{63} + \mathbf{70}$ compose the adjoint representation of $E_7$. The symmetry preserved by the flux is thus $E_7 \times U(1)$. $\square$

In fact we claim that the vector of fluxes associated to the three punctured sphere is (130). To make this statement meaningful we need to make a choice of puncture which will determine the basis of $U(1)$ symmetries. However, since the three punctures appear rather symmetrically the statement above is true for any choice of one out of the three punctures of the trinion. The

flux is $-1$ for $U(1)$ symmetries coming from the baryonic components of the moment maps and the 0s to the mesonic ones. As we saw above this flux preserves $E_7 \times U(1)$ symmetry, consistently with our previous discussion.

Let us perform a consistency check of this claim. We assume that the flux is (130) when computed using the $M_x$ moment maps of (119) and show that the same also holds for $M_y$ (and by symmetry also for $M_z$). To show this we translate the fluxes into the $SU(6) \times U(1)_u \times U(1)_v \times U(1)_w$ basis,

$$6\mathcal{F}_u + 12\mathcal{F}_w = -1, \qquad 6\mathcal{F}_u + 12\mathcal{F}_v = -1, \qquad 4\mathcal{F}_u - 2\mathcal{F}_w - 2\mathcal{F}_v = 0, \tag{132}$$

which gives $\mathcal{F}_u = \mathcal{F}_v = \mathcal{F}_w = -\frac{1}{18}$. By the symmetry of this result it is immediate that (130) is also the flux computed with respect to the other two punctures.

With this understanding of the fluxes let us discuss closing a maximal puncture. A general idea already discussed in the previous Lecture is that given a puncture we can "close" it by turning on a vacuum expectation value to one of the components of the moment map operator. We might also need to introduce certain fields and superpotential terms to match the anomalies with the expectations from the theory associated to the surface without the puncture. Intuitively, the puncture is removed as the vacuum expectation value (vev) breaks the symmetry associated to the puncture.[36] In addition the flux associated with the surface we obtain after giving the vev will be shifted. The value by which we shift the vector of fluxes depends on the charges of the moment map component which obtains the vev. For the E-string the precise procedure of closing a puncture was derived [26]. We will not repeat the computation here but rather state the result.

Let us assume that the charges of the octet of moment map operators are $u_i x^{\pm 1}$ where the $u_i$ are combinations of fugacities for the Cartan of $E_8$ and $x$ is the Cartan of the puncture $SU(2)$ symmetry. Let us then label the components of the moment map operator charged $u_i x^{\pm 1}$ as $M_i^{\pm}$. We can give the vev to any component and for concreteness we will choose to turn on a vev to $M_1^+$. Following the derivation in [26] we will need also to introduce chiral superfields, which we will denote by $F_i$, and couple them with the following superpotential,

$$W = M_1^- F_1 + \sum_{i=2}^{8} M_i^+ F_i. \tag{133}$$

A natural choice for the basis of $U(1)$ symmetries is in terms of the moment maps of the puncture being closed. In this basis the flux of the theory in the IR is shifted,

$$\Delta\mathcal{F} = (2, 0, 0, 0, 0, 0, 0, 0).$$

This vector of shifts is determined by (in fact proportional to) the vector of charges of the operator which received the vev.

Let us apply this procedure choosing different components of the moment maps. We have two different types of components, baryonic and mesonic, and we will discuss them separately. First, we close a puncture giving a vev to a baryonic component of the moment map. Following our rules above the flux of the resulting theory is,

$$\mathcal{F} + \Delta\mathcal{F}_B = (1, -1, 0, 0, 0, 0, 0, 0).$$

This in fact is equivalent to the flux in (130) by action of the Weyl group of $E_8$ and thus preserves the same symmetry. However, if we turn on a vev to one of the mesonic compenents of the moment map and follow the same rules we obtain,

$$\mathcal{F} + \Delta\mathcal{F}_M = (-1, -1, 2, 0, 0, 0, 0, 0).$$

---

[36]More precisely it breaks one combination of $U(1)$ symmetries, the Cartan generator of the puncture symmetries and symmetries coming from six dimensions. We can always choose to parametrize the broken symmetry by the one associated with the puncture.

As we have seen this flux preserves an $E_6 \times SU(2) \times U(1)$ subgroup of $E_8$.

Given the above procedure, since we have field theoretic constructions of the trinions, we can explicitly study the procedure of closing punctures. For example, a baryonic vev will Higgs completely the $SU(3)$ gauge symmetry of the trinion. We will be left with a WZ model in the IR. On the other hand, the mesonic vev Higgses the $SU(3)$ gauge symmetry of the trinion only to $SU(2)$. See Figure 41 for an illustration. We will soon discuss this case in detail in an exercise.

Before proceeding with the detailed discussion of closing punctures let us make an additional comment. Given a puncture with an octet of moment maps $M_i$ we can introduce for example a single chiral field $\Phi$ and flip one of the components, say by ontroducing the superpotential,

$$W = \Phi \cdot M_1. \tag{134}$$

This procedure will remove $M_1$ from the chiral ring and replace it with $\Phi$. Importantly the charges of $\Phi$, which is the new moment map component, are opposite to $M_1$. This flipping of the moment map component does not alter the flux associated to the surface: It just changes the color of puncture. Here let us comment again on the choice of spinor versus co-spinor to compute the flux. Flipping even number of components of the moment map indeed is consistent with our prescription to determine the flux as it is part of the Weyl symmetry of $SO(16)$. However, flipping odd number of components does naively change the flux. This can be remedied by declaring that if we flip odd number of components we compute the flux using the spinor and not the co-spinor. Notice that the $SU(2)$ punctures of the trinion theory have odd number of fundamentals and thus have a Witten anomaly [67]. Flipping even number of moment maps will keep the anomaly while flipping odd number will remove it. Thus we can just say that if we compute the flux relative to puncture symmetry with Witten anomaly we use the co-spinor and if the puncture has no Witten anomaly we compute the flux with the spinor.

Let us now analyze what happens if we close punctures with a flip. Say we flip a mesonic component. Then the vector of fluxes is unchanged as $\mathcal{F}$ of (130) has zeros at the location of the mesonic components. Turning on the vev, the flux is shifted to,

$$\mathcal{F} + \Delta \mathcal{F}_M = (-1, -1, 2, 0, 0, 0, 0, 0),$$

which is the same $E_6 \times SU(2) \times U(1)$ preserving flux as we obtained without flipping. Note however that the vev here is turned on to the flip field which is a singlet under the gauge symmetry. Thus, we do not have any Higgsing but rather generate various mass terms for gauge non-invariant fields. We will then obtain again an $SU(3)$ SQCD albeit with smaller number of flavors, namely $N_f = 5$ $SU(3)$ SQCD. This theory should be then related to the one obtained without the flipping (as the flux and the surface are the same). The latter is $SU(2)$ SQCD with $N_f = 5$, see Figure 41. The two theories are indeed related in a non-trivial way due to a Seiberg duality.

> Exercise: Closing a puncture by giving a vev to a mesonic moment map, with and without flip, derive the theories with fluxes $\mathcal{F} = (-1, -1, +2, 0, 0, 0, 0, 0)$ and $\mathcal{F} = (-1, -1, -2, 0, 0, 0, 0, 0)$. Applying Seiberg duality to the theory with the flip compare the two models.

We will perform the analysis of the flows triggered by vevs using the supersymmetric index. The basic observation [119] is as follows. Given an operator $\mathcal{O}$ in a 4d theory that contributes to the index with weight $X \equiv q^{-j_1^{(\mathcal{O})} + j_2^{(\mathcal{O})} + \frac{1}{2} R^{(\mathcal{O})}} p^{j_1^{(\mathcal{O})} + j_2^{(\mathcal{O})} + \frac{1}{2} R^{(\mathcal{O})}} \prod_i u_i^{\mathfrak{Q}_i^{(\mathcal{O})}}$ and $\mathcal{O}$ can acquire a vev, the index will have a pole when we tune the various fugacities so that $X = 1$. The residue

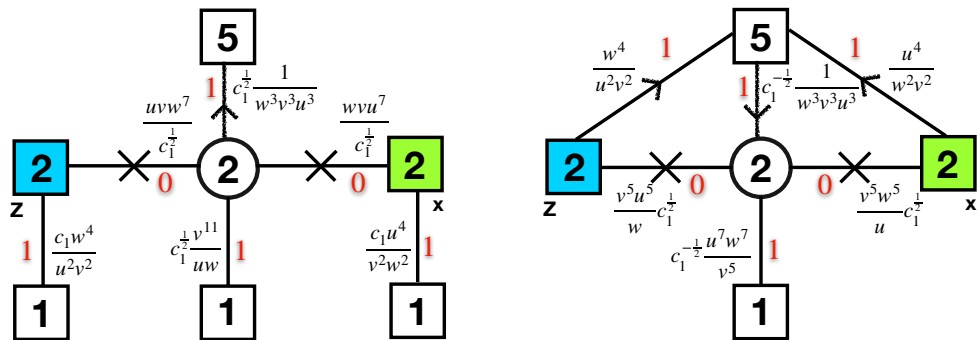

Figure 41: On the left we have the model obtained by closing a puncture with a mesonic vev with no flip. On the right we have the model obtained by closing a puncture with a mesonic vev with a flip. Both theories correspond to a tube with flux breaking $E_8$ to $E_6 \times SU(2) \times U(1)$. The crosses are flip fields flipping the baryons built from the bi-fundamentals. The Figure is taken from [25].

of the pole, removing the Goldstones for the symmetry broken by the vev, is the index of the theory obtained in the IR. Let us apply this for the theories at hand.

The supersymmetric index of the trinion is,

$$I^{(trinion)}(x,y,z) = \frac{(q;q)^2(p;p)^2}{6} \oint \frac{dt_1}{2\pi i t_1} \frac{dt_2}{2\pi i t_2} \frac{1}{\prod_{i \neq j}^{3} \Gamma_e(t_i/t_j)} \tag{135}$$

$$\times \prod_{i=1}^{3} \Gamma_e((qp)^{\frac{1}{6}} u^6 t_i x^{\pm 1}) \Gamma_e((qp)^{\frac{1}{6}} v^6 t_i y^{\pm 1}) \Gamma_e\left((qp)^{\frac{1}{6}} w^6 t_i z^{\pm 1}\right) \prod_{j=1}^{6} \Gamma_e\left((qp)^{\frac{1}{3}} (uvw)^{-2} c_j t_i^{-1}\right).$$

Here $c_i$ parametrize the $SU(6)$ symmetry group ($\prod_{j=1}^{6} c_j = 1$). Let us close the puncture symmetry of which is parameterized by $y$. First we give a vev to the moment map (denote it as $M_3^+$) with weight $X^+ = (qp)^{\frac{1}{2}} \frac{v^4}{w^2 u^2} c_1 y^{+1}$. Note that the integrand of (135) does not have a pole when we tune $X^+$ to be 1. However, the integration contour is pinched leading to a divergence. Let us look at the following poles of the integrand in $t_i$,

$$\text{Outside: } t_i = (qp)^{-\frac{1}{6}} v^{-6} y^{\pm 1},$$
$$\text{Inside: } \quad t_i = (qp)^{\frac{1}{3}} (uvw)^{-2} c_1.$$

Outside/Inside stand for poles outside/inside the unit circle integration contour. Here we assume first that the fugacities for flavor symmetries are on the unit circle and $|q|, |p| < 1$. This determines which poles are inside the unit circle contour for $t_i$ integration variables and which are outside. Setting now $X^+ = 1$ we can take $y = (qp)^{-\frac{1}{2}} \frac{w^2 u^2}{v^4} a_1^{-1}$. Then we have,

$$\text{Outside: } t_i = (qp)^{\frac{1}{3}} (uvw)^{-2} c_1, \quad t_i = (qp)^{-\frac{2}{3}} (uw)^2 v^{-10} c_1^{-1},$$
$$\text{Intside: } t_i = (qp)^{\frac{1}{3}} (uvw)^{-2} c_1.$$

Note then that the poles in say $t_1$ at $t_1 = (qp)^{-\frac{1}{6}} v^{-6} y^{-1}$ and $t_1 = (qp)^{\frac{1}{3}} (uvw)^{-2} c_1$ collide from both sides of the $t_1$ contour pinching it and leading to a divergence. We can consider poles in $t_2$ or $t_3$ which will be related to the above by the action of Weyl symmetry giving an equivalent result. Let us then plug the value $X^+ = 1$ and $t_1 = (qp)^{\frac{1}{3}} (uvw)^{-2} c_1$ in the integrand of (135)

SciPost Phys. Lect. Notes 78 (2024)

to compute the residue,

$$(q;q)(p;p) \times \text{Res}_{X^+ \to 1} I^{(trinion)}(x,y,z) \to \frac{(q;q)(p;p)}{2} \oint \frac{dt}{2\pi it} \frac{1}{\Gamma_e(t^{\pm 2})} \tag{136}$$

$$\times \Gamma_e\left(u^7 vw c_1^{-\frac{1}{2}} t^{\pm 1} x^{\pm 1}\right) \Gamma_e\left(uvw^7 c_1^{-\frac{1}{2}} t^{\pm 1} z^{\pm 1}\right) \Gamma_e\left((qp)^{\frac{1}{2}}(uw)^{-1} v^{11} c_1^{\frac{1}{2}} t^{\pm 1}\right)$$

$$\times \prod_{j=2}^{6} \Gamma_e\left((qp)^{\frac{1}{2}}(uvw)^{-3} c_j c_1^{\frac{1}{2}} t^{\pm 1}\right) \left[ \Gamma_e\left(qp \frac{v^8}{u^4 w^4} c_1^2\right) \prod_{j=2}^{6} \Gamma_e(c_j/c_1) \right] \Gamma_e\left((qp)^{\frac{1}{2}} \frac{u^4}{v^2 w^2} c_1 x^{\pm 1}\right)$$

$$\times \Gamma_e\left((qp)^{\frac{1}{2}} \frac{w^4}{v^2 u^2} c_1 z^{\pm 1}\right).$$

Here we have defined $t = t_2(qp)^{\frac{1}{6}}(uvw)^{-1} c_1^{\frac{1}{2}}$. The multiplication of the residue by $(q;q)(p;p)$ is to remove the Goldstones. Now, notice that the contribution of the last line in the square brackets of (136) is from moment maps $M_3^-$ and the mesonic components of $M_j^+$ ($j = 4\cdots 8$) which are removed by adding additional terms to the superpotential (133) when closing the puncture. We also add flip fields for the two baryonic moment maps $M_{1,2}^+$ and thus the index of the theory we obtain after closing a puncture with the mesonic vev is,

$$I_1^{(tube)}(x,z) = \frac{(q;q)(p;p)}{2} \oint \frac{dt}{2\pi it} \frac{1}{\Gamma_e(t^{\pm 2})} \Gamma_e\left(u^7 vw c_1^{-\frac{1}{2}} t^{\pm 1} x^{\pm 1}\right) \Gamma_e\left(uvw^7 c_1^{-\frac{1}{2}} t^{\pm 1} z^{\pm 1}\right)$$

$$\times \Gamma_e\left((qp)^{\frac{1}{2}}(uw)^{-1} v^{11} c_1^{\frac{1}{2}} t^{\pm 1}\right) \prod_{j=2}^{6} \Gamma_e\left((qp)^{\frac{1}{2}}(uvw)^{-3} c_j c_1^{\frac{1}{2}} t^{\pm 1}\right) \tag{137}$$

$$\times \Gamma_e\left(qp \frac{1}{u^4 v^4 w^{14}} c_1\right) \Gamma_e\left(qp \frac{1}{w^4 v^4 u^{14}} c_1\right) \Gamma_e\left((qp)^{\frac{1}{2}} \frac{u^4}{v^2 w^2} c_1 x^{\pm 1}\right) \Gamma_e\left((qp)^{\frac{1}{2}} \frac{w^4}{v^2 u^2} c_1 z^{\pm 1}\right).$$

This is an $SU(2)$ gauge theory with $N_f = 5$ with additional gauge singlet fields and a superpotential. The quiver is depicted in Figure 41. The flux of this model in terms of the moment map symmetries of the puncture we close is $\mathcal{F} = (-1,-1,2,0,0,0,0,0)$.

Next we can first flip the moment maps $M_3^\pm$ by adding a field $\widetilde{M}_3^\pm$ with the superpotential

$$W = M_3^\pm \cdot \widetilde{M}_3^\mp,$$

and then close the puncture by giving a vev to $\widetilde{M}_3^+$. This operation simply amounts to turning on a mass term to the quarks building $M_3^-$. The resulting pole in the index comes solely from the contribution of $\widetilde{M}_3^+$ which is $\tilde{X}^+ = (qp)^{\frac{1}{2}} \frac{w^2 u^2}{v^4} c_1^{-1} y^{+1}$. The index of the theory in the IR is then just the one obtained by taking (135), setting $\tilde{X}^+ = 1$ and adding the flip fields determined by (133),

$$I_2^{(tube)}(x,z) = \frac{(q;q)^2(p;p)^2}{6} \oint \frac{dt_1}{2\pi it_1} \frac{dt_2}{2\pi it_2} \frac{1}{\prod_{i \neq j}^{3} \Gamma_e(t_i/t_j)} \prod_{i=1}^{3} \Gamma_e((qp)^{\frac{1}{6}} u^6 t_i x^{\pm 1})$$

$$\times \Gamma_e\left((qp)^{-\frac{1}{3}} \frac{v^{10}}{u^2 w^2} t_i c_1\right) \Gamma_e\left((qp)^{\frac{1}{6}} w^6 t_i z^{\pm 1}\right) \prod_{j=2}^{6} \Gamma_e\left((qp)^{\frac{1}{3}}(uvw)^{-2} c_j t_i^{-1}\right) \tag{138}$$

$$\times \Gamma_e\left((qp) \frac{w^2}{v^{10} u^{10}} c_1^{-1}\right) \Gamma_e\left((qp) \frac{u^2}{v^{10} w^{10}} c_1^{-1}\right) \prod_{j=2}^{6} \Gamma_e\left((qp) \frac{u^4 w^4}{v^8} c_j^{-1} c_1^{-1}\right).$$

This is an $SU(3)$ gauge theory with $N_f = 5$ with additional gauge singlet fields and a superpotential. The flux of this model in terms of the moment map symmetries of the puncture we



close is $\mathcal{F} = (-1,-1,-2,0,0,0,0,0)$. Let us perform Seiberg duality for this theory. The dual theory of $SU(3)$ with $N_f = 5$ is an $SU(2)$ with $N_f = 5$ with the mesons flipped. The baryonic symmetry is identified between the two duality frames while the mesonic symmetries are conjugated. The resulting index is,

$$
I_2^{(tube)}(x,z) = \frac{(q;q)(p;p)}{2} \oint \frac{dt}{2\pi i t} \frac{1}{\Gamma_e(t^{\pm 2})} \Gamma_e\left(\frac{v^5 w^5}{u} c_1^{\frac{1}{2}} t^{\pm 1} x^{\pm 1}\right) \Gamma_e\left((qp)^{\frac{1}{2}} \frac{u^7 w^7}{v^5} t^{\pm 1} c_1^{-\frac{1}{2}}\right)
$$
$$
\times \Gamma_e\left(\frac{v^5 u^5}{w} c_1^{\frac{1}{2}} t^{\pm 1} z^{\pm 1}\right) \prod_{j=2}^{6} \Gamma_e\left((qp)^{\frac{1}{2}}(uvw)^{-3} c_j^{-1} c_1^{-\frac{1}{2}} t^{\pm 1}\right) \Gamma_e\left((qp)\frac{w^2}{v^{10}u^{10}} c_1^{-1}\right)
$$
(139)
$$
\times \Gamma_e\left((qp)\frac{u^2}{v^{10}w^{10}} c_1^{-1}\right) \prod_{j=2}^{6} \Gamma_e\left((qp)^{\frac{1}{2}} \frac{u^4}{w^2 v^2} c_j x^{\pm 1}\right) \Gamma_e\left((qp)^{\frac{1}{2}} \frac{w^4}{u^2 v^2} c_j z^{\pm 1}\right).
$$

Note that although $I_1^{(tube)}$ and $I_2^{(tube)}$ are very similar they are not exactly the same. They have exactly the same types of punctures as both are obtained by closing a puncture of a different theory in different ways. However let us compute the flux in terms of the symmetries of say the $x$ puncture. In the basis of the $y$ puncture we started from, as we mentioned above, the flux is $\mathcal{F} = (-1,-1,2,0,0,0,0,0)$. Let us compute this flux in terms of the basis $\{u,v,w,c_i\}$,

$$6\mathcal{F}_v + 12\mathcal{F}_u = -1, \quad 6\mathcal{F}_v + 12\mathcal{F}_w = -1, \tag{140}$$
$$4\mathcal{F}_v - 2\mathcal{F}_u - 2\mathcal{F}_w + \mathcal{F}_{c_1} = 2, \quad 4\mathcal{F}_v - 2\mathcal{F}_u - 2\mathcal{F}_w - \frac{1}{5}\mathcal{F}_{c_1} = 0.$$

This gives us

$$\{\mathcal{F}_u, \mathcal{F}_v, \mathcal{F}_w, \mathcal{F}_{c_1}\} = \left\{-\frac{1}{12}, 0, -\frac{1}{12}, \frac{5}{3}\right\}. \tag{141}$$

From here the flux in terms of the moment maps of the $x$ puncture (119) is,

$$\mathcal{F}_1 = \left(-\frac{1}{2}, -\frac{3}{2}, \frac{3}{2}, -\frac{1}{2}, -\frac{1}{2}, -\frac{1}{2}, -\frac{1}{2}, -\frac{1}{2}\right). \tag{142}$$

Let us repeat now the exercise with flux $\mathcal{F} = (-1,-1,-2,0,0,0,0,0)$,

$$6\mathcal{F}_v + 12\mathcal{F}_u = -1, \qquad 6\mathcal{F}_v + 12\mathcal{F}_w = -1, \tag{143}$$
$$4\mathcal{F}_v - 2\mathcal{F}_u - 2\mathcal{F}_w + \mathcal{F}_{c_1} = -2, \qquad 4\mathcal{F}_v - 2\mathcal{F}_u - 2\mathcal{F}_w - \frac{1}{5}\mathcal{F}_{c_1} = 0.$$

This gives us

$$\{\mathcal{F}_u, \mathcal{F}_v, \mathcal{F}_w, \mathcal{F}_{c_1}\} = \left\{-\frac{1}{36}, -\frac{1}{9}, -\frac{1}{36}, -\frac{5}{3}\right\}. \tag{144}$$

From here the flux in terms of the moment maps of the $x$ puncture (119) is,

$$\mathcal{F}_2 = \left(-\frac{3}{2}, -\frac{1}{2}, -\frac{3}{2}, \frac{1}{2}, \frac{1}{2}, \frac{1}{2}, \frac{1}{2}, \frac{1}{2}\right). \tag{145}$$

Note that although the punctures are the same the fluxes $\mathcal{F}_1$ and $\mathcal{F}_2$ are different. Both fluxes preserve $E_6 \times SU(2) \times U(1)$ and are related by Weyl transformation exchanging the two baryonic symmetries and flipping the signs of the mesonic symmetries.[37] However there is no

---

[37]The fact that $E_6 \times SU(2) \times U(1)$ is preserved follows from $(-\frac{3}{2},-\frac{1}{2},-\frac{3}{2},\frac{1}{2},\frac{1}{2},\frac{1}{2},\frac{1}{2},\frac{1}{2})$ being obtained from $(-1,-1,-2,0,0,0,0,0)$ by change of basis, while for the latter we established the symmetry already. It is though instructive to rederive this fact. Taking the roots of $SO(16)$ the former flux preserves explicitly $SU(2)\times SU(6)\times U(1)^2$. The co-spinorial weights enhance one of the $U(1)$s to $SU(2)$ while also giving $(\mathbf{2},\mathbf{20})$ of $(SU(2),SU(6))$ enhancing it to $E_6$.

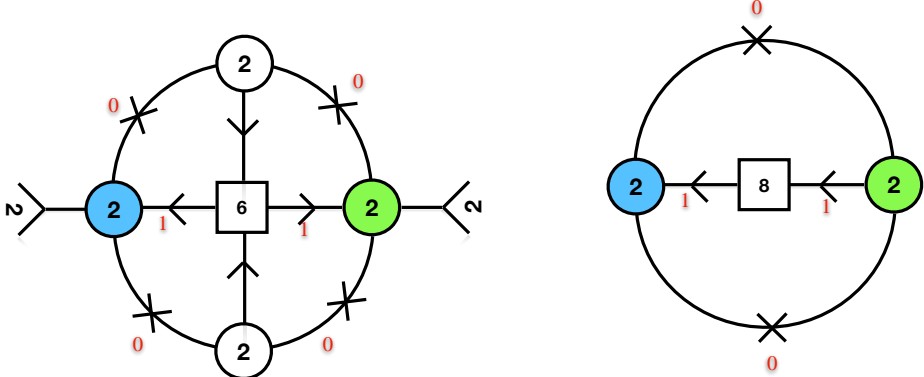

Figure 42: On the left we have the theory resulting in $\Phi$-gluing together either two copies of first or the second tube. The quiver can be thought of as drawn on a sphere with the two half-squares identifies as the same $SU(2)$ symmetry on a pole of that sphere. This theory is associated to the $E_6 \times SU(2)$ preserving flux. On the left we $\Phi$-glued first and second tube to obtain a theory preserving $E_7 \times U(1)$ flux. The explicit quiver drawn is obtained by using Seiberg duality relating $SU(2)$ with three flavors to a WZ model.

redefinition of symmetries which will make both punctures and fluxes of the two theories the same. $\Phi$-gluing two copies of each one of the tube theories into a torus, and thus getting rid of the punctures, we will obtain two completely equivalent models preserving $E_6 \times SU(2) \times U(1)$ symmetry (related by simple redefinitions of symmetries), as is expected. Summing the fluxes, $\mathcal{F}_1 + \mathcal{F}_2 = (-2, -2, 0, 0, 0, 0, 0, 0)$ we obtain $E_7 \times U(1)$ preserving vector and thus $\Phi$-gluing tube one to tube two we will obtain a different theory preserving this symmetry. See Figure 42 and we will discuss these models in more detail in the next Lecture.

Let us see explicitly how the gluing of the two different tubes works. The index of the $\Phi$-glued theory is,

$$
\mathcal{I}_{E_7\,torus} = \left[\frac{(q;q)(p;p)}{2}\right]^2 \oint \frac{dx}{2\pi i x} \frac{1}{\Gamma_e(x^{\pm 2})} \oint \frac{dz}{2\pi i z} \frac{1}{\Gamma_e(z^{\pm 2})} I_1^{(tube)}(x,z) I_2^{(tube)}(x,z)
$$

$$
\times \left( \Gamma_e\left((qp)^{\frac{1}{2}} u^{-6} v^{-12} x^{\pm 1}\right) \Gamma_e\left((qp)^{\frac{1}{2}} u^{-6} w^{-12} x^{\pm 1}\right) \prod_{i=1}^{6} \Gamma_e\left((qp)^{\frac{1}{2}} \frac{v^2 w^2}{u^4} c_i^{-1} x^{\pm 1}\right)\right) \quad (146)
$$

$$
\times \left( \Gamma_e((qp)^{\frac{1}{2}} w^{-6} v^{-12} z^{\pm 1}) \Gamma_e\left((qp)^{\frac{1}{2}} w^{-6} u^{-12} z^{\pm 1}\right) \prod_{i=1}^{6} \Gamma_e\left((qp)^{\frac{1}{2}} \frac{v^2 u^2}{w^4} c_i^{-1} z^{\pm 1}\right)\right).
$$

The contributions from the second and third lines are for the $\Phi$ fields introduced in gluing the two punctures. Let us first perform the $x$ integral. Collecting all the fields and using (60) identity (index of a couple of fields which can form a mass term is equal to 1) we get

$$
\mathcal{I}_x = \frac{(q;q)(p;p)}{2} \oint \frac{dx}{2\pi i x} \quad (147)
$$

$$
\times \frac{\Gamma_e\left(u^7 v w c_1^{-\frac{1}{2}} t^{\pm 1} x^{\pm 1}\right) \Gamma_e(\frac{v^5 w^5}{u} c_1^{\frac{1}{2}} \tilde{t}^{\pm 1} x^{\pm 1}) \Gamma_e\left((qp)^{\frac{1}{2}} u^{-6} v^{-12} x^{\pm 1}\right) \Gamma_e\left((qp)^{\frac{1}{2}} u^{-6} w^{-12} x^{\pm 1}\right)}{\Gamma_e(x^{\pm 2})}.
$$

Here we distinguish the fugacities of the $SU(2)$ gauge groups of the tube theories by denoting one as $t$ and another as $\tilde{t}$. This is an $SU(2)$ SQCD with $N_f = 3$ and thus is dual using Seiberg duality to a WZ model of the gauge invariant mesons and baryons (62),

$$
\mathcal{I}_x = \Gamma_e(u^6 v^6 w^6 t^{\pm 1} \tilde{t}^{\pm 1}) \Gamma_e\left((qp)^{\frac{1}{2}} \frac{uw}{v^{11}} c_1^{-\frac{1}{2}} t^{\pm 1}\right) \Gamma_e\left((qp)^{\frac{1}{2}} \frac{uv}{w^{11}} c_1^{-\frac{1}{2}} t^{\pm 1}\right) \Gamma_e\left((qp)^{\frac{1}{2}} \frac{w^5}{u^7 v^7} c_1^{\frac{1}{2}} \tilde{t}^{\pm 1}\right)
$$
$$
\times \Gamma_e\left((qp)^{\frac{1}{2}} \frac{v^5}{u^7 w^7} c_1^{\frac{1}{2}} \tilde{t}^{\pm 1}\right) \Gamma_e\left(u^{14} v^2 w^2 c_1^{-1}\right) \Gamma_e\left(\frac{v^{10} w^{10}}{w^2} c_1\right) \Gamma_e\left(qp u^{-12} v^{-12} w^{-12}\right).
$$
$$(148)$$

Performing the $z$ integration we obtain the same expression but with $w$ and $u$ exchanged. Combining everything together we obtain the following index,

$$
\mathcal{I}_{E_7 \, torus} = \left[\frac{(q;q)(p;p)}{2}\right]^2 \oint \frac{dt}{2\pi i t} \frac{1}{\Gamma_e(t^{\pm 2})} \oint \frac{d\tilde{t}}{2\pi i \tilde{t}} \frac{1}{\Gamma_e(\tilde{t}^{\pm 2})}
$$
$$
\times \left(\Gamma_e(u^6 v^6 w^6 t^{\pm 1} \tilde{t}^{\pm 1}) \Gamma_e\left(qp u^{-12} v^{-12} w^{-12}\right)\right)^2 \Gamma_e\left((qp)^{\frac{1}{2}} \frac{wu}{v^{11}} c_1^{-\frac{1}{2}} t^{\pm 1}\right) \Gamma_e\left((qp)^{\frac{1}{2}} \frac{vw}{u^{11}} c_1^{-\frac{1}{2}} t^{\pm 1}\right)
$$
$$
\times \Gamma_e\left((qp)^{\frac{1}{2}} \frac{uv}{w^{11}} c_1^{-\frac{1}{2}} t^{\pm 1}\right) \prod_{j=2}^{6} \Gamma_e\left((qp)^{\frac{1}{2}} (uvw)^{-3} c_j c_1^{\frac{1}{2}} t^{\pm 1}\right) \Gamma_e\left((qp)^{\frac{1}{2}} \frac{v^5}{u^7 w^7} c_1^{\frac{1}{2}} \tilde{t}^{\pm 1}\right)
$$
$$
\times \Gamma_e\left((qp)^{\frac{1}{2}} \frac{u^5}{w^7 v^7} c_1^{\frac{1}{2}} \tilde{t}^{\pm 1}\right) \Gamma_e\left((qp)^{\frac{1}{2}} \frac{w^5}{u^7 v^7} c_1^{\frac{1}{2}} \tilde{t}^{\pm 1}\right) \prod_{j=2}^{6} \Gamma_e((qp)^{\frac{1}{2}} (uvw)^{-3} c_j^{-1} c_1^{-\frac{1}{2}} \tilde{t}^{\pm 1}).
$$

The quiver description of this theory is depicted on the right of Figure 42. On the first line we have the bifundamentals of the two $SU(2)$ gauge groups and the flip fields (denoted by crosses on the figure), the second line give an octet of fundamentals under one $SU(2)$ and the last line an octet under the other $SU(2)$ group. Computing the index of this theory one can observe that the protected states form representations of $E_7 \times U(1)$ with the $U(1)$ parametrized by $u^6 v^6 w^6$. For example, taking antisymmetric squares of the octets and building invariants we can obtain operators in $\mathbf{28} + \overline{\mathbf{28}}$ which build the fundamental representation $\mathbf{56}$ of $E_7$. Let us now consider simplifying this theory by deforming it. Consider giving a vev to the flip fields denoted by crosses in the quiver. These are not charged under the $SU(8)$ and thus we do not break the $E_7$ structure. These vevs give mass to the bifundamental fields and we remain with two copies of $SU(2)$ $N_f = 8$ SQCD coupled together through a superpotential $W = M_1 \times M_2$ with $M_i$ being the 28 mesons/baryons of each copy. This theory was discussed in detail in [71] and dubbed the *the $E_7$ surprise* as it was shown to exhibit $E_7$ symmetry. The discussion here shows this surprise in a geometric context.[38]

Let us now consider taking one of the $E_6 \times SU(2) \times U(1)$ tubes and $\Phi$-gluing the two punctures to each other. The two punctures are of different types and thus we will break a part of the symmetry doing so.[39] In $\Phi$-gluing the moment map symmetries of the punctures are identified through the superpotential. The moment map symmetries of our two punctures differ by exchanging $u$ and $w$ and thus these two symmetries are broken to a diagonal one. The broken $u/w$ parametrizes the $SU(2)$ part of the $E_6 \times SU(2) \times U(1)$ symmetry. To see that let us discuss how to derive the character of the adjoint of the preserved symmetry from

---

[38]The more precise statement is that on some locus of the IR conformal manifold of this theory the $SU(8)$ symmetry enhances to $E_7$. In fact it was argued in [72] that on that locus (point) the symmetry is actually $E_7 \times U(1)$. The $U(1)$ is accidental in our discussion.

[39]This has an interesting interpretation in 6d. The flux associated with this tube is not properly quantized, and so to be consistent must be accompanied by certain non-trivial connections in the non-abelian flavor symmetry, here in the $SU(2)$ part. These lead to the breaking of this symmetry once the tube is closed to a torus, see [26, app. C].

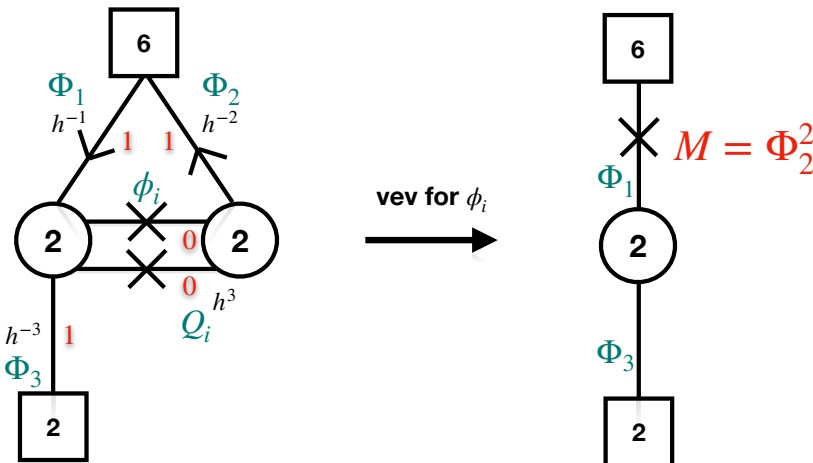

Figure 43: On the left the theory obtained by gluing two punctures of the $E_6 \times SU(2) \times U(1)$ tube together. On the right we give vev to the flip fields and use Seiberg duality of $SU(2)$ with $N_f = 3$ to obtain a simple theory with $E_6$ symmetry. The cross is a field flipping the mesons/baryons in the $\overline{\mathbf{15}}$ of $SU(6)$ built from the bifundamental field.

the vector of fluxes. Let us take the weight of the moment maps of the puncture in terms of which we compute the flux to be $\{a_i\}_{i=1}^8$. These can be thought of as fugacities for the Cartan of $SO(16)$ subgroup of $E_8$. For concreteness let us take the flux (145). The components $a_2^{-1}+a_4+a_5+a_6+a_7+a_8$ is the character of the preserved $U(6) = U(1)_a \times SU(6)$ while $a_1^{-1}+a_3^{-1}$ is the character of preserved $U(2) = U(1)_b \times SU(2)$. The $E_6$ is built from $SU(2) \times SU(6)$. One combination of the $U(1)_{a/b}$ enhances to $SU(2)$. Looking at the co-spinorial weights preserved by the flux, $(a_1 a_3/(a_2^{-1} \prod_{i=4}^8 a_i))^{\frac{1}{2}}$ parametrizes the $U(1)$ while $z = (a_1 a_3(a_2^{-1} \prod_{i=4}^8 a_i))^{\frac{1}{2}}$ the preserved $SU(2)$. Plugging in the values of the $a_i$ for the moment maps of the $x$ puncture we obtain that $z = (w/u)^{12}$ while all the other characters are independent of $u/w$. The resulting theory is depicted in Figure 43. The quiver has explicitly $SU(6) \times SU(2) \times U(1)_h$ symmetry with the $SU(6) \times SU(2)$ expected to enhance to $E_6$. The superpotential of this theory is,

$$W = Q_1 Q_2 \Phi_3^2 + \Phi_1 \Phi_2 Q_1 + \Phi_1 \Phi_2 Q_2 + Q_1^2 \phi_1 + Q_2^2 \phi_2. \tag{149}$$

We can derive the superconformal R-symmetry by parametrizing the trial R-symmettry as $R + \mathfrak{h} q_h$ to compute,

$$a(\mathfrak{h}) = \frac{27}{2} \mathfrak{h} (1 - 9\mathfrak{h}^2), \tag{150}$$

which is minimized for $\mathfrak{h} = \frac{1}{3\sqrt{3}}$. We note that all the operators are above the unitarity bound evaluating their R-charges using this R-symmetry. Let us write the supersymmetric index. We will use the more convenient rational value of $\mathfrak{h} = 2/9$ to write the expression, which is a good approximation to the superconformal one:

$$\mathcal{I}_{E_6 \, torus} = 1 + 3h^{-6}(qp)^{\frac{1}{3}} + \mathbf{27}_{E_6} h^{-4}(qp)^{\frac{5}{9}} + (6h^{-12} + h^6)(qp)^{\frac{2}{3}} \tag{151}$$

$$+ \overline{\mathbf{27}}_{E_6} h^{-2}(qp)^{\frac{7}{9}} + 3h^{-6}(q+p)(qp)^{\frac{1}{3}} + 3 \times \mathbf{27}_{E_6} h^{-10}(qp)^{\frac{8}{9}} + 10h^{-18}(qp) + \cdots.$$

Here we have defined,

$$\mathbf{27}_{E_6} = \mathbf{6}_{SU(6)} \times \mathbf{2}_{SU(2)} + \overline{\mathbf{15}}_{SU(6)}. \tag{152}$$

First we see that the states form representations of $E_6$. Second, we notice that at order $q\,p$ using the superconformal R-symmetry we have 0.[40] This implies in fact that the number of exactly marginal deformations is 7. This follows as we have explicitly preserved $SU(6)\times SU(2)\times U(1)$ symmetry which would contribute corresponding currents with negative signs at this order (see discussion around (56)). To obtain zero we need to have marginal operator in the adjoint of this group. Such single marginal operator will contribute exactly marginal operators number of which is the rank of the group (computing the relevant Kähler quotient this follows immediately). Moreover assuming that somewhere on that conformal manifold the symmetry enhances to $E_6\times U(1)$ we will have to have a marginal operator in the adjoint of this group which again would give a 7 dimensional conformal manifold: Consistently with the above.

Let us now try to simplify the theory of Figure 43 by deforming it but preserving the $E_6$ symmetry. One way to do so is to give a vev to the flip fields $\phi_i$. These fields are only charged under $U(1)_h$ and thus the deformation should preserve the $E_6$ symmetry. Doing so the bifundamental fields $Q_i$ acquire a mass and decouple in the IR. Note also that on a general point of the conformal manifold we would also expect the fields $\Phi_3$ to acquire a vev. One way to see this is by noting that giving a vev to $\phi_i$ we are demanding that $h^{-6}qp=1$. However the weight of the field $\Phi_3$ is $h^{-3}(qp)^{\frac{1}{2}}$ and thus it will be also weighed as $+1$ in this limit. The fact that this vev can be generated also follows from the exactly marginal superpotential term $Q_1Q_2\Phi_3^2$ that we have here: This superpotential breaks no symmetry. Let us assume that however the vev for $\Phi_3$ is not generated (say when the above quartic superpotential is not turned on). Then, as $Q_i$ acquire mass the $SU(2)$ gauge group on the left has only six fundamentals in the IR and thus is dual to a WZ model of the gauge invariants $M=\Phi_2^2$. These fields couple to the rest through $W=\Phi_1^2M+M^3$ which preserves the $SU(6)$ symmetry (and also the $E_6$). The resulting quiver is depicted on the right of Figure 43. Note that this is the theory we analyzed in Section 3.2 as an example of emergence of $E_6$ symmetry. The two models differ by the superpotential term $M^3$ which does not break the $E_6$ symmetry as it is a singlet of $SU(6)$ and $SU(2)$: Moreover we expect this superpotential term to be irrelevant and vanish in the deep IR.[41] Yet again thus we have derived geometrically an instance of emergence of IR symmetry in 4d. Finally if we do generate a vev for $\Phi_3$ this will just lead to a WZ model of fields $\Phi_1$ (bifundamental of $SU(2)\times SU(6)$) and $M$ ($\overline{\mathbf{15}}$ of $SU(6)$) with superpotential $W=\Phi_1^2M+M^3$, which again preserves $E_6$. The superpotential is marginally irrelevant in the IR but if we compactify to 3d it would be relevant leading to an SCFT with $E_6$ global symmetry [133, 134].

$\square$

Finally, let us comment that the fact that we can interpret the $SU(3)$ SQCD in the middle of the conformal window as a trinion of the E-string theory gives us a geometric way to understand its Seiberg (self)duality [2]. See [25] for details. We will see in the next section that one can also find a geometric meaning for $SU(N+2)$ SQCD in the middle of the conformal window: The case of $N=1$ is the E-string theory discussed here, while higher $N$ cases correspond to generalizations to compactifications of the minimal $(D_{N+3}, D_{N+3})$ conformal matter. The bottom line is that the Seiberg duality of $SU(N+2)$ SQCD in the middle of conformal window can be directly related to the Weyl group of $SO(4N+12)$ [25].

---

[40]With $\mathfrak{h}=2/9$ we have at that order 10 states but these are charged under $U(1)_h$ and thus will move away once we stick to the superconformal R-symmetry.

[41]This can be argued for by first turning on the gauge coupling, then the $M\Phi_1^2$ superpotential and then $M^3$ superpotential.

# 6 Lecture V: An algorithm for deriving across dimensional dualities

In the previous Lectures, we discussed how one can try to conjecture various $\mathcal{N} = 1$ four dimensional field theories that result from compactifications of 6d SCFTs. The method used there was to determine the expected 't Hooft anomalies of the model, from integrating the anomaly polynomial of the 6d SCFT on the compactification surface, and then search for a 4d model having the same anomalies. To actually make progress usually one also needs some additional assumptions, like that the model is conformal with a conformal manifold passing through weak coupling. An interesting aspect in this approach is that we do not attempt to tackle the reduction directly. Rather our aim here is to build a 4d model that flows to or sits on the same conformal manifold as the 4d theory expected from the 6d reduction, without the relation between the two being immediately apparent.

This method was used to great effect in the previous Lectures. Nevertheless, it has several shortcomings. First, as we mentioned, it usually requires additional assumptions regarding the nature of the 4d $\mathcal{N} = 1$ theory[42] that may be wrong. Also the reliance on anomalies, while very convenient, is restricted to spacetimes of even dimensions. As such it is appealing to also look for other methods by which 4d $\mathcal{N} = 1$ theories, resulting from the compactification of 6d SCFTs, can be determined.

Here we shall present a different method that can be used to conjecture such 4d $\mathcal{N} = 1$ theories. Unlike the previous method, here we shall actually try to follow the reduction process to determine the 4d theories. However, the reduction itself is in general quite complicated and difficult to follow. Nevertheless, here we can use an observation we made in the previous Lecture. Specifically, we noted that the resulting 4d theories are usually sensitive only to very rough properties of the surface, *e.g.* its genus and total flux. Other properties, *e.g.* the explicit metric of the surface or field strength of the flux, usually affect the result at worst through marginal operators, and in many cases only through irrelevant ones. We can exploit this for our purpose, and choose these to take a very special form for which we can follow the reduction.

We shall next discuss the method, which consists of two parts. In the first one we shall discuss a general method to conjecture the 4d theories resulting from the compactificaion of 6d SCFTs on flat surfaces, that is tori and tubes, with flux. This method follows our previous discussion and relies on following the reduction process in a specific limit where it is easy to analyze. After that, we shall introduce a method that allows us to exploit this to also understand the compactificaion of 6d SCFTs on more generic surfaces, notably spheres with more than two punctures.

## 6.1 The general idea

Consider the compactification of a 6d SCFT on a tube, which is just a sphere with two punctures. To set the stage, we take the coordinates of our 6d spacetime to be $x_1, x_2, \ldots, x_6$, with $x_6$ being the circle direction and $x_5$ the interval direction of the tube. We shall also take the bounaries of the interval to be at $x_5 = \pm a$. We want to determine the reduction of the theory to four dimensions. We can analyze this by first reducing along the circle spanned by $x_6$, to get a 5d theory on the 4d spacetime spanned by $x_1, x_2, x_3, x_4$ and the interval spanned by $x_5$. As we noted in the previous sections, when compactified to 5d, 6d SCFTs can flow to 5d gauge

---

[42]The specific nature of the assumption is usually the R-charges of the chiral fields under the superconformal R-symmetry, from which the $a$ and $c$ central charges of the theory can be determined in terms of the number of vector and chiral fields. The simplest assumption is that all chiral fields have R-charge of $\frac{2}{3}$, that is the theory is conformal at weak coupling. It is also possible to explore other choices where there are chiral fields with more than one value of R-charge, see for instance [82, 83].

theories if a proper holonomy is introduced. Here we shall assume that such a holonomy is incorporated. We shall later on see that it plays an important role in the introduction of flux. We can then reduce along the circle to get the 5d gauge theory on the 4d spacetime times an interval. At the boundaries of the interval we need to put boundary conditions, which can be expressed as boundary conditions on the 5d bulk fields as these approach the boundary, as done in Lectures III and IV. Throughout this Lecture we shall take the boundary conditions to be of the same type as those used in the previous Lecture, that is the ones associated with maximal punctures. As the theory now is just an IR free 5d gauge theory, we can then reduce along the interval and get the 4d theory, which is just built from the 5d matter that is consistent with the boundary conditions.

So far we have discussed the case without flux. However, to deal with interesting cases requires the introduction of flux. It turns out that we can make progress by representing the flux as a variable holonomy. Specifically, we pointed out that we usually need to turn on an holonomy on the circle so that the 6d SCFT reduces to a 5d gauge theory. However, this holonomy is not unique, where in general we can have many different holonomies leading to a 5d gauge theory. For instance, say the holonomy is in a non-abelian flavor symmetry. In that case, it breaks the flavor symmetry to a $U(1)$ group, where the holonomy resides, plus its commutant in the non-abelian group. We can now act on this holonomy with Weyl transformations of the broken group that are not part of the commutant. This should map the holonomy to an equivalent one, but which does not commute with the original holonomy. As such the space of holonomies leading to 5d gauge theories is in general quite large. Additionally, it is possible that a single 6d SCFT can reduce to multiple different 5d gauge theories, depending on the holonomies chosen, and indeed we shall present examples of such behavior later on.

We can consider introducing a variable holonomy. That is we introduce a background vector field coupled to the flavor symmetry and give it the particular profile,

$$A = \left( M_2 + (M_1 - M_2)\theta(x_5) \right)\delta(x_6)dx_6,$$

with $M_1$ and $M_2$ being some matrices in the adjoint representation of the flavor symmetry of the 6d SCFT. This suggests that we have holonomy of $Tr(M_1)$ around the circle direction $x_6$ for $x_5 > 0$, but a holonomy of $Tr(M_2)$ for $x_5 < 0$. As such the holonomy essentially jumps at $x_5 = 0$ between the two values. Finally we note that the presence of the variable holonomy implies the presence of flux on the surface as $F = (M_1 - M_2)\delta(x_5)\delta(x_6)dx_5 \wedge dx_6$.[43] The idea now is to take our flux on the surface to be generated by such a variable holonomy, with $M_1$ and $M_2$ chosen so that the holonomy at both $x_5 > 0$ and $x_5 < 0$ is such that the 6d SCFT flows to a 5d gauge theory.

The advantage of this approach is that we can now make progress on analyzing the reduction to 4d by first reducing to 5d. This is as our system in 5d now reduces to an IR free gauge theory living on the region $x_5 > 0$, and another IR free gauge theory living in the region $x_5 < 0$. These two gauge theories are separated by a domain wall at $x_5 = 0$. The reduction to 4d is now straightforward for the regions in the bulk, and we just expect to get the part of the 5d IR free gauge theory matter that is consistent with the boundary conditions. These should be supplemented by the fields living on the domain wall, *and as such the main problem is reduced to understanding domain walls between 5d gauge theories that are UV completed by 6d SCFTs.*

Let's consider the domain walls in a bit more detail. As we mentioned we expect there to be some 4d QFT living on the domain wall. We will assume that we can derive a description of this QFT in terms of an explicit gauge theory construction. Additionally there should also be

---

[43]We can smear the $\delta$-functions here to make the $F$ continuous. Moreover, later on we will also relate certain constructions with fluxes to punctured surfaces.

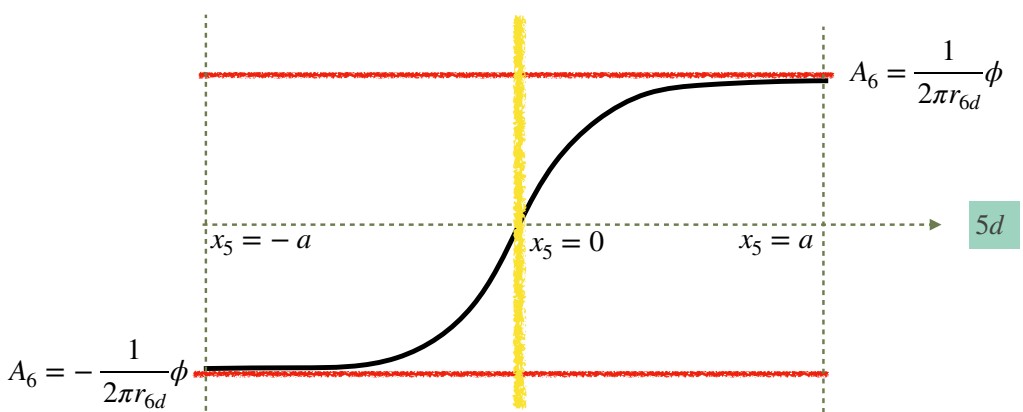

Figure 44: Illustration of a flux domain wall. the horizontal axis is the 5th dimension, $x_5$ and the vertical axis denotes the value of the holonomy on the circle around $x_6$. The holonomy interpolates between two asymptotic values which give rise to effective gauge theory description in 5d.

boundary conditions on the bulk 5d fields as these approach the domain wall. One operator that we expect to generically exist is a chiral operator in the bifundamental representation of the 5d gauge symmetries on the two sides of the domain wall. This is as the domain wall should extrapolate between the two symmetries. Specifically, say our 5d gauge theory has matter hypermultiplets which transform under the flavor symmetry. We should have such hypermultiplets at both sides of the domain walls. However, there should be only one flavor symmetry, so there should be some mechanism that identify the symmetries acting on the two sides of the domain wall. The presence of the bifundamenal operator gives such a mechanism. Specifically, let us denote such a chiral operator by $B$, and break the hypermultiplets to two chiral fields in conjugate representation which we denote as $X_+$, $Y_+$ for one side of the domain wall and $X_-$, $Y_-$ for the other. Then for a hypermultiplet in the fundamental representation of the gauge group, the superpotential $X_+ B Y_-$ does the job. For higher representations, we need the superpotential $X_+ B^n Y_-$.[44] The components $X_-$, $Y_+$ can also be coupled to $B$ with similar superpotentials.

As such, we see that we generically expect to have a bifundamental chiral operator on the domain wall. However, we may have many more fields and such a chiral might actually be a composite. In fact we expect that most domain walls will have rather complicated matter living on them. To illustrate this, consider the case where we have two domain walls. That is we take a variable holonomy such that it has one value for say $-a < x_5 < -b$, another for $-b < x_5 < b$ and another for $b < x_5 < a$. We then have two domain walls, located at $x_5 = \pm b$. Now we can consider the limit where $b \to 0$. In this limit the two domain wall collapse to a single domain wall directly extrapolating between the holonomy at the two edges. We expect the fields living on this domain wall to be the same as the fields living on the two domain walls from which it is made, as well as the fields living on the bulk between the two. We expect the latter to contribute the gauge fields of the 5d theory, and as such we expect the single domain wall to contain gauge fields.

To make progress, it is convenient to make the following assumption. Specifically, we shall assume that there exist a domain wall such that the fields living on it can be described solely in terms of chiral fields interacting through superpotential terms. Of course this assumption may be wrong, and such a domain wall may not exist. However, making this assumption allows us

---

[44]There are some subtle issues here for $SO$ groups with spinor matter that we shall ignore here.

to make progress in analyzing the reduction and as such we shall make it. This assumption will later be checked by a detailed study of the anomalies of the resulting model. Once we determined one domain wall, we can determine more complicated ones by gluing together this basic domain wall. Finally, we need to consider the flux associated with this tube. In principal the flux should be related to the difference in the holonomies at the two sides of the domain wall. However, in this approach we don't actually control the holonomies, rather the domain wall is selected such that the fields living on it are particularly simple. As such we in general don't apriori know what the flux is, and will generally determine it by matching anomalies and symmetries.

Next, we shall illustrate this method with various examples.

## 6.2 Compactification of the rank one E-string SCFT

As our first example, we consider the case of rank one E-string SCFT that we discussed in the previous Lecture, but now from the domain wall point of view. First, recall that the rank 1 E-string is a 6d SCFT with $E_8$ global symmetry. It can be compactified to 5d with a suitable holonomy such that it flows to the $SU(2)$ gauge theory with eight fundamental hypermultiplets. Here we shall rely on this fact to study the dimensional reduction of this 6d SCFT on tubes with flux.

We proceed as outlined previously. Specifically, we first reduce on a circle to 5d. As previously explained, we expect to be able to represent the flux as a variable holonomy, which we take to be such that the theory flows to an $SU(2)$ gauge theory with eight fundamental hypermultiplets on the subspace $x_5 > 0$, a different $SU(2)$ gauge theory with eight fundamental hypermultiplets on the subspace $x_5 < 0$ and a domain wall at $x_5 = 0$. The theory away from the domain wall is thus an IR free 5d gauge theory and its reduction can be easily analyzed.

Here we also need to consider the boundary conditions at the two punctures. Recall that these give Dirichlet boundary conditions to the 4d $\mathcal{N} = 1$ vector multiplet component of the 5d $\mathcal{N} = 1$ vector multiplet at the boundary. Therefore, as before, we expect the $SU(2)$ gauge symmetry to become non-dynamical at the boundary leading to an $SU(2)$ global symmetry that we associate with the punctures. Additionally we have the eight fundamental hypermultiplets. Close to the boundary we can decompose them to two 4d $\mathcal{N} = 1$ chiral fields in opposite representations which we shall denote as $X_i$, $Y_i$, with $i = 1, 2, \ldots, 8$ denoting the chosen hypermultiplet. We then need to give Dirichlet boundary conditions to one of them and Neumann to the other for each $i$. The exact choice doesn't matter too much, as we can transform between them by flipping the fields as explained previously, but for presentational purposes, it is convenient to take the boundary conditions to be such that the fields $Y_i$ receive the Dirichlet boundary conditions for $x_5 > 0$ while the fields $X_i$ receive the Dirichlet boundary conditions for $x_5 < 0$.

Finally we need to address the fields living on the domain wall. For the case at hand these types of domain walls were studied in [135], and we can in principle just use the results found there. However, here we shall take a slightly different approach. Specifically, as we explained previously, we shall assume that the fields living on the domain wall can all be expressed in terms of free chiral fields. These should contain at least a chiral field in the bifundamental representation of the two $SU(2)$ groups at the sides of the domain wall, interacting with the hypermultiplets as we outlined previously. As we shall soon see when we study the anomalies of this model, this is not enough to match the anomalies and we must add one more chiral field, which turns out to be a singlet flipping the quadratic $SU(2) \times SU(2)$ invariant made from the bifundamental. For now, we shall add it to the matter content, but will return to this issue later.

The resulting 4d theory we obtain is shown in Figure 45. Here the two global $SU(2)$ symmetry groups are the ones associated with the punctures and the $SU(8)$ is the subgroup of

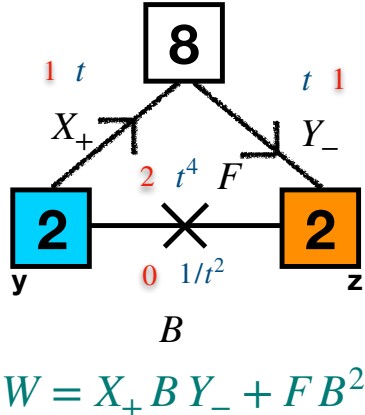

$$W = X_+\, B\, Y_- + F B^2$$

Figure 45: The basic tube theory for the compactifications of the rank one E-string. The $SU(2)$ symmetries correspond to the two punctures; the two fundamental octets come from the bulk fields in the two domains separated by the wall; and the bifundamental field with the flip field come from degrees of freedom residing on the domain wall.

the 6d global symmetry that rotates the fields $X_i$ and $Y_i$. The $SU(2) \times SU(8)$ bifundamentals on the two sides comes from the part of the bulk hypermultiplets receiving Neumann boundary conditions. Finally, the $SU(2) \times SU(2)$ bifundamental and singlet field come from the domain wall. The fields interact trough various cubic superpotentials.

The theory has several global symmetries. First it has the $SU(2) \times SU(2)$ non-abelian symmetries coming from the punctures, and an $SU(8)$ which is a subgroup of the 6d global symmetry. Additionally, we can define two $U(1)$ global symmetries consistent with the superpotential, one of which, denoted as $U(1)_t$ is also associated with the 6d global symmetry. It is defined such that the fields $X_+$ and $Y_-$ have the same charge under it. There is another $U(1)$ global symmetry, under which the fields $X_+$ and $Y_-$ have opposite charges, but it is in general anomalous once we glue the tubes to closed surfaces.[45] Finally, there is a $U(1)$ R-symmetry, which is convenient to define as the one coming from the Cartan of the 6d $SU(2)$ R-symmetry. Since the fields $X_+$ and $Y_-$ came from components of the 5d hypermultiplets, they should have R-charge 1 under it. The superpotential then forces the bifundamental, $B$, to have R-charge 0 and the flip field to have R-charge 2.

> Exercise: Consider the anomalies of the tube and check that these match the 6d expectation. Note that we have not determined the flux associated with this tube yet. However, are there anomalies that are independent of the flux? If so do these match the 6d expectations?

As we mentioned there are two sources of contributions to the 4d anomalies from the 6d ones. One is from the integration of the anomaly polynomial on the surface and the other is from the punctures. The latter are independent of the flux so we are led to consider the former. The only non-trivial contribution can come from the flux, and as such must come from the integration of the $C_2(E_8)$ term. As such there are several anomalies that will not receive contribution from it, and so are independent of the flux. These are the $Tr(U(1)_R)$, $Tr(U(1)_R^3)$

---

[45]Closing the two punctures and obtaining the theory corresponding to a sphere with flux, this $U(1)$ symmetry can be identified [136] with the Cartan generator of the $SU(2)$ isometry of the sphere which becomes a global symmetry in 4d (see [95, 137]).

and $Tr(U(1)_R F^2)$ for $F$ some flavor symmetry. The contribution to all these anomalies will come only from the punctures for which we can evaluate:

$$Tr(U(1)_R) = Tr(U(1)_R^3) = -\frac{3}{2} - \frac{3}{2} = -3\,, \tag{153}$$

$$Tr(U(1)_R U(1)_t^2) = Tr(U(1)_R SU(8)^2) = 0\,. \tag{154}$$

Now let's consider comparing them with the anomalies we observe in the tube. For the ones only involving the R-symmetry we have that:

$$Tr(U(1)_R) = Tr(U(1)_R^3) = -4 + 1 = -3\,, \tag{155}$$

where the first term is the contribution of the bifundamental $B$ and the second term is the contribution of the flip field.

For the one involving the flavor symmetry we have that:

$$Tr(U(1)_R U(1)_t^2) = -2^2 + 4 = 0\,, \ Tr(U(1)_R SU(8)^2) = 0\,, \tag{156}$$

where again the first term in the first equation is the contribution of the bifundamental $B$ and the second one is the contribution of the flip field.

We see that the anomalies indeed match. However, note that for this matching to work it is important that we have the flip field. This is one way to understand why we must add it, as otherwise the anomalies won't match. Furthermore, this necessitates that it has $U(1)_R$ R-charge of 2 and $U(1)_t$ charge of 4. The requirement that the charges have this value then also determines the superpotential.

□

**Gluing two tubes to form a torus**

Having formed a conjecture for the 4d theory associated with the compactification of the 6d E-string SCFT on a tube with flux, we next want to test this conjecture and in the process determine the flux. To do this it is convenient to work with closed surfaces with no punctures, and as such we shall take two such tubes and glue them together. The gluing process is done as we explained in the previous Lectures. Specifically, here the moment map operators associated with the punctures are the $SU(2) \times SU(8)$ bifundamentals. We first note that these are charged differently under the 6d global symmetry for the two punctures so the punctures then have different colors. What we shall do is take two tubes and glue the punctures of same color of each tube together. As we are gluing punctures with same color, the gluing we need to perform is $\Phi$ gluing, so we gauge the $SU(2)$ global symmetry of the two punctures with an $\mathcal{N} = 1$ $SU(2)$ vector multiplet with eight fundamental chiral fields, the fields $\Phi$. These are then coupled to the moment map operators associated with the two glued punctures via a quadratic superpotential. The latter just becomes a mass term, leading to the fields $\Phi$ and some of the moment map operators being integrated out. We end up with the theory shown in Figure 46.

We can next study the resulting theory. We first note that the $U(1)$ R-symmetry giving free R-charge to all the chiral fields is anomaly free, suggesting that the theory might be a conformal theory at weak coupling. As such, we could have in principle arrived at this theory using our previous strategy. We can then ask whether it is actually conformal or not. We first note the two flip fields should decouple as there cannot be any quotient for the symmetries acting only on them. Using the same methods as in the previous Lectures, one can show that the remaining theory has a non-trivial quotient so we get a 4d SCFT and two decoupled free fields.

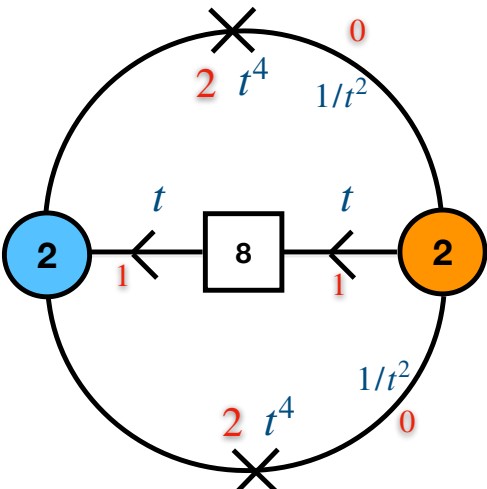

Figure 46: The theory corresponding to gluing two basic tubes into a torus. Note that this is the same model we have obtained on the right side of Figure 42 using a different method.

The first thing we want to do is check whether this theory has the right properties to indeed be the dimensional reduction of the 6d E-string SCFT on a torus with flux, and if so to determine the flux. To do so we first evaluate the superconformal index. As we are mainly interested in matching with expected properties from 6d, we shall calculate the index including the flip fields and refining only with respect to symmetries existing in 6d, that is we shall ignore the fact that the flip fields should decouple. The result we find is,

$$1 + 2t^4(pq)^{\frac{1}{3}} + (pq)^{\frac{2}{3}}\left(\frac{1}{t^4} + 3t^8 + t^2(\mathbf{28} + \overline{\mathbf{28}})\right) + 2t^4(pq)^{\frac{1}{3}}(p+q) \qquad (157)$$
$$+ 2pq(2t^{12} + t^6(\mathbf{28} + \overline{\mathbf{28}})) + \dots$$

One thing we note in this index is that it forms characters of $E_7$. Specifically, we have that $\mathbf{56}_{E_7} \to \mathbf{28} + \overline{\mathbf{28}}$. As such it is tempting to assign to it a flux of value 1 preserving the $U(1) \times E_7$ subgroup of $E_8$. Indeed we have the branching rules: $\mathbf{248}_{E_8} \to \mathbf{1}^0_{E_7} + \mathbf{1}^{\pm 2}_{E_7} + \mathbf{56}^{\pm 1}_{E_7} + \mathbf{133}^0_{E_7}$. This suggests that the 6d global symmetry is related to the symmetry we see in 4d by: $\mathbf{248}_{E_8} \to (t^4 + 1 + \frac{1}{t^4})\mathbf{1} + (t^2 + \frac{1}{t^2})(\mathbf{28} + \overline{\mathbf{28}}) + \mathbf{63} + \mathbf{70}$.

The claim can be further supported by looking at the 't Hooft anomalies of the model. Specifically, we can calculate the 't Hooft anomalies of the model finding:

$$Tr(U(1)_R) = Tr(U(1)_R^3) = Tr(U(1)_R U(1)_t^2) = 0, \quad Tr(U(1)_R^2 U(1)_t) = -8, \qquad (158)$$
$$Tr(U(1)_t) = 24, \quad Tr(U(1)_t^3) = 96, \quad Tr(U(1)_t SU(8)^2) = 2,$$

with the rest vanishing trivially. These anomalies indeed match the anomalies expected from the compactification of the rank one E-string on a torus with unit flux preserving the $U(1) \times E_7$ subgroup of $E_8$ (which can be read off from Appendix F.3). Here we use the expected embedding of the 4d symmetries in the 6d global symmetry suggested by the index, and the following embedding index: $I_{SU(8) \to E_7} = 1$.

All this is consistent with the flux being of value 1 preserving the $U(1) \times E_7$ subgroup of $E_8$. Since we got this theory from gluing together two copies of the tube we introduced, we are lead to associate with the tube a flux of $\frac{1}{2}$ preserving the $U(1) \times E_7$ subgroup of $E_8$. We

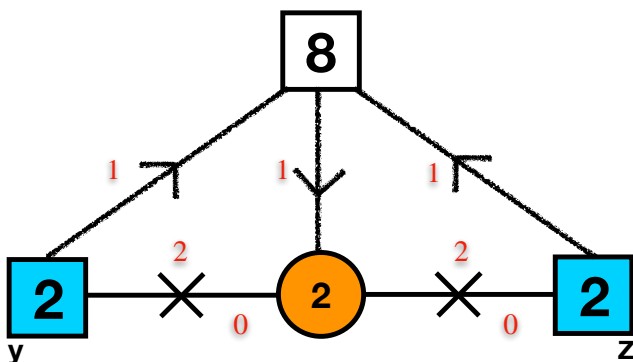

Figure 47: Combining two basic domain walls. This domain wall corresponds to flux which is twice the one for the basic tube.

can write this flux in our flux basis as $\frac{1}{2}(1,1,1,1,1,1,1,1)$.[46]

**Combining domain walls**

We have seen how our method can be used to conjecture 4d theories associated with the compactification of 6d SCFTs on tubes by relating the problem to the behavior of domain walls in 5d and assuming there exist a particularly simple type of the latter. Next, we want to build on that and also get models corresponding to the compactification on tubes with other values of flux. We can do so by combining two domain walls to form more complicated domain walls.

Specifically, consider the configuration where we now have two domain walls, that is we take our holonomy to have the value $M_1$ on one side of the interval, $M_2$ in the middle of the interval, and $M_3$ on the other side. As we explained previously, we can also view this as one domain wall extrapolating directly between $M_1$ and $M_3$, though it is easier to analyze this if we instead view it as multiple domain walls.[47] In particular, as we have determined one domain wall, we can ask how can we combine that one domain wall to form more general ones.

The most straightforward way to do so is to simply chain them together. That is we take the two domain walls to be identical. If we do this then we can repeat our previous analysis and arrive at the theory shown in Figure 47. Here as before, the $SU(2) \times SU(8)$ bifundamentals at the two edges come from the component of the bulk hypermultiplets that receive Neumann boundary conditions. The $SU(2) \times SU(2)$ bifundamental and its flip field come from the domain walls, where now we have two of them. In the middle however, we now have a gauge $SU(2)$ group, as the $\mathcal{N} = 1$ vector receives Neumann boundary conditions at the domain wall. This suggests that the adjoint chiral receives Dirichlet boundary conditions there, explaining its absence. Finally, we again have the $SU(2) \times SU(8)$ bifundamental that comes from the component of the bulk hypermultiplets that receive Neumann boundary conditions, now at the domain wall. The end theory is just the theory we get if we glue two tubes to themselves. This of course is as expected as the resulting surfaces would indeed be itself a tube, but now with two domain walls. As such the flux associated with this tube is just twice the flux of the original tube, which in our basis is $(1,1,1,1,1,1,1,1)$.

The more interesting case is when we glue the tubes with a Weyl symmetry twist. We noted that the flux associated to the tube should be such that it breaks $E_8$ into $U(1) \times E_7$. We have

---

[46]Here we are using the basis of the roots of $E_8$ were its $SO(16)$ spinor roots are taken to have an even number of minus signs. This is as the puncture $SU(2)$ symmetry of the tube theories here does not have Witten anomaly.

[47]If $M_1 = M_3$ the combined domain wall becomes trivial.

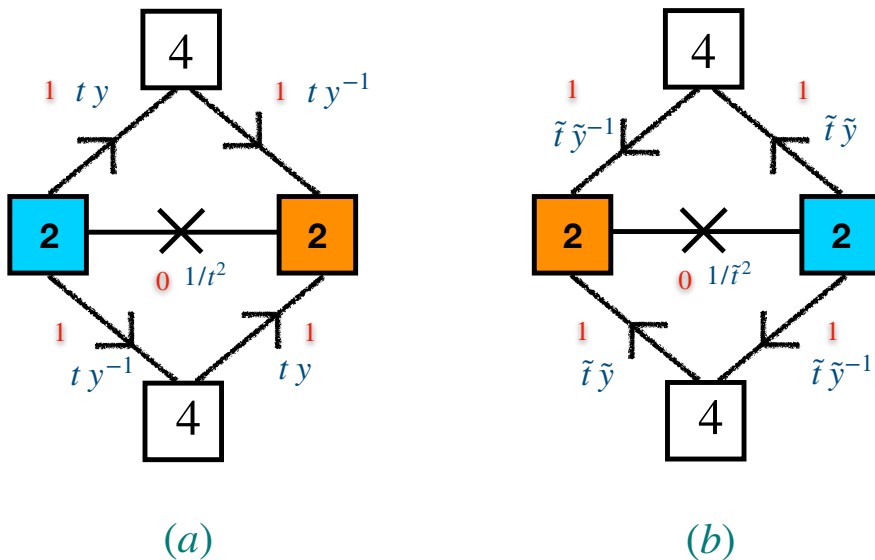

$$(a) \qquad\qquad (b)$$

Figure 48: Combining two basic domain walls with non trivial identification of symmetries between the two copies.

many different choices on how to embed this flux inside $E_8$. For instance, consider one such choice. Then we can get an equivalent, yet different choice by acting on the flux with part of the Weyl group of $E_8$ that is broken by the flux. Since these are equivalent fluxes, the tube theories that we have presented will be the same for both of them, and so also if we glue each one of them to another copy of itself. However, if we glue together two tubes preserving a different $E_7$ inside $E_8$, then the resulting tube will be different. In particular, it will no longer preserve an $E_7$ subgroup. We shall next illustrate how this is done with an example.

We first consider breaking the $SU(8)$ global symmetry to $U(1)_y \times SU(4) \times SU(4)$ as follows: $\mathbf{8} \to y\mathbf{4}_1 + \frac{1}{y}\mathbf{4}_2$. To see why this is interesting it is convenient to also break $E_8$ into these symmetries, where we have:

$$\mathbf{248}_{E_8} \to \left(t^4 + y^4 + 2 + \frac{1}{y^4} + \frac{1}{t^4}\right)\mathbf{1} + \left(t^2 + \frac{1}{t^2}\right)\left(y^2 + \frac{1}{y^2}\right)(\mathbf{6}_1 + \mathbf{6}_2) \qquad (159)$$

$$+ \left(t^2 + \frac{1}{t^2}\right)(\mathbf{4}_1\mathbf{4}_2 + \overline{\mathbf{4}}_1\overline{\mathbf{4}}_2) + \left(y^2 + \frac{1}{y^2}\right)(\mathbf{4}_1\overline{\mathbf{4}}_2 + \overline{\mathbf{4}}_1\mathbf{4}_2) + \mathbf{6}_1\mathbf{6}_2 + \mathbf{15}_1 + \mathbf{15}_2.$$

The interesting property that we shall soon make use of, is that the adjoint of $E_8$ is symmetric under the exchange $t \leftrightarrow y$, $\mathbf{4}_1 \leftrightarrow \overline{\mathbf{4}}_1$. This is part of the Weyl group of $E_8$. To illustrate this, consider the $SO(16)$ subgroup of $E_8$. This decomposition induces also a decomposition of $SO(16) \to U(1)_t \times U(1)_y \times SU(4) \times SU(4)$ such that $\mathbf{16} \to ty\mathbf{4}_2 + \frac{t}{y}\mathbf{4}_1 + \frac{1}{ty}\overline{\mathbf{4}}_2 + \frac{y}{t}\overline{\mathbf{4}}_1$. This implies that this transformation reduces to the Weyl transformation acting as charge conjugation on four out of eight $SO(2)$ independent subgroups of $SO(16)$.

Let's consider two copies of the tube we presented, which we have included in Figure 48. Here we have used different fugacities for the symmetries of the two theories to stress the fact that we have not yet identified the symmetries between the two theories. The simplest identification is to take: $\tilde{y} \to y$, $\tilde{t} \to t$, $\mathbf{4}_1^R \to \mathbf{4}_1^L$ and $\mathbf{4}_2^R \to \mathbf{4}_2^L$. This makes the two tubes identical and we can glue them to get the tube we previously mentioned.

However, in light of what we have seen from the decomposition of the adjoint of $E_8$, we can also make the identification: $\tilde{y} \to t$, $\tilde{t} \to y$, $\mathbf{4}_2^R \to \mathbf{4}_2^L$ and $\mathbf{4}_1^R \to \overline{\mathbf{4}}_1^L$. This will still map the $E_8$ symmetry of the underlying 6d SCFT to itself, but with some action of the Weyl symmetry

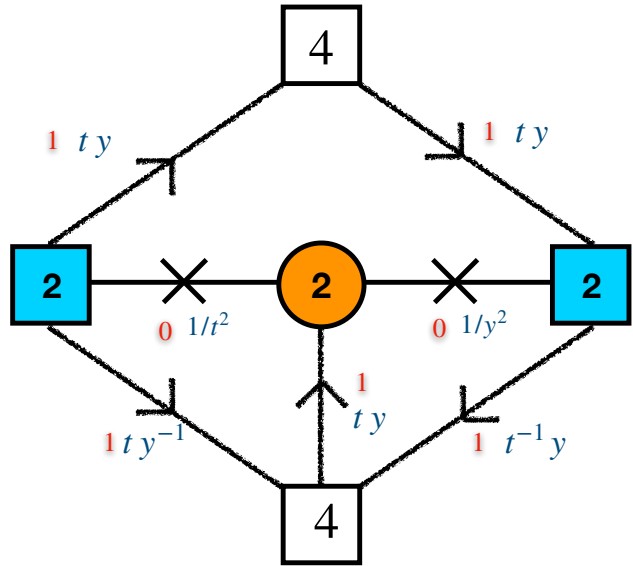

Figure 49: The two domain walls of Figure 48 combined together with half of the boundary conditions being Neumann and half Dirichlet.

of $E_8$. The two tubes are then not completely identical, as can be seen from the fact that the matter representation under the symmetries are slightly different. Specifically, we note that while the $SU(2) \times SU(8)$ bifundamental on the lower right of Figure 48 (a) and lower left of Figure 48 (b) have the same charges, the ones on the upper right of Figure 48 (a) and upper left of Figure 48 (b) have opposite charges.

We can next glue the two tubes together. Here the gluing is done with $\Phi$ gluing for the four $SU(2) \times SU(8)$ bifundamental chiral fields with the same charges, and with S gluing for the four $SU(2) \times SU(8)$ bifundamental chiral fields with opposite charges. This leads to the tube shown in Figure 49. We can also understand this tube as follows. Recall that the $SU(2) \times SU(8)$ bifundamental chiral fields come from the bulk eight hypermultiplets consistent with the boundary conditions. As such, if they have the same charges, then that suggests that the same component of the hyper receives Neumann boundary conditions on both ends, while if they have opposite charges then different components receive boundary conditions at the two ends. As the boundary conditions at the edges should be the same as on the domain wall, this suggests that fields with the same charges will receive Neumann boundary conditions on the two domain walls, and so survive the 4d reduction, while for those with opposite charges every component should receive Dirichlet boundary conditions at least on one of the domain walls.

We next want to determine the flux associated with the new tube in Figure 49. This should be given by the sum of the fluxes of both tubes, when evaluated in the same basis. For the first tube we can take the flux to be $\frac{1}{2}(1, 1, 1, 1, 1, 1, 1, 1)$. For the second one the flux is the same, but in a different basis related to the one of the other tube by the Weyl transformation. As the latter is of order two we can undo it by performing it again. Recall that we determined that the Weyl action we consider here should act on the eight $SO(2)$ subgroup of $SO(16)$ by charge conjugating four and leaving the rest invariant. This suggests that in the basis of the first tube, the flux of the second tube is $\frac{1}{2}(1, 1, 1, 1, -1, -1, -1, -1)$. We then conclude that the flux of the tube in Figure 49 is $(1, 1, 1, 1, 0, 0, 0, 0)$, which is a half flux preserving the $U(1) \times SO(14)$ subgroup of $E_8$.

These results can be checked by gluing two copies of this tube and studying its anomalies and superconformal index. This method can also be generalized to build many more domain walls, and as such 4d theories associated with compactifications on tubes, for many other values of flux. We refer the reader to [26] for more information on both subjects.

## 6.3 Compactification of minimal $(D, D)$ conformal matter theories

Having illustrated the basic idea by studying the compactifications of the E-string SCFT, we next wish to further elaborate on interesting phenomena and considerations that can appear in this construction. For this we consider another example, now involving the compactification of the 6d SCFT known as the minimal $(D_{N+3}, D_{N+3})$ conformal matter on a torus with global symmetry fluxes. For brevity, we shall usually refer to this SCFT as minimal $D$ type conformal matter.

**The minimal $D$ type conformal matter**

The $D$ type conformal matter theories is a family of 6d SCFTs. This family can be engineered in string theory as the theories living on $n$ M5-branes probing a $\mathbb{C}^2/\mathbb{D}_{N+3}$ singularity [10,138]. The specific case of $n = 1$ is known as the minimal case and is the one we will consider here. We note that for $N = 1$, this SCFT is just the rank one E-string theory that we have discussed in detail. As such, we can think of this family as a generalization of the previous case.[48] One advantage of this generalizations is that as we still have only one M5-brane, the tensor branch of these theories is still one dimensional.

These theories have a convenient field theory realization that we shall next study. Specifically, starting from the SCFT point we can deform it by giving a vacuum expectation value to the scalar in the tensor multiplet, that is by going to the tensor branch. This initiates an RG flow that in many cases ends with a 6d IR free gauge theory, whose coupling constant is identified with the scalar vev, see Appendix E. For the case at hand, this 6d tensor branch theory turns out to be a $USp(2N-2)$ gauge theory with $2N+6$ fundamental hypermultiplets. As such we can also think of this 6d SCFT as a UV completion of this 6d gauge theory. The gauge theory as an $SO(4N+12)$ global symmetry algebra rotating the $2N+6$ fundamental hypermultiplets, which turns out to also be the global symmetry algebra of the 6d SCFT.[49]

We can next consider the compactification of this theory to 5d. From our previous discussion, we shall be mainly interested in possible 5d gauge theory descriptions that can emerge from such a reduction with a suitable holonomy. One intriguing property that this family of 6d SCFTs has is that there are actually three different possible 5d gauge theories one can obtain [96, 97]. The first has gauge group $USp(2N)$ and $2N+6$ fundamental hypermultiplets, the second has gauge group $SU(N+1)$, no Chern-Simons term and $2N+6$ fundamental hypermultiplets, and the last has gauge group $SU(2)^N$, bifundamental hypermultiplets connecting the groups into a linear quiver, and four fundamental hypermultiplets for each of the $SU(2)$ groups at the ends of the quiver. Note that these all degenerate to $SU(2) + 8F$ for $N = 1$.

---

[48]There are other possible generalizations. For instance, we can keep $N = 1$, and take $n$ to be arbitrary, leading to non-minimal $D_4$ type conformal matter theories. Alternatively, we can use the constructions of the E-string as the theory living on one M5-brane in the presence of an M9-plane, and generalize to the case of $N$ M5-branes. This class of theories is known as the rank $N$ E-string theories. We note that all theories in the three families are distinct, save for the rank one E-string.

[49]As the 6d SCFT and 6d gauge theory are related by flow, the usual caveats regarding the relation between their symmetries apply. Interestingly, in this case, while the two share the same global symmetry algebra, they don't share the same global symmetry group. Specifically, the gauge theory has an $SO(4N+12)/\mathbb{Z}_2$ global symmetry group. However, the 6d SCFT has a $Spin(4N+12)/\mathbb{Z}_2$ global symmetry group, see the discussion in [19]. This comes about as the 6d SCFT has a Higgs branch operator in a chiral spinor representation of $SO(4N+12)$, which becomes massive once we go on the tensor branch. Finally, we note that for $N = 1$ this Higgs branch operator actually becomes a conserved current multiplet, leading to the enhancement of $SO(16) \rightarrow E_8$.

This phenomena has profound implications on the study of 4d compactification. First, we noted that punctures can be associated with boundary conditions of the 5d gauge theory, but what happens when there are multiple such theories? In that case it appears that we can associate a family of punctures to each 5d gauge theory description. Specifically, we can use the boundary conditions to define a maximal puncture associated with each 5d gauge theory description. Each puncture will have the gauge symmetry of the 5d theory has its associated global symmetry, and carry its own type of moment map operators. We can then choose to partially close the puncture by giving a vev to the moment map operators. This creates a family of punctures starting from each maximal puncture.[50]

Naturally, we can also use the three possible descriptions to construct different boundary conditions. As before we can try to build domain wall theories for domain walls between the same description, as we did for the E-string case, for each of the three gauge theory descriptions. Furthermore, we can consider domain walls where there are entirely different gauge theory descriptions on each side. This leads to six different possibilities. We can then try to conjecture the resulting 4d theories by assuming that the theory living on such domain walls is made of chiral fields, and check whether or not such a conjecture can reproduce the properties expected from theories corresponding to the compactifications of these 6d SCFTs. This analysis has been carried out for this family of theories, and out of the six possibilities only three yield sensible results:[51] $SU(N+1)-SU(N+1)$, $SU(N+1)-USp(2N)$ and $SU(2)^N-SU(2)^N$. Next we shall further illustrate the use of the domain wall method by performing this analysis for the $SU(N+1)-SU(N+1)$ case. For the analysis of the $SU(N+1)-USp(2N)$ case, and for more details on the analysis of the $SU(N+1)-SU(N+1)$ case, we refer the reader to [19], while for the analysis of the $SU(2)^N-SU(2)^N$ case, we refer the reader to [126].

**The $SU-SU$ tube**

Following the previously outlined procedure, we can attempt to conjecture a 4d model associated with the compactification of the minimal $D$ type conformal matter theory on a torus with punctures. Here we shall concentrate on the cases built from the $SU-SU$ domain wall, so we take the punctures at the two ends to be $SU(N+1)$ maximal punctures. Following our previous discussion, we then make the ansatz for the theory shown in Figure 50. Here we have the $SU(N+1)$ global symmetries associated with the two maximal punctures at the two ends of the tube. These come from the 5d gauge symmetry in the bulk that becomes a global symmetry as the vector receives Dirichlet boundary conditions. Additionally, we have the $2N+6$ fundamental hypermultiplets. Half of the chiral fields in them receives Neumann boundary conditions, while the other half receives Dirichlet boundary conditions. As the representation for generic $N$ are complex, the representation under the $SU(N+1)$ puncture global symmetry of the surviving hypermultiplets depends on which component receives the Neumann boundary condition. As we do not know this a priori,[52] we shall leave this question open for now and answer it soon. As such we shall assume that we have $n_+$ hypermultiplets where the fun-

---

[50]It is an open question whether each family is entirely disconnected from the other or rather whether some sub-maximal punctures are shared between the different families.

[51]For the other three cases it seems the domain walls are more complicated. Specifically, we can form $USp(2N)-USp(2N)$ tubes by gluing two $SU(N+1)-USp(2N)$ tubes along the $SU(N+1)$ punctures. These tubes will then contain gauge fields from the gluing. Tubes realizing the $SU(N+1)-SU(2)^N$ or $USp(2N)-SU(2)^N$ combinations have not been worked out to our knowledge, though one should be able to do this using the results of [118].

[52]Technically, we can choose the boundary conditions at the punctures to be what ever we want. However, to maintain the similarity to the previous case, we want to choose the boundary conditions for the bulk hypermultiplets at the punctures to be the same as the one at the domain wall. The issue then stems from our lack of knowledge regarding the boundary conditions on the domain wall for domain walls with only chiral fields living on them.

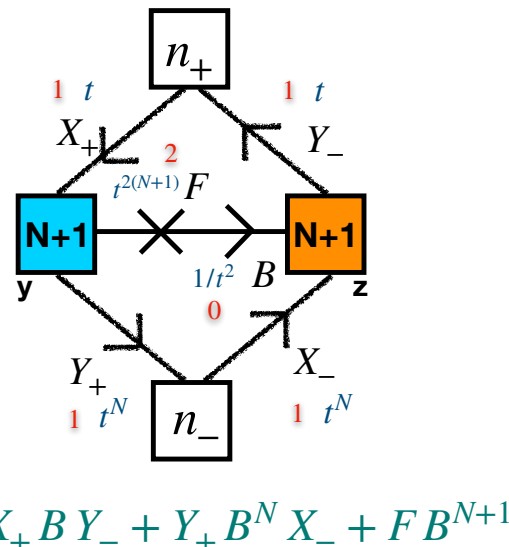

$$W = X_+ \, B \, Y_- + Y_+ \, B^N \, X_- + F \, B^{N+1}$$

Figure 50: The basic tube theory for the compactifications of the minimal $(D_{N+3}, D_{N+3})$ conformal matter theory. The $SU(N+1)$ symmetries correspond to the two punctures; the two collections of fundamental fields come from the bulk fields in the two domains separated by the wall; and the bifundamental field with the flip field come from degrees of freedom residing on the domain wall. The parameters $n_+$ and $n_-$ are fixed to be $2N+4$ and $2$ respectively. In the case of $N=1$ we reproduce the domain wall of Figure 45.

damental receives Neumann boundary condition on one side, and $n_-$ hypermultiplet where the antifundamental receives Neumann boundary condition on the same side. Naturally, we have that $n_+ + n_- = 2N + 6$.

Next we consider the domain wall. As before we shall make the assumption that the fields living on the domain wall can be described using only chiral fields. On general grounds, we expect there to be an $SU(N+1) \times SU(N+1)$ bifundamental chiral field. It should interact with the fundamental and anti-fundamental chiral fields associated with the two punctures via the superpotentials:

$$W = XB\tilde{Y} + YB^N\tilde{X}, \tag{160}$$

where here $B$ refers to the bifundamental chiral, and $X$, $Y$ and $\tilde{X}$, $\tilde{Y}$ refer to the fundamental or anti-fundamental chiral fields associated with the punctures. There might be additional fields associated with the domain wall. To get a better understanding of this, it is good to again consider the anomalies of the tube. Specifically, we want to look at the $Tr(U(1)_R^3)$, $Tr(U(1)_R)$ and $Tr(U(1)_t^2 \times U(1)_R)$ anomalies, where $U(1)_t$ is defined as in Figure 50. These anomalies only receive contributions from the punctures and not from integrating the anomaly polynomial on the surface, and so can be evaluated from the data available to us. The puncture contribution gives:

$$Tr(U(1)_R^3) = Tr(U(1)_R) = -\frac{(N^2-1)}{2} - \frac{(N^2-1)}{2} = -N^2 + 1, \tag{161}$$
$$Tr(U(1)_t^2 \times U(1)_R) = 0.$$

However, the evaluation in the gauge theory gives:

$$Tr(U(1)_R^3) = Tr(U(1)_R) = -N^2, \qquad Tr(U(1)_t^2 \times U(1)_R) = -4(N+1)^2. \tag{162}$$

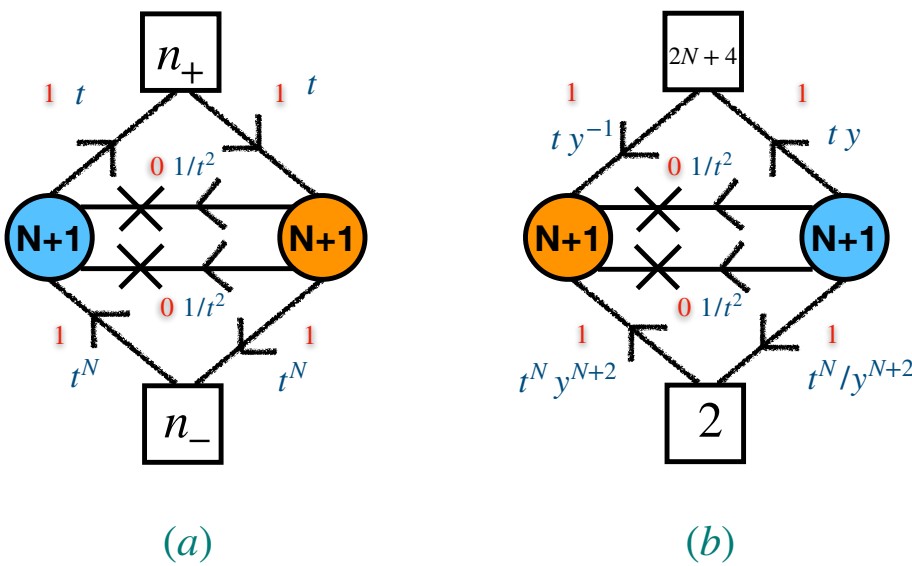

Figure 51: The torus theory for the minimal $(D_{N+3}, D_{N+3})$ conformal matter theory obtained by gluing together two tube theories.

This suggests that we need to add an additional field. The minimal possible addition consistent with our assumption regarding the nature of the domain wall is a single free chiral field with R-charge 2 and $U(1)_t$ charge $2(N + 1)$. This field has a natural interpretation, as its charges are just right for it to flip the baryon made from the $SU(N + 1) \times SU(N + 1)$ bifundamental. As such we conclude that our conjecture for the tube is the theory shown in Figure 50.

**Testing the tube**

We next move to perform more stringent tests on the tube theory that can be used to determine the flux and to provide supporting evidence that it is indeed a tube theory associated with the claimed compactification. As before, it is convenient to do this by gluing two tubes together to form a closed surface. For this case, this leads to the theory show in Figure 51 (a), which we shall next analyze.

The first thing we need to make sure of is that the resulting theory is consistent. Specifically, as the theory is chiral, we need to worry about gauge anomalies. Indeed for generic values of $n_+$, $n_-$ we have a non-zero gauge anomaly $Tr(SU(3)^3_{gauge}) = 2N + 2 + n_- - n_+$. Demanding that this anomaly vanishes, together with the constraint $n_+ + n_- = 2N + 6$, forces $n_+ = 2N + 4$, $n_- = 2$. We therefore see that the consistency of the theory uniquely fixes the tube theory. Additionally, we would like to make sure that the $U(1)_t$ symmetry is anomaly free as we have made use of it when matching the anomalies between the tube and the 6d expectation. Indeed $U(1)_t$ is anomaly free precisely with these values of $n_+$ and $n_-$. This is already an indication that we are on the right track.

The concrete torus theory we need to study then is the one shown in Figure 51 (b). We can next study this theory and compare the results with our 6d expectations. As before the analysis consists of two steps. One is to compute the superconformal index and the other is to compute the anomalies. These are then compared to the 6d expectations and are needed to be consistent with them and with one another. We shall first start with the computation of the superconformal index, which can be used to determine the flux and the mapping of

symmetries, which facilitates the matching of anomalies. For simplicity, we shall here take the case of $N = 2$ and also compute the index of the model without the flip fields, as it appears the flip fields decouple in the IR. This is seen by looking for the expected superconformal R-symmetry via $a$-maximization. We then find for the superconformal index:

$$1 + t^3(pq)^{\frac{5}{8}}\mathbf{2}_{SU(2)}\left(y^3\mathbf{8}_{SU(8)} + \frac{1}{y^3}\overline{\mathbf{8}}_{SU(8)}\right) + \frac{4}{t^6}(pq)^{\frac{3}{4}} \tag{163}$$

$$+ (\mathbf{3}_{SU(2)} - 1)pq + t^3(pq)^{\frac{5}{8}}(p+q)\mathbf{2}_{SU(2)}\left(y^3\mathbf{8}_{SU(8)} + \frac{1}{y^3}\overline{\mathbf{8}}_{SU(8)}\right)$$

$$+ t^3(pq)^{\frac{9}{8}}\left(\frac{1}{y^9}\mathbf{8}_{SU(8)} + \frac{1}{y^3}\mathbf{56}_{SU(8)} + y^3\overline{\mathbf{56}}_{SU(8)} + y^9\overline{\mathbf{8}}_{SU(8)}\right) + \dots$$

One interesting thing to note about this index is that it forms characters of $SU(2) \times SO(16)$. Specifically, we have that $\mathbf{16}_{SO(16)} \rightarrow y^3\mathbf{8}_{SU(8)} + \frac{1}{y^3}\overline{\mathbf{8}}_{SU(8)}$, $\mathbf{128}'_{SO(16)} \rightarrow \frac{1}{y^9}\mathbf{8}_{SU(8)} + \frac{1}{y^3}\mathbf{56}_{SU(8)} + y^3\overline{\mathbf{56}}_{SU(8)} + y^9\overline{\mathbf{8}}_{SU(8)}$, so in terms of characters the index reads:

$$1 + t^3(pq)^{\frac{5}{8}}\mathbf{2}_{SU(2)}\mathbf{16}_{SO(16)} + \frac{4}{t^6}(pq)^{\frac{3}{4}} + (\mathbf{3}_{SU(2)} - 1)pq \tag{164}$$

$$+ t^3(pq)^{\frac{5}{8}}(p+q)\mathbf{2}_{SU(2)}\mathbf{16}_{SO(16)} + t^3(pq)^{\frac{9}{8}}\mathbf{128}'_{SO(16)} + \dots$$

This suggests that this theory should be associated with the minimal flux preserving a $U(1) \times SU(2) \times SO(16)$ subgroup of $SO(20)$, which is the global symmetry algebra of the 6d SCFT for $N = 2$. We can as before introduce a flux basis for $SO(4N + 12)$ where the roots are spanned by $\frac{1}{2}(1, 1, 0, 0, \dots, 0)$ + permutations + even number of reflections. In this basis, we can associate with this theory a flux given by $(-2, -2, 0, 0, 0, 0, 0, 0, 0, 0)$.[53] This term arises in the gauge theory from the mesons from the fundamental and antifundamental chirals at the two sides.

For general $N$, we expect the enhancement of $U(1) \times SU(2N + 4) \rightarrow SO(4N + 8)$. This can be seen by again looking at the mesons and noting that we now get a state in the $t^{N+1}\mathbf{2}_{SU(2)}(y^{N+1}\mathbf{2N+4}_{SU(2N+4)} + \frac{1}{y^{N+1}}\overline{\mathbf{2N+4}}_{SU(2N+4)})$. This fits with the minimal flux preserving a $U(1) \times SU(2) \times SO(4N + 8)$ subgroup of $SO(4N + 12)$, which is given by $(-2, -2, 0, 0, \dots, 0, 0)$ in our flux basis. This also determines the embedding of the symmetries.

We can next compare anomalies. For this we first need the anomaly polynomial of the 6d SCFT. For the case at hand, this was computed in [94] (see also [19]), and we can just use their results. Alternatively, it can be computed from the gauge theory description similarly to the computation in Lecture III, though now we also need to take into account the contribution of the hypermultiplets (see [94, 117] for the relevant expressions). Either way, we have that,

$$I_{DCM} = \frac{N(10N+3)C_2^2(R)}{24} - \frac{N(2N+9)}{48}p_1(T)C_2(R) - \frac{N}{2}C_2(R)C_2(SO(4N+12))_{\mathbf{4N+12}}$$

$$+ \frac{(N+2)}{24}p_1(T)C_2(SO(4N+12))_{\mathbf{4N+12}} + \frac{(2N+1)}{24}C_2^2(SO(4N+12))_{\mathbf{4N+12}} \tag{165}$$

$$- \frac{(N-1)}{6}C_4(SO(4N+12))_{\mathbf{4N+12}} + (29 + (N-1)(2N+13))\frac{7p_1^2(T) - 4p_2(T)}{5760}.$$

---

[53]Here it is important that the group is actually $Spin(4N+12)/\mathbb{Z}_2$ so this gives the minimal flux preserving the $U(1) \times SU(2) \times SO(16)$ subalgebra. Indeed this is further supported using the logic of [113] (which we do not review here) by noting that the index contain the states $t^3(pq)^{\frac{5}{8}}\mathbf{2}_{SU(2)}\mathbf{16}_{SO(16)}$, which have the precise charges to be coming from the broken currents of $SO(20)$ under that subgroup. We also expect states in the $2t^6(pq)^{\frac{1}{4}}$ from the broken currents. These are just the flip fields which we have not included in the index.

Next we need to introduce the flux. As we mentioned, we expect the theory to be associated with compactifications having a unit of flux preserving the $U(1) \times SU(2) \times SO(4N+8)$ subgroup of $SO(4N+12)$. From our study of the index, we can infer that the expected relation between the 4d and 6d global symmetries should be,

$$\mathbf{4N+12}_{SO(4N+12)} \rightarrow (t^{N+1}+\frac{1}{t^{N+1}})\mathbf{2}_{SU(2)}+y^{N+1}\mathbf{2N+4}_{SU(2N+4)}+\frac{1}{y^{N+1}}\overline{\mathbf{2N+4}}_{SU(2N+4)}. \quad (166)$$

This induces the following relation among the characteristic classes:

$$\begin{aligned}
C_2(SO(4N+12))_{\mathbf{4N+12}} = {} & 2C_2(SU(2))_{\mathbf{2}} + 2C_2(SU(2N+4))_{\mathbf{2N+4}} - 2(N+1)^2 C_1(U(1)_t)^2 \\
& - 2(N+2)(N+1)^2 C_1(U(1)_y)^2,
\end{aligned} \quad (167)$$

$$\begin{aligned}
C_4(SO(4N+12))_{\mathbf{4N+12}} = {} & C_2^2(SU(2))_{\mathbf{2}} + C_2^2(SU(2N+4))_{\mathbf{2N+4}} \\
& + 4C_2(SU(2))_{\mathbf{2}}C_2(SU(2N+4))_{\mathbf{2N+4}} + (N+1)^4 C_1(U(1)_t)^4 \\
& + 2(N+1)^2 C_2(SU(2))_{\mathbf{2}}C_1(U(1)_t)^2 \\
& - 4(N+1)^2 C_2(SU(2N+4))_{\mathbf{2N+4}}C_1(U(1)_t)^2 \\
& - 4(N+2)(N+1)^2 C_2(SU(2))_{\mathbf{2}}C_1(U(1)_y)^2 \\
& - 2(2N+1)(N+1)^2 C_2(SU(2N+4))_{\mathbf{2N+4}}C_1(U(1)_y)^2 \\
& + 4(N+2)(N+1)^4 C_1(U(1)_y)^2 C_1(U(1)_t)^2 \\
& + (N+2)(2N+3)(N+1)^4 C_1(U(1)_y)^4 \\
& + 2C_4(SU(2N+4))_{\mathbf{2N+4}} \\
& - 6(N+1)C_1(U(1)_y)C_3(SU(2N+4))_{\mathbf{2N+4}}.
\end{aligned}$$

Here the flux is inside $U(1)_t$, so we need to take $C_1(U(1)_t) = \frac{1}{N+1}\hat{t} + C_1(U(1)_t^{4d})$, where $C_1(U(1)_t^{4d})$ is the 4d part of the characteristic class. Note that the minimal charge under $U(1)_t$ here is $N+1$, which leads to the normalization factor in front of $t$, which ensures that the flux is of unit value in a normalization where the minimal charge is 1. Inserting all of this into (165) and integrating on the Riemann surface, we arrive at the expected 4d anomaly polynomial:

$$\begin{aligned}
I_{4d} = {} & \frac{2(N+2)(N+1)^3}{3}C_1^3(U(1)_t^{4d}) - \frac{(N+1)(N+2)}{6}p_1(T)C_1(U(1)_t^{4d}) \quad (168) \\
& - 2N(N+1)C_1^2(U(1)_R^{6d})C_1(U(1)_t^{4d}) + 2(N+2)(N+1)^3 C_1^2(U(1)_y)C_1(U(1)_t^{4d}) \\
& - 2N(N+1)C_2(SU(2))_{\mathbf{2}}C_1(U(1)_t^{4d}) - 2(N+1)C_2(SU(2N+4))_{\mathbf{2N+4}}C_1(U(1)_t^{4d}),
\end{aligned}$$

where here we used the decomposition $C_2(R) \rightarrow -C_1^2(U(1)_R^{6d})$ for the characteristic classes of the $SU(2)$ R-symmetry bundle in terms of those of its $U(1)$ Cartan.

We can next compare that against the results from the gauge theory. Specifically, we find that:

$$\begin{aligned}
Tr(U(1)_t^3) = Tr(U(1)_t \times U(1)_y^2) = {} & 4(N+1)^3(N+2), \\
Tr(U(1)_t) = 4(N+1)(N+2), \quad Tr(U(1)_t \times (U(1)_R^{6d})^2) = {} & -4N(N+1), \\
Tr(U(1)_t SU(2)^2) = N(N+1), \quad Tr(U(1)_t SU(2N+4)^2) = {} & N+1, \quad (169)
\end{aligned}$$

with the rest of the anomalies vanishing either trivially or non-trivially. This matches what we expect from 6d, particularly, the anomalies encoded in (168).

## 6.4 From tubes to trinions

So far we discussed a method to conjecture and test 4d theories associated with the compactification of certain 6d SCFTs on tubes with flux. We have noted how we can capitalize on the

understanding of even just one tube, and build from it a wealth of other tubes leading to a potentially extensive understanding of the compactification of 6d SCFTs on flat surfaces with flux, that is tubes and tori. While we shall not discuss this here, we can further extend our understanding to compactifications on surfaces with positive Euler number, like spheres with no punctures, by closing down the punctures (See [136, 139–141].). This, however, leaves the question of understanding the compactification on surfaces with negative Euler number, notably Riemann surfaces of genus $g > 1$, a question which we shall try to address next. The discussion follows the results of [27] (See also [118, 142].).

It turns out that we can make surprising progress on tackling this problem by considering an altogether different and seemingly unrelated problem. The specific problem turns out to be understanding the relationship between RG flows of 6d SCFTs and related RG flows of the 4d theories resulting from their compactifications. As such, we shall next switch gears and consider this problem, after which we shall elucidate on how this new understanding can be put to use in tackling the problem of finding trinions.

**A relationship between flows in different dimensions**

The problem we wish to consider can be simply stated as follows. We have considered the compactification of 6d SCFTs. These are field theories in 6d and can be related to one another via various flows. For flows preserving supersymmetry, we are limited to ones triggered by giving vevs to various fields. This can be separated into two classes: Tensor branch flows and Higgs branch flows. The former are done by giving a vev to a scalar in the tensor multiplet, while the latter are done by giving a vev to a scalar in the hypermultiplet. There are no scalars in the 6d $(1, 0)$ vector multiplet leading to the lack of a "Coulomb" branch for 6d $(1, 0)$ SCFTs. The vevs break conformal invariance and initiate an RG flow starting from the 6d SCFT and ending with a new theory, that in some cases is also a 6d SCFT. Let us assume that is the case and denote the starting 6d SCFT as $T_{6d}^{UV}$, the final 6d SCFT as $T_{6d}^{IR}$, and the operator to which we give the vev as $\mathcal{O}_{6d}$.

Now consider compactifying $T_{6d}^{UV}$ on a Riemann surface to 4d, potentially with flux in its global symmetry. This should result in a 4d $\mathcal{N} = 1$ theory which we denote as $T_{4d}^{UV}$. We know that the 6d SCFT contains the operator $\mathcal{O}_{6d}$, and we expect that this operator might reduce to a 4d operator, $\mathcal{O}_{4d}$. Of course this may not always be the case, but let's for the moment assume it does.[54] Then we can try to give the 4d operator, $\mathcal{O}_{4d}$, a vev as well and thus initiate a 4d RG flow. Again, there is the possibility that the operator $\mathcal{O}_{4d}$ cannot be given a vev, but we shall for the moment assume that this is not the case. We then expect this to cause a 4d RG flow between the 4d theory $T_{4d}^{UV}$, and the end point of the flow, which is some 4d theory that we denote as $T_{4d}^{IR}$. It is then natural to ask does the theory $T_{4d}^{IR}$ also has a higher dimensional interpretation as a compactification of $T_{6d}^{IR}$ and if so what is the compactification data? In other words, instead of asking what happens to specific theories upon dimensional reduction, it is also natural to ask what happens to RG flows upon dimensional reduction.

So far we have been somewhat general regarding our chosen deformation. However, next we wish to concentrate on a specific class of deformations, the ones associated with Higgs branch flows. The main reason for that is the special place occupied by the operators whose vevs cause these flows. Specifically, such operators are short representation of the 6d $(1, 0)$ superconformal group, that is they are BPS operators, and are in fact the shortest possible representations. As such, they form a chiral ring. This is in contrast with tensor multiplets that don't enjoy any such protection. This is useful as it suggests that the resulting 4d operator may also be BPS, and as such enjoy some protection. This reduces the chances that these operators disappear during the flow, and furthermore allows for their detection and study via

---

[54]As in any RG flow the spectrum of operator in the IR does not have to be the same as in UV.

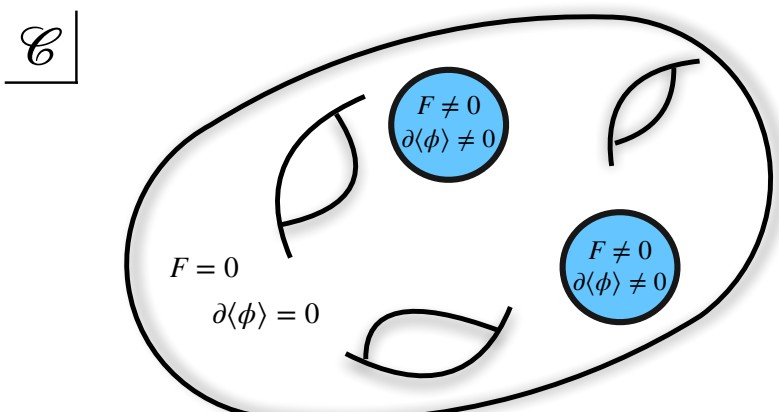

Figure 52: We consider a 6d SCFT on surface $\mathcal{C}$ with some value of flux. We then give a vev to an operator $\phi$ charged under a $U(1)$ symmetry with the flux. Assuming the non trivial gauge fields giving rise to the flux are localized on the surfaces we can turn on a constant vev almost everywhere, but the regions with non trivial gauge fields. The special loci on the surface with non-constant vev turn out to be equivalent to inserting minimal punctures in the IR theory in 6d. The number of such loci is determined by the flux for the $U(1)$ under which $\phi$ is charged.

the superconformal index, which is very convenient for our purposes as it is not too sensitive to potential 4d strong coupling phenomena. As such, we shall concentrate here on the study of the dimensional reduction of such Higgs branch flows.

Consider a 6d SCFT, which can be deformed to a 6d gauge theory by going on the tensor branch. We can go on the Higgs branch by giving a vev to a scalar field, $\phi$, which in the gauge theory description appears as the lowest components of a hypermultiplet.[55] Naturally, the vev must be such that it does not change the energy. Due to the kinetic terms, this necessitates that $\partial < \phi >= 0$, that is the vev is a constant.

So far we have considered the theory in flat spacetime with no background fields. However, we next want to consider the case where the 6d SCFT is compactified on a Riemann surface coupled to background gauge fields. For simplicity, we shall first take the Riemann surface to be a torus supporting non-trivial fluxes associated with global symmetries. The main difference is that we now expect the standard partial derivatives in the kinetic term to be replaced with covariant derivatives, $\nabla$, and as such that the requirement that the vev does not change the energy to be replaced with $\nabla < \phi >= 0$. The latter, however, is in general not satisfied just by $< \phi >$ being a constant.

Naively then, the presence of the flux eliminates the vev, and we may wonder if such flows actually have an image in the dimensionally reduced theory. However, we can try and examine this in more detail by using the domain wall picture. Here we use a specific realization of the flux, such that it is localized at a discrete number of points, relying on the observation that the compactification result generally only depends on the total value of flux and not so much on its explicit realization. We have seen how this can be used to better analyze the reduction process, and we can employ it again for our purposes here.

The main advantage that we gain from this is that we can now choose the flux configuration so that it is localized at only few points. Away from these points, the solution $< \phi >=$ constant

---

[55]More abstractly, these would be lowest components of Higgs branch chiral ring operators. For simplicity, we shall phrase things here mostly using the Lagrangian 6d gauge theory, though we expect things to hold also for the actual 6d SCFT.

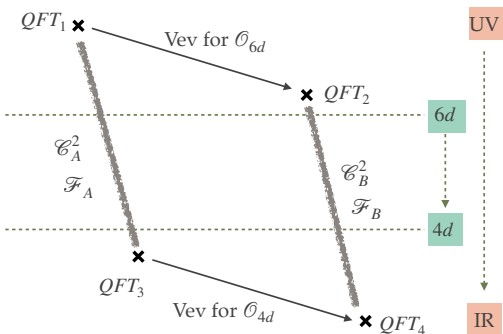

Figure 53: Commuting diagram of flows in 4d, in 6d triggered by vevs, and across dimensions triggered by geometry.

should still be valid so we expect that we should be able to turn on such vevs, with potential modification at the points were the flux is non-vanishing. Such solutions are actually quite common in physics, and are known as vortex solutions. These are ubiquitous in spontaneously broken $U(1)$ gauge theories, and describe configurations in which the Higgs field, whose vev triggers the breaking of the $U(1)$ gauge symmetry, approaches its vev value at infinity, only significantly deviating from it in a localized neighborhood. At this neighborhood the value of the Higgs field approaches zero. Additionally, the flux of the $U(1)$ approaches zero at infinity, only significantly deviating from this value at the localized neighborhood where the Higgs field is nearly vanishing. This neighborhood is then where the flux is localized, and its size defines the size of the vortex. The solution we are proposing then can be thought of as such a vortex solution in the limit where the size of the vortex becomes zero. This analogy is useful as it gives some idea on how these space dependent vevs would look like if one considers more generic flux configurations

We then expect that these vevs can still be turned on but now instead of being strictly constant, they should look like a vortex solution, at least in the limit where the flux is localized around small disjoint patches in space. Now consider the resulting 4d theory. We can look at it from two different perspectives. Consider first performing the compactification and then giving the vev. We shall also assume that the operator acquiring the vev desends to an operator or a collection of operators in the 4d theory. We would then naively expect that giving a vev to this operator should mimic the effect of the 6d vev. See Figure 52 for an illustration.

Now let us consider the case where we give the vev already at the 6d level. We then flow to a new 6d SCFT. We expect that the theory we got by giving a vev to the 4d theory to be also realizable by the compactification of this 6d SCFT. The subtle issue here is the nature of the compactification surface. It turns out that this surface is given by the initial surface with extra punctures, which can be attributed to the points where the vev is not constant. In other words, it seems that the non-constant vevs are manifested in this reduction as additional punctures. In addition the flux of the compactification may change.

The picture emerging from this study is as follows. Consider the theories $T_{6d}^{UV}$, $T_{6d}^{IR}$, $T_{4d}^{UV}$ and $T_{4d}^{IR}$ that we mentioned previously. Recall that $T_{4d}^{UV}$ is the result of the compactification of the theory $T_{6d}^{UV}$ on some Riemann surface with flux. Then we have that $T_{4d}^{IR}$ is also the compactification, of now theory $T_{6d}^{IR}$, on the same surface, but with potentially more punctures and a different value of flux. This idea is illustrated in Figure 5 (which we reproduce here in Figure 53 for convenience).

These results were motivated in [27] by the study of the compactification of the 6d SCFTs living on $N$ M5-branes probing a $\mathbb{C}^2/\mathbb{Z}_k$ singularity. Particularly, these SCFTs possess Higgs

branch flows leading from theories with one value of $k$ to ones with smaller values,[56] and one can study these flows. In this class of theories, a string theoretic picture can be used to argue from the domain wall picture that one indeed gets extra minimal punctures in the theory after the flow. Finally, this can be explicitly checked in various models. The results are that indeed the two 4d theories are related as explained with the number of additional minimal punctures being proportional to the flux felt by the operator receiving the vev.

Interestingly, this picture suggests a curious connection between punctures and and domain walls. Specifically, we noted that if the operator receiving a vev is charged under the symmetry receiving a flux, which we can insert in 5d through a domain wall, then we should get a puncture left at the location of the domain wall. We noted that the domain walls are a specific tractable limit of representing fluxes, and there are in principle various deformations one can take of them. These deformations usually don't affect the IR, or affect it only through marginal operators. An example of these types of deformations is the moving around of the domain walls, where we can even merge several domain walls together to form a different domain wall. Indeed, we have used this previously to construct more complicated domain walls from known ones. We can then wonder what these relations imply for the puncture. Specifically, the motion and union of domain wall is something that also apply to puncture, where we can move punctures around and even in some cases merge them together to form different punctures. The exact relation between the two and what can it teach us about them is something that has not been explored in detail yet. See for related discussion [24,25,116,118].

**Application to the construction of trinions**

So far we have discussed what happens to Higgs branch flows under the process of dimensional reduction. We noted that if two 6d SCFTs are related by a Higgs branch flow then a similar relation may exist for the compactified theory. Specifically, if the 4d theory, resulting from the compactification of the 6d SCFT from which the flow is initiated, has a BPS operator which comes from the 6d Higgs branch chiral ring operator receiving the vev, and if such an operator can receive a vev, then giving it a vev in 4d should produce the image of this flow in 4d. This flow should then lead to a new 4d theory which also possesses a higher dimensional interpretation as the compactification of a 6d SCFT, now the one at the end of the 6d RG flow. However, the details of the relation are sensitive to the flux, specifically the one felt by the operator. If the latter is zero then most of the compactification details are expected to remain unchanged, up to the fact that part of the symmetry is broken by the vev. On the other hand, if the flux felt by the operator is non-zero then some of the details of the compactification will change. Notably, the compactification surface will change by the addition of punctures, number of which is expected to be the number of flux quanta felt by the operators, and whose nature may depend on details involving the flux. Additionally, the flux associated with the compctification will change due to the breaking and identification of symmetries caused by the vev. This is motivated by the domain wall picture of the flux, and has been tested in several models, notably the family of $A$ and $D$ type conformal matter 6d SCFTs, although it is still unclear whether it will hold for arbitrary 6d SCFTs.

This observation has an interesting application that we shall next consider. Say we have an established conjecture for the 4d theories resulting from the tube compactification with global symmetry fluxes of a family of 6d SCFTs, which are related to one another via Higgs branch flows. Such flows are triggered by a vev to a BPS operator in 6d. We can then consider looking for the 4d operator expected to come from such 6d operator. This can be done by reading the global symmetry charges of this operator, translating them to the charges of symmetries in the

---

[56]This class of theories actually possess a very rich structure of Higgs branch flows. Specifically, besides the flows reducing $k$, there are also flows that reduce $N$. Finally, there is a rich structure of additional Higgs branch flows leading to new 6d SCFTs. We shall not consider such flows here.

4d theories and looking for a chiral operator with the same charges.[57] We can then give such 4d operator a vev, assuming this is indeed possible in the 4d theory. If so then this initiates an RG flow in 4d which should be the image of the 6d RG flow.

This flow should end with a 4d theory associated with the compactification of a different 6d SCFT in this family, but now on a surface with different flux and more puncture. The latter point is of special interest to us, as it allows us to gain insight into the compactification of 6d SCFTs in this family on spheres with more than two punctures just from knowledge of the compactification on spheres with two punctures. This is the main application of this observation to understanding the compactification on generic surfaces.

We next want to comment on the number and nature of the puncture. In the spirit of the discussion in this section so far, we shall assume that the 4d theories resulting from the tube compactification of this family of 6d SCFTs were derived using the domain wall method. This would then suggest that the basic models here are associated with very simple domain wall theories, that is ones whose fields living on them are just chiral fields. These domain walls are then thought of as being somewhat elementary on account of not being separable to two simpler domain walls. This then first suggests that they should correspond to minimal flux, and as such it should be possible to get a sphere with three puncture for some tube theory in this family. Second, the simplicity of such domain walls would also lead us to expect the resulting puncture to be of minimal type. This is indeed generically what has been observed in this construction.

More complicated domain walls may lead to other types of punctures. However, in our construction, we opted to construct such domain walls by chaining simpler domain walls together. This has lead to the interpretation where you can represent a flux by a single domain wall, or by chaining of multiple domain walls corresponding to smaller flux. For the case at hand, this suggests that the same may be applied to the punctures, that is that it may be possible to merge multiple punctures to form a puncture of a different type. This all relies on the observation that only a handful of parameters are relevant in the IR theory.

This method can then be used to get theories corresponding to spheres with more than two punctures just from knowledge of the theories associated to spheres with two punctures. However, we don't have that much control on what type of additional punctures would appear. Generically, we expect to get two maximal punctures, coming from the original punctures of the tube and a collection of minimal punctures, roughly associated with the number of basic domain walls needed to engineer the flux. Nevertheless, it might be possible to merge the punctures to form more general punctures, which would then be interpreted as coming from a single domain wall which is the merger of multiple domain walls. In some cases, we can actually form a maximal puncture using this construction, and thus achieve a derivation of a trinion theory [118,142].

We shall next illustrate this method with an example.

**From tubes to trinions for $D$ type conformal matter theories**

Let us briefly apply the general algorithm of deriving spheres with more than two punctures by flowing from $(D_{N+3}, D_{N+3})$ minimal conformal matter tori to $(D_{N+2}, D_{N+2})$ minimal conformal matter tori with punctures. We will rederive the results of [25] using the technology described

---

[57]As we previously mentioned, the Higgs branch chiral ring operators, to which we give a vev to start the flow, are in short representations. Specifically, if the scalar primary is in the **R** dimensional representation of $SU(2)_R$, they obey a shortening condition where the application of the supercharge annihilates the state if the $SU(2)_R$ representation of the final state is of dimension **R** + **1**, see [143] for the details. This suggests that the top (bottom) $SU(2)_R$ component should be annihilated by the $SU(2)_R$ component of the supercharge that raises (lowers) the $SU(2)_R$ spin. These components should then descend to $\mathcal{N} = 1$ chiral fields. Additionally, they should carry the same charges under the flavor symmetry as the primary of the Higgs branch supermultiplet.

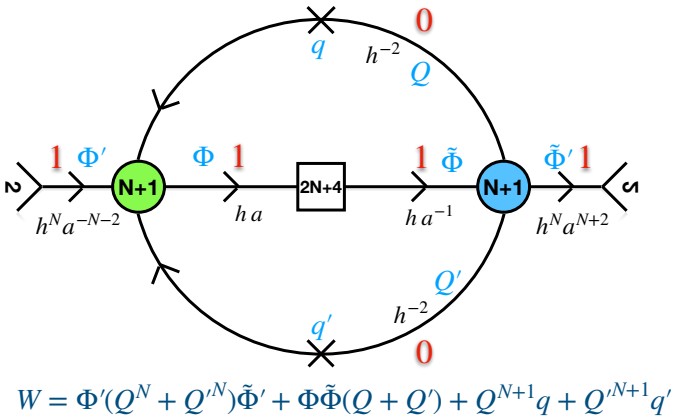

$$W = \Phi'(Q^N + Q'^N)\tilde{\Phi}' + \Phi\tilde{\Phi}(Q + Q') + Q^{N+1}q + Q'^{N+1}q'$$

Figure 54: Two tubes $\Phi$-glued together to form a torus with flux $(-2, -2, 0, 0, \cdots)$. Some terms in the superpotential are irrelevant for general $N$.

above and following [118]. We begin by considering the flow in 6d from the $(D_{N+3}, D_{N+3})$ minimal conformal matter to $(D_{N+2}, D_{N+2})$ minimal conformal matter in flat space. For this, we recall that we can think of the minimal $D_{N+3}$ type conformal matter as the UV completion of the 6d gauge theory with gauge group $USp(2N-2)$ and $2N+6$ fundamental hypermultiplets. The Higgs branch flow in question is described in the gauge theory by giving a vev to two fundamental hypermultiplets, or more correctly, to the quadratic gauge invariant made from them. This gauge invariant is indeed a Higgs branch chiral ring operator, which furthermore is the conserved current supermultiplet.

Next, we look at the 4d theory we associated to the torus compactification of this theory. Specifically, we take the simple torus model obtained by gluing together two tube theories, here reproduced in Figure 54.

We next seek the 4d operator expected to descend from the 6d Higgs branch chiral ring operator in question. In this case, as the operator is just part of the conserved current multiplet, it is easy to identify a possible choice. Specifically, we noted previously that the operator $\Phi'\Phi$, in the notations of Figure 54, matches the 6d expectation as being an operator descending from the broken current multiplet, and as such we shall take this operator.

We then consider giving a vacuum expectation value to this operator. As it is a mesonic operator, it breaks one of the $SU(N+1)$ gauge nodes down to $SU(N)$. We can analyze the flow triggered by the vev using a variety of techniques, *e.g.* the supersymmetric index, as was done in Section 5.4. We will not do so in full detail but just outline the initial important steps. The weight of the operator $\Phi'\Phi$ in the index (which encodes its symmetry) is $x = qp\left(\frac{h}{a}\right)^{N+1}b_1\epsilon^{-1}$. Here $b_i$ ($\prod_{i=1}^{2N+4} b_i = 1$) parametrize the $SU(2N+4)$ global symmetry and $\epsilon$ the $SU(2)$ global symmetry. We thus should set $x = 1$ *e.g.* by setting $\epsilon$ appropriately. We need then to find the locus of the pinching of the $SU(N+1)$ contour integral. Parametrizing the $SU(N+1)$ gauge fugacities by $z_i$ ($\prod_{i=1}^{N+1} z_i = 1$) the contour integral setting $x = 1$ is pinched *e.g.* at $z_1 = (qp)^{\frac{1}{2}} h\, a\, b_1$.[58] Using these values for $z_1$ and $\epsilon$ (following from $x = 1$) one obtains the quiver on the left side of Figure 55.[59]

It is useful to redefine the symmetries in the following way,

$$u^{2N+4} = (a b_1)^{-\frac{1}{N+1}} h^{-\frac{2N+3}{N+1}} (qp)^{-\frac{1}{2N+2} - \frac{1}{2N+4}}, \tag{170}$$

$$v^{2N+4} = h^{-1} a^{N+3} (qp)^{-\frac{1}{2N+4}}, \qquad \tilde{\epsilon} = a^{N+2} b_1^{-1} (qp)^{-\frac{1}{2}}.$$

---

[58]To be precise we parametrize the fields $\Phi'$ as $(qp)^{\frac{1}{2}} h^N a^{-N-2} z_i \epsilon^{\pm 1}$.

[59]The preserved $SU(N)$ gauge symmetry is parametrized by $u_i$ so that $\prod_{i=1}^{N} u_i = 1$ and that $z_i = z^{\frac{1}{N}} u_{i-1}$ for $i > 1$.

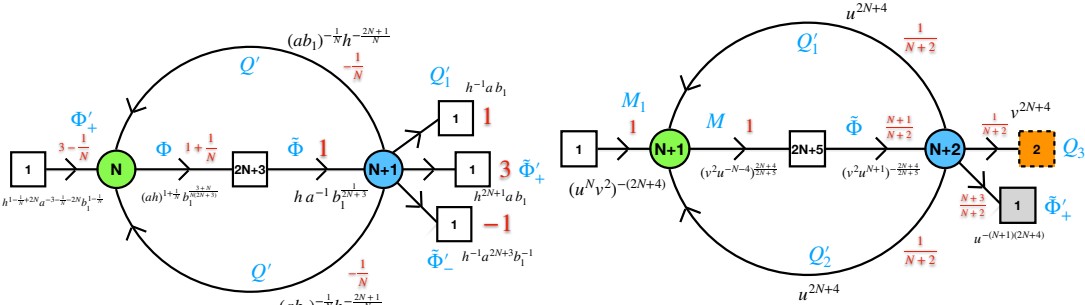

Figure 55: On the left the quiver obtained after the flow using the UV fugacities (dropping the gauge singlet fields). On the right the quiver with redefined (and simplified) definitions of symmetries. We have also shifted $N$ by one on the right side to switch again to the notations with $N = 1$ corresponding to the E-string.

Here $\widetilde{\epsilon}$ is a fugacity parametrizing an $SU(2)$ symmetry appearing on the quiver on the right side of Figure 55: The combination of $U(1)$ symmetries giving $\widetilde{\epsilon}$ enhances to $SU(2)$ in the IR of the flow triggered by the vev. One should turn on all the superpotential terms consistent with the symmetries detailed in Figure 55.

Now we want to interpret the theory on the right side of Figure 55 as a torus compactification of $(D_{N+3}, D_{N+3})$ minimal conformal matter with a single $SU(2)$ minimal puncture. In particular we want to interpret it as a three punctured sphere, with two maximal $SU(N+1)$ punctures and one minimal $SU(2)$ puncture, $\Phi$-glued to itself to form the torus. The quiver of Figure 55 is naturally interpreted as such. The fields $M_1$ and $M$ form $2N+6$-plet of fundamental fields which flip the moment maps when $\Phi$-gluing. The $SU(2)_{\widetilde{\epsilon}}$ is the the symmetry we will associate to the minimal puncture: Note that this is a combination of 6d symmetries of the higher $N$ minimal conformal theory we started with. The gauge symmetry $SU(N+1)$ can be interpreted as gluing the two maximal $SU(N+1)$ punctures. This gives us the quiver of Figure 57 as a candidate for a three punctured sphere for $(D_{N+3}, D_{N+3})$ minimal conformal matter theory.

The trinion we obtain here for $N = 1$ is directly related to the one we discussed in Section 5.1. The trinion of Figure 35 is simply obtained by taking the trinion of Figure 57 and gluing to it the two punctured sphere. The gluing is done by $S$-gluing all the moment maps except for one of the mesonic ones of Figure 57 to one of the $SU(2)$ moment maps of the tube, see Figure 58. The fluxes of the various parts are related as,

$$(-2, 0, 0, \cdots) + (1, -1, 0, \cdots) = (-1, -1, 0, \cdots). \tag{171}$$

We thus obtain here a trinion theory for general values of $N$. For more details of the resulting theory and nature of the minimal puncture see [25].

# 7 Discussion and comments

We will briefly discuss the salient features of the geometric constructions of 4d $\mathcal{N} = 1$ QFTs in 4d by compactifying 6d SCFTs overviewing the relevant literature. We will also comment on some research directions not covered in our Lectures, but which bare direct relevance to the subject. We will end with some general remarks.

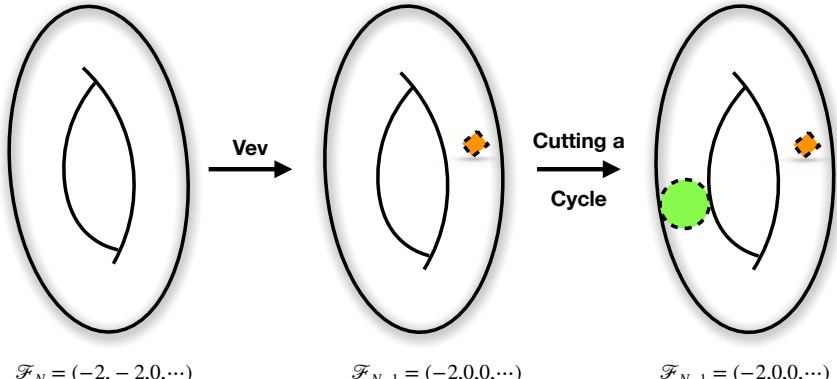

$\mathcal{F}_N = (-2,-2,0,\cdots)$    $\mathcal{F}_{N-1} = (-2,0,0,\cdots)$    $\mathcal{F}_{N-1} = (-2,0,0,\cdots)$

Figure 56: The sequence of operations: We start with a $(D_{N+3}, D_{N+3})$ torus with flux $\mathcal{F}_N = (-2, -2, 0, \cdots)$ preserving $U(1) \times SO(4N+8) \times SU(2)$ symmetry. We turn on then a vev to an operator which is charged under all three factors and breaks the symmetry to $U(1)_{punct.} \times [U(1) \times SO(4N+6)]$ and corresponds to the following flux of the IR theory, $\mathcal{F}_{N-1} = (-2, 0, 0, \cdots)$. We obtain a torus with one puncture. The puncture parametrized by a combination of fugacities of the $D_{2N+6}$ symmetry of the UV theory, $U(1)_{punct.}$, which are not parametrizing the $D_{2N+4}$ symmetry of the IR 6d theory. Finally, we cut the trous along a cycle to obtain a sphere with two maximal and one minimal punctures with the flux equal to the one of the torus.

## Across dimension dualities

We have discussed two examples of across dimension dualities, the 6d minimal $SU(3)$ SCFT [24] and the compactifications of $(D_{N+3}, D_{N+3})$ minimal conformal matter [19, 25, 26, 118], with the $N = 1$ E-string case being our main example. There are several other examples for which such dualities can be derived systematically. First, one can consider compactifications of $A_1$ $(2, 0)$ theory analyzed in the seminal work of Gaiotto [8]. One can repeat our analysis in this case verbatim. For example in this case turning $\mathcal{N} = 2$ preserving flux the 4d theories turn out to be described by conformal Lagrangians and one can explicitly find them using the general technology of Section 3.3. One can analyze domain walls in 5d and obtain tube theories implementing $\mathcal{N} = 1$ preserving flux from which (and the $\mathcal{N} = 2$ trinions) $\mathcal{N} = 1$ $A_1$ class $\mathcal{S}$ theories [128] can be constructed.[60] The 5d domain walls, and the corresponding 4d theories obtained by compactifying on two punctured spheres with some value of flux, of 6d $(ADE, ADE)$ conformal matter SCFTs [19, 126, 130] (see Figure 59 for an example), $D$-type $(2, 0)$ theory probing $A$ type singularity [144], and higher rank E-string theory [132, 136, 145] have been also analyzed giving rise to a large class of across dimension dualities. Using then flows between these theories one can produce [27] models with two maximal and one minimal punctures [118, 142], and in particular class $\mathcal{S}$ theories with a minimal puncture can be derived in this way.[61] There are many other examples of across dimension dualities obtained using various methods: *e.g.* $\mathcal{N} = 1$ deformations of $\mathcal{N} = 2$ Lagrangian theories leading to strongly coupled $\mathcal{N} = 2$ theories again [12, 146–148]; a large set of $\mathcal{N} = 2$ theories which have been shown to have weakly coupled conformal Lagrangians, free or interacting, see *e.g.* [115, 149, 150]; a search over Lagrangians with restricted set of R-charges to describe theories with $\mathcal{N} = 2$ [22, 82] and $\mathcal{N} = 3$ supersymmetry [83].

The evidence that one can derive for dualities across six and four dimensions is similar in

---

[60]The tube models were discussed in [131, 139–141].

[61]Of course historically these models were originally derived using very different methods [8, 125] and these results have been used as checks of the more sophisticated derivations.

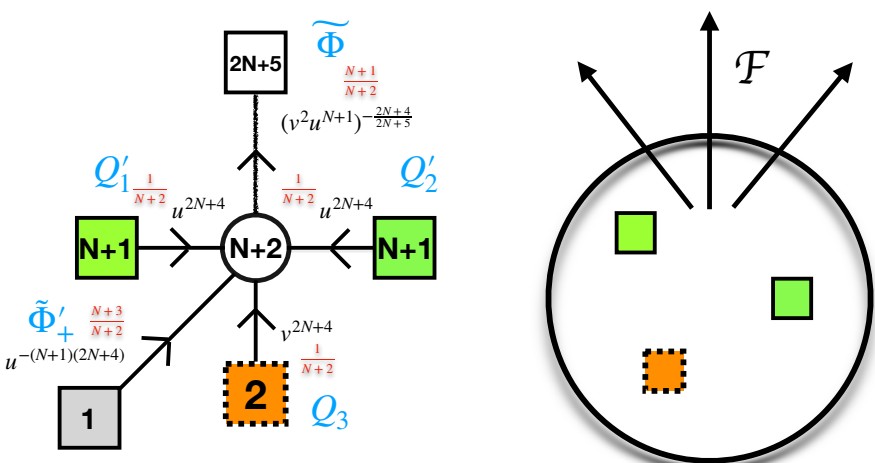

Figure 57: The three punctured sphere obtained by cutting the cycle of the one punctured torus. We have a baryonic superpotential $\tilde{\Phi}'_+ \cdot (Q'^{N+1}_1 + Q'^{N+1}_2)$, which is irrelevant but the case of $N = 1$. The trinion is associated to flux $(-2, 0, \cdots)$ which preserves $U(1) \times SO(4N + 10)$ symmetry. The Figure is taken partially from [25].

nature to the evidence one typically gives for 4d IR duality. First one can match IR symmetries. As both start and end dimensions are even one can also match the 't Hooft anomalies for various continuous symmetries. One can also study consistency of various deformations, and other field theoretic operations, with duality. For example, combining two theories by gauging some symmetries, such that the fluxes associated to the two theories are consistent with bigger symmetry, we should observe the enhanced symmetry in the IR. Certain relevant superpotential deformations in 4d breaking some of the 6d global symmetries should produce a theory with the flux for the broken symmetry vanishing [113, 117]. Finally one can try to map operators across the flow [113, 151]. Local operators in 4d come from local operators and surface operators wrapping the surface in 6d. At least for low quantum numbers such a map can be performed for local operators [113], and it is plausible that a more complete procedure can be devised.

Although many 4d theories obtained in compactifications have an across dimensional dual description, we do not have such a description for the most general compactification. For this reason theories without known across dimension dual are referred to as being currently *non-Lagrangian*. Nevertheless, many of these non-Lagrangian theories are connected to Lagrangian ones by gauging a subgroup of their global symmetry. The canonical example is the Argyres-Seiberg duality [152, 153] relating $\mathcal{N} = 2$ strongly coupled SCFTs to cusps of conformal manifolds of $\mathcal{N} = 2$ Lagrangians. Such $\mathcal{N} = 2$ constructions have a geometric interpretation in terms of pair of pants decompositions of a Riemann surface [8]. Similarly the $\mathcal{N} = 2$ strongly coupled SCFTs can appear at cusps of $\mathcal{N} = 1$ Lagrangian theories [154]. Such relations between Lagrangian and strongly coupled theories, though not giving a description of the strongly coupled SCFT itself, often can be "inverted" to acquire a lot of useful information about protected quantities, such as indices, of the SCFTs [155]. Such inversion procedures have a field theoretical meaning of gauging symmetries emergent at strong coupling [156, 157], an important procedure we will soon discuss.

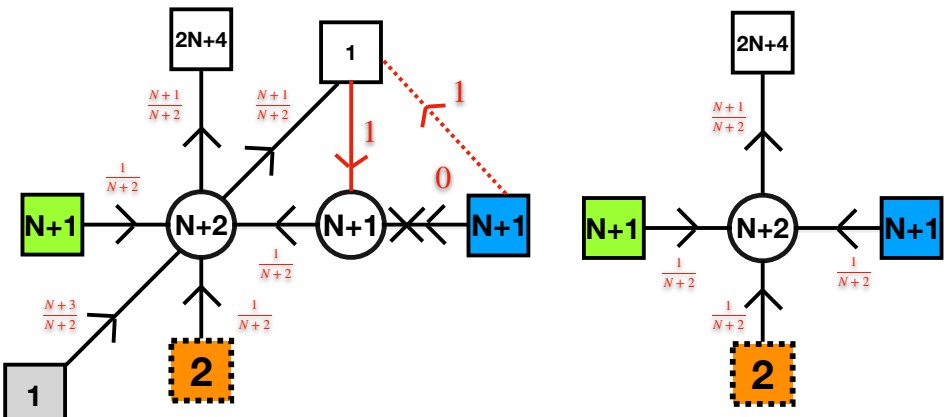

Figure 58: The relations between two trinions. On the left the trinion of Figure 57 glued to the two punctured sphere. The $SU(N+1)$ gauging has $N+2$ flavors and thus it S-confines to give a WZ model of mesons anbaryons. The mesons charged under $SU(N+1)$ puncture symmetry becomes a bifundamental under puncture and $SU(N+2)$ gauge symmetry. The meson built from the $\Phi$-field (denoted by red line) gives mass to one of the $2N+5$ fundamentals. The baryon gives mass to the field charged under $U(1)$ symmetry (denoted by shaded grey on the quiver). Finally, we have a baryon built from the bifundamental and the $\Phi$ field denoted by red solid line which gets a mass with the dashed red line. This theory then flows to the one on the right side. The Figure is taken from [25].

## Atypical degeneration

Analyzing the interplay between 6d and 4d flows one can generate 4d theories corresponding to compactifications on surfaces with additional minimal punctures as we discussed. However, an interesting question is whether one can also obtain surfaces with more general punctures and in particular maximal ones. Maximal punctures are such that we can glue theories together along these punctures by gauging the symmetry associated to them. Thus this will allow to construct theories associated to arbitrary surfaces. A useful observation here is that collections of sub-maximal punctures can collide and form bigger punctures: This effect was dubbed *atypical degenerations* in the case of class $\mathcal{S}$ in [116]. We have encountered such an effect in Section 4.4 discussing compactifications of $SU(3)$ minimal SCFT with triplets of empty punctured building a maximal one. Here the empty punctures have no symmetry while maximal ones have $SU(3)$ symmetry. Similar effects were observed [24] studying compactifications of minimal $SO(8)$ SCFTs: In both of these cases the rank of the symmetry of the maximal puncture is emerging at special loci of the conformal manifolds.

In compactifications of $(D_{N+3}, D_{N+3})$ minimal conformal matter collections of $N$ minimal punctures were argued to form a $USp(2N)$ maximal puncture [118]. Similarly, studying next to minimal $A_k$ conformal matter (2 M5 branes on $\mathbb{Z}_k$ orbifold) it was argued that collections of $k$ minimal punctures form a maximal puncture [118] of a novel type: In both these cases the rank of the maximal puncture is the same as the collection of colliding minimal punctures but the non-abelian structure is enhanced. The new puncture in the $A_k$ case can be obtained by studying compactifications on a circle of the $A_k$ conformal matter with no holonomies turned on [17]. The resulting QFT in 5d can be thought of as a gauging of a strongly coupled SCFT in 5d. In turn this SCFT has a real mass deformation leading to a gauge theory description. Thus in terms of the 5d Lagrangian constructions this puncture corresponds to gauging an emergent UV symmetry. This underscores the need to understand and classify *all* possible 5d

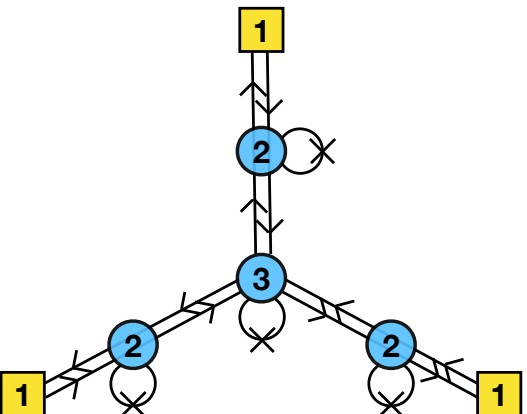

Figure 59: Compactification of minimal $(E_6, E_6)$ conformal matter theory on a torus with 1/6 unit of flux breaking $E_6 \times E_6$ 6d symmetry down to $U(1) \times E_6$ results in the depicted conformal quiver theory [126]. This is in fact $\mathcal{N} = 2$ quiver shaped as the affine Dynkin diagram of $E_6$ with four decoupled free chiral fields (denoted by the crosses).

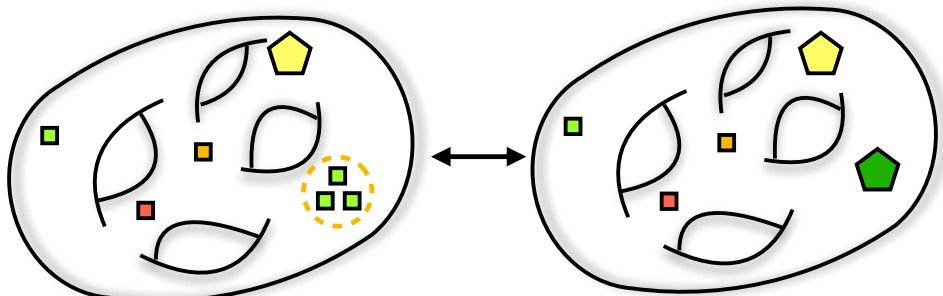

Figure 60: Atypical degenerations of the surface. In numerous examples in certain limits of collision of some types of punctures one can obtain a new type of a puncture. The symmetry of the new "composite" puncture might be different than the symmetry of the punctures building it. Conversely, there are examples of exactly marginal deformations breaking the puncture symmetry and not corresponding to naive geometric deformations of the surface.

effective descriptions of 6d SCFTs when studying compactifications to 4d. To build higher genus surfaces gauging symmetries of colliding minimal punctures we thus need to gauge emergent symmetries on special loci of the conformal manifold (or in the IR).

Conversely considering a theory with a puncture (maximal or non-maximal) there might be exactly marginal deformations which break subgroups of the puncture symmetry. We have seen an example of this in Appendix D: $A_2$ class $\mathcal{S}$ compactifications with minimal punctures have exactly marginal deformations which break the puncture symmetries. These exactly marginal deformations preserve only $\mathcal{N} = 1$ supersymmetry and do not correspond to complex structure moduli or the flat connections on the surface.[62] It would be interesting to understand whether these directions on the conformal manifold can be understood geometrically.

---

[62]For more general $A_{N>1}$ class $\mathcal{S}$ compactifications one also has $\mathcal{N} = 1$ preserving deformations when certain non maximal punctures appear on the surface with the puncture symmetries broken to a subgroup on a general locus of the conformal manifold [158].

## Gauging emergent symmetries

An important procedure which appeared several times in our discussion is the utility of gauging emergent symmetries. In particular some symmetries enhance in the IR of an RG flow or at some strong coupling loci of the conformal manifold allowing to consider coupling the emergent conserved currents to dynamical gauge fields. This is a manifestly strongly coupled procedure. Although the gauge coupling might be weak the starting point of the construction is strongly coupled. Nevertheless, if the rank of the symmetry is manifest in the UV, or at weak coupling of the conformal manifold, a lot of the protected information of the gauge theory can be explicitly computed. This follows from the fact that the supersymmetric partition functions related to counting problems are independent of the RG flow and the continuous parameters of the theory. Several of the 4d theory constructions corresponding to compactifications of 6d SCFTs obtained till now involve such gaugings [24, 26, 117, 118, 125, 132, 156, 157]. In some cases the emergent symmetry in one duality frame might be explicit symmetry of the Lagrangian in another duality frame and thus the gauging would be completely weakly coupled. An example is $(D_5, D_5)$ minimal conformal matter where the trinion is constructed as a four punctured sphere with two maximal $SU(3)$ punctures and two minimal $SU(2)$ punctures colliding to form a $USp(4)$ maximal puncture somewhere on the conformal manifold [118]. However, using a sequence of Seiberg dualities one can obtain a description with the the two $SU(3)$ symmetries and the $USp(4)$ symmetry manifest [127]. An interesting question in this context is whether there are any fundamental obstructions for having a weakly coupled description of a given theory with a given symmetry manifest. We have discussed several examples of theories with weakly coupled descriptions and emergent symmetry in 4d such that the 6d dual description has the symmetry manifest but is not weakly coupled; and the question is whether we can or cannot find a description with manifest symmetry in 4d. For $\mathcal{N} = 2$ theories in 4d many examples of theories with no Lagrangians with manifest symmetry are known just by listing all the possible manifestly $\mathcal{N} = 2$ supersymmetric Lagrangians [9]. A related question would be then under which conditions we can find say $\mathcal{N} = 1$ Lagrangians with manifest global symmetry for such models.[63]

## General types of punctures: 5d UV dualities

A useful knob in constructing 4d theories corresponding to a surface is through gluing theories corresponding to smaller surfaces with the gluing done along a puncture. A given 6d SCFT might have a variety of types of maximal punctures along which we can glue.[64] Such maximal punctures correspond as we have discussed to circle compactifications of the 6d SCFT, *i.e.* 5d gauge theories UV completed by the 6d SCFT. Compactifying on a circle, if the 6d (1, 0) SCFT has a global symmetry, one has a choice of a holonomy in this global symmetry to be turned on around the circle. Different such holonomies might lead to different 5d descriptions, see *e.g.* [96–98, 112, 164–169]. Such different 5d effective descriptions are UV dual to each other in the sense that they are different deformations of the same UV SCFT. We have mentioned above the relatively simple constructions of three punctured spheres *e.g.* for the $(D_{N+3}, D_{N+3})$ minimal conformal matter with two $SU(N+1)$ maximal punctures and one $USp(2N)$ puncture, although the trinions with three punctures of the same type are more complicated (These can be obtained from the former by gluing in two punctured spheres.). The 5d UV dualities can lead to 4d IR dualities by constructing same surfaces from building blocks having different

---

[63]Such questions can be phrased also in other dimensions. For example let us mention here a simple example in 3d where there are (at least) three different descriptions manifesting only two of three symmetries: $\mathcal{N} = 2$ supersymmetry, $SU(3)$ global symmetry, or time reversal symmetry [159–163].

[64]One might have also more general "fixtures" connecting different types of maximal and non-maximal punctures. See *e.g.* [149].

types of maximal punctures, see Figure 61 for an example. It is thus very useful to completely map out all the possible 5d theories UV completed by 6d SCFTs and interrelations between these.

## Integrable models

Yet another interconnection between physics in different dimensions goes through the appearance of quantum mechanical integrable models in various counting problems. The integrable models have a long history of interconnections with gauge theories in various dimensions, see *e.g.* [170]. The connection most relevant to us here goes as follows. One considers a 6d $(1,0)$ SCFT (eight supercharges), compactifies to 4d breaking half of the supercharges, and considers surface defects in the 4d theory. In particular one can count protected operators (the $\mathbb{S}^3 \times \mathbb{S}^1$ partition function, the index) of the 4d theories in the presence of such defects. Using general considerations the resulting partition function can be obtained by acting on the partition function of the theory without the defect with an analytic difference operator [119].[65] We have different operators labeled by the surface defect we want to introduce. These operators can be thought of in general as Hamiltonians of an integrable relativistic quantum mechanical system. The choices of the defects we are considering are determined by the 6d SCFT and are the same for all the 4d theories obtained by compactifications of this SCFT. In particular these integrable models are thus labeled by a theory with eight supercharges, which bares a direct relation to the constructions of [172]. These integrable systems can be explicitly derived by studying analytic structure of the indices. In particular they are obtained by computing the residues of poles the index can have in presence of *minimal punctures* and are acting on the fugacities of a *maximal puncture*. As such these operators are then also labeled by the choice of the *maximal puncture i.e.* the choice of the 5d effective gauge theory description.

The various duality properties of the 4d theories obtained in compactifications imply mathematical properties of the operators introducing defects [119].[66] For example, the operators have to commute and the indices themselves give *Kernel functions* of these operators,

$$\mathcal{O}_\ell^{(6d\,CFT;5d\,QFT^1)}(\mathbf{x}_{5d}^1|\mathbf{y}_{6d}) \cdot \mathcal{I}(\mathbf{x}_{5d}^1, \mathbf{x}_{5d}^2|\mathbf{y}_{6d}) = \mathcal{O}_\ell^{(6d\,CFT;5d\,QFT^2)}(\mathbf{x}_{5d}^2|\mathbf{y}_{6d}) \cdot \mathcal{I}(\mathbf{x}_{5d}^1, \mathbf{x}_{5d}^2|\mathbf{y}_{6d}).$$

Here $\mathcal{O}_\ell^{(6d\,CFT;5d\,QFT^1)}(\mathbf{x}_{5d}^1|\mathbf{y}_{6d})$ is the operator corresponding to the introduction of a defect labeled by $\ell$ acting on a maximal puncture coming from the 5d $QFT^1$ description of the 6d $CFT$ on a circle. The supersymmetric index $\mathcal{I}(\mathbf{x}_{5d}^1, \mathbf{x}_{5d}^2|\mathbf{y}_{6d})$ corresponds to a surface with at least two maximal punctures of types (5d $QFT^1$) and (5d $QFT^2$), and $\mathbf{y}_{6d}$ label the Cartan of the 6d global symmetry. Such Kernel functions relations are highly non trivial and thus give important mathematical checks of the conjectured across dimension dualities.

Explicitly for compactifications of the ADE $(2,0)$ theories the relevant models turn out to be elliptic ADE Ruijsenaars-Schneider systems [119,173]; for the rank $N$ E-string this is the $BC_N$ van Diejen model [127,174] (this was shown explicitly for $N = 1$ using the index methods and conjectured for higher $N$); for minimal 6d SCFTs with $SU(3)$ and $SO(8)$ gauge groups these were discussed in [120,121]; for $A$ type conformal matter the operators were discussed in [119,175]; for minimal $D$ type conformal matter the operators were computed in [127]. These operators can be obtained independently by studying Seiberg-Witten curves [176,177] directly in 6d [178]. One way to understand this correspondence between indices and integrable models is as a version of AGT correspondence where we view the 6d theory on $\mathbb{S}^3 \times \mathbb{S}^1 \times \mathcal{C}$ (with $\mathcal{C}$ being a Riemann surface) and take either $\mathcal{C}$ small to obtain the index in 4d, or the $\mathbb{S}^3 \times \mathbb{S}^1$ to be small to obtain a TQFT on $\mathcal{C}$ (which should be related to the integrable

---

[65]One can also construct in principle such operators by explicitly coupling 2d degrees of freedom to 4d ones [171].

[66]See [127] for a recent overview.

model) [179]. In the case of (2,0) theories and a particularly simple version of the index, the Schur index [180, 181], the TQFT turns out to be the q-deformed YM [180, 182–187]. The Schur index then has further relations to beautiful mathematical structures such as chiral algebras in 2d [188].[67]

## 6d dualities

Studying compcatifications of different 6d SCFTs on different surfaces to 4d one at times obtains same IR 4d theories, or more generally same 4d theories up to decoupled free fields. Such 6d dualities were analyzed for the relations between (2,0) compactifications on punctured spheres and (1,0) on tori with no flux [16–18]. Let us give here three examples of such dualities between two (1,0) theories.

First, let us discuss the example mentioned in the bulk of the paper: *rank one E-string on a genus two surface with no flux is the same as minimal $SU(3)$ (1,0) 6d SCFT on a sphere with four maximal punctures (and $\mathbb{Z}_3$ twist lines)*. The resulting theory in 4d has a large conformal manifold and the statement of the duality is that the two theories reside on the same manifold. In particular, considering the E-string compactification the number of the marginal operators minus the conserved currents is given by,

$$3g_1 - 3 + \mathbf{248}_{E_8}(g_1 - 1) + 1\big|_{g_1=2} \quad \rightarrow \quad \dim \mathcal{M}_c = 252. \tag{172}$$

Here $g_1 = 2$ is the compactification genus. The first two terms are the expected marginal operators minus the current coming from complex structure moduli and the flat connections for the $E_8$ symmetry. The last deformation is accidental for this compactification. On the other hand considering the same quantity for the $SU(3)$ minimal SCFT on the four punctured sphere we obtain,

$$3g_2 - 3 + s_2 + \sum_{i=1}^{s_2}(\mathbf{10}_i - \mathbf{8}_i) + 3 \times (\mathbf{3}_1, \mathbf{3}_2, \mathbf{3}_3, \mathbf{3}_4)\Big|_{g_2=0, s_2=4} \quad \rightarrow \quad \dim \mathcal{M}_c = 252. \tag{173}$$

Here $\mathbf{R}_i$ denotes the representation $\mathbf{R}$ of the $i$th $SU(3)$ puncture symmetry group. The first term comes from the complex structure moduli and the second from the punctures, while the last one is accidental for this compactification. Going along this large conformal manifold, likely including the accidental exactly marginal directions in the two cases, the two theories can be then connected.

Second example is as follows [19]. *Compactifications of conformal matter theories residing on 2 M5 branes probing $\mathbb{Z}_{z(N+1)}$ singularity on a sphere with $N + 1$ minimal and two maximal punctures and flux $\mathcal{F} = z\frac{N+1}{2}$ in the $U(1)_t$ subgroup of the $SU(k) \times SU(k) \times U(1)_t \subset SU(2k)$ symmetry of the 6d theory is dual (up to $N(N+1)$ decoupled free fields) to the compactification of $(D_{N+3}, D_{N+3})$ minimal conformal matter theory on a torus with flux z breaking the $SO(4N + 12)$ symmetry of the 6d SCFT to $U(1) \times SU(N+1) \times SO(10+2N)$ for integer z.* See Figure 61 This duality produces field theories which are 6d dual to each other up to free fields. The simplest case of this duality is taking $N = 1$ and $z = \frac{1}{2}$ where $\mathcal{N} = 2$ $SU(2)$ $N_f = 4$ SCFT with a decoupled chiral field, *i.e.* $A_1$ (2,0) theory on a four punctured sphere (plus a decoupled field), is equivalent to the rank one E-string on a torus with half a unit of flux breaking $E_8$ to $U(1) \times SO(8) \subset U(1) \times E_7$ [26].[68]

---

[67]Let us mention here that in the case that an $\mathcal{N} = 1$ theory is not chiral and has rationally quantized R-charges one can define a simplified version of the $\mathcal{N} = 1$ index [154] which coincides with the Schur index for the $\mathcal{N} = 2$ theories.

[68]This compactifcation of the E-string is actually naively consistent with $SO(8)$ exchanged with any rank four subgroup of $E_7$, *e.g.* $G = F_4, USp(8)$ [26]. This is in particular responsible for the observations that the protected spectrum of the theory forming irreps of either of these groups, see *e.g.* [189].

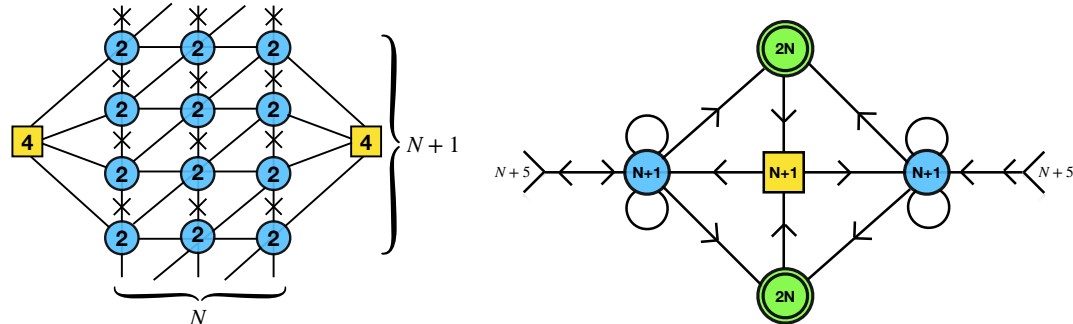

Figure 61: On the left we have the theory obtained by compactifying $(D_{N+3}, D_{N+3})$ minimal conformal matter theory on a torus with one unit of flux breaking the $SO(4N + 12)$ symmetry of the 6d SCFT to $U(1) \times SU(N + 1) \times SO(10 + 2N)$. On the right we have a dual theory [19] (double circles correspond to $USp$ groups and the lines starting and ending on the same vertex are in antisymmetric representation). We assume that $N$ is an odd integer. Note the two dual theories manifest different subgroups of the global symmetry group. The model on the right is obtained by gluing two tubes with $USp(2N)$ and $SU(N+1)$ punctures while on the left we glue tubes with $SU(2)^N$ punctures. This duality follows from a 5d UV duality. The theory on the left (removing the free flip fields) can be alternatively obtained by compactifying the SCFT on two M5 branes probing $\mathbb{Z}_{N+1}$ singularity on a sphere with two maximal and $N + 1$ minimal punctures: This is thus an example of a 6d duality.

The last example is as follows. *Taking $A_{N-1}$ (2, 0) theory on a sphere with two maximal and two minimal punctures (half positive and half negative in the notations of [128]) and zero flux is dual to compactification of $(D_{N+1}, D_{N+1})$ minimal conformal matter theory on a sphere with two maximal and a minimal puncture, and half a unit of flux breaking $SO(4 + 4N)$ 6d symmetry to $U(1) \times SU(2) \times SO(4N)$*. In both of these cases we obtain $\mathcal{N} = 1$ $SU(N)$ SQCD with $2N$ flavors. Again the explicit symmetry of the theories in the two compactifications is different: In former the symmetry is $SU(N)^2 \times U(1)^3$ and in the latter it is $SU(N-1)^2 \times SU(2) \times SU(2N) \times U(1)^3$, and the two compactifications are dual to each other exploring the large conformal manifold.

It will be extremely interesting to understand these and other 6d dualities by embedding them for example in string theory.

## More comments

Let us briefly mention several additional related avenues of research. First, one can consider compactification scenarios in the holographic regime. In particular when the 6d SCFTs are obtained on collections of M5 branes one can consider taking the number of branes to be large and study the corresponding supergravity backgrounds. This for example was done in the class $\mathcal{S}$ context in [128, 190–192] and studying branes on orbifolds in [137, 193, 194]. Another property which we did not address in this review is the higher form/higher group/categorical symmetries [28, 29, 195] of the 4d theories derived in compactifications. For some studies of these issues see [194, 196–199].

It is interesting to consider compactifications to lower dimensions (3d and 2d). This can be done in a variety of ways. For example, one can consider compactifications on a circle of the 4d theories obtained by first considering 6d SCFTs on a surface $\mathcal{C}$. The dualities following from the compactifciations in 4d will then give rise to a rich set dualities in 3d [13, 200]. Moreover the resulting 3d theories might have novel 3d dual descriptions which might not trivially lift

to 4d.[69] These typically take the form of *mirror* duality [201]. The point is that 3d dynamics is richer than the 4d one: For example abelian gauge theories typically lead to interesting SCFTs and there is no upper bound on the matter representations to lead to interacting theories in the IR. Although for compactifications of $(2,0)$ theories such mirror dualities are extensively studied, see *e.g.* [141, 201–206], for compactifications of $(1,0)$ theories on $\mathcal{C} \times \mathbb{S}^1$ not much is known.

One can also study compactifications on surfaces starting from 5d [207, 208] or 4d [209–211]. Similarly we can consider compactifying on circles starting from 5d [212] and 3d [213–216]. One can motivate through these constructions various dualities in 3d. See *e.g.* [217]. Finally one can consider compactifications on higher dimensional surfaces. Starting from 6d we can compactify on three manifolds [15, 218–222] or four manifolds [223, 224].

## Final remarks

Let us make several more philosophical remarks. Our discussion of 4d supersymmetric dynamics has been performed using standard Lagrangian techniques in 4d and 5d. The starting point in 6d is a strongly coupled SCFT, and although some field theoretic Lagrangian tools (such as tensor branch descriptions) can be utilized, at the moment to fully understand such models one usually includes string theoretic constructions (see [10] for a review). One can draw several general lesson from our considerations. First, it is often the case that if one wants to understand a weakly coupled UV description of a given IR physics it is useful to give up some of the symmetry and let it emerge only in the IR. In our discussion the symmetry was typically continuous global symmetry, but this can be also supersymmetry. Related to this there are various suggestions to understand 6d SCFTs themselves by giving up some symmetry, in this case space-time symmetry. These include engineering 6d SCFTs as limits of 4d quiver theories through the procedure of deconstruction [225, 226], trying to decode the full 6d physics from its circle compactifications [108, 109], or DLCQ matrix model [227, 228].[70] Giving up symmetry we can gain many insights and it will be interesting to understand the limits of such an approach. A complimentary way of thinking, at least about the conformal fixed points, is giving up description in terms of weakly coupled fields completely and making an emphasis on extracting all the possible physics from symmetries, including the conformal symmetry. This bootstrap approach has led to many beautiful results in recent years, see *e.g.* [230–232]. Another more abstract approach applicable to theories with well defined Coulomb branches, such as $\mathcal{N} = 2$ theories, is to study the Coulomb branches abstractly without use of a Lagrangian using the Seiberg-Witten curve associated to them [233–235].[71] The program of systematically studying the curves and extracting physics from them is pursued *e.g.* in [242–245]. Even using the geometric techniques one can search for general expressions of various quantities describing SCFTs such that the geometry will be manifest and no use of Lagrangians made. Examples of this are the AGT correspondence [44] (see [246] for a review) relating the $\mathbb{S}^4$ partitions function of $\mathcal{N} = 2$ theories to 2d CFT correlators and the relation of the supersymmetric index to integrable models and to 2d TQFTs [179] we discussed above.[72]

---

[69]However, some mirror dualities have a surprizing lift to 4d. For example, a canonical example of mirror selfdual theory, the 3d $T[SU(N)]$ model residing on S-duality domain wall of $\mathcal{N} = 4$ $SU(N)$ SYM in 4d, can be uplifted to 4d [132, 145]. In fact the 4d theory is the one obtained by compactifying rank $N$ E-string on a two punctured sphere: The mirror duality exchanges the two punctures. One of the puncture symmetries is emergent in the IR and thus the exchange of punctures is not a trivial operation from the point of view of the UV Lagrangian.

[70]See [229] discussing all three approaches.

[71]This approach can be applied in certain cases also for $\mathcal{N} = 1$ theories [236], which was studied for theories obtained from M5 branes on $A$ type orbifolds in [237–241].

[72]Sometimes these types of relations are referred to as BPS/CFT correspondence, see *e.g.* [247, 248].

Another comment we want to make is that although all of our discussion is $\mathcal{N} = 1$ supersymmetric in 4d, that is relies on having four supercharges, the starting point in 6d has eight supercharges, that is has $\mathcal{N} = 2$ supersymmetry in 4d counting. By considering such theories on surfaces and turning on various knobs and levers we can derive a plethora of $\mathcal{N} = 1$ dynamics. An interesting question is then to what extent all supersymmetric phenomena have their origin in theories with eight supercharges. This question can be of course extended by adding more supercharges and imbedding everything in string theory, if one is willing to leave the cradle of purely QFT discussion.[73] A way to phrase a similar question is to what extent all non trivial dynamics in lower dimensions follows from QFT constructions involving geometric deformations starting in 6d, and/or going beyond QFT in string theory: *e.g.* are there CFTs in lower dimensions which cannot be constructed as limits and deformations of supersymmetric CFTs in 6d and/or of string theoretic setups?

## Acknowledgments

We would like to thank our numerous amazing collaborators for sharing with us their ideas and insights. SSR is grateful to the organizers of the "Cargese Summer School: Quantum Gravity, Strings and Fields" (2018), "ICTP spring school on superstring theories and related topics" (2019), and "The 15th Kavli Asian Winter School on Strings, Particles and Cosmology" (2021) where parts of the material discussed here were presented.

**Funding information**    The research of SSR, OS, and ES was supported in part by Israel Science Foundation under grant no. 2289/18, by I-CORE Program of the Planning and Budgeting Committee, by a Grant No. I-1515-303./2019 from the GIF, the German-Israeli Foundation for Scientific Research and Development, and by BSF grant no. 2018204. The research of SSR was also supported by the IBM Einstein fellowship of the Institute of Advanced Study and by the Ambrose Monell Foundation. The research of ES was also supported by the European Union's Horizon 2020 Framework: ERC grant 682608 and the "Simons Collaboration on Special Holonomy in Geometry, Analysis and Physics". The research of OS was also supported by the Clore Scholars Program and by the Mani L. Bhaumik Institute for Theoretical Physics at UCLA. GZ is supported in part by the ERC-STG grant 637844-HBQFTNCER, by the INFN and by the Simons Foundation grant 815892.

## A    Superconformal algebra in 4d

In this appendix we consider the spacetime symmetry algebra $\mathfrak{sl}(4|1)$ of four dimensional conformal field theories with $\mathcal{N} = 1$ supersymmetry, and its representations.

We begin with the bosonic conformal subalgebra. Written using spinorial notation with $\alpha, \dot{\alpha} = \pm, \dot{\pm}$, the nonvanishing commutators are given by

$$
\begin{aligned}
\left[ (J_1)_\alpha^{\ \beta}, (J_1)_\gamma^{\ \delta} \right] &= \delta_\gamma^\beta (J_1)_\alpha^{\ \delta} - \delta_\alpha^\delta (J_1)_\gamma^{\ \beta} \,, \\
\left[ (J_2)_{\dot{\beta}}^{\ \dot{\alpha}}, (J_2)_{\dot{\delta}}^{\ \dot{\gamma}} \right] &= \delta_{\dot{\delta}}^{\dot{\alpha}} (J_2)_{\dot{\beta}}^{\ \dot{\gamma}} - \delta_{\dot{\beta}}^{\dot{\gamma}} (J_2)_{\dot{\delta}}^{\ \dot{\alpha}} \,, \\
\left[ (J_1)_\alpha^{\ \beta}, P_{\gamma\dot{\gamma}} \right] &= \delta_\gamma^\beta P_{\alpha\dot{\gamma}} - \frac{1}{2} \delta_\alpha^\beta P_{\gamma\dot{\gamma}} \,,
\end{aligned}
\tag{A.1}
$$

---

[73] Note that this is a different question than the one discussed in the context of the swampland program [249–251]. In the swampland context one constrains $D$-dimensional low energy QFTs coupled in the UV to $D$-dimensional gravity. Here we allow $D$-dimensional QFTs to be coupled to higher dimensional gravity in the string constructions.

$$\left[(J_2)^{\dot\alpha}_{\dot\beta}, P_{\gamma\dot\gamma}\right] = \delta^{\dot\alpha}_{\dot\gamma} P_{\gamma\dot\beta} - \frac{1}{2}\delta^{\dot\alpha}_{\dot\beta} P_{\gamma\dot\gamma},$$

$$\left[(J_1)^{\beta}_{\alpha}, K^{\dot\gamma\gamma}\right] = -\delta^{\gamma}_{\alpha} K^{\dot\gamma\beta} + \frac{1}{2}\delta^{\beta}_{\alpha} K^{\dot\gamma\gamma},$$

$$\left[(J_2)^{\dot\alpha}_{\dot\beta}, K^{\dot\gamma\gamma}\right] = -\delta^{\dot\gamma}_{\dot\beta} K^{\dot\alpha\gamma} + \frac{1}{2}\delta^{\dot\alpha}_{\dot\beta} K^{\dot\gamma\gamma},$$

$$[H, P_{\alpha\dot\alpha}] = P_{\alpha\dot\alpha},$$

$$\left[H, K^{\dot\alpha\alpha}\right] = -K^{\dot\alpha\alpha},$$

$$\left[K^{\dot\alpha\alpha}, P_{\beta\dot\beta}\right] = \delta^{\alpha}_{\beta}\delta^{\dot\alpha}_{\dot\beta} H + \delta^{\alpha}_{\beta}(J_2)^{\dot\alpha}_{\dot\beta} + \delta^{\dot\alpha}_{\dot\beta}(J_1)^{\alpha}_{\beta}.$$

We next turn to the part of the algebra involving fermionic generators. There are eight such generators, the four Poincare supercharges $Q_\alpha, \tilde{Q}_{\dot\alpha}$ and the four conformal supercharges $S^\alpha, \tilde{S}^{\dot\alpha}$, with the following nonvanishing anticommutators,

$$\{Q_\alpha, \tilde{Q}_{\dot\alpha}\} = P_{\alpha\dot\alpha},$$

$$\{S^\alpha, \tilde{S}^{\dot\alpha}\} = K^{\dot\alpha\alpha},$$

$$\{Q_\alpha, S^\beta\} = \frac{1}{2}\delta^{\beta}_{\alpha}\left(H + \frac{3}{2}R\right) + (J_1)^{\beta}_{\alpha}, \tag{A.2}$$

$$\{\tilde{Q}_{\dot\alpha}, \tilde{S}^{\dot\beta}\} = \frac{1}{2}\delta^{\dot\beta}_{\dot\alpha}\left(H - \frac{3}{2}R\right) + (J_2)^{\dot\beta}_{\dot\alpha}.$$

We are left with the following commutators of the fermionic generators with the bosonic ones,

$$\left[(J_1)^{\beta}_{\alpha}, Q_\gamma\right] = \delta^{\beta}_{\gamma} Q_\alpha - \frac{1}{2}\delta^{\beta}_{\alpha} Q_\gamma,$$

$$\left[(J_2)^{\dot\alpha}_{\dot\beta}, \tilde{Q}_{\dot\gamma}\right] = \delta^{\dot\alpha}_{\dot\gamma} \tilde{Q}_{\dot\beta} - \frac{1}{2}\delta^{\dot\alpha}_{\dot\beta} \tilde{Q}_{\dot\gamma},$$

$$\left[(J_1)^{\beta}_{\alpha}, S^\gamma\right] = -\delta^{\gamma}_{\alpha} S^\beta + \frac{1}{2}\delta^{\beta}_{\alpha} S^\gamma,$$

$$\left[(J_2)^{\dot\alpha}_{\dot\beta}, \tilde{S}^{\dot\gamma}\right] = -\delta^{\dot\gamma}_{\dot\beta} \tilde{S}^{\dot\alpha} + \frac{1}{2}\delta^{\dot\alpha}_{\dot\beta} \tilde{S}^{\dot\gamma},$$

$$[H, Q_\alpha] = \frac{1}{2}Q_\alpha, \qquad [H, \tilde{Q}_{\dot\alpha}] = \frac{1}{2}\tilde{Q}_{\dot\alpha},$$

$$[H, S^\alpha] = -\frac{1}{2}S^\alpha, \qquad [H, \tilde{S}^{\dot\alpha}] = -\frac{1}{2}\tilde{S}^{\dot\alpha}, \tag{A.3}$$

$$[R, Q_\alpha] = -Q_\alpha, \qquad [R, \tilde{Q}_{\dot\alpha}] = \tilde{Q}_{\dot\alpha},$$

$$[R, S^\alpha] = S^\alpha, \qquad [R, \tilde{S}^{\dot\alpha}] = -\tilde{S}^{\dot\alpha},$$

$$\left[K^{\dot\alpha\alpha}, Q_\beta\right] = \delta^{\alpha}_{\beta}\tilde{S}^{\dot\alpha},$$

$$\left[K^{\dot\alpha\alpha}, \tilde{Q}_{\dot\beta}\right] = \delta^{\dot\alpha}_{\dot\beta}S^\alpha,$$

$$\left[P_{\alpha\dot\alpha}, S^\beta\right] = -\delta^{\beta}_{\alpha}\tilde{Q}_{\dot\alpha},$$

$$\left[P_{\alpha\dot\alpha}, \tilde{S}^{\dot\beta}\right] = -\delta^{\dot\beta}_{\dot\alpha}Q_\alpha.$$

Once the superconformal algebra is known, the next natural thing to examine is its representation theory. That is, we wish to classify the possible multiplets and understand their properties.

A generic multiplet of the algebra is characterized by the charges $(\Delta, R, j_1, j_2)$ of the superconformal primary state with respect to the generators $(H, R, j_1, j_2)$, where $j_1$ and $j_2$ are the following Cartans:

$$j_1 = (J_1)^{+}_{+} = -(J_1)^{-}_{-}, \quad j_2 = (J_2)^{\dot+}_{\dot+} = -(J_2)^{\dot-}_{\dot-}. \tag{A.4}$$

The primary state is annihilated by the conformal supercharges $S^\alpha, \tilde{S}^{\dot\alpha}$,[74] and the multiplet is constructed by acting on it with the Poincare supercharges $Q_\alpha, \tilde{Q}_{\dot\alpha}$. In some cases, a combination of these supercharges annihilates the primary as well, resulting in a short multiplet and a relation between the charges of the primary. These relations are just a saturation of the corresponding unitarity bounds, derived from requiring the absence of negative-norm states in the multiplet, and determine the conformal dimension $\Delta$ in terms of the other charges ($j_1$, $j_2$ and $R$). This, in turn, means that the conformal dimension is protected against changing the parameters of the theory. We list the possible such shortening conditions (classified by the corresponding null state in the multiplet) and their common names in the table below, as well as the associated unitarity bounds [20, 41, 143].[75]

| Shortening Condition | Name | Primary | Null State | Unitarity Bound |
|---|---|---|---|---|
| – | $L$ | $[j_1; j_2]^{(R)}_\Delta$ | – | $\Delta > 2 + 2j_1 - \frac{3}{2}R$ |
| – | $\overline{L}$ | $[j_1; j_2]^{(R)}_\Delta$ | | $\Delta > 2 + 2j_2 + \frac{3}{2}R$ |
| $\epsilon^{\alpha\beta}Q_\alpha\|\text{Primary}\rangle_\beta = 0$ | $A_1$ | $[j_1; j_2]^{(R)}_\Delta,\ j_1 \geq \frac{1}{2}$ | $[j_1 - \frac{1}{2}; j_2]^{(R-1)}_{\Delta+1/2}$ | $\Delta = 2 + 2j_1 - \frac{3}{2}R$ |
| $(Q)^2\|\text{Primary}\rangle = 0$ | $A_2$ | $[0; j_2]^{(R)}_\Delta$ | $[0; j_2]^{(R-2)}_{\Delta+1}$ | $\Delta = 2 - \frac{3}{2}R$ |
| $\epsilon^{\dot\alpha\dot\beta}\tilde{Q}_{\dot\alpha}\|\text{Primary}\rangle_{\dot\beta} = 0$ | $\overline{A}_1$ | $[j_1; j_2]^{(R)}_\Delta,\ j_2 \geq \frac{1}{2}$ | $[j_1; j_2 - \frac{1}{2}]^{(R+1)}_{\Delta+1/2}$ | $\Delta = 2 + 2j_2 + \frac{3}{2}R$ |
| $(\tilde{Q})^2\|\text{Primary}\rangle = 0$ | $\overline{A}_2$ | $[j_1; 0]^{(R)}_\Delta$ | $[j_1; 0]^{(R+2)}_{\Delta+1}$ | $\Delta = 2 + \frac{3}{2}R$ |
| $Q_\alpha\|\text{Primary}\rangle = 0$ | $B_1$ | $[0; j_2]^{(R)}_\Delta$ | $[\frac{1}{2}; j_2]^{(R-1)}_{\Delta+1/2}$ | $\Delta = -\frac{3}{2}R$ |
| $\tilde{Q}_{\dot\alpha}\|\text{Primary}\rangle = 0$ | $\overline{B}_1$ | $[j_1; 0]^{(R)}_\Delta$ | $[j_1; \frac{1}{2}]^{(R+1)}_{\Delta+1/2}$ | $\Delta = \frac{3}{2}R$ |

As can be seen, unbarred letters denote shortening conditions with respect to the supercharges $Q_\alpha$ while barred letters are with respect to $\tilde{Q}_{\dot\alpha}$. Moreover, the letters $L$ and $\overline{L}$ correspond to the absence of shortening conditions with respect to $Q_\alpha$ and $\tilde{Q}_{\dot\alpha}$, respectively.

To completely specify a multiplet, one needs to impose both $Q_\alpha$ and $\tilde{Q}_{\dot\alpha}$ shortening conditions, and as a result each kind of multiplet is denoted by two letters – an unbarred one and a barred one. For example, $L\overline{L}[j_1; j_2]^{(R)}_\Delta$ corresponds to a long multiplet with charges $(\Delta, R, j_1, j_2)$ for the primary state, while $L\overline{B}_1[j_1; 0]^{(R)}_\Delta$ to a multiplet with a chiral primary.[76] The constraints on the charges of the primary state resulting from the various combinations of shortening conditions are detailed in the following table [143].

| | $\overline{L}$ | $\overline{A}_1$ | $\overline{A}_2$ | $\overline{B}_1$ |
|---|---|---|---|---|
| $L$ | $\Delta > 2 + \max\left\{2j_1 - \frac{3}{2}R, 2j_2 + \frac{3}{2}R\right\}$ | $\Delta = 2 + 2j_2 + \frac{3}{2}R$ $j_2 \geq \frac{1}{2}, R > \frac{2}{3}(j_1 - j_2)$ | $\Delta = 2 + \frac{3}{2}R$ $j_2 = 0, R > \frac{2}{3}j_1$ | $\Delta = \frac{3}{2}R$ $j_2 = 0, R > \frac{2}{3}(j_1 + 1)$ |
| $A_1$ | $\Delta = 2 + 2j_1 - \frac{3}{2}R$ $j_1 \geq \frac{1}{2}, R < \frac{2}{3}(j_1 - j_2)$ | $\Delta = 2 + j_1 + j_2$ $j_1, j_2 \geq \frac{1}{2}, R = \frac{2}{3}(j_1 - j_2)$ | $\Delta = 2 + j_1$ $j_1 \geq \frac{1}{2}, j_2 = 0, R = \frac{2}{3}j_1$ | $\Delta = 1 + j_1$ $j_1 \geq \frac{1}{2}, j_2 = 0, R = \frac{2}{3}(j_1 + 1)$ |
| $A_2$ | $\Delta = 2 - \frac{3}{2}R$ $j_1 = 0, R < -\frac{2}{3}j_2$ | $\Delta = 2 + j_2$ $j_1 = 0, j_2 \geq \frac{1}{2}, R = -\frac{2}{3}j_2$ | $\Delta = 2$ $j_1 = j_2 = 0, R = 0$ | $\Delta = 1$ $j_1 = j_2 = 0, R = \frac{2}{3}$ |
| $B_1$ | $\Delta = -\frac{3}{2}R$ $j_1 = 0, R < -\frac{2}{3}(j_2 + 1)$ | $\Delta = 1 + j_2$ $j_1 = 0, j_2 \geq \frac{1}{2}, R = -\frac{2}{3}(j_2 + 1)$ | $\Delta = 1$ $j_1 = j_2 = 0, R = -\frac{2}{3}$ | $\Delta = 0$ $j_1 = j_2 = R = 0$ |

To illustrate how these constraints between the charges are obtained from the algebra, let us consider as an example a multiplet of the type $L\overline{B}_1[0; 0]^{(R)}_\Delta$. Here, the superconformal primary state $|\chi\rangle$ has $j_1 = j_2 = 0$ and satisfies

$$\tilde{Q}_{\dot\alpha}|\chi\rangle = S^\alpha|\chi\rangle = \tilde{S}^{\dot\alpha}|\chi\rangle = 0. \tag{A.5}$$

---

[74]Notice that as a result, the primary state is also annihilated by $K^{\dot\alpha\alpha}$.

[75]Notice that we use different conventions for $j_1$ and $j_2$ in comparison to [20, 143], such that $j_\text{there} = 2j_\text{here}$.

[76]Note that a chiral free field multiplet is represented by $A_k\overline{B}_1[j_1; 0]^{(R)}_\Delta$.

As a result, the expectation value of the fourth equation in (A.2) is given by

$$\langle\chi|\{\tilde{Q}_{\dot{\alpha}},\tilde{S}^{\dot{\beta}}\}|\chi\rangle = 0 = \frac{1}{2}\delta^{\dot{\beta}}_{\dot{\alpha}}\left(\Delta - \frac{3}{2}R\right),\tag{A.6}$$

implying $\Delta = \frac{3}{2}R$. In addition, since the dimension $\Delta$ is greater than 1 (otherwise it would be equal to the free-state value 1, corresponding to the primary of the multiplet $A_2\overline{B}_1[0;0]^{(R)}_{\Delta}$), we have $R > \frac{2}{3}$.

Let us next note that even though the short multiplets are protected in the sense we mentioned before, most of them are not absolutely protected since they can recombine as we change the parameters of the theory with other short multiplets to form a long multiplet, which is no longer protected. This can happen *e.g.* as we move along a conformal manifold from one superconformal field theory to another, and short multiplets recombine to form a long multiplet such that the operators residing in the short multiplets are not protected in the new theory. In the other direction, a long multiplet at one point of the conformal manifold can hit the unitarity bound at another point, decomposing into a collection of short multiplets. As an example, let us consider the long multiplet $L\overline{L}[0;0]^{(R=0)}_{\Delta=2+\epsilon}$ as $\epsilon \to 0$, where it hits the unitarity bound. It then splits into three short multiplets in the following way,

$$L\overline{L}[0;0]^{(R=0)}_{\Delta=2} \to A_2\overline{A}_2[0;0]^{(0)}_2 \oplus B_1\overline{L}[0;0]^{(-2)}_3 \oplus L\overline{B}_1[0;0]^{(2)}_3,\tag{A.7}$$

where $A_2\overline{A}_2[0;0]^{(0)}_2$ contains a conserved current while $B_1\overline{L}[0;0]^{(-2)}_3$ and $L\overline{B}_1[0;0]^{(2)}_3$ contain a marginal operator.[77] This recombination corresponds to the fact that a marginal operator can fail to be exactly marginal only if it combines with a conserved current corresponding to a broken global symmetry [40], and is the reason that marginal operators and conserved currents contribute to the superconformal index with opposite signs (see Eq. (56)). In particular, the difference between the numbers of conserved-current and marginal-operator multiplets is invariant under a change in the parameters of the theory which might be accompanied by the recombination (A.7).

For each of the multiplets listed above, one can compute the superconformal index and use it to extract information about the operator content of a given theory from its index. The multiplets that contribute nontrivially to the index are collected in the following table, along with the corresponding expressions for the index.

---

[77]Note that in this discussion we refer to operators instead of states, which is possible due to the state/operator correspondence in theories with conformal symmetry.

| Multiplet | Superconformal Index |
|---|---|
| $L\bar{A}_1[j_1;j_2]^{(R)}_\Delta$ | $(-1)^{2(j_1+j_2)+1}\dfrac{(pq)^{\frac{R}{2}+j_2+1}\chi_{j_1}(\sqrt{p/q})}{(1-p)(1-q)}$ |
| $L\bar{A}_2[j_1;0]^{(R)}_\Delta$ | $(-1)^{2j_1+1}\dfrac{(pq)^{\frac{R}{2}+1}\chi_{j_1}(\sqrt{p/q})}{(1-p)(1-q)}$ |
| $L\bar{B}_1[j_1;0]^{(R)}_\Delta$ | $(-1)^{2j_1}\dfrac{(pq)^{\frac{R}{2}}\chi_{j_1}(\sqrt{p/q})}{(1-p)(1-q)}$ |
| $A_1\bar{A}_1[j_1;j_2]^{(R)}_\Delta$ | $(-1)^{2(j_1+j_2)+1}\dfrac{(pq)^{\frac{1}{3}(j_1+2j_2)+1}\chi_{j_1}(\sqrt{p/q})}{(1-p)(1-q)}$ |
| $A_1\bar{A}_2[j_1;0]^{(R)}_\Delta$ | $(-1)^{2j_1+1}\dfrac{(pq)^{\frac{1}{3}j_1+1}\chi_{j_1}(\sqrt{p/q})}{(1-p)(1-q)}$ |
| $A_2\bar{A}_1[0;j_2]^{(R)}_\Delta$ | $(-1)^{2j_2+1}\dfrac{(pq)^{\frac{2}{3}j_2+1}}{(1-p)(1-q)}$ |
| $A_2\bar{A}_2[0;0]^{(R)}_\Delta$ | $-\dfrac{pq}{(1-p)(1-q)}$ |
| $A_1\bar{B}_1[j_1;0]^{(R)}_\Delta$ | $\dfrac{(-1)^{2j_1}(pq)^{\frac{1}{3}(j_1+1)}}{(1-p)(1-q)}\left[\chi_{j_1}(\sqrt{p/q})-(pq)^{\frac{1}{2}}\chi_{j_1-\frac{1}{2}}(\sqrt{p/q})\right]$ |
| $A_2\bar{B}_1[0;0]^{(R)}_\Delta$ | $\dfrac{(pq)^{\frac{1}{3}}}{(1-p)(1-q)}$ |
| $B_1\bar{A}_1[0;j_2]^{(R)}_\Delta$ | $-\dfrac{(pq)^{\frac{2}{3}(j_2+1)}}{(1-p)(1-q)}$ |
| $B_1\bar{A}_2[0;0]^{(R)}_\Delta$ | $-\dfrac{(pq)^{\frac{2}{3}}}{(1-p)(1-q)}$ |

Here, $\chi_j(z)$ is the character of the spin-$j$ representation of $SU(2)$. We can immediately see from this table that as stated in the two bullets around Eq. (56), the only operators that can contribute at order $(pq)^n$ with $n < 1$ are relevant operators (corresponding to primaries of $L\bar{B}_1[0;0]^{(R)}_\Delta$ multiplets with $R < 2$) and that at order $pq$ the only operators which can contribute are marginal ones (contributing with a positive sign) and certain fermionic components from the conserved current multiplet $A_2\bar{A}_2[0;0]^{(R)}_\Delta$, which contribute with a negative sign. As mentioned above, the signs of these contributions at order $pq$ correspond to the recombination rule (A.7). Note, in addition, that there is no recombination rule involving relevant operators, meaning that the multiplets $L\bar{B}_1[0;0]^{(R)}_\Delta$ with $R < 2$ are absolutely protected. Let us finally emphasize that this discussion about the contributions of various operators to the index excludes free multiplets, that is we concentrate only on interacting theories.

## B  Leigh-Strassler argument for conformal manifolds

Let us here detail briefly yet another method to study the conformal manifold introduced in a seminal paper by Leigh and Strassler [51]. This method is very Lagrangian in nature on one hand but on another hand it is rather intuitive. The basic observation is that as the superpotential is not renormalized due to holomprphy arguments [1] in a supersymmetric theory, the beta-function of the superpotential coupling is determined by the anomalous dimensions of the fields in the superpotential,

$$\beta_{\lambda_i}(\{\lambda,g\}) \sim n_i - 3 + \frac{1}{2}\sum_{j=1}^{n_i}\gamma_{\alpha_i(j)}(\{\lambda,g\}), \qquad W = \sum_{i=1}^{L}\lambda_i\prod_{j=1}^{n}Q_{\alpha_i(j)}. \qquad (B.1)$$

Here $\alpha_i \in S_L$ is some permutation of the $L$ chiral superfields $Q_i$. Moreover also the gauge beta function is related to the anomalous dimensions of the fields transforming under the gauge

symmetry [252, 253],[78]

$$\beta_{g_i} \propto - \left( 3C_2(G_i) - \sum_{k \in h_i} T(R_k) + \sum_{k \in h_i} T(R_k)\gamma_k(\{\lambda, g\}) \right), \tag{B.2}$$

where the proportionality coefficient depends on couplings, $C_2(G)$ is the quadratic Casimir of the adjoint, $T(R)$ is quadratic Casimir of irrep $R$, and the set $h_k$ is the set of representatives of fields $Q_i$ from every non trivial irrep of $G_k$. Now say we have $L$ superpotential couplings $\lambda$ and $M$ gauge couplings $g$ such that the one loop beta functions for the gauge fields vanish, $3C_2(G_i) - \sum_{k \in h_i} T(R_k) = 0$, and the superpotentials are marginal, $n_i = 3$. Then the vanishing of the beta functions are in general $L + M$ equations for $L + M$ independent variables. As such they will typically have only isolated solutions. However this is not the case if the equations are linearly dependent: And this can happen easily here as they are written in terms of anomalous dimensions corresponding to fields which might be related by symmetries. The dimension of the conformal manifold is thus given by the number of couplings minus the number of independent beta functions (modulo global symmetry).

As an example consider the $SU(3)$ SQCD with $N_f = 9$ with the marginal superpotential (29). because of the symmetry of the superpotential the anomalous dimensions of all the (anti)fundamental fields are the same. Thus all of the beta functions are proportional to this anomalous dimension which is a function of two couplings $\lambda$ and the gauge coupling. We have one function depending on two parameters which vanishes at the origin, and thus we expect it in general to have a line of solutions. This gives a one dimensional conformal manifold on which some symmetry is preserved.

## C  Computing Kähler quotients with Hilbert series

One can compute in principle in a straight-forward way the Kähler quotients needed for the determination of the conformal manifold using the Hilbert series techniques.[79] Let us assume that we have an SCFT with global symmetry (commuting with any gauge symmetry, including anomalous symmetries) $G_F$ and that all marginal operators, including gauge couplings charged under the anomalous symmetry, are in representation $\mathcal{R}$ of $G_F$. Let $\chi_{\overline{\mathcal{R}}}(\{a\})$ be the character of $\overline{\mathcal{R}}$, the complex conjugate representation of $\mathcal{R}$, and $\{a\}$ being a set of complex parameters corresponding to the Cartan generators of $G_F$. We define a plethystic exponential of a function $f(\{x\})$ ($\{x\}$ is some collection of parameters),

$$\mathrm{PE}[f\{x\}] = \exp\left( \sum_{n=1}^{\infty} \frac{1}{n} f(\{x^n\}) \right). \tag{C.1}$$

Then we compute the following integral,

$$HS_{\mathcal{M}_c}(x) \triangleq \oint \prod_{i=1}^{\mathrm{Rank}\, G_F} \frac{da_i}{2\pi i a_i} \Delta_{Haar}^{(G_F)}(\{a\}) \mathrm{PE}\left[ x \cdot \chi_{\overline{\mathcal{R}}}(\{a\}) \right]. \tag{C.2}$$

Here $\Delta_{Haar}^{(G_F)}(\{a\})$ is the Haar measure of $G_F$ and $x$ is some complex parameter with $|x| < 1$. The function $HS_{\mathcal{M}_c}(x)$ is the Hilbert series associated with the conformal manifold and as such in particular it captures its dimension. The dimension can be extracted by taking the

---

[78]Note that for the free theory the R-charges of all the fields $\frac{2}{3}$ and the one loop contribution in this formula is precisely $-3 \operatorname{Tr} R\, G_i^2$.

[79]See *e.g.* [254] for uses of the Hilbert series in supersymmetric QFTs.

limit $x \to 1$ and is equal to the degree of divergence of $HS_{\mathcal{M}_c}(x)$ in this limit. By definition the plethystic exponent generates all the symmetric products of the argument function, and the integrations with the Haar measure project on $G_F$ invariants. Thus $HS_{\mathcal{M}_c}(x)$ is just given by a sum of all the symmetrized invariants of the couplings weighed by powers of $x$. This sum is generated by the independent such invariants and there might be algebraic relations. The result thus can in general be written as a rational function in $x$. The denominator which is product of terms of the form $1 - x_i^n$ with $n_i$ being a power in which some singlet can appear. The numerator is a polynomial encoding algebraic relations between the various generators.

Although this procedure is mathematically straightforward, technically when the group $G_F$ is large, it is hard to implement. Let us give an example. Consider we have a marginal deformation in $\mathbf{10}$ of $SU(3)$. This is a three index completely symmetric representation. First we compute its character. To compute this we note that the coefficient of $x^n$ in the expansion of $\text{PE}[x \cdot \chi_{\mathcal{R}}(\{a\})]$ is precisely the character of the $n$th symmetric power of $\mathcal{R}$.[80] Taking the character of fundamental of $SU(3)$ to be $\chi_{\mathbf{3}} = a_1 + a_2 + 1/(a_1 a_2)$ we then obtain,

$$\chi_{\mathbf{10}}(a_1, a_2) = 1 + a_1^3 + \frac{1}{a_1^3 a_2^3} + \frac{1}{a_1 a_2^2} + \frac{1}{a_1^2 a_2} + \frac{a_1}{a_2} + \frac{a_2}{a_1} + \frac{a_1^2}{a_2} + \frac{a_1}{a_2^2} + a_2^3. \tag{C.3}$$

Then we compute,

$$\oint \prod_{i=1}^{2} \frac{da_i}{2\pi i a_i} \frac{1}{6} \prod_{i \neq j} (1 - a_i/a_j), \text{PE}\left[x \cdot \chi_{\mathbf{10}}(a_1^{-1}, a_2^{-1})\right] = \frac{1}{1 - x^4} \frac{1}{1 - x^6}. \tag{C.4}$$

Here $a_3 = 1/(a_1 a_2)$. We find thus that we have a two dimensional conformal manifold as the degree of divergence is two. We have one singlet in fourth symmetric product and one in sixths and there are no algebraic relations involving them.

Another example we will compute here is the case of $\mathbf{6}$, two index symmetric, of $SU(3)$. Following same procedure as above we have,

$$\chi_{\mathbf{6}}(a_1, a_2) = \frac{1}{a_1} + \frac{1}{a_2} + a_1 a_2 + a_1^2 + a_2^2 + \frac{1}{a_1^2 a_2^2}. \tag{C.5}$$

Then we compute,

$$\oint \prod_{i=1}^{2} \frac{da_i}{2\pi i a_i} \frac{1}{6} \prod_{i \neq j} (1 - a_i/a_j), \text{PE}\left[x \cdot \chi_{\mathbf{6}}(a_1^{-1}, a_2^{-1})\right] = \frac{1}{1 - x^3}. \tag{C.6}$$

We get a single invariant which is the determinant of the two-index symmetric matrix definition of $\mathbf{6}$.

# D  Class $\mathcal{S}$ interpretation of the exercise

The theory discussed in the exercise of Section 3.4 has a class $\mathcal{S}$ origin. We will discuss the definition of this theory here. We can consider class $\mathcal{S}$ theory of type $A_2$, that is three $M5$ branes compactified on a Riemann surface. These models have $\mathcal{N} = 2$ supersymmetry. The building block of the four dimensional theories of this kind is the so called $T_3$ model corresponding to compactification on a sphere with three maximal punctures. The symmetry associated to every puncture is $SU(3)$ (as 5d compactification of $A_2$ $(2, 0)$ theory gives maximally supersymmetric

---

[80]The character of $n$th antisymmetric power is the coefficient of $(-1)^n x^n$ in expansion of $\text{PE}[-x \cdot \chi_{\mathcal{R}}(\{a\})]$.

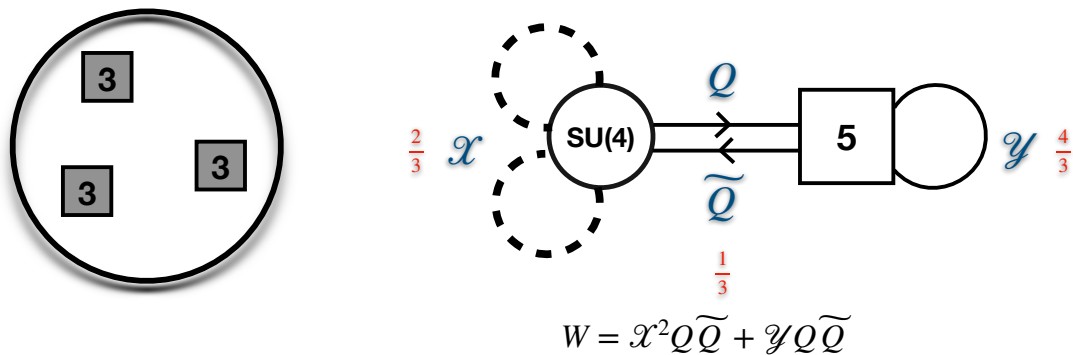

$$W = \mathcal{X}^2 Q\widetilde{Q} + \mathcal{Y}Q\widetilde{Q}$$

Figure 62: The $T_3$ model. On the right we have a $\mathcal{N} = 1$ non-conformal Lagrangian flowing to $T_3$ obtained in [82]. (See [255] for a different Lagrangian description.) The fields $X$ form a doublet of two-index antisymmetrics of $SU(4)$, while $Y$ is adjoint plus a singlet of $SU(5)$. The Lagrangian has manifest $SU(5) \times U(1) \times SU(2)$ subgroup of $E_6$ manifest in the UV. The $U(1)_t$ symmetry is conjectured to emerge in the IR.

YM theory with gauge group $SU(3)$) and for the $T_3$ model the symmetry enhances to $E_6$ such that,

$$\mathbf{78}_{E_6} \to (\mathbf{8,1,1}) + (\mathbf{1,8,1}) + (\mathbf{1,1,8}) + (\mathbf{3,3,3}) + (\bar{\mathbf{3}}, \bar{\mathbf{3}}, \bar{\mathbf{3}}), \tag{D.1}$$

where on the right we have the decomposition in terms of $SU(3)^3$ symmetry. This theory has an additional $U(1)_t$ symmetry in $\mathcal{N} = 1$ language coming from the extended R-symmetry of $\mathcal{N} = 2$. The $T_3$ theory has relevant operators $M$ with $\mathcal{N} = 1$ R-charge $\frac{4}{3}$, $U(1)_t$ charge $+1$, and in $\mathbf{78}_{E_6}$: These are the moment map operators. In addition to these there is a marginal Coulomb branch operator $X$ with $U(1)_t$ charge $-3$ and R-charge 2. These are the only marginal and relevant operators of $T_3$. One constructs more general theories corresponding to surfaces with maximal punctures by combining together $T_3$ models gauging diagonal combinations of puncture $SU(3)$ symmetries with $\mathcal{N} = 2$ vector multiplets. In $\mathcal{N} = 1$ language in addition to the $\mathcal{N} = 1$ vector multiplet one adds a chiral adjoint superfield $\Phi$ coupling through a superpotential to the moment maps of the glued punctures, $W = \Phi M - \Phi M'$. This discussion is the $(2,0)$ version of the $\Phi$-gluing.

An additional building block of theories in this class is the free trinion, which can be obtained from $T_3$ by an RG flow giving a vacuum expectation value to one of the $SU(3)$ moment maps breaking the $SU(3)$ down to a $U(1)$. We thus obtain a theory corresponding to a sphere with two maximal $SU(3)$ punctures and one $U(1)$ puncture. The theory is just a bifundamental hypermultiplet of two $SU(3)$ symmetries with the $U(1)$ symmetry charging oppositely the two half-hypermultiplets. Let us denote these half-hypermultiplets $q$ and $\widetilde{q}$.

Gluing the $T_3$ theories together the resulting models have the $\mathcal{N} = 2$ preserving conformal manifolds parametrized by the $\mathcal{N} = 2$ gauge couplings used to glue the spheres together. If the surface has at least one minimal puncture one can build also $\mathcal{N} = 1$ preserving conformal manifolds on which the $U(1)_t$ symmetry and the symmetry of one minimal puncture is broken. We go to a duality frame in which we glue the free trinion to a generic class $\mathcal{S}$ theory of type $A_2$. Then, let us analyze the conformal manifold. The marginal operators involving the glued puncture operators and the free trinion fields are: The $\mathcal{N} = 2$ gauging (which is exactly marginal), the Coulomb branch operator $\Phi^3$ charged $-3$ under $U(1)_t$, and the baryons $q^3$ and $\widetilde{q}^3$ charged $+\frac{3}{2}$ under $U(1)_t$ and $\pm 1$ under the minimal puncture $U(1)$ by definition. Thus we can build a Kähler quotient as $(\Phi^3)(q^3)(\widetilde{q}^3)$ is not charged under any symmetry. This adds a one dimensional $\mathcal{N} = 1$ preserving conformal manifold. Moreover as $U(1)_t$ is now broken the dimension three Coulomb branch operators coming from the adjoint chirals involved in gaug-

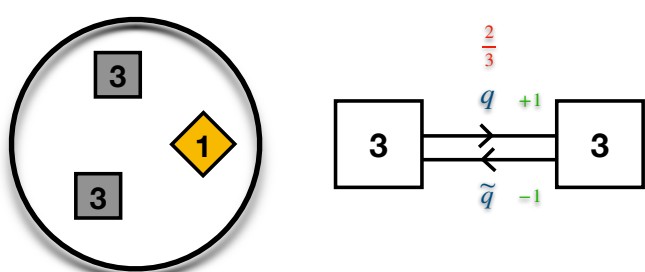

Figure 63: The free trinion model corresponding to a sphere with two maximal and one minimal punctures.

ings used to construct the theory we glued to the free trinion become also exactly marginal.

Finally in special cases the conformal manifold is even richer. Let us consider one $T_3$ glued to one free trinion. We obtain a sphere with three maximal and one minimal punctures. The conformal anomalies of this theory are $a = \frac{15}{4}$ and $c = \frac{17}{4}$. As in the general discussion above we have exactly marginal deformations corresponding to the $\mathcal{N} = 2$ gluing, $X$, and the $\mathcal{N} = 1$ deformation involving the baryons. However here we have additional deformations. We can take the component of the $E_6$ moment map charged $(\mathbf{3}, \mathbf{3}, \mathbf{3})$ and contract it with $\widetilde{q}$ building a gauge invariant operator $H$ in $(\mathbf{3}, \mathbf{3}, \mathbf{3})$ of the three maximal punctures of the four punctured sphere which is marginal. In a same manner by contracting $(\bar{\mathbf{3}}, \bar{\mathbf{3}}, \bar{\mathbf{3}})$ of the moment map with $q$ we get a marginal operator $\widetilde{H}$ in representation $(\bar{\mathbf{3}}, \bar{\mathbf{3}}, \bar{\mathbf{3}})$ of the three maximal puncture symmetries. In fact as $H\widetilde{H}$ is a singlet of all the symmetries we have additional directions of the conformal manifold on which $SU(3)^3$ is broken to $SU(2)^3 \times U(1)^2$ such that $\mathbf{3}_i \to \mathbf{2}_i a_i + \mathbf{1}a_i^{-2}$, $\mathbf{8}_i \to \mathbf{3} + \mathbf{2}(a_i^3 + a_i^{-3}) + \mathbf{1}$, and $\prod_{i=1}^3 a_i = 1$. On this conformal manifold we have operators in representation,

$$4\,(\mathbf{1},\mathbf{1},\mathbf{1})_{0,0} + 2\,(\mathbf{2},\mathbf{2},\mathbf{2})_{0,0} + (\mathbf{1},\mathbf{2},\mathbf{2})_{\pm3,0} + (\mathbf{2},\mathbf{1},\mathbf{2})_{0,\pm3} + (\mathbf{2},\mathbf{2},\mathbf{1})_{\pm1,\pm1}\,. \tag{D.2}$$

We can next turn on the last deformations breaking the two $U(1)$s to a diagonal one and locking the two $SU(2)$ symmetries onto each other. This adds another dimension to the conformal manifold and leaves us with $SU(2)^2 \times U(1)$ symmetry and marginal deformations,

$$5\,(\mathbf{1},\mathbf{1})_0 + 2\,(\mathbf{2},\mathbf{3})_0 + 2\,(\mathbf{2},\mathbf{1})_0 + 2\,(\mathbf{2},\mathbf{2})_{\pm3} + (\mathbf{3},\mathbf{1})_0\,. \tag{D.3}$$

We can break the first $SU(2)$ to the Cartan using the last deformation preserving $SU(2) \times U(1)^2$ with marginals being,

$$6\,\mathbf{1}_{0,0} + 2\,\mathbf{3}_{0,\pm1} + 2\,\mathbf{1}_{0,\pm1} + 2\,\mathbf{2}_{\pm_13,\pm_21}\,. \tag{D.4}$$

Next we break the Cartan $U(1)$ using the next to last operators preserving $SU(2) \times U(1)$ and having marginal operators in,

$$9\,\mathbf{1}_0 + 4\,\mathbf{3}_0 + 4\,\mathbf{2}_{\pm3}\,. \tag{D.5}$$

We use the second operator to break the remaining $SU(2)$ to the Cartan preserving $U(1)^2$ and having marginal operators in,

$$13\,\mathbf{1}_{0,0} + 3\,\mathbf{1}_{0,\pm2} + 4\,\mathbf{1}_{\pm_13,\pm_21}\,. \tag{D.6}$$

Next second type of the operators is turned on to break the second $U(1)$ with only one $U(1)$ preserved and marginal operators being,

$$18\,\mathbf{1}_0 + 8\,\mathbf{1}_{\pm3}\,, \tag{D.7}$$

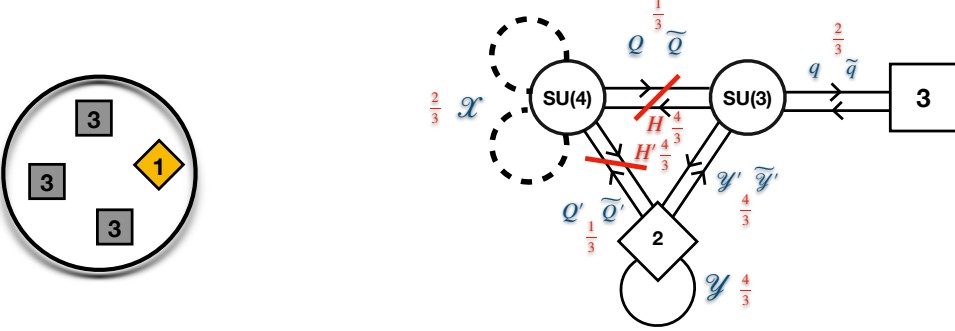

$$W = \mathcal{X}^2(Q\widetilde{Q} + Q'\widetilde{Q'}) + \mathcal{Y}Q\widetilde{Q'} + Q\mathcal{Y'}Q' + \widetilde{Q}\,\widetilde{\mathcal{Y'}}\,\widetilde{Q'} + Q\widetilde{Q}q\widetilde{q} + HQ\widetilde{Q} + H'Q'\widetilde{Q'}$$

Figure 64: The theory corresponding to a sphere with three maximal punctures and one minimal puncture obtained by gluing to a $T_3$ the free trinion. On the right we have a non-conformal description of the theory using the Lagrangian of [82] for the $T_3$. The fields $H$ and $H'$ are gauge singlets, and $Y$ is adjoint of $SU(2)$. One maximal puncture $SU(3)$ and the minimal puncture $U(1)$ are manifest in the UV, while the remaining two maximal puncture $SU(3)$ symmetries emerge in the IR enhancing an $SU(2)^2 \times U(1)^2$ symmetry of the Lagrangian. The $U(1)_t$ symmetry is conjectured to emerge in the IR. One can further turn on marginal deformations consistent with the R-symmetry assignment in the Figure breaking completely all the global symmetry.

and at last the second operators can be used to construct additional 15 exactly marginal operators breaking all the symmetry to obtain 33 dimensional conformal manifold.

The relevant deformations of the model are the moment maps of the two maximal punctures of the $T_3$, the operator $\mathrm{Tr}\,\Phi^2$, and $q\widetilde{q}$: This gives rise to $8 + 8 + 1 + 9 = 26$ relevant operators. The conformal Lagrangian we derive in the exercise exactly reproduces the features of this strongly coupled model. We can compute the supersymmetric index either using the theory of Figure 64,

$$\mathcal{I}_A = \frac{(q;q)^5(p;p)^5}{3! \times 4!} \oint \prod_{i=1}^{2} \frac{dz_i}{2\pi i z_i} \oint \prod_{a=1}^{3} \frac{du_a}{2\pi i u_a} \frac{\prod_{i=1}^{3}\left(\Gamma_e((qp)^{\frac{1}{3}}z_i^{\pm 1})\right)^3\left(\Gamma_e((qp)^{\frac{2}{3}}z_i^{\pm 1})\right)^2}{\prod_{i \neq j}^{3}\Gamma_e(z_i/z_j)\prod_{a\neq b}^{4}\Gamma_e(u_a/u_b)} \quad (\text{D.8})$$

$$\times \prod_{a=1}^{4}\left(\Gamma_e((qp)^{\frac{1}{6}}u_a^{\pm 1})\right)^2 \prod_{a<b}^{4}\left(\Gamma_e((qp)^{\frac{1}{3}}u_au_b)\right)^2 \prod_{a=1}^{4}\prod_{i=1}^{3}\left(\Gamma_e((qp)^{\frac{1}{6}}(u_a/z_i)^{\pm 1})\right)\left(\Gamma_e((qp)^{\frac{2}{3}})\right)^5,$$

or the one of Figure 21,

$$\mathcal{I}_B = \frac{(q;q)^3(p;p)^3}{8 \times 2} \oint \prod_{i=1}^{2} \frac{dz_i}{2\pi i z_i} \oint \frac{du}{2\pi i u} \frac{\prod_{i=1}^{2}\left(\Gamma_e((qp)^{\frac{1}{3}}z_i^{\pm 1})\right)^6\left(\Gamma_e((qp)^{\frac{1}{3}}z_i^{\pm 1}u^{\pm 1})\right)^2}{\Gamma_e(u^{\pm 2})\Gamma_e(z_1^{\pm 2})\Gamma_e(z_2^{\pm 2})\Gamma_2(z_1^{\pm 1}z_2^{\pm 1})} \quad (\text{D.9})$$

$$\times \left(\Gamma_e((qp)^{\frac{1}{3}}u^{\pm 1})\right)^4\left(\Gamma_e((qp)^{\frac{1}{3}}z_1^{\pm 1}z_2^{\pm 1})\right)^2\left(\Gamma_e((qp)^{\frac{1}{3}})\right)^3\Gamma_e\left((qp)^{\frac{1}{3}}z_1^{\pm 2}\right)\Gamma_e\left((qp)^{\frac{1}{3}}z_2^{\pm 2}\right),$$

and obtain the same result,

$$\mathcal{I}_A = \mathcal{I}_B = 1 + 26(qp)^{\frac{2}{3}} - (q+p)(qp)^{\frac{1}{3}} + 33qp + 25(q+p)(qp)^{\frac{2}{3}} + (348 - q^2 - p^2)(qp)^{\frac{1}{3}} + \cdots.$$
$$(\text{D.10})$$

We do not have a proof of the equality but have checked it to the stated order in expansion in $q$ and $p$. As a side remark, we can also compute the so called Schur limit of the index [180, 181]. The Schur index of the $\mathcal{N} = 2$ theory is a limit of the supersymmetric index

computed using $\mathcal{N} = 1$ superconformal R-symmetry by setting $p = q^{\frac{1}{2}}$ [22, 154]. This index has an alternative description using the geometric data, $A_2$ theory on a sphere with three maximal and one minimal puncture in our case. The expression is [181],

$$\mathcal{I}_C = \frac{(q^2; q)^2 (q^3; q)}{(q; q)^{25} (q^{\frac{3}{2}}; q)^2} \sum_{\lambda_1 = 0}^{\infty} \sum_{\lambda_2 = 0}^{\lambda_1} \frac{(\dim(\lambda_1, \lambda_2))^3 \dim_{q^{\frac{1}{2}}}(\lambda_1, \lambda_2)}{(\dim_q(\lambda_1, \lambda_2))^2}.$$

Here,

$$\dim_q(\lambda_1, \lambda_2) = \frac{q^{-\lambda_1}(1 - q^{1+\lambda_2})(1 - q^{2+\lambda_1})(1 - q^{1+\lambda_1-\lambda_2})}{(1 - q^2)(1 - q)^2}, \tag{D.11}$$

$$\dim(\lambda_1, \lambda_2) = \lim_{q \to 1} \dim_q(\lambda_1, \lambda_2) = \frac{1}{2}(1 + \lambda_2)(2 + \lambda_1)(1 + \lambda_1 - \lambda_2).$$

The sum here is over the finite dimensional representations of $A_2$; three factors of $\dim(\lambda_1, \lambda_2)$ in the numerator come from maximal punctures and the factor of $\dim_{q^{\frac{1}{2}}}(\lambda_1, \lambda_2)$ comes from the minimal puncture; the power of $\dim_q(\lambda_1, \lambda_2)$ in the numerator is $2g-2+s = 2\times0-2+4 = 2$ with $g$ the genus and $s$ the number of punctures. The overall factor involving the q-Pocchhammer symbols $((z; q) = \prod_{j=0}^{\infty}(1 - zq^j))$ also is determined by the geometric data. Using these expressions and either one of the Lagrangians we obtain in the Schur limit,

$$\mathcal{I}_C = \mathcal{I}_A(p = q^{\frac{1}{2}}) = \mathcal{I}_B(p = q^{\frac{1}{2}}) = 1 + 25q + 56q^{3/2} + 403q^2 + 1348q^{5/2} + 5844q^3 \tag{D.12}$$
$$+ 19756q^{7/2} + 70702q^4 + 225640q^{9/2} + 719961q^5 + 2155220q^{11/2} + 6338818q^6 + \cdots.$$

Here we do not have a proof of the equality but have checked it to the stated order in expansion in $q$. These are additional indications that the proposed duality is correct.

# E   Basics of 6d SCFTs

In this appendix we will review some aspects of 6d $\mathcal{N} = (1, 0)$ theories. First, we note that due to the fact that spinors are pseudo real in 6d, supercharges of different chiralities are inequivalent. Thus, we have several possible supersymmetries including the chiral $\mathcal{N} = (1, 0), (2, 0)$ with eight and 16 real supercharges and the non-chiral $\mathcal{N} = (1, 1)$ with 16 real supercharges. The $\mathcal{N} = (1, 0), (2, 0)$ can be extended to include also conformal symmetry to superconformal algebras [85]. For a detailed discussion of 6d superconformal algebra we refer the reader to [86] as well as the more modern exposition in [143].

Let us summarize the basic facts we need for our discussions. The $(2, 0)$ algebra is a special case of $(1, 0)$ and we will focus only on the latter. Six dimensional $\mathcal{N} = (1, 0)$ theories have eight real supercharges and contain the following field multiplets,

- **Hypermultiplet:** The bosonic part of the hypermultiplet contains four real scalar (or two complex scalars $q, \widetilde{q}^{\dagger}$ transforming in the doublet of $SU(2)_R$) that parameterize the Higgs branch of vacua. In addition, its fermionic part contains a Weyl fermion $\psi_\alpha$ transforming as a spinor of $SO(5, 1)$.

- **Vector multiplet:** The bosonic part of the vector multiplet contains only a vector field $A$ transforming as a vector of $SO(5, 1)$; therefore, these theories don't have a Coulomb branch of vacua. Its fermionic part contains a single Weyl fermion $\overline{\lambda}^{\dot{\alpha}}$ transforming as a co-spinor of $SO(5, 1)$ and the doublet of $SU(2)_R$.

- **Tensor multiplet:** The bosonic part of the tensor multiplet contains a two-form $B$ with a self dual field strength $H = dB$, $H = \star H$. In addition there is a real scalar $\phi$ that parameterizes the tensor branch of vacua. Both the scalar and the two-form are invariant under the R-symmetry. The fermionic part contains a single Weyl fermion $\chi_\alpha$ transforming as a spinor of $SO(5,1)$ and the doublet of $SU(2)_R$.

All Lorentz invariant Lagrangians we can write are IR free in 6d. Nevertheless mainly due to string theoretic constructions there is strong evidence that 6d interacting SCFTs exist [89] (see [10] for a review). Let us consider a gauge theory with a simple gauge group, hyper multiplets, and tensor multiplets. The standard kinetic term for the gauge fields,

$$\frac{1}{g_{YM}^2} \int \mathrm{tr}\,(F \wedge \star F)\,, \tag{E.1}$$

in 6d implies that the gauge coupling $g_{YM}$ scales as length, meaning this theory is trivial in the IR. Regarding tensor multiplet it is problematic to write the kinetic term for the self dual field strength $H$ as,

$$\int H \wedge \star H = \int H \wedge H = - \int H \wedge H\,, \tag{E.2}$$

implying the simple kinetic term vanishes. Nevertheless, we can treat this theory as if it has a Lagrangian describing the interactions with the constraint $H = \star H$. In this description the scalar in the tensor multiplet can couple to the gauge field as,

$$c \int \phi \, \mathrm{tr}\,(F \wedge \star F)\,, \tag{E.3}$$

with constant $c$. This term allows us to absorb the gauge coupling into a vacuum expectation value of $\phi$ with the effective gauge coupling of the theory given by,

$$\frac{1}{g_{eff}^2} = c \langle \phi \rangle\,, \tag{E.4}$$

meaning the gauge coupling as well as the BPS instanton tension are controlled by the tensor modulus and are non-vanishing on the tensor branch. The interpretation of these observations, mainly coming from string theoretic constructions, is that the underlying UV theory is a 6d SCFT that has been deformed by a non-zero vev to the scalar in its tensor multiplet. This initiates an RG flow leading at low-energies to a 6d gauge theory with inverse coupling squared proportional to the size of the vev.

## E.1 The Green-Schwarz mechanism in 6d

The Green-Schwartz mechanism in 6d arises from the interaction term coupling the tensor multiplet $B$ and the field strength $F$

$$c \int B \wedge \mathrm{tr}\,(F \wedge F)\,. \tag{E.5}$$

This term is implied form the coupling of $\phi$ in (E.3) and the fact that $\phi$ is a part of the tensor multiplet. The related $B$ equation of motion is[81]

$$d \star H = c \, \mathrm{tr}\,(F \wedge F)\,. \tag{E.6}$$

---

[81]This equation of motion can be derived using the interaction term together with the kinetic term of $H$ as if it exists.

Using the self-duality constraint then modifies the Bianchi identity of $H$ as

$$dH = c\,\mathrm{tr}\,(F \wedge F)\,, \tag{E.7}$$

meaning the instantons are charged under the two-form $B$. Using the the Dirac quantization condition for higher forms one finds that $c^2$ must be quantized.

It's important to note that the $B$-field is not invariant under gauge transformations. It is required from $\widetilde{H} = dB - cI_3(A, F)$ to be gauge invariant where $I_3$ is the Chern-Simons 3-form. The descent equations together with the modification of the Bianchi identity give

$$dI_3 = I_4 = \mathrm{tr}\,(F \wedge F)\,, \qquad \delta I_3 = dI_2^1\,. \tag{E.8}$$

We find that the invariance of $\widetilde{H}$ requires

$$d(\delta B) = c\,\delta I_3 = c\,dI_2^1\,, \tag{E.9}$$

meaning $\delta B = -cI_2^1$. The contribution of the modified Bianchi identity to the anomaly polynomial eight-form is then given by

$$\begin{aligned}
I_6 &\equiv \delta\,(c\,B \wedge \mathrm{tr}\,(F \wedge F)) = c\,\delta B \wedge \mathrm{tr}\,(F \wedge F) = c^2 I_2^1 \wedge \mathrm{tr}\,(F \wedge F)\,, \\
I_7 &\equiv dI_6 = c^2 dI_2^1 \wedge \mathrm{tr}\,(F \wedge F) = c^2 \delta I_3 \wedge \mathrm{tr}\,(F \wedge F)\,, \\
I_8 &\equiv dI_7 = c^2 dI_3 \wedge \mathrm{tr}\,(F \wedge F) = c^2 \delta I_4 \wedge \mathrm{tr}\,(F \wedge F) = c^2\,(\mathrm{tr}\,(F \wedge F))^2\,.
\end{aligned} \tag{E.10}$$

This means that gauge theories with non-vanishing quadratic part of the 1-loop anomaly polynomial can be made gauge anomaly free by the addition of a tensor multiplet as long as the coefficient of the quadratic part is negative definite. In addition, since $c^2$ needs to be properly quantized, so is the coefficient of the quadratic part of the 1-loop anomaly polynomial.

## E.2 The 6d anomaly polynomial eight-form

One way to compute the 6d anomaly polynomial is by considering the description of the 6d theory on its tensor branch. Such descriptions are given by specifying the gauge symmetry (with the various GS terms) and the matter content which can consist of three types of multiplets: Vectors, hypermultiplets, and tensor multiplets. Let us here describe the contributions of each of the matter multiplets. As usual to do so we will need to identify the contributions of the Weyl fermions residing in each multiplet.

The computation proceeds as follows.[82] The contribution of the Weyl fermion in 6d to the anomaly polynomial can be shown to be given by the 8-form part of

$$\hat{A}(T)\,ch(B)\,. \tag{E.11}$$

Here $\hat{A}(T)$ is the so called *Dirac A-roof genus*. This has the following general structure,

$$\hat{A}(T) = 1 - \frac{p_1(T)}{24} + \frac{7p_1^2(T) - 4p_2(T)}{5760} + \dots \tag{E.12}$$

The characteristic classes appearing in the expression, $p_i(T)$, are the $i$-th Pontryagin classes of the tangent bundle $T$. Concretely, the first two Pontryagin classes are given by

$$p_1(T) = -\frac{2}{(4\pi)^2}\mathrm{tr}R^2\,, \quad p_2(T) = -\frac{4}{(4\pi)^4}\left(\mathrm{tr}R^4 - \frac{1}{2}\left(\mathrm{tr}R^2\right)^2\right)\,. \tag{E.13}$$

---

[82]See *e.g.* [30, 94, 256] for more comprehensive discussion. We mainly follow the notations of [117].

These are given in terms of the curvature two form $R^a_{\ b} = \frac{1}{2} R^a_{\ b\mu\nu} dx^\nu dx^\mu$ defining $\mathrm{tr} R^{2n} = R^{a_1}_{\ a_2} \ldots R^{a_{2n}}_{\ c_1}$. Finally, $ch(B)$ appearing in (E.11) is the Chern character of the entire bundle for all the gauge and global symmetries that are not included in the Dirac A-roof genus. The Chern character can be expanded in terms of the $B$ bundle Chern classes as follows

$$ch(B) = \mathrm{rank}(B) + C_1(B) + \frac{C_1^2(B) - 2C_2(B)}{2} + \frac{C_1^3(B) - 3C_1(B)C_2(B) + 3C_3(b)}{6}$$
$$+ \frac{C_1^4(B) + 4C_1(B)C_3(B) - 4C_1^2(B)C_2(B) + 2C_2^2(B) - 4C_4(B)}{24} + \cdots . \quad (E.14)$$

Here $C_n(B)$ are the $n$-th Chern classes of the bundle $B$ defined by,

$$\sum_n C_n(B) t^k = \det\left(\frac{itF}{2\pi} + I\right) \quad (E.15)$$

$$= I + i\frac{\mathrm{tr}F}{(2\pi)}t + \frac{\mathrm{tr}F^2 - (\mathrm{tr}F)^2}{2(2\pi)^2}t^2 - i\frac{2\mathrm{tr}F^3 - 3\mathrm{tr}F\mathrm{tr}F^2 + (\mathrm{tr}F)^3}{6(2\pi)^3}t^3 + \ldots$$

Here $F$ is the curvature form of $B$. The Chern character obeys

$$ch(B_1 \oplus B_2) = ch(B_1) + ch(B_2), \quad ch(B_1 \otimes B_2) = ch(B_1)ch(B_2). \quad (E.16)$$

We are then ready to write the polynomial for each one of the three types of multiplets. First, we need to understand the global symmetries of the 6d $(1,0)$ SCFTs. These include the $SU(2)_R$ R-symmetry and global symmetry which we will denote by $F_{6d}$. In addition we will also need the contribution of the gauge symmetries, which we will denote by $G_{6d}$. Thus, according to (E.16) we decompose

$$ch(B) = ch(R)ch(G_{6d})ch(F_{6d}).$$

Let us start with the hypermultiplet $\mathcal{H}$. This is specified by its representation under the global symmetry, $r_{F_{6d}}$, and the gauge symmetry, $r_{G_{6d}}$. The Weyl fermion residing in the hypermultiplet is a singlet of the 6d R-symmetry. The anomaly polynomial is then given by,

$$I_8^{\mathcal{H}} = d_{r_{G_{6d}}} d_{r_{F_{6d}}} \frac{7p_1(T)^2 - 4p_2(T)}{5760} + \frac{d_{r_{F_{6d}}}}{12}\left(C_2(G_{6d})^2_{r_{G_{6d}}} - 2C_4(G_{6d})_{r_{G_{6d}}}\right)$$
$$+ d_{r_{F_{6d}}} \frac{p_1(T)C_2(G_{6d})_{r_{G_{6d}}}}{24} - d_{r_{G_{6d}}} \frac{p_1(T)}{48}\left(C_1(F_{6d})^2_{r_{F_{6d}}} - 2C_2(F_{6d})_{r_{F_{6d}}}\right)$$
$$- \frac{1}{2}C_2(G_{6d})_{r_{G_{6d}}}\left(C_1(F_{6d})^2_{r_{F_{6d}}} - 2C_2(F_{6d})_{r_{F_{6d}}}\right) + \frac{1}{2}C_1(F_{6d})_{r_{F_{6d}}}C_3(G_{6d})_{r_{G_{6d}}}$$
$$+ \frac{d_{r_{G_{6d}}}}{24}\left(C_1(F_{6d})^4_{r_{F_{6d}}} + 4C_1(F_{6d})_{r_{F_{6d}}}C_3(F_{6d})_{r_{F_{6d}}} - 4C_1(F_{6d})^2_{r_{F_{6d}}}C_2(F_{6d})_{r_{F_{6d}}}\right.$$
$$\left. + 2C_2(F_{6d})^2_{r_{F_{6d}}} - 4C_4(F_{6d})_{r_{F_{6d}}}\right). \quad (E.17)$$

The $r_B$ index of the Chern classes indicates the representation in which the corresponding traces in (E.15) are computed, while $d_{r_B}$ is the dimension of $r_B$. Note that all the gauge groups in all our constructions are simple and thus the first Chern class is always zero for us.

Second, let us discuss the vector multiplet $\mathcal{V}$. The Weyl fermion in this multiplet is now a doublet of the 6d R-symmetry and its chirality is opposite to the one in the hypermultiplet (and the one in the tensor multiplet to be discussed next). We then compute the polynomial to be,

$$I_8^{\mathcal{V}} = -d_{Adj}\frac{7p_1(T)^2 - 4p_2(T)}{5760} - \frac{1}{12}\left(C_2(G_{6d})^2_{Adj} - 2C_4(G_{6d})_{Adj}\right) - d_{Adj}\frac{C_2(R)_2^2}{24}$$
$$- \frac{p_1(T)C_2(G_{6d})_{Adj}}{24} - d_{Adj}\frac{p_1(T)C_2(R)_2}{48} - \frac{C_2(R)_2 C_2(G_{6d})_{Adj}}{2}. \quad (E.18)$$

Note that here the Chern classes for the gauge bundles are computed in adjoint representation. In addition, note that since we have a single Weyl fermion in the doublet of $SU(2)_R$ and not two Weyl fermions that form a doublet the contribution is divided by 2.

Finally, we compute the contribution of a tensor multiplet $\mathcal{T}$. The tensor contains a single Weyl fermion in the doublet of the R-symmetry with the same chirality as the one in the hypermultiplet.[83] The tensor multiplet also contains a self dual tensor which is chiral. Such chiral fields in general also contribute to gravitational anomaly [257]. Collecting all the contributions we obtain,

$$I_8^{\mathcal{T}} = \frac{23 p_1(T)^2 - 116 p_2(T)}{5760} + \frac{C_2(R)^2}{24} + \frac{p_1(T) C_2(R)}{48}. \tag{E.19}$$

As discussed in the previous sub-section to specify the anomaly polynomial we will also need to account for the contributions of the Green-Schwartz terms.

### E.3 Example: The rank one E-string anomaly polynomial

Here we look at the simple, but important example of the rank one E-string theory anomaly polynomial. The rank $Q$ E-string theory is the low-energy theory on $Q$ M5-branes on top of the end-of-the-world $E_8$ brane. It was shown in [94] that the anomaly polynomial of the rank one E-string theory is given by

$$I_8^{E-string} = \frac{13}{24} C_2(R)_{\mathbf{2}}^2 - \frac{11}{48} C_2(R)_{\mathbf{2}} p_1(T) - \frac{1}{60} C_2(R)_{\mathbf{2}} C_2(E_8)_{\mathbf{248}} + \frac{1}{7200} C_2(E_8)_{\mathbf{248}}^2$$
$$+ \frac{1}{240} p_1(T) C_2(E_8)_{\mathbf{248}} + \frac{29(7 p_1(T) - 4 p_2(T))}{5760}. \tag{E.20}$$

The tensor branch description of this theory contains only one tensor multiplet. Thus, we can write this anomaly polynomial as the contribution of a single tensor multiplet and a Green-Schwartz term involving global and spacetime symmetries [258],

$$I_8^{E-string} = I_8^{tensor} + \frac{1}{2}\left( C_2(R) - \frac{1}{60} C_2(E_8)_{\mathbf{248}} + \frac{1}{4} p_1(T) \right)^2. \tag{E.21}$$

## F Twisting, fluxes, and integrating anomaly polynomials

In this appendix we will show how one can compactify 6d SCFTs to 4d theories preserving half the supersymmetry. In general in order to preserve all the supersymmetry when compactifying a supersymmetric theory the compactification surface needs to be flat. Another option to preserve supersymmetry is to perform a topological twist that preserves part of the supersymmetry.

### F.1 The topological twist and compactification

Topological twisting [259] of a $d$ dimensional supersymmetric theory is performed by a choice of subgroup of the rotation symmetry $SO(d)$ and mapping it into the R-symmetry group. Then given a choice of a supersymmetry generator $Q$ which is invariant under the combined action of $SO(d)$ rotations and the R-symmetry we can consider the subspace of observables which are in the kernel of $Q$. This subspace defines a topological field theory.

---

[83]As in the vector multiplet case we will need to divide its contribution by 2.

In our context we will use topological twisting on the compactifying surface to preserve part of the supersymmetry. We decompose the supersymmetry spinors under $SO(d-1,1)_L \times G_R \to SO(d-p-1,1)_L \times SO(p)_M \times G_R$ and look for a choice of subgroup inclusion of $SO(p)_M$ into $G_R$ that will leave some of the $SO(d-p-1,1)_L$ spinors invariant under the new twisted $SO(p)$ symmetry. Specifically for compactifications of 6d $\mathcal{N} = (1,0)$ theories with $OSp(6,2|2)$ superconformal group the bosonic subgroup is the 6d $\mathcal{N} = (1,0)$ conformal group times $USp(2) \simeq SU(2)$ R-symmetry. In particular, the supercharges $Q$ transform as $(\mathbf{4}, \mathbf{2})$ under $SO(5,1)_L \times SU(2)_R$. Decomposing the supercharge in terms of 4d symmetries we obtain,

$$SO(5,1)_L \times SU(2)_R \to SO(3,1)_L \times SU(2)_R \times SO(2)_M \,,$$
$$Q : (\mathbf{4}, \mathbf{2}) \to (\mathbf{2}, \mathbf{1}; \mathbf{2})_{+\frac{1}{2}} \oplus (\mathbf{1}, \mathbf{2}; \mathbf{2})_{-\frac{1}{2}} \,. \tag{F.1}$$

In this case in order to preserve half the supersymmetry we break $SU(2)_R \to U(1)_R$ and then twist the $SO(2)_M = U(1)_M$ as $U(1)_{M'} = U(1)_M - U(1)_R$. Under the original decomposition and the twist we find

$$SO(5,1)_L \times SU(2)_R \to SO(3,1)_L \times U(1)_{M'} \times U(1)_R \,,$$
$$Q : (\mathbf{4}, \mathbf{2}) \to (\mathbf{2}, \mathbf{1})_{+1,+\frac{1}{2}} \oplus (\mathbf{2}, \mathbf{1})_{0,+\frac{1}{2}} \oplus (\mathbf{1}, \mathbf{2})_{0,-\frac{1}{2}} \oplus (\mathbf{1}, \mathbf{2})_{-1,-\frac{1}{2}} \,. \tag{F.2}$$

Thus, the preserved supercharges are $Q = (\mathbf{2}, \mathbf{1})_{0,+\frac{1}{2}}$ with conjugate $Q^\dagger = (\mathbf{1}, \mathbf{2})_{0,-\frac{1}{2}}$ which means we preserve $\mathcal{N} = 1$ supersymmetry. The 4d R-symmetry is identified with twice $U(1)_R$.

When compactifying the 6d $(1,0)$ theories to 4d there are additional knobs and levers one can utilize due to the fact that many $(1,0)$ theories have flavor symmetries. Specifically we can turn on background configurations for gauge fields for the flavor symmetries supported on the Riemann surface in a way that preserves $\mathcal{N} = 1$ supersymmetry in 4d. One can demonstrate this for example by considering a compactification of a free hypermultiplet in 6d on a torus. Since this is a free field theory the computation can be performed very explicitly. Specifically, one reduces the theory first to 5d on a circle with a holonomy for the global symmetry around the compactification circle that varies along the fifth dimension. Such a configuration amounts to turning on flux for the global symmetry supported on the torus and allows analyzing explicitly the preserved supersymmetry. This analysis was performed by Chan, Ganor, and Krogh in [260] (Section 2), to which we refer the reader interested in the details of such computations.

## F.2 Integrating the anomaly polynomial

When considering the general compactification case on a genus $g$ Riemann surface with non trivial fluxes to abelian subgroups of the 6d flavor symmetry we need to break the flavor symmetry as $G \to U(1)^k \times H$ with $H$ a subgroup of $G$ commuting with all the $U(1)$s we give flux to.

Decomposing the group means we need to decompose a representation of the group $\mathbf{R}$ that appears in the 6d anomaly polynomial as $C_i(G)_\mathbf{R}$, to the representations of $U(1)^k \times H$.

$$\mathbf{R}_\mathfrak{g} \to \sum_i \left(\mathbf{R}_\mathfrak{h}^i\right)^{\left(q_1^i, \ldots, q_k^i\right)} \,, \tag{F.3}$$

where $\mathbf{R}_h^i$ and $q_a^i$ are the $i$-th summand representation of $H$ and charge of $U(1)_a$, respectively, in the decomposition sum.

Writing the Chern character of the 6d global symmetries up to eight forms using (E.14) we get

$$ch(\mathfrak{g}_\mathbf{R}) = \dim\left(\mathbf{R}_\mathfrak{g}\right) - C_2(\mathfrak{g})_{\mathbf{R}_\mathfrak{g}} + \frac{1}{2}C_3(\mathfrak{g})_{\mathbf{R}_\mathfrak{g}} + \frac{1}{12}\left(C_2(\mathfrak{g})_{\mathbf{R}_\mathfrak{g}}^2 - 2C_4(\mathfrak{g})_{\mathbf{R}_\mathfrak{g}}\right) \,, \tag{F.4}$$

where we set $C_1(\mathfrak{g})_{R_\mathfrak{g}} = 0$ as the global symmetries we are interested in 6d have semisimple Lie algebras. Using the properties in (E.16) we can write the Chern character after the above decomposition as

$$
\begin{aligned}
ch(\mathfrak{g}_R) \rightarrow ch\left(\bigoplus_i u(1)_{q_1^i} \otimes \cdots \otimes u(1)_{q_k^i} \otimes \mathfrak{h}_{R_\mathfrak{h}^i}\right) &= \sum_i ch\left(\mathfrak{h}_{R_\mathfrak{h}^i}\right) \prod_{a=1}^{k} ch\left(u(1)_{q_a^i}\right) \\
&= \sum_i \left(\dim\left(R_\mathfrak{h}^i\right) - C_2(\mathfrak{h})_{R_\mathfrak{h}^i} + \frac{1}{2}C_3(\mathfrak{h})_{R_\mathfrak{h}^i} + \frac{1}{12}\left(C_2(\mathfrak{h})_{R_\mathfrak{h}^i}^2 - 2C_4(\mathfrak{h})_{R_\mathfrak{h}^i}\right) + \dots\right) \\
&\quad \times \prod_{a=1}^{k} \sum_{n=0}^{\infty} \left(\frac{1}{n!}\left(q_a^i C_1(u(1)_a)\right)^n\right),
\end{aligned}
\tag{F.5}
$$

where $C_1(\mathfrak{h})_{R_\mathfrak{h}^i} = 0$ as $H$ has a semisimple Lie algebra. Comparing forms of equal dimension we find how the Chern classes decompose

$$
\begin{aligned}
C_2(\mathfrak{g})_{R_\mathfrak{g}} \;\rightarrow\; & -\frac{1}{2}\sum_{a,b=1}^{k}\sum_i \dim\left(R_\mathfrak{h}^i\right) q_a^i q_b^i C_1(u(1)_a) C_1(u(1)_b) + \sum_i C_2(\mathfrak{h})_{R_\mathfrak{h}^i}, \\
C_3(\mathfrak{g})_{R_\mathfrak{g}} \;\rightarrow\; & \frac{1}{3}\sum_{a,b,c=1}^{k}\sum_i \dim\left(R_\mathfrak{h}^i\right) q_a^i q_b^i q_c^i C_1(u(1)_a) C_1(u(1)_b) C_1(u(1)_c) \\
& -2\sum_{a=1}^{k}\sum_i q_a^i C_1(u(1)_a) C_2(\mathfrak{h})_{R_\mathfrak{h}^i} + \sum_i C_3(\mathfrak{h})_{R_\mathfrak{h}^i}, \\
C_4(\mathfrak{g})_{R_\mathfrak{g}} \;\rightarrow\; & -\frac{1}{4}\sum_{a,b,c,d=1}^{k}\sum_i \dim\left(R_\mathfrak{h}^i\right) q_a^i q_b^i q_c^i q_d^i C_1(u(1)_a) C_1(u(1)_b) C_1(u(1)_c) C_1(u(1)_d) \\
& +3\sum_{a,b=1}^{k}\sum_i \left(1 - \frac{1}{6}\dim\left(R_\mathfrak{h}^i\right)\right) q_a^i q_b^i C_1(u(1)_a) C_1(u(1)_b) C_2(\mathfrak{h})_{R_\mathfrak{h}^i} \\
& -3\sum_{a=1}^{k}\sum_i q_a^i C_1(u(1)_a) C_3(\mathfrak{h})_{R_\mathfrak{h}^i} - \frac{1}{2}\sum_i C_2(\mathfrak{h})_{R_\mathfrak{h}^i}^2 + \sum_i C_4(\mathfrak{h})_{R_\mathfrak{h}^i} \\
& +\frac{1}{2}\left(-\frac{1}{2}\sum_{a,b=1}^{k}\sum_i \dim\left(R_\mathfrak{h}^i\right) q_a^i q_b^i C_1(u(1)_a) C_1(u(1)_b) + \sum_i C_2(\mathfrak{h})_{R_\mathfrak{h}^i}\right)^2. \tag{F.6}
\end{aligned}
$$

Next we wish to fix the flux on the compactification Riemann surface $\Sigma_g$ of genus $g$ for the selected $U(1)$ subgroups. Thus we set

$$
\int_{\Sigma_g} C_1(U(1)_a) = -z_a. \tag{F.7}
$$

In addition, we perform the aforementioned topological twist which translates to setting the R-symmetry Chern class as

$$
C_2(R) \rightarrow -\left(C_1(R') - \frac{t}{2}\right)^2, \tag{F.8}
$$

where $t$ is a two form related to the compactification surface symmetry and integrates to

$$
\int_{\Sigma_g} t = 2(1-g). \tag{F.9}
$$

We also set the Pontryagin classes

$$p_1\left(T_{6d}\right) \to p_1\left(T_{4d}\right) + t^2\,, \quad p_2\left(T_{6d}\right) \to p_2\left(T_{4d}\right) + p_1\left(T_{4d}\right)t^2\,, \tag{F.10}$$

and to meet the flux constraint

$$C_1\left(U(1)_i\right) \to -z_i \frac{t}{2(1-g)} + \epsilon_i C_1\left(U(1)_R\right) + C_1\left(U(1)_{F_i}\right)\,, \tag{F.11}$$

where the first term gives the required flux, the second term takes into account possible mixing of the $U(1)$ symmetry with the superconformal R-symmetry, where $\epsilon$ is a parameter to be determined via $a$-maximization [76], and the third term is the 4d curvature term of the $U(1)$ symmetry. Note that when we compactify the theory on a generic Riemann surface, only terms linear in $t$ will contribute to the 4d anomaly polynomial six-form.[84]

Finally, with the 4d anomaly polynomial at hand one can calculate the anomalies of various symmetries. For $U(1)$ symmetries one can extract the anomalies from the anomaly polynomial using the relations

$$Tr\left(U(1)_x\right) = -24\partial_{C_1(U(1)_x)}\partial_{p_1(T_4)}I_6\,,$$
$$Tr\left(U(1)_x U(1)_y U(1)_z\right) = \partial_{C_1(U(1)_x)}\partial_{C_1(U(1)_y)}\partial_{C_1(U(1)_z)}I_6\,. \tag{F.12}$$

For mixed or cubic non-abelian anomalies one finds the relations

$$Tr\left(G^3\right) = 2C_{\mathbf{r}}\partial_{C_3(G)_{\mathbf{r}}}I_6\,,$$
$$Tr\left(G^2 U(1)_x\right) = -T_{\mathbf{r}}\partial_{C_1(U(1)_x)}\partial_{C_2(G)_{\mathbf{r}}}I_6\,, \tag{F.13}$$

where $T_{\mathbf{r}}$ and $C_{\mathbf{r}}$ denote the Dynkin and cubic index of the representation $\mathbf{r}$, respectively. These relations can be derived by comparing the contribution of Weyl fermions to the anomalies on one side and to the anomaly polynomial on the other side.

## F.3 Examples

Here we will examine the E-string theory and its compactification possibilities preserving 4d $\mathcal{N} = 1$ supersymmtry discussed in this appendix. We start with a simple compactification on a genus $g$ Riemann surface with no fluxes. We first write the full 6d anomaly polynomial for a rank $Q$ E-string theory

$$
\begin{aligned}
I_8^{E-string} = {}& \frac{Q\left(4Q^2 + 6Q + 3\right)}{24}C_2^2(R) + \frac{(Q-1)\left(4Q^2 - 2Q + 1\right)}{24}C_2^2(L) \\
& - \frac{Q\left(Q^2 - 1\right)}{3}C_2(R)C_2(L) - \frac{(Q-1)(6Q+1)}{48}C_2(L)p_1(T) \\
& - \frac{Q(6Q+5)}{48}C_2(R)p_1(T) + \frac{Q(Q-1)}{120}C_2(L)C_2(E_8)_{\mathbf{248}} \\
& - \frac{Q(Q+1)}{120}C_2(R)C_2(E_8)_{\mathbf{248}} + \frac{Q}{240}p_1(T)C_2(E_8)_{\mathbf{248}} \\
& + \frac{Q}{7200}C_2(E_8)_{\mathbf{248}}^2 + (30Q-1)\frac{7p_1^2(T) - 4p_2(T)}{5760}\,,
\end{aligned} \tag{F.14}
$$

where in our notations $C_i(G)_{\mathbf{r}_G}$ denoted the $i$-th Chern class of the $G$-bundle in the $\mathbf{r}_G$ representation, $C_2(R) \equiv C_2(R)_{\mathbf{2}}$ and $C_2(L) \equiv C_2(L)_{\mathbf{2}}$, and $p_i(T)$ is the $i$-th Pontryagin class of the tangent bundle.

---

[84]Terms with higher powers of $t$ can still contribute to other anomalies. Notable examples include: Anomalies in symmetries originating from isometries of the Riemann surface [137], and anomalies associated with couplings on the conformal manifold [261].

Since we wish to compactify on a genus $g$ Riemann surface we need to use a topological twist to preserve half the supersymmetry as shown previously. This translates to the assignment

$$C_2(R) \to -\left(C_1\left(R'\right) - \frac{t}{2}\right)^2 . \tag{F.15}$$

In addition, we need to set the Pontryagin classes

$$p_1\left(T_{6d}\right) \;\to\; p_1\left(T_{4d}\right) + t^2 , \quad p_2\left(T_{6d}\right) \to p_2\left(T_{4d}\right) + p_1\left(T_{4d}\right) t^2 , \tag{F.16}$$

and finally set

$$\int_{\Sigma_g} t = 2(1-g) . \tag{F.17}$$

Effectively this means that in the resulting 4d anomaly polynomial only the terms linear in $t$ survive. This results in

$$
\begin{aligned}
I_6^{E-string} =\; & \frac{Q\left(4Q^2 + 6Q + 3\right)}{6} (g-1) C_1^3(R) + \frac{Q(6Q+5)}{24} (g-1) C_1(R) p_1\left(T_4\right) \\
& + \frac{2Q\left(Q^2 - 1\right)}{3} (g-1) C_1(R) C_2(L) + \frac{Q(Q+1)}{60} (g-1) C_1(R) C_2(E_8)_{\mathbf{248}} .
\end{aligned}
\tag{F.18}
$$

From the anomaly polynomial we can extract that the only nontrivial anomalies are

$$
\begin{aligned}
Tr\left(U(1)_R^3\right) &= Q\left(4Q^2 + 6Q + 3\right)(g-1) , & Tr\left(U(1)_R\right) &= -Q(6Q+5)(g-1) , \\
Tr\left(U(1)_R SU(2)_L^2\right) &= -\frac{Q\left(Q^2 - 1\right)}{3}(g-1) , & Tr\left(U(1)_R E_8^2\right) &= -\frac{Q(Q+1)}{2}(g-1) .
\end{aligned}
\tag{F.19}
$$

In the next example we will again compactify the E-string theory only this time on a torus with flux preserving a $U(1) \times E_7$ subgroup of the $E_8$ group. This requires breaking $E_8 \to E_7 \times U(1)$ using the branching rule

$$\mathbf{248} \to \mathbf{133} \oplus \mathbf{56}_{+1} \oplus \mathbf{56}_{-1} \oplus \mathbf{1}_{+2} \oplus \mathbf{1} \oplus \mathbf{1}_{-2} . \tag{F.20}$$

This translates to the following Chern class assignments using (F.6)

$$C_2(E_8)_{\mathbf{248}} \to C_2\left(E_7\right)_{\mathbf{133}} + 2C_2\left(E_7\right)_{\mathbf{56}} - 60 C_1(U(1))^2 = 5 C_2\left(E_7\right)_{\mathbf{56}} - 60 C_1(U(1))^2 , \tag{F.21}$$

where in the second equality we used the relation implied by (F.13). Since we are dealing with a torus, no topological twist is required, but due to the flux we still break half the supersymmetry. This is consistent with the assignments we used in the former example when taking $g = 1$. In addition we need to set in accordance with the flux

$$C_1(U(1)) \;\to\; -z \frac{t}{2(1-g)} + \epsilon C_1\left(U(1)_R\right) + C_1\left(U(1)_F\right) , \tag{F.22}$$

where here we will take the limit of $g \to 1$ after the compactification.

$$
\begin{aligned}
I_6^{E-string} =\; & z\left(C_1(F) + \epsilon C_1(R)\right)\left(\frac{Q}{2} p_1\left(T_4\right) + Q(Q-1) C_2(L) + \frac{Q}{6} C_2\left(E_7\right)_{\mathbf{56}}\right) \\
& + Q(Q+1) z\left(C_1(F) + \epsilon C_1(R)\right) C_1^2(R) - 2Q z\left(C_1(F) + \epsilon C_1(R)\right)^3 .
\end{aligned}
\tag{F.23}
$$

In this case since we have an additional $U(1)$ flavor symmetry which can mix with the R-symmetry in the IR fixed point. The mixing can be calculated using $a$-maximization, where the trial $a$-anomaly is

$$a_{trial} = \frac{3}{32}\left(18Q\epsilon\left(Q - 2\epsilon^2 + 1\right)z + 12Q\epsilon z\right).$$ (F.24)

This anomaly is maximized for

$$\epsilon = \sqrt{\frac{3Q+5}{18}}\text{sign}(z).$$ (F.25)

For our final example we will consider a compactification on a genus $g$ Riemann surface with flux preserving $U(1) \times E_7 \subset E_8$. In this case we use the same process as the former example only without setting the genus to one. The resulting anomaly polynomial setting the mixing coefficient to zero for brevity is

$$
\begin{aligned}
I_6^{E-string} &= \frac{Q\left(4Q^2 + 6Q + 3\right)}{6}(g-1)C_1^3(R) + \frac{Q(6Q+5)}{24}(g-1)C_1(R)p_1\left(T_4\right) \\
&+ \frac{2Q\left(Q^2 - 1\right)}{3}(g-1)C_1(R)C_2(L) + \frac{Q(Q+1)}{12}(g-1)C_1(R)C_2\left(E_7\right)_{\mathbf{56}} \\
&+ Q(Q+1)z_t C_1(t)C_1^2(R) + Q(Q+1)(g-1)C_1^2(t)C_1(R) - 2Qz_t C_1^3(t) \\
&+ \frac{Q}{2}z_t C_1(t)p_1\left(T_4\right) + Q(Q-1)z_t C_1(t)C_2(L) + \frac{Q}{6}z_t C_1(t)C_2\left(E_7\right)_{\mathbf{56}}.
\end{aligned}
$$ (F.26)

As before one can find the mixing of the $U(1)$ with the R-symmetry in the IR.

# G  Supersymmetric boundary conditions

In this section we focus on five dimensional quantum field theories with $\mathcal{N} = 1$ supersymmetry and discuss the most basic boundary conditions that one can impose while preserving half of the supersymmetry.

Five dimensional $\mathcal{N} = 1$ gauge theories have eight real supercharges and two kinds of multiplets, a vector multiplet containing a real scalar $\Phi$, fermions $\Psi$ and a vector field $A$, and a hypermultiplet containing four real scalars $\phi$ and fermions $\psi$. The Lorentz group is given by $SO(4,1)$ and there is an $SU(2)_R$ R-symmetry. In fact, there is an $SO(4)$ symmetry rotating the four real scalars in the hypermultiplet which is the product of $SU(2)_R$ and an additional $SU(2)_F$ flavor symmetry. Denoting the representations under the bosonic $SO(4,1) \times SU(2)_R \times SU(2)_F$ symmetry by $(\mathbf{R}_{SO(4,1)}, \mathbf{R}_{SU(2)_R}, \mathbf{R}_{SU(2)_F})$, the supercharges $Q_\alpha^i$ transform as $(\mathbf{4}, \mathbf{2}, \mathbf{1})$, the scalars $\phi^{ia}$ and fermions $\psi_\alpha^a$ of the hypermultiplet as $(\mathbf{1}, \mathbf{2}, \mathbf{2})$ and $(\mathbf{4}, \mathbf{1}, \mathbf{2})$ (respectively), and the fermions $\Psi_\alpha^i$ of the vector multiplet as $(\mathbf{4}, \mathbf{2}, \mathbf{1})$ (with the real scalar and the vector field transforming trivially under $SU(2)_R \times SU(2)_F$). Here the indices $i, a = 1, 2$ and $\alpha = 1, \ldots, 4$ correspond to $SU(2)_R$, $SU(2)_F$ and $SO(4,1)$, respectively.

Our goal is to find the most simple boundary conditions one can impose in a 5d $\mathcal{N} = 1$ gauge theory such that half of the supersymmetry, corresponding to $\mathcal{N} = 1$ in four dimensions, is preserved. Concretely, we will focus on free hyper and vector multiplets which appear in the low-energy limit of such gauge theories.

We begin with addressing the case of a free hypermultiplet, where the theory is considered in the domain $x^4 < 0$ and a boundary condition is given at $x^4 = 0$ (where the coordinates are $x^M$ with $M = 0, \ldots, 4$). In order to find which kinds of simple boundary conditions preserve half of the supersymmetry, we first describe the theory in terms of four dimensional $\mathcal{N} = 1$

multiplets. Then, finding boundary conditions that respect this multiplet structure (and therefore preserve the desired supersymmetry) will be natural.[85]

The action of a free hypermultiplet is given by

$$S = \int d^5 x \left( -\frac{1}{4} \epsilon_{ij} \epsilon_{ab} \partial_M \phi^{ia} \partial^M \phi^{jb} + \frac{1}{2} \epsilon_{ab} \psi^a_\alpha \Gamma^{M\alpha\beta} \partial_M \psi^b_\beta \right), \tag{G.1}$$

where $\Gamma^M$ are the Gamma matrices in five dimensions. When written using the 4d $\mathcal{N} = 1$ sub-supersymmetry, such a hypermultiplet decomposes into an $SU(2)_F$ doublet of chiral multiplets $(X, Y)$. The Cartan of $SU(2)_R$ becomes the $U(1)_R$ R-symmetry in four dimensions, under which both $X$ and $Y$ have R-charge 1. We would like to write the action (G.1) in terms of the chiral multiplets $X$ and $Y$. It is easy to see that in this description we should include a superpotential, since the standard kinetic terms,

$$\int dx^\perp d^4 x d^4 \theta \left( X^\dagger X + Y^\dagger Y \right) \tag{G.2}$$

(where we denote $x^\perp = x^4$), do not reproduce the full action (G.1). As an example, notice that only the $\epsilon_{ij} \epsilon_{ab} \partial_\mu \phi^{ia} \partial^\mu \phi^{jb}$ part of the scalar-field kinetic term, where $\mu = 0, \ldots, 3$, is recovered. In order to obtain the missing piece, $\epsilon_{ij} \epsilon_{ab} \partial_\perp \phi^{ia} \partial^\perp \phi^{jb}$, it is easy to see that we should add the following superpotential,

$$\int dx^\perp d^4 x d^2 \theta \, Y \partial_\perp X, \tag{G.3}$$

that includes it as well as the missing part of the rest of the action (G.1) . Indeed, the superpotential (G.3) sets the $F$-term of the $X$ multiplet to be (up to a coefficient) $F_{(X)} = \partial_\perp \overline{Y}$, where $\overline{Y}$ is the scalar component of the anti-chiral multiplet $\overline{Y}$. Similarly, it sets $F_{(Y)} = \partial_\perp \overline{X}$ such that the total contribution of the superpotential (G.3) to the scalar-field potential is given by the missing part, which involves the derivatives of the scalars in the $x^\perp$ direction.

Now that we have expressed the action (G.1) in terms of the chiral multiplets $X$ and $Y$, we can look for simple boundary conditions that preserve the 4d $\mathcal{N} = 1$ sub-supersymmetry. A common choice is to give the scalar of $X$ a Dirichlet boundary condition, setting it equal to zero at $x^\perp = 0$. By (4d $\mathcal{N} = 1$) supersymmetry, this implies that the rest of the $X$ multiplet vanishes at the boundary, including the $F$-term $\partial_\perp \overline{Y}$. This enforces Neumann boundary conditions for $Y$, resulting in

$$X|_{x^\perp=0} = 0, \quad \partial_\perp Y|_{x^\perp=0} = 0. \tag{G.4}$$

In a similar way, one can give Dirichlet boundary conditions for $Y$, enforcing Neumann boundary conditions for $X$,

$$Y|_{x^\perp=0} = 0, \quad \partial_\perp X|_{x^\perp=0} = 0. \tag{G.5}$$

Let us next turn to discuss vector multiplets, focusing for simplicity on a $U(1)$ gauge group. As in the case of the hypermultiplet, we would first like to express the theory in terms of 4d $\mathcal{N} = 1$ multiplets. In this case, the 5d vector multiplet decomposes into a 4d vector multiplet $V$ and a chiral multiplet $S$, such that the bottom component of the chiral multiplet is given by the complex scalar field $\Phi + iA_\perp$. Recall that we do not consider the 5d vector multiplet as coupled to matter in an interacting theory; however, in such a case the covariant derivative in the $x^\perp$ direction is given by $\partial_\perp - S$. Notice also that the chiral multiplet $S$ has an unusual gauge transformation due to the appearance of $A_\perp$ in its scalar component, given by $S \rightarrow S + i\partial_\perp \Lambda$

---

[85]Notice that the full 5d $\mathcal{N} = 1$ supersymmetry cannot be preserved since the anti-commutator of some of the supercharges yields a translation in the $x^4$ direction, which is broken by the boundary condition at $x^4 = 0$.

for an abelian gauge transformation with chiral-multiplet parameter $\Lambda$. The 5d action is reproduced by the standard kinetic term for $V$, plus a slightly-modified kinetic term for $S$ that takes into account its unusual gauge transformation. In contrast to the previous case of the hypermultiplet, no superpotential for $S$ is needed here (and therefore its $F$ field vanishes). The kinetic terms of $V$ and $S$ set the $D$ field of $V$ to be (up to a coefficient) $D = \partial_\perp \Phi$, resulting in a contribution to the potential of $\Phi$ that reproduces the part of its kinetic term involving derivatives in the $x^\perp$ direction.

Turning to possible boundary conditions that preserve the 4d $\mathcal{N} = 1$ sub-supersymmetry, a simple choice would be to give the gauge field $A$ a Dirichlet boundary condition, setting $A_\mu = 0$ at $x^\perp = 0$ (where recall that $\mu = 0, \ldots, 3$). Since this condition is only preserved by gauge transformations with parameters which are constant along the boundary $x^\perp = 0$, the gauge symmetry actually breaks to a global symmetry there.[86] The (4d $\mathcal{N} = 1$) supersymmetric completion of this boundary condition simply sets the entire 4d vector multiplet $V$ to zero on the boundary, including its $D$-term. This results in the boundary conditions

$$V|_{x^\perp=0} = 0 \;\Rightarrow\; A_\mu\big|_{x^\perp=0} = 0, \qquad \lambda_\sigma|_{x^\perp=0} = 0, \, \partial_\perp \Phi|_{x^\perp=0} = 0, \tag{G.6}$$

where we denoted the fermion of $V$ by $\lambda_\sigma$.

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
