# Peer review of "Aspects of 4d supersymmetric dynamics and geometry"

_SciPost Physics Lecture Notes, doi:SciPost Phys. Lect. Notes 78 (2024)_

## Round 1 · Referee Report · Anonymous (Referee 1) · 2022-10-5

Strengths

1- The topic has seen outstanding developments in the past few years and have not appeared in comprehensive lecture notes yet. 2- The authors are the most qualified to write about the subject.

Weaknesses

No particular weakness.

Report

These Lecture notes cover the dynamics of 4d $\mathcal{N}=1$ SCFTs with an emphasis on dualities and conformal manifolds. While this topic has been studied since the 90s, the past 10 years or so have witnessed impressive developments with a novel geometric understanding, in particular using compactifications from 6d SCFTs, with an optional stop in 5d. This in turn harnesses the recent understanding and classification of 6d and 5d SCFTs using geometry and string/M/F theory. The authors have played a central role in these developments, and their mastery of the subject and its subtleties is breathtaking.

The lecture notes fill a gap in the literature and will certainly become a reference on this active topic. The presence of corrected exercises and several appendices reviewing basic material is highly appreciated, and I have enjoyed reading them in detail.

As such, I highly recommend it for publication in SciPost Physics Lecture Notes.

Requested changes

Here are a few (mostly cosmetic) remarks about the text, along with points that could be clarified. - p. 9: "lets" $\rightarrow$ "let's". - p. 9: "possess" $\rightarrow$ "possesses". - p. 10: The notation in (2.3) could be explained, as this is the first time anomalies are computed. - p. 11: Maybe explain what weakly gauging means. - p. 11: Can you comment on the assumption that the theory has R symmetry? Are there known counter examples? - p. 12: It could be good to explain what (2.7) becomes when the U(1) is taken to be R-symmetry. - p. 15: Say before (2.20) that P refers to the SCFT under study. - p. 17: How does tau transform under the anomalous symmetry in (2.23)? - p. 26: "its invariant" $\rightarrow$ "it's invariant" - p. 26: "the number of the short multiplets" $\rightarrow$ "the number of short multiplets". - p. 28: Capital letter after (2.54). - p. 28: The sentence after (2.55) seems to be truncated. - p. 30: Before delving in the $N_f > N$ window it could be interesting to state in one sentence what happens for $N_f \leq N$. - p. 32: In the exercise $N_f=4$ should be added. And a capital letter is missing. - p. 33: "in the last Lecture" $\rightarrow$ "in the next Lecture"? - p. 33: "in-equivalent" $\rightarrow$ "inequivalent"? - p. 35: What is the argument showing that the character of the adjoint appearing at order $pq$ implies $E_6$ symmetry of the whole theory? - p. 37: The dimension of the conformal manifold is presented as an observable that is constant on $\mathcal{M}_c$, what is the argument? In other words why is $\mathcal{M}_c$ a manifold (and not e.g. the union of manifolds of different dimensions)? - p. 37: "be still wrong" $\rightarrow$ "still be wrong". - p. 38: the "9" below (3.8) should have another color. - p. 48: above (4.2) the notation $\phi$ and $B$ should be introduced. (or refer to Appendix E). - p. 57: In the first line the "$+s$" should be removed for a closed Riemann Surface. - p. 64: For pedagogical reasons it could be helpful to define and give a simple example of holonomy, which is introduced briefly in Lecture IV on a complex example with $E_8$ symmetry without much explanation. The importance of the concept is repeated in Lecture V but proceeds immediately with variable holonomy. - p. 95: What are the conditions on $M_1$ and $M_2$ to get the desired breaking of flavor symmetry? Similarly on p. 103 when $M_3$ is introduced. E.g. what happens if $M_1 = M_3$? - p. 95: the choice of the flux proportional to $\delta(x_5) \delta (x_6)$ can be disconcerting at this point. It could be mentioned that there will later be an interpretation of this point as a puncture.
- p. 113: In which circumstances would a 6d operator not reduce to a 4d one? - p. 113: "appear" $\rightarrow$ "appears" - p. 115: "get" $\rightarrow$ "gets" - p. 146: below (E.16) $F_{6d}$ should be $G_{6d}$. Below, the notation for representations is $\mathcal{R}$ in the text and $r$ in the equations.

  • validity: top
  • significance: top
  • originality: -
  • clarity: high
  • formatting: excellent
  • grammar: good

Author:  Shlomo Razamat  on 2022-11-29  [id 3084]

(in reply to Report 1 on 2022-10-05)

We would like to thank the referee for the thorough reading of the paper and the thoughtful report.

We will make the suggested corrections and expand the lecture notes with the requested explanations in an amended version. For example:

-- page 11: The R-symmetry might be broken by superpotentials or be anomalous

-- page 35: Character of adjoint of $E_6$ appearing at order $qp $ with negative sign can come only from conserved currents following the analysis of representation theory of the superconformal group (discussed around (2.52)).

-- page 37: An argument why the space of exactly marginal couplings in a supersymmetric theory is a manifold is that the number of supersymmetric exactly marginal deformations is an invariant under continuous deformations as it corresponds to counting certain protected operators in the theory.

-- page 95: The symmetry preserved by the holonomies is generated by elements of the 6d symmetry commuting with $M_i$. If the two matrices are equal the symmetry is still broken but there is no flux.

--page 113: One can view compactification as RG flow between $6d$ theory and $4d$ theory. Then as in a usual RG flow we can trace protected operators along RG flow modulo recombinations. An example is a free massive theory which becomes gapped in the IR.

We will expand on all of these questions in the revised version and correct all the other issues raised by the referee.

---

## Round 2 · Referee Report · Anonymous (Referee 1) · 2024-1-10

Strengths

1- The topic has seen outstanding developments in the past few years and have not appeared in comprehensive lecture notes yet. 2- The authors are the most qualified to write about the subject.

Weaknesses

No particular weakness.

Report

The authors have addressed all the points I raised in the previous report. I highly recommend it for publication in SciPost Physics Lecture Notes.

---

## Round 2 · Author Response

We thank the referee again for thoroughly going over the paper and for the very useful comments.

We have edited the file according to referee's comments, correcting the typos and expanding a bit on some of the points.

---

## Editorial Decision

published